# Learning under Quantization for High-Dimensional Linear Regression

**Dechen Zhang**
Institute of Data Science
The University of Hong Kong
dechenzhang@connect.hku.hk

**Junwei Su**
School of Computing & Data Science
The University of Hong Kong
junweisu@connect.hku.hk

**Difan Zou**
School of Computing & Data Science and Institute of Data Science
The University of Hong Kong
dzou@hku.hk

## Abstract

The use of low-bit quantization has emerged as an indispensable technique for enabling the efficient training of large-scale models. Despite its widespread empirical success, a rigorous theoretical understanding of its impact on learning performance remains notably absent, even in the simplest linear regression setting. We present the first systematic theoretical study of this fundamental question, analyzing finite-step stochastic gradient descent (SGD) for high-dimensional linear regression under a comprehensive range of quantization targets: data, label, parameter, activation, and gradient. Our novel analytical framework establishes precise algorithm-dependent and data-dependent excess risk bounds that characterize how different quantization affects learning: parameter, activation, and gradient quantization amplify noise during training; data quantization distorts the data spectrum and introduces additional approximation error. Crucially, we distinguish the effects of two quantization schemes: we prove that for additive quantization (with constant quantization steps), the noise amplification benefits from a suppression effect scaled by the batch size, while multiplicative quantization (with input-dependent quantization steps) largely preserves the spectral structure, thereby reducing the spectral distortion. Furthermore, under common polynomial-decay data spectra, we quantitatively compare the risks of multiplicative and additive quantization, drawing a parallel to the comparison between FP and integer quantization methods. Our theory provides a powerful lens to characterize how quantization shapes the learning dynamics of optimization algorithms, paving the way to further explore learning theory under practical hardware constraints.

## 1 Introduction

Quantization has garnered widespread attention as an essential technique for deploying large-scale deep learning models, particularly large language models (LLMs) (Lang et al., 2024; Shen et al., 2024). In line with this low-precision paradigm, a new frontier of research has emerged: quantization scaling laws, which seek to formalize the trade-offs between model size, dataset size, and computational bit-width. Seminal work by Kumar et al. (2024) treated bit-width as a discrete measure of precision. This was extended by Sun et al. (2025), who established a more comprehensive scaling law for floating-point (FP) quantization (Kuzmin et al., 2022) by separately accounting for the distinct roles of exponent and mantissa bits. Going further, Chen et al. (2025) proposed a unified scaling law that models quantized error as a function of model size, training data volume, and quantization group size. Collectively, these studies provide rigorous understandings to guide the joint allocation of fixed compute or memory budgets across data size, model size and precision (quantization bit-width).

The empirical understanding of low-precision training has advanced rapidly, yet a significant theory-practice gap persists. Theoretical research remains predominantly restricted to analyzing *convergence guarantees on the training loss* for quantized optimizers (Nadiradze et al., 2021; Liu et al., 2023; Markov et al., 2023; Xin et al., 2025). For example, Markov et al. (2023) derived convergence guarantee for the communication-efficient variant of Fully-Shared Data-Parallel distributed training under parameter and gradient quantization. While these studies offer crucial insights into optimization, they overlook a more fundamental question: *how does quantization affect the model's learning performance?* Specifically, a rigorous characterization of the interplay between quantization, model dimension, dataset size, and their joint effect on the *population risk* remains largely unexplored. A notable step in this direction is Zhang et al. (2022), which analyzed the generalization of quantized two-layer networks through the lens of neural tangent kernel (NTK). However, their work is limited in three key aspects: it only considers parameter quantization; its analysis is confined to the lazy-training regime; and it fails to provide explicit generalization bounds in terms of core parameters like sample size, dimension, and quantization error. These limitations restrict its applicability to modern low-precision training practices.

Motivated by recent theoretical advances in scaling laws (Lin et al., 2024; 2025; Li et al., 2025), we analyze the learning performance of quantized training using a high-dimensional linear model. This model serves as a powerful and well-established testbed for isolating phenomena like learning rate and batch size effects (Kunstner & Bach, 2025; Luo et al., 2025; Zhang et al., 2024b; Xiao, 2024; Ren et al., 2025; Bordelon et al., 2025). Its simplicity provides the analytical flexibility necessary to derive precise relationships between generalization error and critical parameters such as dimension, sample size, and quantization error (or bit-width).

**Our setting.** In this paper, we consider SGD for linear regression under quantization. We first iterate the standard linear regression problem as follows:

$$\min_{\mathbf{w}} L(\mathbf{w}), \text{ where } L(\mathbf{w}) = \frac{1}{2}\mathbb{E}_{\mathbf{x},y}\left[\left(y - \langle \mathbf{w}, \mathbf{x} \rangle\right)^2\right].$$

Here $\mathbf{x} \in \mathcal{H}$ is the feature vector, $\mathcal{H}$ is some (finite $d$-dimensional or countably infinite dimensional) Hilbert space, $y \in \mathbb{R}$ is the response, $\mathcal{D}$ is an unknown distribution over $\mathbf{x}$ and $y$, and $\mathbf{w} \in \mathcal{H}$ is the weight vector to be optimized. We consider the constant step size SGD under quantization: at each iteration $t$, an i.i.d. batch (with batch size $B$) of examples $(\mathbf{X}_t, \mathbf{y}_t) \in \mathbb{R}^{B \times d} \times \mathbb{R}^B$ is observed, and the weight $\mathbf{w}_t \in \mathbb{R}^d$ is updated according to following quantized SGD algorithm.

$$\mathbf{w}_t = \mathbf{w}_{t-1} + \gamma\frac{1}{B}\mathcal{Q}_d(\mathbf{X}_t)^\top \mathcal{Q}_o\Big(\mathcal{Q}_l(\mathbf{y}_t) - \mathcal{Q}_a\big(\mathcal{Q}_d(\mathbf{X}_t)\mathcal{Q}_p(\mathbf{w}_{t-1})\big)\Big), \quad t = 1, ..., N,$$

(quantized SGD)

where $\gamma > 0$ is a constant stepsize, $N$ is the number of sample batches observed, the master weights is initialized at $\mathbf{w}_0$, and $\mathcal{Q}_d, \mathcal{Q}_l, \mathcal{Q}_p, \mathcal{Q}_a, \mathcal{Q}_o$ are independent general quantization operations for data feature, label, model parameter, activation and output gradient respectively. Notably, for theoretical simplicity, we assume all matrix operations (e.g., addition and multiplication) are computed in full precision, with quantization applied subsequently to obtain low-precision values. Then, we consider the average iterate as the algorithm output, i.e., $\overline{\mathbf{w}}_N := \frac{1}{N}\sum_{t=0}^{N-1}\mathbf{w}_t$. Without loss of generality, we assume the initial parameter is $\mathbf{w}_0 = 0$.

The goal of this work is to characterize the learning performance of the quantized SGD via evaluating the population risk $L(\overline{\mathbf{w}}_N)$, and more importantly, its relationship with the quantization error. Let $\mathbf{w}^* = \arg\min L(\mathbf{w})$, we define the following excess risk as a surrogate of the population risk:

$$\mathcal{E}(\overline{\mathbf{w}}_N) = L(\overline{\mathbf{w}}_N) - L(\mathbf{w}^*). \tag{Excess Risk}$$

**Our contributions.** We perform a novel theoretical study on the learnability of the quantized SGD algorithm for high-dimensional linear regression problems. Our contributions are summarized as follows:

- We perform systematic analysis and establish a theoretical bound for the excess risk of quantized SGD. This bound is explicitly formulated as a function of the full eigen-spectrum of the quantized data feature covariance, sample size, and quantization errors (see Theorem 4.1 for details). Our results precisely reveal how quantization applied to different model components impacts learning performance: quantization of data distorts the spectrum of effective data covariance and introduces

an additional approximation error; while the quantization of parameter, activation, and output gradient amplify noise throughout the training process on the quantized feature space.

- We analyze two standard quantization error models: additive and multiplicative, which conceptually relate to the integer and FP quantization techniques. For additive quantization, our theoretical bounds indicate that the noise amplification stemming from activation and output gradient quantization diminishes as the batch size increases, whereas the spectrum of the effective data covariance is distorted by a constant noise floor (see Corollary 4.1 for details). Conversely, for multiplicative quantization, our results demonstrate that data quantization preserves the intrinsic spectral structure of the effective covariance, thereby reducing spectral distortion; however, the resulting noise amplification remains independent of the batch size (see Theorem 4.2 for details).

- We further derive the conditions on the quantization errors such that the learning performance of the full-precision SGD can be maintained (in orders). Our results indicate that compared with multiplicative quantization, additive quantization necessitates stricter spectral constraints on data quantization but allows for more relaxed conditions on activation and output gradient quantization, benefiting from the batch-averaging effect (see Corollary 4.2 for details). By applying our excess risk bounds to polynomial decay spectrum, we show that multiplicative quantization is applicable even in high-dimensional settings, whereas additive quantization is not (see Corollary 4.3 for details). These simplified theoretical results also draw implications for comparing integer and FP quantization, allowing us to identify the conditions under which each type is likely to yield superior performance.

**Notations.** For two positive-valued functions $f(x)$ and $g(x)$, we write $f(x) \lesssim g(x)$ or $f(x) \gtrsim g(x)$ if $f(x) \leq cg(x)$ or $f(x) \geq cg(x)$ holds for some absolute (if not otherwise specified) constant $c > 0$ respectively. We write $f(x) \asymp g(x)$ if $f(x) \lesssim g(x) \lesssim f(x)$. For two vectors $\mathbf{u}$ and $\mathbf{v}$ in a Hilbert space, we denote their inner product by $\langle \mathbf{u}, \mathbf{v} \rangle$ or $\mathbf{u}^\top \mathbf{v}$. For two matrices $\mathbf{A}$ and $\mathbf{B}$ of appropriate dimensions, we define their inner product by $\langle \mathbf{A}, \mathbf{B} \rangle := \mathrm{tr}\left(\mathbf{A}^\top \mathbf{B}\right)$. We use $\| \cdot \|$ to denote the operator norm for matrices and $\ell_2$-norm for vectors. For a positive semi-definite (PSD) matrix $\mathbf{A}$ and a vector $\mathbf{v}$ of appropriate dimension, we write $\|\mathbf{v}\|_{\mathbf{A}}^2 = \mathbf{v}^\top \mathbf{A} \mathbf{v}$.

## 2 RELATED WORKS

**High-dimensional linear regression via SGD.** Theoretical guarantees for the generalization property have garnered significant attention in machine learning and deep learning. Seminal work by Bartlett et al. (2020); Tsigler & Bartlett (2023) derived nearly tight upper and lower excess risk bounds in linear (ridge) regression for general regularization schemes. With regards to the classical underparameterized regime, a large number of works studied the learnability of iterate averaged SGD in linear regression (Polyak & Juditsky, 1992; Défossez & Bach, 2015; Bach & Moulines, 2013; Dieuleveut et al., 2017; Jain et al., 2018; 2017). With regards to modern overparameterized setting, one-pass SGD in linear regression has also been extensively studied (Dieuleveut & Bach, 2015; Berthier et al., 2020; Varre et al., 2021; Zou et al., 2023; Wu et al., 2022a;b; Zhang et al., 2024a), providing a framework to characterize how the optimization algorithm affects the generalization performance for various data distributions. Another line of work analyzed the behavior of multi-pass SGD on a high-dimensional $\ell^2$-regularized least-squares problem, characterizing excess risk bounds (Lei et al., 2021; Zou et al., 2022) and the exact dynamics of excess risk (Paquette et al., 2024a). From a technical perspective, our work builds on the sharp finite-sample and dimension-free analysis of SGD developed by Zou et al. (2023). However, these works did not concern the practical quantization operations. It remains unclear how quantization error affects the learning behavior of SGD for linear regression.

**Theoretical analysis for quantization.** As a powerful technique for deploying large-scale deep learning models, quantization has attracted significant attention. From the theoretical perspective, a line of works focus on the convergence guarantee in both quantized training (SGD) algorithms (De Sa et al., 2015; Alistarh et al., 2017; Faghri et al., 2020; Gorbunov et al., 2020; Gandikota et al., 2021; Markov et al., 2023; Xin et al., 2025) and post-training quantization methods (Lybrand & Saab, 2021; Zhang & Saab, 2023; Zhang et al., 2023; 2025). For low-precision training (SGD), De Sa et al. (2015) was the first to consider the convergence guarantees. Assuming unbiased stochastic quantization, convexity, and gradient sparsity, they gave upper bounds on the error probability of SGD. Alistarh et al. (2017) refined these results by focusing on the trade-off between commu-

nication and convergence and proposed Quantized SGD (QSGD). Faghri et al. (2020) extended the fixed quantization scheme (Alistarh et al., 2017) to two adaptive quantization schemes, providing a more general convergence guarantee for quantized training. For post-training quantization, Lybrand & Saab (2021) derived an error bound for ternary weight quantization under independent Gaussian data distribution. Zhang et al. (2023) extended these results to more general quantization grids and a wider range of data distributions using a different proof technique. More recently, Zhang et al. (2025) presented the first quantitative error bounds for OPTQ post-training algorithm framework. However, no prior work provides explicit generalization bounds.

**Linear models for theory of scaling law.** Several recent studies have sought to formalize and explain the empirical scaling laws using conceptually simplified linear models (Bahri et al., 2024; Atanasov et al., 2024; Paquette et al., 2024b; Bordelon et al., 2024; Lin et al., 2024; 2025). Among them, Bahri et al. (2024) considered a linear teacher-student model with power-law spectrum and showed that the test loss of the ordinary least square estimator decreases following a power law in sample size $N$ (or model size $M$) when the other parameter goes to infinity. Bordelon et al. (2024) analyzed the test error of the solution found by gradient flow in a linear random feature model and established power-law scaling in one of $N$, $M$ and training time $T$ while the other two parameters go to infinity. Building on the technique in Zou et al. (2023), Lin et al. (2024) analyzed the test error of the last iterate of one-pass SGD in a sketched linear model. They presented the first systematic study to establish a finite-sample joint scaling law (in $M$ and $N$) for linear models that aligns with empirical observations (Kaplan et al., 2020). More recently, Lin et al. (2025) extended the scaling law analysis to the setting with data reuse (i.e., multi-pass SGD) in data-constrained regimes.

## 3 PRELIMINARY

### 3.1 QUANTIZATION OPERATIONS

For all quantization operations in (quantized SGD), we employ the stochastic quantization method (Markov et al., 2023), which unbiasedly rounds values using randomly adjusted probabilities. This stochastic quantization is widely used in both empirical and theoretical analysis of quantization (Modoranu et al., 2024; Ozkara et al., 2025). We summarize this in the following assumption.

**Assumption 3.1.** *Let $\mathcal{Q}_i, i \in \{d, l, p, a, o\}$ be the coordinate-wise quantization operation for data feature, label, model parameter, activation, and output gradient, respectively. We assume that the quantization operation is unbiased, i.e., for any $\mathbf{u}$,*

$$\mathbb{E}\left[\mathcal{Q}_i(\mathbf{u})|\mathbf{u}\right] = \mathbf{u}.$$

Furthermore, to better uncover the effect of quantization, we consider the following two types of quantization error: multiplicative quantization and additive quantization, which are motivated by abstracting the behavior of prevalent numerical formats used in practice.

**Definition 3.1.** *Let $\mathcal{Q}$ be an unbiased quantization operation. We categorize it based on the structure of its error variance:*

- *Multiplicative quantization. We call the quantization is $\epsilon$-multiplicative if the conditional second moment of quantization error is proportional to the outer product of raw data itself, i.e.,*

$$\mathbb{E}\left[\left(\mathcal{Q}(\mathbf{x}) - \mathbf{x}\right)\left(\mathcal{Q}(\mathbf{x}) - \mathbf{x}\right)^\top \Big| \mathbf{x}\right] = \epsilon \mathbf{x}\mathbf{x}^\top.$$

- *Additive quantization. We call the quantization is $\epsilon$-additive if the conditional second moment of quantization error is proportional to identity, i.e.,*

$$\mathbb{E}\left[\left(\mathcal{Q}(\mathbf{x}) - \mathbf{x}\right)\left(\mathcal{Q}(\mathbf{x}) - \mathbf{x}\right)^\top \Big| \mathbf{x}\right] = \epsilon \mathbf{I}.$$

This theoretical distinction is grounded in practical quantization schemes. For instance, integer quantization (e.g., INT8, INT16) uses a fixed bin length, resulting in an error that is largely independent of the value's magnitude. This characteristic aligns with our definition of additive quantization, where the error variance is uniform across coordinates. Conversely, floating-point quantization (e.g.,

FP8, FP32) employs a value-aware bin length via its exponent and mantissa bits (e.g., the E4M3 format in FP8). This structure causes the quantization error to scale with the magnitude of the value itself, corresponding to the model of multiplicative quantization.

To precisely capture the quantization error, we further introduce some relevant notations on quantization errors during training. Denote the activation and output gradient at time $t$ as

$$\mathbf{a}_t = \mathcal{Q}_d(\mathbf{X}_t)\mathcal{Q}_p(\mathbf{w}_{t-1}), \quad \mathbf{o}_t = \mathcal{Q}_l(\mathbf{y}_t) - \mathcal{Q}_a\left(\mathcal{Q}_d(\mathbf{X}_t)\mathcal{Q}_p(\mathbf{w}_{t-1})\right).$$

Then we are ready to define quantization errors.

**Definition 3.2.** *The quantization error on data $\boldsymbol{\epsilon}^{(d)}$, on label $\epsilon^{(l)}$, on parameter $\boldsymbol{\epsilon}_t^{(p)}$ at time $t$, on activation $\boldsymbol{\epsilon}_t^{(a)}$ at time $t$ and on output gradient $\boldsymbol{\epsilon}_t^{(o)}$ at time $t$ are defined as follows.*

$$\boldsymbol{\epsilon}^{(d)} := \mathcal{Q}_d(\mathbf{x}) - \mathbf{x}, \quad \epsilon^{(l)} := \mathcal{Q}_l(y) - y, \quad \boldsymbol{\epsilon}_t^{(p)} := \mathcal{Q}_p(\mathbf{w}_t) - \mathbf{w}_t,$$
$$\boldsymbol{\epsilon}_t^{(a)} := \mathcal{Q}_a(\mathbf{a}_t) - \mathbf{a}_t, \quad \boldsymbol{\epsilon}_t^{(o)} := \mathcal{Q}_o(\mathbf{o}_t) - \mathbf{o}_t.$$

### 3.2 DATA MODEL

We then state the regularity assumptions on the data distribution, which align with those common in prior works (Zou et al., 2023; Lin et al., 2024). A key distinction in our setting is that the training process is performed on quantized data, i.e., $\mathcal{Q}_d(\mathbf{x})$ and $\mathcal{Q}_l(y)$. Consequently, we formulate these assumptions directly on the quantized data rather than the full-precision versions.

**Assumption 3.2** (Data covariance). *Let $\mathbf{H} = \mathbb{E}[\mathbf{x}\mathbf{x}^\top]$ be the data covariance matrix and*

$$\mathbf{H}^{(q)} := \mathbb{E}[\mathcal{Q}_d(\mathbf{x})\mathcal{Q}_d(\mathbf{x})^\top], \quad \mathbf{D} := \mathbb{E}[(\mathcal{Q}_d(\mathbf{x}) - \mathbf{x})(\mathcal{Q}_d(\mathbf{x}) - \mathbf{x})^\top],$$

*be the covariance matrices of the quantized data feature and quantization error of data covariance, respectively. Then we assume that $\mathrm{tr}(\mathbf{H})$ and $\mathrm{tr}(\mathbf{H}^{(q)})$ are finite.*

Further let $\mathbf{H} = \sum_i \lambda_i \mathbf{v}_i \mathbf{v}_i^\top$ be the eigen-decomposition of $\mathbf{H}$, where $\{\lambda_i\}_{i=1}^\infty$ are the eigenvalues of $\mathbf{H}$ sorted in non-increasing order and $\mathbf{v}_i$ are the corresponding eigenvectors. As in Zou et al. (2023), we denote

$$\mathbf{H}_{0:k} := \sum_{i=1}^k \lambda_i \mathbf{v}_i \mathbf{v}_i^\top, \quad \mathbf{H}_{k:\infty} := \sum_{i>k} \lambda_i \mathbf{v}_i \mathbf{v}_i^\top, \quad \mathbf{I}_{0:k} := \sum_{i=1}^k \mathbf{v}_i \mathbf{v}_i^\top, \quad \mathbf{I}_{k:\infty} := \sum_{i>k} \mathbf{v}_i \mathbf{v}_i^\top.$$

Similarly, we denote the eigen-decomposition of $\mathbf{H}^{(q)}$ as $\mathbf{H}^{(q)} = \sum_i \lambda_i^{(q)} \mathbf{v}_i^{(q)} \mathbf{v}_i^{(q)\top}$ and correspondingly obtain $\mathbf{H}_{0:k}^{(q)}, \mathbf{H}_{k:\infty}^{(q)}, \mathbf{I}_{0:k}^{(q)}, \mathbf{I}_{k:\infty}^{(q)}$. We then extend the fourth moment and noise assumptions in Zou et al. (2023); Lin et al. (2024) to the low-precision setting.

**Assumption 3.3** (Fourth-order moment). *Let $\mathbf{x}^{(q)} = \mathcal{Q}_d(\mathbf{x})$. Then for any PSD matrix $\mathbf{A}$, there exists a constant $\alpha_B > 0$ such that*

$$\mathbb{E}\left[\mathbf{x}^{(q)}\mathbf{x}^{(q)\top}\mathbf{A}\mathbf{x}^{(q)}\mathbf{x}^{(q)\top}\right] \preceq \alpha_B \, \mathrm{tr}(\mathbf{H}^{(q)}\mathbf{A})\mathbf{H}^{(q)}.$$

To extend the model noise assumption in Zou et al. (2023) to the low-precision setting, we define the optimal model weight regarding the quantized data feature and label:

$$\mathbf{w}^{(q)*} = \mathrm{argmin}_{\mathbf{w}} \, \mathbb{E}\left[(\mathcal{Q}_l(y) - \langle \mathbf{w}, \mathcal{Q}_d(\mathbf{x})\rangle)^2\right].$$

Then we are ready to make the assumption on the model noise $\xi := \mathcal{Q}_l(y) - \langle \mathbf{w}^{(q)*}, \mathcal{Q}_d(\mathbf{x})\rangle$.

**Assumption 3.4.** *Assume there exists a positive constant $\sigma > 0$ such that*

$$\mathbb{E}\left[\xi^2 \mathcal{Q}_d(\mathbf{x})\mathcal{Q}_d(\mathbf{x})^\top\right] \preceq \sigma^2 \mathbf{H}^{(q)}.$$

In fact, Assumptions 3.3 and 3.4 can be directly inferred from the standard assumptions on the full-precision data (Assumptions 2.1 and 2.2 in Zou et al. (2023)) under specific quantization schemes. We defer the discussion to Section E.

## 4 MAIN THEORETICAL RESULTS

We first derive excess risk upper bounds for quantized SGD in Section 4.1, then compare these rates with the full-precision SGD (in orders) in Section 4.2 and perform specific case study in Section 4.3.

### 4.1 EXCESS RISK BOUNDS

We now provide excess risk bounds under general quantization, multiplicative quantization and additive quantization. Denote the effective dimension for $\mathbf{H}^{(q)}$: $k^* = \max\left\{k : \lambda_k^{(q)} \geq \frac{1}{N\gamma}\right\}$.

**Theorem 4.1** (**General quantization**). *Consider general quantization. Denote* $\mathbf{D}_1^{\mathbf{H}} = \mathbf{D}(\mathbf{H} + \mathbf{D})^{-1}\mathbf{H}(\mathbf{H} + \mathbf{D})^{-1}\mathbf{D}$, $\mathbf{D}_2^{\mathbf{H}} = \mathbf{H}(\mathbf{H}^{(q)})^{-1}\frac{1}{N\gamma}\left(\mathbf{I} - (\mathbf{I} - \gamma\mathbf{H}^{(q)})^N\right)(\mathbf{H}^{(q)})^{-1}\mathbf{D}(\mathbf{H}^{(q)})^{-1}\mathbf{H}$. *Under Assumption 3.1, 3.2, 3.3 and 3.4, if the stepsize* $\gamma < \frac{1}{\alpha_B \mathrm{tr}(\mathbf{H}^{(q)})}$*, then it holds,*

$$\mathbb{E}[\mathcal{E}(\overline{\mathbf{w}}_N)] \leq 2\mathrm{VarErr} + 2\mathrm{BiasErr} + \mathrm{ApproxErr},$$

*where*

$$\mathrm{VarErr} \leq \frac{2\alpha_B\left(\frac{\|\mathbf{w}^{(q)*}\|_{\mathbf{I}_{0:k^*}^{(q)}}^2}{N\gamma} + \|\mathbf{w}^{(q)*}\|_{\mathbf{H}_{k^*:\infty}^{(q)}}^2\right) + \sigma_G^{(q)2}}{1 - \gamma\alpha_B\mathrm{tr}(\mathbf{H}^{(q)})}\left(\frac{k^*}{N} + N\gamma^2 \cdot \sum_{i>k^*}(\lambda_i^{(q)})^2\right),$$

$$\mathrm{BiasErr} \leq \frac{1}{\gamma^2 N^2} \cdot \|\mathbf{w}^{(q)*}\|_{(\mathbf{H}_{0:k^*}^{(q)})^{-1}}^2 + \|\mathbf{w}^{(q)*}\|_{\mathbf{H}_{k^*:\infty}^{(q)}}^2,$$

$$\mathrm{ApproxErr} \leq \|\mathbf{w}^*\|_{\mathbf{D}_1^{\mathbf{H}}}^2 + \|\mathbf{w}^*\|_{\mathbf{D}_2^{\mathbf{H}}}^2,$$

*with* $\sigma_G^{(q)2} = \frac{\sigma^2 + \sup_t\left\{\left\|\mathbb{E}\left[\boldsymbol{\epsilon}_t^{(o)}\boldsymbol{\epsilon}_t^{(o)\top}|\mathbf{o}_t\right] + \mathbb{E}\left[\boldsymbol{\epsilon}_t^{(a)}\boldsymbol{\epsilon}_t^{(a)\top}|\mathbf{a}_t\right]\right\|\right\}}{B} + \alpha_B\sup_t\mathbb{E}\left[\mathrm{tr}\left(\mathbf{H}^{(q)}\boldsymbol{\epsilon}_{t-1}^{(p)}\boldsymbol{\epsilon}_{t-1}^{(p)\top}\right)\right].$

Theorem 4.1 establishes the first excess risk bound for quantized SGD under a general quantization paradigm. The excess risk is decomposed into three components: variance error, bias error, and approximation error. Notably, the variance and bias errors mirror those of full-precision SGD (Zou et al., 2023) and exact equivalence is recovered when the quantization error vanishes. The key role that quantization plays is two-fold: data quantization significantly influences the effective (quantized) data covariance $\mathbf{H}^{(q)}$, while activation, output gradient and parameter quantization amplify the effective noise variance $\sigma_G^{(q)}$ (which will be further characterized in the subsequent theorems when given specific quantization type). Specifically, the quantized data covariance arises from performing SGD in quantized data feature space and the quantized noise variance corresponds to additional quantization error introduced in the parameter update rule. We also note that the additional approximation error, resulting from quantization of data, can be interpreted as the discrepancy between the global optimum in full-precision data space and quantized data feature space.

Crucially, in the absence of quantization, our excess risk bound reduces exactly to the standard results presented in Zou et al. (2023). It is also worth noting that under the unbiased quantization assumption, the quantization of parameter, output gradient, and activation do not affect bias error [1].

To further elucidate the effects of quantization, we examine two specific schemes: multiplicative and additive quantization. The result for additive quantization can be derived directly from Theorem 4.1 and is summarized below.

**Corollary 4.1** (**Additive quantization**). *Under Assumption 3.1, 3.2, 3.3 and 3.4, if there exist* $\epsilon_d, \epsilon_l, \epsilon_p, \epsilon_a$ *and* $\epsilon_o$ *such that for any* $i \in \{d, l, p, a, o\}$*, quantization* $\mathcal{Q}_i$ *is* $\epsilon_i$*-additive, and the stepsize satisfies* $\gamma < \frac{1}{\alpha_B[\mathrm{tr}(\mathbf{H}) + d\epsilon_d]}$*, then*

$$\mathbb{E}[\mathcal{E}(\overline{\mathbf{w}}_N)] \lesssim \mathrm{ApproxErr} + \mathrm{VarErr} + \mathrm{BiasErr},$$

---

[1]For theoretical tractability and simplicity, our framework employs the unbiased quantization assumption (Assumption 3.1). Without this assumption, the conditional expectations of the parameter, output gradient, and activation quantization errors (i.e., quantization bias) would contribute to the bias error. We believe our framework is readily extendable to this general biased quantization setting.

*where*

$$\text{ApproxErr} \lesssim \frac{\epsilon_d}{\lambda_d + \epsilon_d} \|\mathbf{w}^*\|_{\mathbf{H}}^2, \quad \text{BiasErr} \lesssim \frac{1}{\gamma^2 N^2} \cdot \|\mathbf{w}^{(q)*}\|_{(\mathbf{H}_{0:k^*}^{(q)})^{-1}}^2 + \|\mathbf{w}^{(q)*}\|_{\mathbf{H}_{k^*:\infty}^{(q)}}^2,$$

$$\text{VarErr} \lesssim \frac{\alpha_B \|\mathbf{w}^*\|_{\mathbf{H}}^2 + \frac{\sigma^2 + \epsilon_o + \epsilon_a}{B} + \alpha_B \epsilon_p [\text{tr}(\mathbf{H}) + d\epsilon_d]}{1 - \gamma \alpha_B [\text{tr}(\mathbf{H}) + d\epsilon_d]} \left( \frac{k^*}{N} + N\gamma^2 \cdot \sum_{i>k^*} (\lambda_i + \epsilon_d)^2 \right).$$

Corollary 4.1 explicitly demonstrates how data quantization distorts effective data covariance spectrum and how parameter, activation and output gradient quantization amplify noise during training under additive quantization scheme. A key observation concerns the scaling with respect to the batch size $B$. Consistent with the label noise $\sigma^2$, the noise amplification from activation and output gradient quantization ($\epsilon_a, \epsilon_o$) are scaled by a factor of $1/B$. In contrast, the noise amplification from parameter quantization ($\epsilon_p$) scales with the trace of the quantized data covariance and is independent of batch size.

The interpretation is that additive quantization imposes a constant bound on the conditional second moment of the quantization error. Consequently, the underlying data structure inherent within the activation quantization error $\boldsymbol{\epsilon}_t^{(a)}$ and output gradient quantization error $\boldsymbol{\epsilon}_t^{(o)}$ is effectively neutralized. Formally, the noise amplification from these terms is characterized as $\frac{1}{B^2}\mathbb{E}[\mathbf{X}^{q\top}\boldsymbol{\epsilon}\boldsymbol{\epsilon}^\top\mathbf{X}^q]$. Under additive quantization, since the error variance is bounded by a constant, the dependency on data within $\boldsymbol{\epsilon}$ vanishes. However, the noise amplification from parameter quantization, which is characterized as $\frac{1}{B^2}\mathbb{E}[\mathbf{X}^{q\top}\mathbf{X}^q\boldsymbol{\epsilon}^{(p)}\boldsymbol{\epsilon}^{(p)\top}\mathbf{X}^{q\top}\mathbf{X}^q]$, preserves the underlying dependency on data, even if the error variance itself is constant.

Moreover, a critical consequence of additive quantization is the distortion of the data covariance spectrum $\mathbf{H}^{(q)}$. Specifically, a fixed constant $\epsilon_d$ is added across the entire spectrum, effectively imposing a noise floor that prevents the tail eigenvalues from decaying. This spectral flattening severely impedes learnability, as it leads to substantial risk accumulation within the high-dimensional tail subspace.

We next examine the multiplicative quantization scheme. Unlike additive quantization, multiplicative quantization exhibits an inherent structural alignment with the full-precision dynamics, as the error scales relative to the signal magnitude. Exploiting this property allows us to derive a refined excess risk bound through a direct analysis, rather than relying on a generic application of the general result in Theorem 4.1. Our theoretical findings are summarized below.

**Theorem 4.2** (**Multiplicative quantization**). *Under Assumption 3.1, 3.2, 3.3 and 3.4, if there exist $\epsilon_d, \epsilon_l, \epsilon_p, \epsilon_a$ and $\epsilon_o$ such that for any $i \in \{d, l, p, a, o\}$, quantization $\mathcal{Q}_i$ is $\epsilon_i$-multiplicative, and the stepsize satisfies $\gamma < \frac{1}{\alpha_B(1+\epsilon_o)[1+\epsilon_p+\epsilon_a(1+\epsilon_p)](1+\epsilon_d)\text{tr}(\mathbf{H})}$, then the excess risk can be upper bounded as follows.*

$$\mathbb{E}[\mathcal{E}(\overline{\mathbf{w}}_N)] \lesssim \text{ApproxErr} + \text{VarErr} + \text{BiasErr},$$

*where*

$$\text{ApproxErr} \lesssim \frac{\epsilon_d}{1 + \epsilon_d} \|\mathbf{w}^*\|_{\mathbf{H}}^2, \quad \text{BiasErr} \lesssim \frac{1}{\gamma^2 N^2} \cdot \|\mathbf{w}^{(q)*}\|_{(\mathbf{H}_{0:k^*}^{(q)})^{-1}}^2 + \|\mathbf{w}^{(q)*}\|_{\mathbf{H}_{k^*:\infty}^{(q)}}^2,$$

$$\text{VarErr} \lesssim \left( \frac{k^*}{N} + N\gamma^2(1+\epsilon_d)^2 \sum_{i>k^*} \lambda_i^2 \right) \frac{\frac{(1+\epsilon_o)\sigma^2}{B} + \alpha_B(1+\epsilon_o)[1 + \epsilon_p + \epsilon_a(1+\epsilon_p)] \|\mathbf{w}^*\|_{\mathbf{H}}^2}{1 - \gamma\alpha_B(1+\epsilon_o)[1 + \epsilon_p + \epsilon_a(1+\epsilon_p)](1+\epsilon_d)\text{tr}(\mathbf{H})}.$$

Theorem 4.2 characterizes the spectrum distortion and noise amplification effects induced by multiplicative quantization. Notably, in stark contrast to additive quantization, which severely flattens the tail spectrum by imposing a constant floor, multiplicative quantization largely preserves the intrinsic spectral structure. Specifically, it acts as a linear transformation that scales the entire spectrum by a factor of $(1 + \epsilon_d)$ without altering the relative distribution of eigenvalues. This preservation of the spectral decay property ensures superior learnability compared to the additive quantization scheme.

Regarding noise amplification, Theorem 4.2 reveals a critical divergence from the additive quantization scheme. While the contribution from intrinsic label noise ($\sigma^2$) is still suppressed by the batch size factor $1/B$, the quantization noise stemming from activation and output gradients ($\epsilon_a, \epsilon_o$) is coupled with the model parameter $\|\mathbf{w}^*\|_{\mathbf{H}}^2$ and does not scale with $1/B$. This phenomenon arises

because multiplicative quantization error (scales proportionally with the signal strength) is inherently signal-dependent and is intrinsically tied to the data structure.

We provide further analysis of quantized SGD with quantized master weights in Section F. Training with quantized master weights necessitates stricter step size conditions to ensure convergence and introduces additional error terms into the excess risk bounds, thereby degrading generalization performance.

## 4.2 Comparisons with Standard Excess Risk Bound

In this part, we will provide a detailed comparison with standard excess risk bounds and identify the conditions on the quantization error such that the excess risk bound will not be largely affected. First, let $k_0^* = \max\{k : \lambda_k \geq \frac{1}{N\gamma}\}$, we recall the standard excess risk bound (Zou et al., 2023):

$$R_0 = \left( \frac{k_0^*}{N} + N\gamma^2 \cdot \sum_{i > k_0^*} \lambda_i^2 \right) \frac{\alpha_B \left( \frac{1}{N\gamma} \|\mathbf{w}^*\|_{\mathbf{I}_{0:k_0^*}}^2 + \|\mathbf{w}^*\|_{\mathbf{H}_{k_0^*:\infty}}^2 \right) + \frac{\sigma^2}{B}}{1 - \gamma\alpha_B \mathrm{tr}(\mathbf{H})}$$
$$+ \frac{1}{\gamma^2 N^2} \cdot \|\mathbf{w}^*\|_{(\mathbf{H}_{0:k_0^*})^{-1}}^2 + \|\mathbf{w}^*\|_{\mathbf{H}_{k_0^*:\infty}}^2.$$

The following corollary derives the conditions on the quantization errors such that the learning performance of the full-precision SGD can be maintained (in orders).

**Corollary 4.2.** *To ensure that $\mathbb{E}[\mathcal{E}(\overline{\mathbf{w}}_N)] \lesssim R_0$, conditions on the quantization error are as follows:*

- *For multiplicative quantization, under the conditions in Theorem 4.2, we require*

$$\epsilon_d \lesssim 1 \wedge \frac{R_0}{\|\mathbf{w}^*\|_{\mathbf{H}}^2}, \quad \epsilon_o, \epsilon_a, \epsilon_p \lesssim \left( \frac{\sigma^2}{B\alpha_B\|\mathbf{w}^*\|_{\mathbf{H}}^2} + \frac{\frac{1}{N\gamma}\|\mathbf{w}^*\|_{\mathbf{I}_{0:k_0^*}}^2 + \|\mathbf{w}^*\|_{\mathbf{H}_{k_0^*:\infty}}^2}{\|\mathbf{w}^*\|_{\mathbf{H}}^2} \right) \wedge 1.$$

- *For additive quantization, under the conditions in Corollary 4.1, we require*

$$\epsilon_d \lesssim \sqrt{\frac{\frac{k_0^*}{N} + N\gamma^2 \cdot \sum_{i > k_0^*} \lambda_i^2}{N\gamma^2(d - k_0^*)}} \wedge \frac{R_0\lambda_d}{\|\mathbf{w}^*\|_{\mathbf{H}}^2}, \quad \epsilon_a, \epsilon_o \lesssim \sigma^2 + B\alpha_B \left( \frac{\|\mathbf{w}^*\|_{\mathbf{I}_{0:k_0^*}}^2}{N\gamma} + \|\mathbf{w}^*\|_{\mathbf{H}_{k_0^*:\infty}}^2 \right),$$

$$\epsilon_p \lesssim \frac{\sigma^2}{B\alpha_B[\mathrm{tr}(\mathbf{H}) + d\epsilon_d]} + \frac{\frac{\|\mathbf{w}^*\|_{\mathbf{I}_{0:k_0^*}}^2}{N\gamma} + \|\mathbf{w}^*\|_{\mathbf{H}_{k_0^*:\infty}}^2}{\mathrm{tr}(\mathbf{H}) + d\epsilon_d}.$$

Corollary 4.2 identifies the conditions under which the quantized excess risk matches the full-precision baseline $R_0$. Regarding data quantization ($\epsilon_d$), the additive scheme imposes stringent spectrum-dependent constraints compared to the multiplicative quantization scheme. Specifically, the precision requirements are notably strict to prevent the constant quantization noise floor from overwhelming weak spectral components. Conversely, for activation and output gradient quantization ($\epsilon_a, \epsilon_o$), the additive scheme exhibits a favorable dependence on the batch size. As indicated by the scaling with $B$ in the bounds for $\epsilon_a$ and $\epsilon_o$, larger batch sizes effectively relax the precision requirements for these components. In contrast, larger batch sizes may essentially tighten the requirements under the multiplicative quantization scheme.

These findings validate our core insights: (1) multiplicative data quantization is superior in maintaining the spectral structure of $\mathbf{H}$, thus tolerating larger data quantization errors; (2) additive quantization benefits from the fact that the noise variance in activation and output gradient is independent of the signal magnitude, allowing these errors to be effectively suppressed by increasing the batch size.

## 4.3 Case Study on Data Distribution with Polynomial-decay Spectrum

Following Lin et al. (2024; 2025), we study the excess risk bounds assuming optimal parameter prior and the power-law spectrum for more concise theoretical results. In particular, we make the following assumption.

**Assumption 4.1.** *There exists $a > 1$ such that the eigenvalues of $\mathbf{H}$ satisfy $\lambda_i \approx i^{-a}$, $i > 0$. We also assume that $\mathbb{E}\left[\mathbf{w}^*\mathbf{w}^{*\top}\right] = \mathbf{I}$ and $\sigma^2 \lesssim 1$.*

**Corollary 4.3.** *Taking expectation on $\mathbf{w}^*$, under Assumption 4.1, we have:*

- *For multiplicative quantization, under the conditions in Theorem 4.2,*

$$\mathbb{E}\left[\mathcal{E}(\overline{\mathbf{w}}_N)\right] \lesssim \frac{\epsilon_d}{1 + \epsilon_d} + d^{1-a} + N^{1/a-1}(1 + \epsilon_o)[1 + \epsilon_p + \epsilon_a(1 + \epsilon_p)](1 + \epsilon_d)^{1/a}.$$

- *For additive quantization, under the conditions in Corollary 4.1,*

$$\mathbb{E}[\mathcal{E}(\overline{\mathbf{w}}_N)] \lesssim \left(1 + \frac{\epsilon_o + \epsilon_a}{B} + \epsilon_p(1 + d\epsilon_d)\right)\left(1 + \frac{(d^a\epsilon_d)^2}{1 + d^a\epsilon_d}\right)d^{1-a} + \frac{d^a\epsilon_d}{1 + d^a\epsilon_d}$$
$$+ \left(1 + \frac{\epsilon_o + \epsilon_a}{B} + \epsilon_p(1 + d\epsilon_d)\right)(1 + d^a\epsilon_d)^{1/a}N^{1/a-1}.$$

Our findings in polynomial-decay data spectrum scenarios reveal distinct scaling behaviors under multiplicative and additive quantization. Specifically, the excess risk induced by additive data quantization exhibits a detrimental dependency on data dimension $d$, whereas the risk under multiplicative data quantization remains dimension-independent. This dependence has critical implications for learnability: in high-dimensional regimes ($d \to \infty$), the risk bound for additive quantization diverges, rendering the generalization guarantee vacuous. In contrast, the dimension-free nature of multiplicative quantization ensures its applicability even in infinite-dimensional settings.

Intuitively, this disparity stems from how each scheme interacts with the spectral structure. Multiplicative quantization preserves the intrinsic spectral decay, thereby retaining the utility of the effective dimension ($k^*$) cut-off. This allows the learning complexity to be controlled by the intrinsic data properties rather than the data dimension. Conversely, additive quantization employs a uniform quantization strength across all dimensions. This constant noise floor prevents the tail spectrum from decaying effectively and accumulates across the entire high-dimensional tail, rendering the effective dimension mechanism failed.

**Implications for integer and FP quantization.** These simplified theoretical results (Corollary 4.3) draw critical implications for comparing integer and floating-point (FP) quantization, allowing us to identify the conditions under which each type yields superior performance. Specifically, in practical integer quantization with bit-width $b$ and FP quantization with mantissa bit-width $m$, the quantization step size for a value $x$ are approximately $\delta(x) \approx 2^{-b}$ and $\delta(x) = |x|2^{-m}$ [2], respectively. Since the conditional second moment of quantization error $\mathbb{E}[(\mathcal{Q}(x) - x)^2|x]$ is roughly proportional to the square of the quantization step size ($\delta(x)^2$), the quantization error parameters in our bounds can be characterized as $\epsilon_{\text{add}} \approx 2^{-2b}$ for the additive (integer) quantization scheme and $\epsilon_{\text{mul}} \approx 2^{-2m}$ for the multiplicative (FP) quantization scheme.

Equipped with this mapping, practitioners can directly apply Corollary 4.3 to determine the optimal quantization scheme for specific scenarios. A notable observation concerns the distinct role of the dimension $d$ in data quantization. Roughly, FP quantization becomes preferable when $m_d \geq b_d - \frac{a}{2}\log_2 d$ whereas integer quantization is favored when $b_d \geq m_d + \frac{a}{2}\log_2 d$ [3]. This means FP quantization can outperform integer quantization even when its mantissa bit-width is smaller than the integer bit-width by $\frac{a}{2}\log_2 d$, highlighting the advantage of FP quantization in high-dimensional settings.

**Numerical experiments.** We evaluate constant–stepsize SGD with iterate averaging on a Gaussian least–squares model. The feature distribution has covariance matrix with eigenvalues $\lambda_i = i^{-2}$. The ground–truth parameter is $\mathbf{w}^*$ with entries $\mathbf{w}^*[i] = 1$, and the observation noise variance is $\sigma^2 = 1$. This study answers two questions: **Q1**: How do *additive* vs. *multiplicative* quantization errors affect learning? **Q2**: How does *dimension* $d$ interact with these two quantization types?

---

[2] We assume the exponent bits in FP quantization can cover the scaling of $x$. For integer quantization, we assume the dynamic range (i.e., $x_{\max} - x_{\min}$) is normalized to constant level.

[3] Here $b_d$ and $m_d$ are the bit-width for integer data quantization and the mantissa bit-width for FP data quantization respectively.

(a) **Multiplicative** (FP-like) (b) **Additive** (INT-like) (c) **Multiplicative** (FP-like) (d) **Additive** (INT-like)

Figure 1: **Generalization under quantization.** Population risk ($\mathbb{E}_{\mathbf{x},y}[(y - \langle \mathbf{w}, \mathbf{x} \rangle)^2]$) for quantized SGD with iterate averaging under multiplicative (FP-like) vs. additive (INT-like) quantization. (a) and (b): vary the quantization level at fixed dimension. (c) and (d): vary dimension at fixed quantization level.

**Q1 (Quantization level).** We fix $d = 200$ and $B = 1$, and vary the quantization error level $\varepsilon \in \{0.001, 0.005, 0.01\}$ for each scheme. Results are shown in Fig. 1(a,b). Under multiplicative quantization, quantized SGD largely retains the generalization performance of full-precision SGD across a wide range of quantization levels. Conversely, under additive quantization, performance degrades as the quantization level increases. These empirical observations validate our theoretical findings: with a batch size of $B = 1$, additive quantization requires stricter conditions (lower quantization level) to match the performance of full-precision SGD.

**Q2 (Dimension).** We fix the quantization level at $\varepsilon = 0.01$ and $B = 1$, and vary $d \in \{50, 100, 200, 400\}$. Results are shown in Fig. 1(c,d). Under multiplicative quantization, generalization performance is preserved even in high-dimensional settings; conversely, under additive quantization, performance deteriorates as the data dimension increases. These empirical results corroborate our theoretical findings: multiplicative quantization remains effective in high-dimensional contexts, whereas additive quantization is ill-suited for such scenarios.

Furthermore, we conduct additional experiments on the real-world `Communities and Crime` dataset, as well as settings with larger batch sizes and exponential-decay spectra. These results, presented in Section G, consistently align with our theoretical analysis.

## 5 CONCLUSION AND LIMITATIONS

In this work, we presented a comprehensive theoretical framework to analyze the excess risk of quantized SGD in high-dimensional linear regression. Our analysis disentangles the distinct impacts of various quantization targets: while parameter, activation, and gradient quantization primarily serve as noise amplifiers, data quantization fundamentally distorts the effective feature covariance spectrum. Crucially, we show that multiplicative quantization excels at preserving the spectral structure of the data, thereby maintaining learnability even in high-dimensional settings. In contrast, additive quantization leverages the independence of noise variance from signal magnitude, allowing activation and gradient noise to be effectively suppressed by large batch sizes. Furthermore, our theory establishes the conditions on quantization errors required to maintain full-precision SGD performance, and identifies the scenarios under which FP and integer quantization are each likely to yield superior performance under polynomial decay spectrum.

**Future work.** Our work lays a solid foundation for several promising research avenues. Firstly, developing a lower bound analysis for the excess risk of quantized SGD. Secondly, extending single-pass SGD to more practical training configurations, such as data reuse (i.e., multi-pass SGD), learning rate scheduling, momentum, and preconditioning. Thirdly, extending training the linear models to the training of over-parameterized neural networks. Fourthly, deriving scaling laws for low-precision training.

## ACKNOWLEDGMENTS

We would like to thank the anonymous reviewers and area chairs for their helpful comments. We acknowledge the support from NSFC 62306252, Hong Kong ECS award 27309624, Guangdong NSF 2024A1515012444, and the central fund from HKU IDS.

ETHICS STATEMENT

We have carefully reviewed the Code of Ethics and find that our work does not raise any significant ethical concerns. Our research does not involve human subjects, sensitive data, or potentially harmful applications. We believe our methodology and contributions align with principles of fairness, transparency, and research integrity.

REPRODUCIBILITY STATEMENT

The paper fully discloses all the information, including training and testing details, needed to reproduce the main experimental results of the paper to the extent that it affects the main conclusions of the paper, as described in Section 4.3. All data are synthetically generated with detailed description.

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

APPENDIX

CONTENTS

The appendix is organized as follows. In Section A, we begin the analysis of excess risk bounds for the iteratively averaged quantized SGD by firstly deriving the update rule for the parameter deviation $\mathbf{w}_t - \mathbf{w}^{(q)*}$ (detailed in Section A.1) and secondly performing an excess risk decomposition (detailed in Section A.2):

$$\mathbb{E}[\mathcal{E}(\overline{\mathbf{w}}_N)] = \underbrace{\frac{1}{2}\langle \mathbf{H}, \mathbb{E}[\overline{\boldsymbol{\eta}}_N \otimes \overline{\boldsymbol{\eta}}_N]\rangle}_{R_N} + \text{ApproxErr}.$$

We then conduct a refined analysis for ApproxErr in Sections B. For $R_N$, we extend techniques from Zou et al. (2023) in Section C. In particular, we first introduce useful notations in Section C.1 and then present a comprehensive analysis of the update rule for $\mathbb{E}[\boldsymbol{\eta}_t \boldsymbol{\eta}_t^\top]$ in Section C.2. This analysis is crucial for adapting previous proof techniques to the quantized SGD setting. Based on these results, we perform a bias–variance decomposition in Section C.3, and analyze the bias and variance errors separately in Section C.4 and C.5. In Section F, we include bounds when master weight is quantized.

The following proof dependency graph visually encapsulates the logical structure and organizational architecture of the theoretical results in our paper. In particular, the arrow from element $X$ to element $Y$ means the proof of $Y$ relies on $X$.

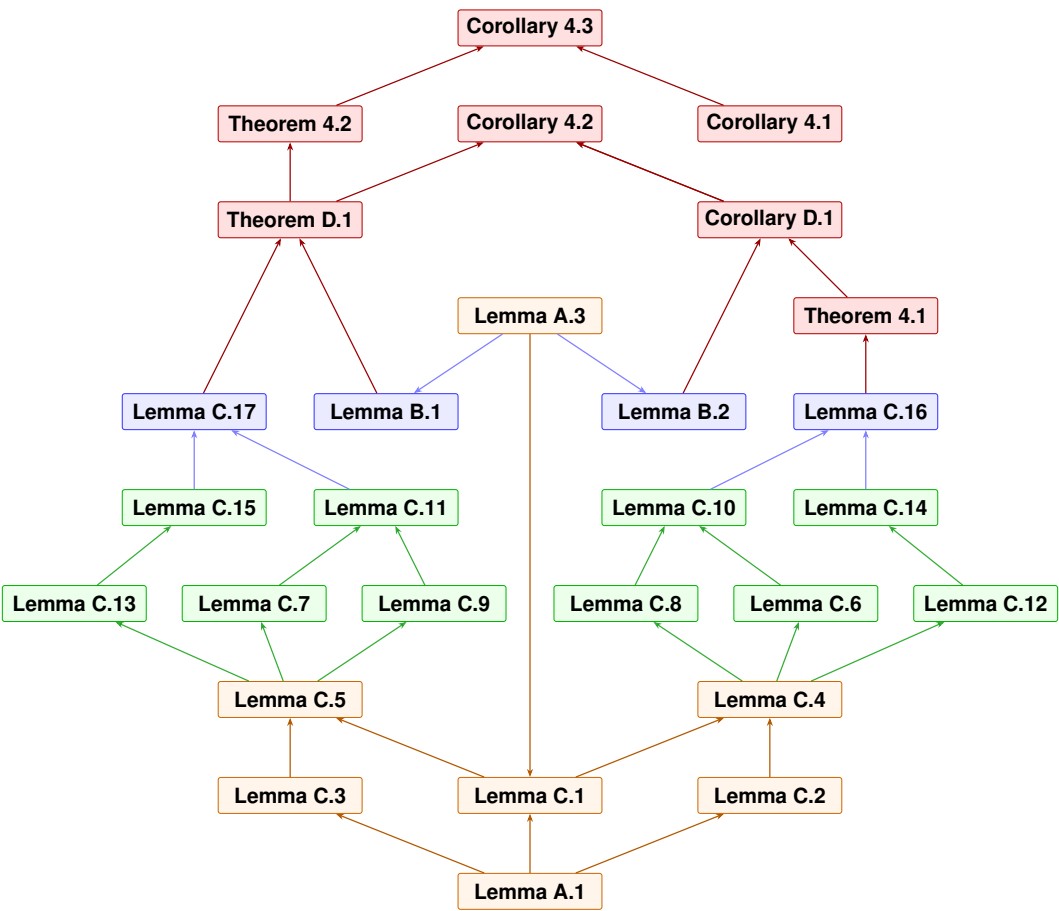

## A    INITIAL STUDY

For simplicity, we denote $y^{(q)} = \mathcal{Q}_l(y), \mathbf{w}_t^{(q)} = \mathcal{Q}_p(\mathbf{w}_t), \mathbf{x}^{(q)} = \mathcal{Q}_d(\mathbf{x})$. For convenience, we assume that $\mathbf{H}$ is strictly positive definite and that $L(\mathbf{w})$ admits a unique global optimum as Zou et al. (2023). We first recall the definition of the global minima $\mathbf{w}^*$ and $\mathbf{w}^{(q)^*}$:

$$\mathbf{w}^* = \operatorname{argmin}_{\mathbf{w}} \mathbb{E}\left[(y - \langle \mathbf{w}, \mathbf{x} \rangle)^2\right], \quad \mathbf{w}^{(q)^*} = \operatorname{argmin}_{\mathbf{w}} \mathbb{E}\left[(\mathcal{Q}_l(y) - \langle \mathbf{w}, \mathcal{Q}_d(\mathbf{x}) \rangle)^2\right].$$

The first order optimality shows that

$$\mathbb{E}[(y - \langle \mathbf{w}^*, \mathbf{x} \rangle)\mathbf{x}] = \mathbf{0}, \quad \mathbb{E}[(\mathcal{Q}_l(y) - \langle \mathbf{w}^{(q)^*}, \mathcal{Q}_d(\mathbf{x}) \rangle)\mathcal{Q}_d(\mathbf{x})] = \mathbf{0}, \qquad \text{(A.1)}$$

which implies that

$$\mathbf{w}^* = \mathbf{H}^{-1}\mathbb{E}_{(\mathbf{x},y)\sim\mathcal{D}}[y\mathbf{x}], \quad \mathbf{w}^{(q)^*} = (\mathbf{H}^{(q)})^{-1}\mathbb{E}\left[\mathcal{Q}_l(y)\mathcal{Q}_d(\mathbf{x})\right] = (\mathbf{H}^{(q)})^{-1}\mathbb{E}_{(\mathbf{x},y)\sim\mathcal{D}}[y\mathbf{x}].$$

Hence, by denoting $\mathbf{H}^{(q)} = \mathbf{H} + \mathbf{D}$, we can characterize the difference between $\mathbf{w}^{(q)^*}$ and $\mathbf{w}^*$ as:

$$\begin{aligned}
\mathbf{w}^{(q)^*} - \mathbf{w}^* &= \left[(\mathbf{H}^{(q)})^{-1} - \mathbf{H}^{-1}\right]\mathbb{E}_{(\mathbf{x},y)\sim\mathcal{D}}[y\mathbf{x}] \\
&= (\mathbf{H}^{(q)})^{-1}\left(\mathbf{H} - \mathbf{H}^{(q)}\right)\mathbf{H}^{-1}\mathbb{E}_{(\mathbf{x},y)\sim\mathcal{D}}[y\mathbf{x}] \\
&= (\mathbf{H}^{(q)})^{-1}\left(\mathbf{H} - \mathbf{H}^{(q)}\right)\mathbf{w}^* \\
&= -(\mathbf{H}^{(q)})^{-1}\mathbf{D}\mathbf{w}^* \\
&= -(\mathbf{H} + \mathbf{D})^{-1}\mathbf{D}\mathbf{w}^*.
\end{aligned} \qquad \text{(A.2)}$$

### A.1    DEVIATION OF THE UPDATE RULE

In this section, we derive the evolution of parameter deviation $\boldsymbol{\eta}_t := \mathbf{w}_t - \mathbf{w}^{(q)^*}$.

**Lemma A.1** (Error propagation).

$$\boldsymbol{\eta}_t = \left(\mathbf{I} - \frac{1}{B}\gamma\mathcal{Q}_d(\mathbf{X}_t)^\top\mathcal{Q}_d(\mathbf{X}_t)\right)\boldsymbol{\eta}_{t-1} + \gamma\frac{1}{B}\mathcal{Q}_d(\mathbf{X}_t)^\top\left[\boldsymbol{\xi}_t + \boldsymbol{\epsilon}_t^{(o)} - \boldsymbol{\epsilon}_t^{(a)} - \mathcal{Q}_d(\mathbf{X}_t)\boldsymbol{\epsilon}_{t-1}^{(p)}\right],$$

*where the quantization errors are*

$$\begin{aligned}
\boldsymbol{\epsilon}_t^{(o)} &:= \mathcal{Q}_o\left(\mathcal{Q}_l(\mathbf{y}_t) - \mathcal{Q}_a\left(\mathcal{Q}_d(\mathbf{X}_t)\mathcal{Q}_p(\mathbf{w}_{t-1})\right)\right) - \left[\mathcal{Q}_l(\mathbf{y}_t) - \mathcal{Q}_a\left(\mathcal{Q}_d(\mathbf{X}_t)\mathcal{Q}_p(\mathbf{w}_{t-1})\right)\right], \\
\boldsymbol{\epsilon}_t^{(a)} &:= \mathcal{Q}_a\left(\mathcal{Q}_d(\mathbf{X}_t)\mathcal{Q}_p(\mathbf{w}_{t-1})\right) - \mathcal{Q}_d(\mathbf{X}_t)\mathcal{Q}_p(\mathbf{w}_{t-1}), \\
\boldsymbol{\epsilon}_{t-1}^{(p)} &:= \mathcal{Q}_p(\mathbf{w}_{t-1}) - \mathbf{w}_{t-1}, \\
\boldsymbol{\xi}_t &:= \mathcal{Q}_l(\mathbf{y}_t) - \mathcal{Q}_d(\mathbf{X}_t)\mathbf{w}^{(q)^*}.
\end{aligned}$$

*Proof.* The lemma can be proved directly by the parameter update rule. By definition and the update rule of $\mathbf{w}_t$ (quantized SGD),

$$\begin{aligned}
\boldsymbol{\eta}_t &= \mathbf{w}_t - \mathbf{w}^{(q)^*} \\
&= \mathbf{w}_{t-1} - \mathbf{w}^{(q)^*} + \gamma\frac{1}{B}\mathcal{Q}_d(\mathbf{X}_t)^\top\mathcal{Q}_o\left(\mathcal{Q}_l(\mathbf{y}_t) - \mathcal{Q}_a\left(\mathcal{Q}_d(\mathbf{X}_t)\mathcal{Q}_p(\mathbf{w}_{t-1})\right)\right) \\
&= \boldsymbol{\eta}_{t-1} + \gamma\frac{1}{B}\mathcal{Q}_d(\mathbf{X}_t)^\top\mathcal{Q}_o\left(\mathcal{Q}_l(\mathbf{y}_t) - \mathcal{Q}_a\left(\mathcal{Q}_d(\mathbf{X}_t)\mathcal{Q}_p(\mathbf{w}_{t-1})\right)\right).
\end{aligned}$$

We then introduce quantization errors to better characterize each quantization operation $\mathcal{Q}(\cdot)$. In particular, define quantization erros:

$$\begin{aligned}
\boldsymbol{\epsilon}_t^{(o)} &:= \mathcal{Q}_o\left(\mathcal{Q}_l(\mathbf{y}_t) - \mathcal{Q}_a\left(\mathcal{Q}_d(\mathbf{X}_t)\mathcal{Q}_p(\mathbf{w}_{t-1})\right)\right) - \left[\mathcal{Q}_l(\mathbf{y}_t) - \mathcal{Q}_a\left(\mathcal{Q}_d(\mathbf{X}_t)\mathcal{Q}_p(\mathbf{w}_{t-1})\right)\right], \\
\boldsymbol{\epsilon}_t^{(a)} &:= \mathcal{Q}_a\left(\mathcal{Q}_d(\mathbf{X}_t)\mathcal{Q}_p(\mathbf{w}_{t-1})\right) - \mathcal{Q}_d(\mathbf{X}_t)\mathcal{Q}_p(\mathbf{w}_{t-1}), \\
\boldsymbol{\epsilon}_{t-1}^{(p)} &:= \mathcal{Q}_p(\mathbf{w}_{t-1}) - \mathbf{w}_{t-1}, \\
\boldsymbol{\xi}_t &:= \mathcal{Q}_l(\mathbf{y}_t) - \mathcal{Q}_d(\mathbf{X}_t)\mathbf{w}^{(q)^*}.
\end{aligned}$$

Then the update rule for the parameter deviation can be expressed as:

$$
\begin{aligned}
\boldsymbol{\eta}_t =& \boldsymbol{\eta}_{t-1} + \gamma \frac{1}{B} \mathcal{Q}_d(\mathbf{X}_t)^\top \mathcal{Q}_o \left( \mathcal{Q}_l(\mathbf{y}_t) - \mathcal{Q}_a \left( \mathcal{Q}_d(\mathbf{X}_t) \mathcal{Q}_p(\mathbf{w}_{t-1}) \right) \right) \\
=& \boldsymbol{\eta}_{t-1} + \gamma \mathcal{Q}_d(\mathbf{X}_t)^\top \frac{1}{B} \left[ \mathcal{Q}_l(\mathbf{y}_t) - \mathcal{Q}_a \left( \mathcal{Q}_d(\mathbf{X}_t) \mathcal{Q}_p(\mathbf{w}_{t-1}) \right) \right] + \gamma \frac{1}{B} \mathcal{Q}_d(\mathbf{X}_t)^\top \boldsymbol{\epsilon}_t^{(o)} \\
=& \boldsymbol{\eta}_{t-1} + \gamma \mathcal{Q}_d(\mathbf{X}_t)^\top \frac{1}{B} \left[ \mathcal{Q}_l(\mathbf{y}_t) - \mathcal{Q}_d(\mathbf{X}_t) \mathcal{Q}_p(\mathbf{w}_{t-1}) \right] + \gamma \frac{1}{B} \mathcal{Q}_d(\mathbf{X}_t)^\top (\boldsymbol{\epsilon}_t^{(o)} - \boldsymbol{\epsilon}_t^{(a)}) \\
=& \boldsymbol{\eta}_{t-1} + \gamma \frac{1}{B} \mathcal{Q}_d(\mathbf{X}_t)^\top (\boldsymbol{\epsilon}_t^{(o)} - \boldsymbol{\epsilon}_t^{(a)}) + \gamma \mathcal{Q}_d(\mathbf{X}_t)^\top \frac{1}{B} \\
& \left[ \mathcal{Q}_l(\mathbf{y}_t) - \mathcal{Q}_d(\mathbf{X}_t) \mathbf{w}^{(q)^*} - \mathcal{Q}_d(\mathbf{X}_t) \boldsymbol{\eta}_{t-1} - \mathcal{Q}_d(\mathbf{X}_t) \mathcal{Q}_p(\mathbf{w}_{t-1}) + \mathcal{Q}_d(\mathbf{X}_t) \mathbf{w}_{t-1} \right] \\
=& \boldsymbol{\eta}_{t-1} + \gamma \mathcal{Q}_d(\mathbf{X}_t)^\top (\boldsymbol{\epsilon}_t^{(o)} - \boldsymbol{\epsilon}_t^{(a)}) + \gamma \mathcal{Q}_d(\mathbf{X}_t)^\top \frac{1}{B} \\
& \left[ \mathcal{Q}_l(\mathbf{y}_t) - \mathcal{Q}_d(\mathbf{X}_t) \mathbf{w}^{(q)^*} - \mathcal{Q}_d(\mathbf{X}_t) \boldsymbol{\eta}_{t-1} - \mathcal{Q}_d(\mathbf{X}_t) \boldsymbol{\epsilon}_{t-1}^{(p)} \right] \\
=& \boldsymbol{\eta}_{t-1} + \gamma \frac{1}{B} \mathcal{Q}_d(\mathbf{X}_t)^\top (\boldsymbol{\epsilon}_t^{(o)} - \boldsymbol{\epsilon}_t^{(a)} + \boldsymbol{\xi}_t) - \gamma \mathcal{Q}_d(\mathbf{X}_t)^\top \frac{1}{B} \left[ \mathcal{Q}_d(\mathbf{X}_t) \boldsymbol{\eta}_{t-1} + \mathcal{Q}_d(\mathbf{X}_t) \boldsymbol{\epsilon}_{t-1}^{(p)} \right] \\
=& \left( \mathbf{I} - \frac{1}{B} \gamma \mathcal{Q}_d(\mathbf{X}_t)^\top \mathcal{Q}_d(\mathbf{X}_t) \right) \boldsymbol{\eta}_{t-1} + \gamma \frac{1}{B} \mathcal{Q}_d(\mathbf{X}_t)^\top \left[ \boldsymbol{\xi}_t + \boldsymbol{\epsilon}_t^{(o)} - \boldsymbol{\epsilon}_t^{(a)} - \mathcal{Q}_d(\mathbf{X}_t) \boldsymbol{\epsilon}_{t-1}^{(p)} \right].
\end{aligned}
$$

$\square$

## A.2 Decomposition of the Excess Risk

In this section, we take the initial step to analyze the excess risk of averaged SGD iterate $\overline{\mathbf{w}}_N$. In particular, we define the deviation of the averaged SGD iterate as $\overline{\boldsymbol{\eta}}_N := \frac{1}{N} \sum_{t=0}^{N-1} \boldsymbol{\eta}_t$. We decompose the excess risk as follows.

**Lemma A.2** (Excess risk decomposition). *Under Assumption 3.1 and Assumption 3.2,*

$$
\mathbb{E}[\mathcal{E}(\overline{\mathbf{w}}_N)] = R_1 + R_2 + R_3 + R_4,
$$

*where*

$$
\begin{aligned}
R_1 =& -\frac{1}{2} \mathbb{E} \left[ \langle \overline{\mathbf{w}}_N, \mathcal{Q}_d(\mathbf{x}) - \mathbf{x} \rangle^2 \right], \\
R_2 =& \frac{1}{2} \langle \mathbf{H}^{(q)}, \mathbb{E}[\overline{\boldsymbol{\eta}}_N \otimes \overline{\boldsymbol{\eta}}_N] \rangle, \\
R_3 =& \frac{1}{2} \mathbb{E} \left[ \langle \mathbf{w}^{(q)^*}, \mathcal{Q}_d(\mathbf{x}) - \mathbf{x} \rangle^2 \right], \\
R_4 =& \frac{1}{2} \left\langle \mathbf{H}, (\mathbf{w}^* - \mathbf{w}^{(q)^*}) \otimes (\mathbf{w}^* - \mathbf{w}^{(q)^*}) \right\rangle.
\end{aligned}
$$

*Proof.* By the definition of the excess risk (Excess Risk),

$$
\begin{aligned}
\mathbb{E}[\mathcal{E}(\overline{\mathbf{w}}_N)] =& \frac{1}{2} \mathbb{E} \left[ (y - \langle \overline{\mathbf{w}}_N, \mathbf{x} \rangle)^2 \right] - \frac{1}{2} \mathbb{E} \left[ (y - \langle \mathbf{w}^*, \mathbf{x} \rangle)^2 \right] \\
=& \underbrace{\frac{1}{2} \mathbb{E} \left[ (y - \langle \overline{\mathbf{w}}_N, \mathbf{x} \rangle)^2 \right] - \frac{1}{2} \mathbb{E} \left[ (\mathcal{Q}_l(y) - \langle \overline{\mathbf{w}}_N, \mathcal{Q}_d(\mathbf{x}) \rangle)^2 \right]}_{E_1} \\
& + \underbrace{\frac{1}{2} \mathbb{E} \left[ (\mathcal{Q}_l(y) - \langle \overline{\mathbf{w}}_N, \mathcal{Q}_d(\mathbf{x}) \rangle)^2 \right] - \frac{1}{2} \mathbb{E} \left[ (\mathcal{Q}_l(y) - \langle \mathbf{w}^{(q)^*}, \mathcal{Q}_d(\mathbf{x}) \rangle)^2 \right]}_{E_2} \\
& + \underbrace{\frac{1}{2} \mathbb{E} \left[ (\mathcal{Q}_l(y) - \langle \mathbf{w}^{(q)^*}, \mathcal{Q}_d(\mathbf{x}) \rangle)^2 \right] - \frac{1}{2} \mathbb{E} \left[ \left( y - \langle \mathbf{w}^{(q)^*}, \mathbf{x} \rangle \right)^2 \right]}_{E_3} \\
& + \underbrace{\frac{1}{2} \mathbb{E} \left[ \left( y - \langle \mathbf{w}^{(q)^*}, \mathbf{x} \rangle \right)^2 \right] - \frac{1}{2} \mathbb{E} \left[ (y - \langle \mathbf{w}^*, \mathbf{x} \rangle)^2 \right]}_{E_4},
\end{aligned}
$$

where $E_1$ captures the gap of the averaged SGD iterate between the full-precision and quantized domains, $E_2$ characterizes the distance from the averaged SGD iterate to the quantized optimal solution within the quantized domain, $E_3$ represents the mismatch of the quantized optimal solution in full-precision data space and quantized data space and $E_4$ defines the discrepancy between the averaged SGD iterate and the quantized optimal solution in the full-precision domain.

We would like to remark that the quantization operations $\mathcal{Q}_l(y)$ and $\mathcal{Q}_d(\mathbf{x})$ introduced in excess risk decomposition are independent of those quantization operators introduced in the training stage, i.e., $\overline{\mathbf{w}}_N$. Next, we analyze $E_1, E_2, E_3$ and $E_4$ respectively. These computations are mainly based on the first order optimality condition (A.1) and the unbiased quantization Assumption 3.1. For $E_4$,

$$
\begin{aligned}
E_4 =& \frac{1}{2}\mathbb{E}\left[\left(y - \langle \mathbf{w}^{(q)^*}, \mathbf{x}\rangle\right)^2\right] - \frac{1}{2}\mathbb{E}\left[(y - \langle \mathbf{w}^*, \mathbf{x}\rangle)^2\right]\\
=& \frac{1}{2}\mathbb{E}\left[\langle \mathbf{w}^* - \mathbf{w}^{(q)^*}, \mathbf{x}\rangle \cdot \left(2y - \langle \mathbf{w}^* + \mathbf{w}^{(q)^*}, \mathbf{x}\rangle\right)\right]\\
=& \frac{1}{2}\mathbb{E}\left[\langle \mathbf{w}^* - \mathbf{w}^{(q)^*}, \mathbf{x}\rangle^2\right]\\
=& \frac{1}{2}\left(\mathbf{w}^* - \mathbf{w}^{(q)^*}\right)^{\top}\mathbf{H}\left(\mathbf{w}^* - \mathbf{w}^{(q)^*}\right)\\
=& \frac{1}{2}\langle \mathbf{H}, (\mathbf{w}^* - \mathbf{w}^{(q)^*}) \otimes (\mathbf{w}^* - \mathbf{w}^{(q)^*})\rangle,
\end{aligned}
\tag{A.3}
$$

where the third equality uses the first order optimality condition that $\mathbb{E}_{(\mathbf{x},y)\sim\mathcal{D}}[(y - \langle\mathbf{w}^*, \mathbf{x}\rangle)\mathbf{x}] = \mathbf{0}$.

For $E_2$, similarly by the first order optimality condition (A.1) with respect to $\mathbf{w}^{(q)^*}$, it holds

$$
\begin{aligned}
E_2 =& \frac{1}{2}\mathbb{E}\left[(\mathcal{Q}_l(y) - \langle \overline{\mathbf{w}}_N, \mathcal{Q}_d(\mathbf{x})\rangle)^2\right] - \frac{1}{2}\mathbb{E}\left[(\mathcal{Q}_l(y) - \langle \mathbf{w}^{(q)^*}, \mathcal{Q}_d(\mathbf{x})\rangle)^2\right]\\
=& \frac{1}{2}\mathbb{E}\left[\langle \mathbf{w}^{(q)^*} - \overline{\mathbf{w}}_N, \mathcal{Q}_d(\mathbf{x})\rangle \cdot \left(2\mathcal{Q}_l(y) - \langle \mathbf{w}^{(q)^*} + \overline{\mathbf{w}}_N, \mathcal{Q}_d(\mathbf{x})\rangle\right)\right]\\
=& \frac{1}{2}\mathbb{E}\left[\langle \mathbf{w}^{(q)^*} - \overline{\mathbf{w}}_N, \mathcal{Q}_d(\mathbf{x})\rangle^2\right]\\
=& \frac{1}{2}\langle \mathbf{H}^{(q)}, \mathbb{E}[\overline{\boldsymbol{\eta}}_N \otimes \overline{\boldsymbol{\eta}}_N]\rangle.
\end{aligned}
\tag{A.4}
$$

For $E_3$,

$$
\begin{aligned}
E_3 =& \frac{1}{2}\mathbb{E}\left[(\mathcal{Q}_l(y) - \langle \mathbf{w}^{(q)^*}, \mathcal{Q}_d(\mathbf{x})\rangle)^2\right] - \frac{1}{2}\mathbb{E}\left[\left(y - \langle \mathbf{w}^{(q)^*}, \mathbf{x}\rangle\right)^2\right]\\
=& \frac{1}{2}\mathbb{E}\left[\left(\mathcal{Q}_l(y) - y - \langle \mathbf{w}^{(q)^*}, \mathcal{Q}_d(\mathbf{x}) - \mathbf{x}\rangle\right) \cdot \left(\mathcal{Q}_l(y) + y - \langle \mathbf{w}^{(q)^*}, \mathcal{Q}_d(\mathbf{x}) + \mathbf{x}\rangle\right)\right]\\
=& \frac{1}{2}\mathbb{E}\left[\mathcal{Q}_l(y)^2 - y^2 + \langle \mathbf{w}^{(q)^*}, \mathcal{Q}_d(\mathbf{x}) - \mathbf{x}\rangle\langle \mathbf{w}^{(q)^*}, \mathcal{Q}_d(\mathbf{x}) + \mathbf{x}\rangle\right],
\end{aligned}
$$

where the last equality utilizes the unbiased quantization Assumption 3.1.

For $E_1$, similarly by the unbiased quantization Assumption 3.1, it holds

$$
\begin{aligned}
E_1 =& \frac{1}{2}\mathbb{E}\left[(y - \langle \overline{\mathbf{w}}_N, \mathbf{x}\rangle)^2\right] - \frac{1}{2}\mathbb{E}\left[(\mathcal{Q}_l(y) - \langle \overline{\mathbf{w}}_N, \mathcal{Q}_d(\mathbf{x})\rangle)^2\right]\\
=& \frac{1}{2}\mathbb{E}\left[(y - \mathcal{Q}_l(y) - \langle \overline{\mathbf{w}}_N, \mathbf{x} - \mathcal{Q}_d(\mathbf{x})\rangle) \cdot (y + \mathcal{Q}_l(y) - \langle \overline{\mathbf{w}}_N, \mathbf{x} + \mathcal{Q}_d(\mathbf{x})\rangle)\right]\\
=& \frac{1}{2}\mathbb{E}\left[y^2 - \mathcal{Q}_l(y)^2\right] + \frac{1}{2}\mathbb{E}\left[\langle \overline{\mathbf{w}}_N, \mathbf{x} - \mathcal{Q}_d(\mathbf{x})\rangle\langle \overline{\mathbf{w}}_N, \mathbf{x} + \mathcal{Q}_d(\mathbf{x})\rangle\right].
\end{aligned}
$$

Hence,

$$
E_1 + E_3 = \frac{1}{2}\mathbb{E}\left[\langle \mathbf{w}^{(q)^*}, \mathcal{Q}_d(\mathbf{x}) - \mathbf{x}\rangle^2\right] - \frac{1}{2}\mathbb{E}\left[\langle \overline{\mathbf{w}}_N, \mathbf{x} - \mathcal{Q}_d(\mathbf{x})\rangle^2\right].
\tag{A.5}
$$

Therefore, combining (A.3), (A.4) and (A.5) we have

$$
\begin{aligned}
\mathbb{E}[\mathcal{E}(\overline{\mathbf{w}}_N)] =& \frac{1}{2}\langle \mathbf{H}^{(q)}, \mathbb{E}[\overline{\boldsymbol{\eta}}_N \otimes \overline{\boldsymbol{\eta}}_N]\rangle + \frac{1}{2}\langle \mathbf{H}, (\mathbf{w}^* - \mathbf{w}^{(q)^*}) \otimes (\mathbf{w}^* - \mathbf{w}^{(q)^*})\rangle\\
& + \frac{1}{2}\mathbb{E}\left[\langle \mathbf{w}^{(q)^*}, \mathcal{Q}_d(\mathbf{x}) - \mathbf{x}\rangle^2\right] - \frac{1}{2}\mathbb{E}\left[\langle \overline{\mathbf{w}}_N, \mathbf{x} - \mathcal{Q}_d(\mathbf{x})\rangle^2\right].
\end{aligned}
$$

$\square$

**Lemma A.3** (Refine excess risk decomposition). *Under Assumption 3.1 and Assumption 3.2, if the stepsize $\gamma < \frac{1}{\lambda_1^{(q)}}$,*

$$\mathbb{E}[\mathcal{E}(\overline{\mathbf{w}}_N)] = \underbrace{\frac{1}{2}\langle \mathbf{H}, \mathbb{E}[\overline{\boldsymbol{\eta}}_N \otimes \overline{\boldsymbol{\eta}}_N]\rangle}_{R_N} + \text{ApproxErr},$$

*where*

$$\begin{aligned}
\text{ApproxErr} =& \frac{1}{2}\langle \mathbf{H}, (\mathbf{w}^* - \mathbf{w}^{(q)^*}) \otimes (\mathbf{w}^* - \mathbf{w}^{(q)^*})\rangle \\
&+ \left(\mathbf{w}^{(q)^*}\right)^\top \frac{1}{N\gamma}\left(\mathbf{I} - (\mathbf{I} - \gamma\mathbf{H}^{(q)})^N\right)(\mathbf{H}^{(q)})^{-1}\left(\mathbf{H}^{(q)} - \mathbf{H}\right)\mathbf{w}^{(q)^*}.
\end{aligned}$$

*Proof.* By Lemma A.2,

$$\begin{aligned}
\mathbb{E}[\mathcal{E}(\overline{\mathbf{w}}_N)] =& \frac{1}{2}\langle \mathbf{H}^{(q)}, \mathbb{E}[\overline{\boldsymbol{\eta}}_N \otimes \overline{\boldsymbol{\eta}}_N]\rangle \\
&+ \frac{1}{2}\langle \mathbf{H}, (\mathbf{w}^* - \mathbf{w}^{(q)^*}) \otimes (\mathbf{w}^* - \mathbf{w}^{(q)^*})\rangle \\
&+ \frac{1}{2}\mathbb{E}\left[\langle\mathbf{w}^{(q)^*}, \mathcal{Q}_d(\mathbf{x}) - \mathbf{x}\rangle^2\right] - \frac{1}{2}\mathbb{E}\left[\langle\overline{\mathbf{w}}_N, \mathbf{x} - \mathcal{Q}_d(\mathbf{x})\rangle^2\right].
\end{aligned}$$

We then focus on $\frac{1}{2}\mathbb{E}\left[\langle\overline{\mathbf{w}}_N, \mathbf{x} - \mathcal{Q}_d(\mathbf{x})\rangle^2\right]$. Recall that

$$\overline{\mathbf{w}}_N = \overline{\mathbf{w}}_N - \mathbf{w}^{(q)^*} + \mathbf{w}^{(q)^*} = \overline{\boldsymbol{\eta}}_N + \mathbf{w}^{(q)^*},$$

we have

$$\begin{aligned}
\frac{1}{2}\mathbb{E}\left[\langle\overline{\mathbf{w}}_N, \mathbf{x} - \mathcal{Q}_d(\mathbf{x})\rangle^2\right] =& \frac{1}{2}\mathbb{E}\left[\overline{\mathbf{w}}_N^\top\left(\mathbf{H}^{(q)} - \mathbf{H}\right)\overline{\mathbf{w}}_N\right] \\
=& \frac{1}{2}\mathbb{E}\left[\overline{\boldsymbol{\eta}}_N^\top\left(\mathbf{H}^{(q)} - \mathbf{H}\right)\overline{\boldsymbol{\eta}}_N\right] + \frac{1}{2}\mathbb{E}\left[\left(\mathbf{w}^{(q)^*}\right)^\top\left(\mathbf{H}^{(q)} - \mathbf{H}\right)\mathbf{w}^{(q)^*}\right] \\
&+ \mathbb{E}\left[\overline{\boldsymbol{\eta}}_N^\top\left(\mathbf{H}^{(q)} - \mathbf{H}\right)\mathbf{w}^{(q)^*}\right].
\end{aligned}$$

Hence,

$$\begin{aligned}
\mathbb{E}[\mathcal{E}(\overline{\mathbf{w}}_N)] =& \frac{1}{2}\langle \mathbf{H}, \mathbb{E}[\overline{\boldsymbol{\eta}}_N \otimes \overline{\boldsymbol{\eta}}_N]\rangle + \frac{1}{2}\langle \mathbf{H}, (\mathbf{w}^* - \mathbf{w}^{(q)^*}) \otimes (\mathbf{w}^* - \mathbf{w}^{(q)^*})\rangle \\
&- \mathbb{E}\left[\overline{\boldsymbol{\eta}}_N^\top\left(\mathbf{H}^{(q)} - \mathbf{H}\right)\mathbf{w}^{(q)^*}\right].
\end{aligned} \tag{A.6}$$

Noticing that by Lemma A.1,

$$\boldsymbol{\eta}_t = \left(\mathbf{I} - \frac{1}{B}\gamma\mathcal{Q}_d(\mathbf{X}_t)^\top\mathcal{Q}_d(\mathbf{X}_t)\right)\boldsymbol{\eta}_{t-1} + \gamma\frac{1}{B}\mathcal{Q}_d(\mathbf{X}_t)^\top\left[\boldsymbol{\xi}_t + \boldsymbol{\epsilon}_t^{(o)} - \boldsymbol{\epsilon}_t^{(a)} - \mathcal{Q}_d(\mathbf{X}_t)\boldsymbol{\epsilon}_{t-1}^{(p)}\right],$$

it follows by Assumption 3.1 that

$$\mathbb{E}[\boldsymbol{\eta}_t] = \mathbb{E}[\mathbb{E}[\boldsymbol{\eta}_t|\boldsymbol{\eta}_{t-1}]] = \mathbb{E}\left[\left(\mathbf{I} - \gamma\mathbf{H}^{(q)}\right)\boldsymbol{\eta}_{t-1}\right] = \left(\mathbf{I} - \gamma\mathbf{H}^{(q)}\right)\mathbb{E}[\boldsymbol{\eta}_{t-1}] = \left(\mathbf{I} - \gamma\mathbf{H}^{(q)}\right)^t\boldsymbol{\eta}_0.$$

Hence,

$$\begin{aligned}
-\mathbb{E}\left[\overline{\boldsymbol{\eta}}_N^\top\left(\mathbf{H}^{(q)} - \mathbf{H}\right)\mathbf{w}^{(q)^*}\right] =& -\boldsymbol{\eta}_0^\top\frac{1}{N}\sum_{t=0}^{N-1}\left(\mathbf{I} - \gamma\mathbf{H}^{(q)}\right)^t\left(\mathbf{H}^{(q)} - \mathbf{H}\right)\mathbf{w}^{(q)^*} \\
=& \left(\mathbf{w}^{(q)^*}\right)^\top\frac{1}{N}\sum_{t=0}^{N-1}\left(\mathbf{I} - \gamma\mathbf{H}^{(q)}\right)^t\left(\mathbf{H}^{(q)} - \mathbf{H}\right)\mathbf{w}^{(q)^*} \\
=& \left(\mathbf{w}^{(q)^*}\right)^\top\frac{1}{N\gamma}\left(\mathbf{I} - (\mathbf{I} - \gamma\mathbf{H}^{(q)})^N\right)(\mathbf{H}^{(q)})^{-1}\left(\mathbf{H}^{(q)} - \mathbf{H}\right)\mathbf{w}^{(q)^*}.
\end{aligned}$$

Combining (A.6) completes the proof. $\square$

# B  ANALYSIS OF APPROXIMATION ERROR

In this section, we analyze ApproxErr under multiplicative quantization and additive quantization, respectively. We first apply the definition of $\mathbf{w}^{(q)^*}$:

$$\mathbf{w}^{(q)^*} = (\mathbf{H}^{(q)})^{-1}\mathbf{H}\mathbf{w}^*.$$

We first handle $\frac{1}{2}\langle \mathbf{H}, (\mathbf{w}^* - \mathbf{w}^{(q)^*}) \otimes (\mathbf{w}^* - \mathbf{w}^{(q)^*})\rangle$. Recall $\mathbf{D} = \mathbf{H}^{(q)} - \mathbf{H}$, we have

$$
\begin{aligned}
\frac{1}{2}\langle \mathbf{H}, (\mathbf{w}^* - \mathbf{w}^{(q)^*}) \otimes (\mathbf{w}^* - \mathbf{w}^{(q)^*})\rangle =& \frac{1}{2}\mathbb{E}\left[\operatorname{tr}\left(\mathbf{H}(\mathbf{H}+\mathbf{D})^{-1}\mathbf{D}\mathbf{w}^*\mathbf{w}^{*\top}\mathbf{D}(\mathbf{H}+\mathbf{D})^{-1}\right)\right] \\
=& \frac{1}{2}\mathbb{E}\left[\operatorname{tr}\left(\mathbf{w}^{*\top}\mathbf{D}(\mathbf{H}+\mathbf{D})^{-1}\mathbf{H}(\mathbf{H}+\mathbf{D})^{-1}\mathbf{D}\mathbf{w}^*\right)\right] \\
=& \frac{1}{2}\|\mathbf{w}^*\|^2_{\mathbf{D}(\mathbf{H}+\mathbf{D})^{-1}\mathbf{H}(\mathbf{H}+\mathbf{D})^{-1}\mathbf{D}}.
\end{aligned}
$$
(B.1)

We then handle $\left(\mathbf{w}^{(q)^*}\right)^\top \frac{1}{N\gamma}\left(\mathbf{I} - (\mathbf{I} - \gamma\mathbf{H}^{(q)})^N\right)(\mathbf{H}^{(q)})^{-1}\left(\mathbf{H}^{(q)} - \mathbf{H}\right)\mathbf{w}^{(q)^*}$.

$$
\begin{aligned}
&\left(\mathbf{w}^{(q)^*}\right)^\top \frac{1}{N\gamma}\left(\mathbf{I} - (\mathbf{I} - \gamma\mathbf{H}^{(q)})^N\right)(\mathbf{H}^{(q)})^{-1}\left(\mathbf{H}^{(q)} - \mathbf{H}\right)\mathbf{w}^{(q)^*} \\
=&\mathbf{w}^{*\top}\mathbf{H}(\mathbf{H}^{(q)})^{-1}\frac{1}{N\gamma}\left(\mathbf{I} - (\mathbf{I} - \gamma\mathbf{H}^{(q)})^N\right)(\mathbf{H}^{(q)})^{-1}\left(\mathbf{H}^{(q)} - \mathbf{H}\right)(\mathbf{H}^{(q)})^{-1}\mathbf{H}\mathbf{w}^*.
\end{aligned}
$$
(B.2)

**Lemma B.1** (Approximation error under multiplicative quantization). *If there exists $\epsilon_d$ such that $\mathcal{Q}_d$ is $\epsilon_d$-multiplicative, under the assumptions and notations in Lemma A.3,*

$$\operatorname{ApproxErr} \le \frac{\epsilon_d^2}{2(1+\epsilon_d)^2}\|\mathbf{w}^*\|^2_{\mathbf{H}} + \frac{\epsilon_d}{(1+\epsilon_d)^2}\|\mathbf{w}^*\|^2_{\mathbf{H}}.$$

*Proof.* Under multiplicative quantization,

$$\mathbf{H}^{(q)} = (1+\epsilon_d)\mathbf{H}, \quad \mathbf{D} = \epsilon_d\mathbf{H}.$$

It follows by (B.1) that

$$\frac{1}{2}\langle \mathbf{H}, (\mathbf{w}^* - \mathbf{w}^{(q)^*}) \otimes (\mathbf{w}^* - \mathbf{w}^{(q)^*})\rangle = \frac{1}{2}\|\mathbf{w}^*\|^2_{\mathbf{D}(\mathbf{H}+\mathbf{D})^{-1}\mathbf{H}(\mathbf{H}+\mathbf{D})^{-1}\mathbf{D}} = \frac{\epsilon_d^2}{2(1+\epsilon_d)^2}\|\mathbf{w}^*\|^2_{\mathbf{H}}.$$

Similarly, by (B.2),

$$
\begin{aligned}
&\left(\mathbf{w}^{(q)^*}\right)^\top \frac{1}{N\gamma}\left(\mathbf{I} - (\mathbf{I} - \gamma\mathbf{H}^{(q)})^N\right)(\mathbf{H}^{(q)})^{-1}\left(\mathbf{H}^{(q)} - \mathbf{H}\right)\mathbf{w}^{(q)^*} \\
=&\mathbf{w}^{*\top}\mathbf{H}(\mathbf{H}^{(q)})^{-1}\frac{1}{N\gamma}\left(\mathbf{I} - (\mathbf{I} - \gamma\mathbf{H}^{(q)})^N\right)(\mathbf{H}^{(q)})^{-1}\left(\mathbf{H}^{(q)} - \mathbf{H}\right)(\mathbf{H}^{(q)})^{-1}\mathbf{H}\mathbf{w}^* \\
=&\frac{\epsilon_d}{N\gamma(1+\epsilon_d)}\mathbf{w}^{*\top}\mathbf{H}(\mathbf{H}^{(q)})^{-1}\left(\mathbf{I} - (\mathbf{I} - \gamma\mathbf{H}^{(q)})^N\right)(\mathbf{H}^{(q)})^{-1}\mathbf{H}\mathbf{w}^* \\
\le&\frac{\epsilon_d}{N\gamma(1+\epsilon_d)^2}\|\mathbf{w}^*\|^2_{\mathbf{H}}\left\|(\mathbf{H}^{(q)})^{-1/2}\left(\mathbf{I} - (\mathbf{I} - \gamma\mathbf{H}^{(q)})^N\right)(\mathbf{H}^{(q)})^{-1/2}\right\| \\
\le&\frac{\epsilon_d}{N\gamma(1+\epsilon_d)^2}\|\mathbf{w}^*\|^2_{\mathbf{H}}\max_i \frac{\min\{1, N\gamma\lambda_i^{(q)}\}}{\lambda_i^{(q)}} \\
\le&\frac{\epsilon_d}{(1+\epsilon_d)^2}\|\mathbf{w}^*\|^2_{\mathbf{H}}.
\end{aligned}
$$

$\square$

**Lemma B.2** (Approximation error under additive quantization). *If there exists $\epsilon_d$ such that $\mathcal{Q}_d$ is $\epsilon_d$-additive, under the assumptions and notations in Lemma A.3,*

$$\operatorname{ApproxErr} \le \frac{\epsilon_d^2}{2(\lambda_d + \epsilon_d)^2}\|\mathbf{w}^*\|^2_{\mathbf{H}} + \frac{\lambda_1\epsilon_d}{(\lambda_d + \epsilon_d)(\lambda_1 + \epsilon_d)}\|\mathbf{w}^*\|^2_{\mathbf{H}}.$$

*Proof.* Under additive quantization,

$$\mathbf{H}^{(q)} = \mathbf{H} + \epsilon_d\mathbf{I}, \quad \mathbf{D} = \epsilon_d\mathbf{I}.$$

It follows by (B.1) that

$$\frac{1}{2}\langle\mathbf{H}, (\mathbf{w}^* - \mathbf{w}^{(q)*}) \otimes (\mathbf{w}^* - \mathbf{w}^{(q)*})\rangle = \frac{1}{2}\|\mathbf{w}^*\|^2_{\mathbf{D}(\mathbf{H}+\mathbf{D})^{-1}\mathbf{H}(\mathbf{H}+\mathbf{D})^{-1}\mathbf{D}} \leq \frac{\epsilon_d^2}{2(\lambda_d + \epsilon_d)^2}\|\mathbf{w}^*\|^2_{\mathbf{H}}.$$

Similarly, by (B.2),

$$\left(\mathbf{w}^{(q)*}\right)^\top \frac{1}{N\gamma}\left(\mathbf{I} - (\mathbf{I} - \gamma\mathbf{H}^{(q)})^N\right)(\mathbf{H}^{(q)})^{-1}\left(\mathbf{H}^{(q)} - \mathbf{H}\right)\mathbf{w}^{(q)*}$$

$$= \mathbf{w}^{*\top}\mathbf{H}(\mathbf{H}^{(q)})^{-1}\frac{1}{N\gamma}\left(\mathbf{I} - (\mathbf{I} - \gamma\mathbf{H}^{(q)})^N\right)(\mathbf{H}^{(q)})^{-1}\left(\mathbf{H}^{(q)} - \mathbf{H}\right)(\mathbf{H}^{(q)})^{-1}\mathbf{H}\mathbf{w}^*$$

$$\leq \frac{\epsilon_d}{N\gamma(\lambda_d + \epsilon_d)}\mathbf{w}^{*\top}\mathbf{H}(\mathbf{H}^{(q)})^{-1}\left(\mathbf{I} - (\mathbf{I} - \gamma\mathbf{H}^{(q)})^N\right)(\mathbf{H}^{(q)})^{-1}\mathbf{H}\mathbf{w}^*$$

$$\leq \frac{\lambda_1\epsilon_d}{N\gamma(\lambda_d + \epsilon_d)(\lambda_1 + \epsilon_d)}\|\mathbf{w}^*\|^2_{\mathbf{H}}\left\|(\mathbf{H}^{(q)})^{-1/2}\left(\mathbf{I} - (\mathbf{I} - \gamma\mathbf{H}^{(q)})^N\right)(\mathbf{H}^{(q)})^{-1/2}\right\|$$

$$\leq \frac{\lambda_1\epsilon_d}{N\gamma(\lambda_d + \epsilon_d)(\lambda_1 + \epsilon_d)}\|\mathbf{w}^*\|^2_{\mathbf{H}}\max_i \frac{\min\{1, N\gamma\lambda_i^{(q)}\}}{\lambda_i^{(q)}}$$

$$\leq \frac{\lambda_1\epsilon_d}{(\lambda_d + \epsilon_d)(\lambda_1 + \epsilon_d)}\|\mathbf{w}^*\|^2_{\mathbf{H}}.$$

$\square$

# C   ANALYSIS OF $R_N$

## C.1   PRELIMINARY

We first define the following linear operators as in Zou et al. (2023):

$$\mathcal{I} = \mathbf{I} \otimes \mathbf{I}, \quad \mathcal{M}^{(q)} = \mathbb{E}[\mathbf{x}^{(q)} \otimes \mathbf{x}^{(q)} \otimes \mathbf{x}^{(q)} \otimes \mathbf{x}^{(q)}], \quad \widetilde{\mathcal{M}}^{(q)} = \mathbf{H}^{(q)} \otimes \mathbf{H}^{(q)},$$

$$\mathcal{T}^{(q)} = \mathbf{H}^{(q)} \otimes \mathbf{I} + \mathbf{I} \otimes \mathbf{H}^{(q)} - \gamma\mathcal{M}^{(q)}, \quad \widetilde{\mathcal{T}}^{(q)} = \mathbf{H}^{(q)} \otimes \mathbf{I} + \mathbf{I} \otimes \mathbf{H}^{(q)} - \gamma\mathbf{H}^{(q)} \otimes \mathbf{H}^{(q)}.$$

For a symmetric matrix $\mathbf{A}$, the above definitions result in:

$$\mathcal{I} \circ \mathbf{A} = \mathbf{A}, \quad \mathcal{M}^{(q)} \circ \mathbf{A} = \mathbb{E}[(\mathbf{x}^{(q)\top}\mathbf{A}\mathbf{x}^{(q)})\mathbf{x}^{(q)}\mathbf{x}^{(q)\top}], \quad \widetilde{\mathcal{M}}^{(q)} \circ \mathbf{A} = \mathbf{H}^{(q)}\mathbf{A}\mathbf{H}^{(q)},$$

$$(\mathcal{I} - \gamma\mathcal{T}^{(q)}) \circ \mathbf{A} = \mathbb{E}[(\mathbf{I} - \gamma\mathbf{x}^{(q)}\mathbf{x}^{(q)\top})\mathbf{A}(\mathbf{I} - \gamma\mathbf{x}^{(q)}\mathbf{x}^{(q)\top})], \quad (\mathcal{I} - \gamma\widetilde{\mathcal{T}}^{(q)}) \circ \mathbf{A} = (\mathbf{I} - \gamma\mathbf{H}^{(q)})\mathbf{A}(\mathbf{I} - \gamma\mathbf{H}^{(q)}).$$

Further, we generalize the linear operators from Zou et al. (2023) to account for batch size effects. For a symmetric matrix $\mathbf{A}$, we define

$$\mathcal{M}_B^{(q)} \circ \mathbf{A} = \mathbb{E}\left[\frac{1}{B^2}\mathbf{X}^{(q)\top}\mathbf{X}^{(q)}\mathbf{A}\mathbf{X}^{(q)\top}\mathbf{X}^{(q)}\right],$$

$$(\mathcal{I} - \gamma\mathcal{T}_B^{(q)}) \circ \mathbf{A} = \mathbb{E}\left[\left(\mathbf{I} - \gamma\frac{1}{B}\mathbf{X}^{(q)\top}\mathbf{X}^{(q)}\right)\mathbf{A}\left(\mathbf{I} - \gamma\frac{1}{B}\mathbf{X}^{(q)\top}\mathbf{X}^{(q)}\right)\right].$$

## C.2   INITIAL STUDY

To analyze $R_N$, we firstly utilize the fact that

$$R_N = \frac{1}{2}\langle\mathbf{H}, \mathbb{E}[\overline{\boldsymbol{\eta}}_N \otimes \overline{\boldsymbol{\eta}}_N]\rangle \leq \mu_{\max}\left(\mathbf{H}(\mathbf{H}^{(q)})^{-1}\right)\frac{1}{2}\langle\mathbf{H}^{(q)}, \mathbb{E}[\overline{\boldsymbol{\eta}}_N \otimes \overline{\boldsymbol{\eta}}_N]\rangle \leq \underbrace{\frac{1}{2}\langle\mathbf{H}^{(q)}, \mathbb{E}[\overline{\boldsymbol{\eta}}_N \otimes \overline{\boldsymbol{\eta}}_N]\rangle}_{R_N^{(0)}}.$$
(C.1)

We secondly substitute $\overline{\boldsymbol{\eta}}_N$ with the summation of $\boldsymbol{\eta}_t$. This step mainly based on the propagation in Lemma A.1, the unbiased quantization Assumption 3.1 and the first order optimality condition (A.1). We summarize as the following lemma.

**Lemma C.1.** *Under Assumption 3.1 and Assumption 3.2,*

$$R_N^{(0)} \leq \frac{1}{N^2} \cdot \sum_{t=0}^{N-1} \sum_{k=t}^{N-1} \left\langle (\mathbf{I} - \gamma \mathbf{H}^{(q)})^{k-t} \mathbf{H}^{(q)}, \mathbb{E}[\boldsymbol{\eta}_t \otimes \boldsymbol{\eta}_t] \right\rangle.$$

*Proof.* Recall that by (C.1),

$$R_N^{(0)} = \frac{1}{2} \langle \mathbf{H}^{(q)}, \mathbb{E}[\overline{\boldsymbol{\eta}}_N \otimes \overline{\boldsymbol{\eta}}_N] \rangle,$$

we then focus on $\mathbb{E}[\overline{\boldsymbol{\eta}}_N \otimes \overline{\boldsymbol{\eta}}_N]$. By definition $\overline{\boldsymbol{\eta}}_N = \frac{1}{N} \sum_{t=0}^{N-1} \boldsymbol{\eta}_t$,

$$\begin{aligned}
\mathbb{E}[\overline{\boldsymbol{\eta}}_N \otimes \overline{\boldsymbol{\eta}}_N] = & \frac{1}{N^2} \cdot \left( \sum_{0 \leq k \leq t \leq N-1} \mathbb{E}[\boldsymbol{\eta}_t \otimes \boldsymbol{\eta}_k] + \sum_{0 \leq t < k \leq N-1} \mathbb{E}[\boldsymbol{\eta}_t \otimes \boldsymbol{\eta}_k] \right) \\
\preceq & \frac{1}{N^2} \cdot \left( \sum_{0 \leq k \leq t \leq N-1} \mathbb{E}\left[\mathbb{E}[\boldsymbol{\eta}_t \otimes \boldsymbol{\eta}_k | \boldsymbol{\eta}_k]\right] + \sum_{0 \leq t \leq k \leq N-1} \mathbb{E}\left[\mathbb{E}[\boldsymbol{\eta}_t \otimes \boldsymbol{\eta}_k | \boldsymbol{\eta}_t]\right] \right).
\end{aligned} \tag{C.2}$$

Note that by the unbiased Assumption 3.1,

$$\mathbb{E}\left[ \gamma \mathcal{Q}_d(\mathbf{X}_t)^\top \left( \boldsymbol{\epsilon}_t^{(o)} - \boldsymbol{\epsilon}_t^{(a)} - \mathcal{Q}_d(\mathbf{X}_t) \boldsymbol{\epsilon}_{t-1}^{(p)} \right) \bigg| \boldsymbol{\eta}_{t-1} \right] = \mathbf{0}.$$

Further, by the optimality (A.1),

$$\mathbb{E}\left[ \gamma \mathcal{Q}_d(\mathbf{X}_t)^\top \boldsymbol{\xi}_t \bigg| \boldsymbol{\eta}_{t-1} \right] = \mathbb{E}\left[ \gamma \mathcal{Q}_d(\mathbf{X}_t)^\top \left[ \mathcal{Q}_l(\mathbf{y}_t) - \mathcal{Q}_d(\mathbf{X}_t) \mathbf{w}^{(q)*} \right] \bigg| \boldsymbol{\eta}_{t-1} \right] = \mathbf{0}.$$

Hence, by Lemma A.1,

$$\mathbb{E}\left[ \boldsymbol{\eta}_t | \boldsymbol{\eta}_{t-1} \right] = \left( \mathbf{I} - \gamma \mathbf{H}^{(q)} \right) \boldsymbol{\eta}_{t-1}. \tag{C.3}$$

Therefore, by (C.2) and (C.3),

$$\begin{aligned}
& \mathbb{E}[\overline{\boldsymbol{\eta}}_N \otimes \overline{\boldsymbol{\eta}}_N] \\
\preceq & \frac{1}{N^2} \cdot \left( \sum_{0 \leq k \leq t \leq N-1} \mathbb{E}\left[\mathbb{E}[\boldsymbol{\eta}_t \otimes \boldsymbol{\eta}_k | \boldsymbol{\eta}_k]\right] + \sum_{0 \leq t \leq k \leq N-1} \mathbb{E}\left[\mathbb{E}[\boldsymbol{\eta}_t \otimes \boldsymbol{\eta}_k | \boldsymbol{\eta}_t]\right] \right) \\
= & \frac{1}{N^2} \cdot \left( \sum_{0 \leq k \leq t \leq N-1} (\mathbf{I} - \gamma \mathbf{H}^{(q)})^{t-k} \mathbb{E}[\boldsymbol{\eta}_k \otimes \boldsymbol{\eta}_k] + \sum_{0 \leq t \leq k \leq N-1} \mathbb{E}[\boldsymbol{\eta}_t \otimes \boldsymbol{\eta}_t](\mathbf{I} - \gamma \mathbf{H}^{(q)})^{k-t} \right) \\
= & \frac{1}{N^2} \cdot \sum_{t=0}^{N-1} \sum_{k=t}^{N-1} \left( (\mathbf{I} - \gamma \mathbf{H}^{(q)})^{k-t} \mathbb{E}[\boldsymbol{\eta}_t \otimes \boldsymbol{\eta}_t] + \mathbb{E}[\boldsymbol{\eta}_t \otimes \boldsymbol{\eta}_t](\mathbf{I} - \gamma \mathbf{H}^{(q)})^{k-t} \right).
\end{aligned} \tag{C.4}$$

Applying (C.4) into $R_N^{(0)}$, we have

$$\begin{aligned}
R_N^{(0)} = & \frac{1}{2} \langle \mathbf{H}^{(q)}, \mathbb{E}[\overline{\boldsymbol{\eta}}_N \otimes \overline{\boldsymbol{\eta}}_N] \rangle \\
\leq & \frac{1}{2N^2} \cdot \sum_{t=0}^{N-1} \sum_{k=t}^{N-1} \left\langle \mathbf{H}^{(q)}, (\mathbf{I} - \gamma \mathbf{H}^{(q)})^{k-t} \mathbb{E}[\boldsymbol{\eta}_t \otimes \boldsymbol{\eta}_t] + \mathbb{E}[\boldsymbol{\eta}_t \otimes \boldsymbol{\eta}_t](\mathbf{I} - \gamma \mathbf{H}^{(q)})^{k-t} \right\rangle \\
= & \frac{1}{N^2} \cdot \sum_{t=0}^{N-1} \sum_{k=t}^{N-1} \left\langle (\mathbf{I} - \gamma \mathbf{H}^{(q)})^{k-t} \mathbf{H}^{(q)}, \mathbb{E}[\boldsymbol{\eta}_t \otimes \boldsymbol{\eta}_t] \right\rangle,
\end{aligned}$$

where the last equality holds since $\mathbf{H}^{(q)}$ and $(\mathbf{I} - \gamma \mathbf{H}^{(q)})^{k-t}$ commute. This completes the proof. $\square$

Lemma C.1 implies that, to bound $R_N^{(0)}$, the main goal is to bound $\mathbb{E}[\boldsymbol{\eta}_t \otimes \boldsymbol{\eta}_t]$. Recall that by Lemma A.1,

$$\boldsymbol{\eta}_t = \left(\mathbf{I} - \frac{1}{B}\gamma \mathcal{Q}_d(\mathbf{X}_t)^\top \mathcal{Q}_d(\mathbf{X}_t)\right)\boldsymbol{\eta}_{t-1} + \gamma\frac{1}{B}\mathcal{Q}_d(\mathbf{X}_t)^\top\left[\boldsymbol{\xi}_t + \boldsymbol{\epsilon}_t^{(o)} - \boldsymbol{\epsilon}_t^{(a)} - \mathcal{Q}_d(\mathbf{X}_t)\boldsymbol{\epsilon}_{t-1}^{(p)}\right].$$

Denote

$$\boldsymbol{\eta}_t^{\text{bias}} = \left(\mathbf{I} - \frac{1}{B}\gamma \mathcal{Q}_d(\mathbf{X}_t)^\top \mathcal{Q}_d(\mathbf{X}_t)\right)\boldsymbol{\eta}_{t-1}^{\text{bias}}, \quad \boldsymbol{\eta}_0^{\text{bias}} = \boldsymbol{\eta}_0,$$

$$\boldsymbol{\eta}_t^{\text{var}} = \left(\mathbf{I} - \frac{1}{B}\gamma \mathcal{Q}_d(\mathbf{X}_t)^\top \mathcal{Q}_d(\mathbf{X}_t)\right)\boldsymbol{\eta}_{t-1}^{\text{var}} + \gamma\frac{1}{B}\mathcal{Q}_d(\mathbf{X}_t)^\top\left[\boldsymbol{\xi}_t + \boldsymbol{\epsilon}_t^{(o)} - \boldsymbol{\epsilon}_t^{(a)} - \mathcal{Q}_d(\mathbf{X}_t)\boldsymbol{\epsilon}_{t-1}^{(p)}\right],$$

with $\boldsymbol{\eta}_0^{\text{var}} = \mathbf{0}$. Then

$$\boldsymbol{\eta}_t = \boldsymbol{\eta}_t^{\text{var}} + \boldsymbol{\eta}_t^{\text{bias}},$$

and

$$\mathbb{E}\left[\boldsymbol{\eta}_t \otimes \boldsymbol{\eta}_t\right] \preceq 2\left(\underbrace{\mathbb{E}\left[\boldsymbol{\eta}_t^{\text{bias}} \otimes \boldsymbol{\eta}_t^{\text{bias}}\right]}_{\mathbf{B}_t} + \underbrace{\mathbb{E}\left[\boldsymbol{\eta}_t^{\text{var}} \otimes \boldsymbol{\eta}_t^{\text{var}}\right]}_{\mathbf{C}_t}\right). \tag{C.5}$$

Regarding $\mathbf{B}_t$,

$$\mathbf{B}_t = \mathbb{E}\left[\left(\mathbf{I} - \gamma\frac{1}{B}\mathcal{Q}_d(\mathbf{X})^\top\mathcal{Q}_d(\mathbf{X})\right)\mathbf{B}_{t-1}\left(\mathbf{I} - \gamma\frac{1}{B}\mathcal{Q}_d(\mathbf{X})^\top\mathcal{Q}_d(\mathbf{X})\right)\right]. \tag{C.6}$$

Regarding $\mathbf{C}_t$, by the unbiased quantization Assumption 3.1 and $\boldsymbol{\eta}_0^{\text{var}} = \mathbf{0}$, it holds,

$$\mathbf{C}_t = \mathbb{E}\left[\left(\mathbf{I} - \gamma\frac{1}{B}\mathcal{Q}_d(\mathbf{X})^\top\mathcal{Q}_d(\mathbf{X})\right)\mathbf{C}_{t-1}\left(\mathbf{I} - \gamma\frac{1}{B}\mathcal{Q}_d(\mathbf{X})^\top\mathcal{Q}_d(\mathbf{X})\right)\right] + \gamma^2\boldsymbol{\Sigma}_t, \tag{C.7}$$

where

$$\boldsymbol{\Sigma}_t := \frac{1}{B^2}\mathbb{E}\left[\mathcal{Q}_d(\mathbf{X}_t)^\top\left[\boldsymbol{\xi}_t + \boldsymbol{\epsilon}_t^{(o)} - \boldsymbol{\epsilon}_t^{(a)} - \mathcal{Q}_d(\mathbf{X}_t)\boldsymbol{\epsilon}_{t-1}^{(p)}\right]\left[\boldsymbol{\xi}_t + \boldsymbol{\epsilon}_t^{(o)} - \boldsymbol{\epsilon}_t^{(a)} - \mathcal{Q}_d(\mathbf{X}_t)\boldsymbol{\epsilon}_{t-1}^{(p)}\right]^\top\mathcal{Q}_d(\mathbf{X}_t)\right]$$

$$= \underbrace{\frac{1}{B^2}\mathbb{E}\left[\mathcal{Q}_d(\mathbf{X}_t)^\top\boldsymbol{\xi}_t\boldsymbol{\xi}_t^\top\mathcal{Q}_d(\mathbf{X}_t)\right]}_{\boldsymbol{\Sigma}_t^\xi} + \underbrace{\frac{1}{B^2}\mathbb{E}\left[\mathcal{Q}_d(\mathbf{X}_t)^\top\boldsymbol{\epsilon}_t^{(o)}\boldsymbol{\epsilon}_t^{(o)\top}\mathcal{Q}_d(\mathbf{X}_t)\right]}_{\boldsymbol{\Sigma}_t^{\epsilon^{(o)}}}$$

$$+ \underbrace{\frac{1}{B^2}\mathbb{E}\left[\mathcal{Q}_d(\mathbf{X}_t)^\top\boldsymbol{\epsilon}_t^{(a)}\boldsymbol{\epsilon}_t^{(a)\top}\mathcal{Q}_d(\mathbf{X}_t)\right]}_{\boldsymbol{\Sigma}_t^{\epsilon^{(a)}}} + \underbrace{\frac{1}{B^2}\mathbb{E}\left[\mathcal{Q}_d(\mathbf{X}_t)^\top\mathcal{Q}_d(\mathbf{X}_t)\boldsymbol{\epsilon}_{t-1}^{(p)}\boldsymbol{\epsilon}_{t-1}^{(p)\top}\mathcal{Q}_d(\mathbf{X}_t)^\top\mathcal{Q}_d(\mathbf{X}_t)\right]}_{\boldsymbol{\Sigma}_t^{\epsilon^{(p)}}}.$$

We then summarize the update rule for $\mathbb{E}\left[\boldsymbol{\eta}_t \otimes \boldsymbol{\eta}_t\right]$ as follows.

**Lemma C.2** (Update rule under general quantization). *Under Assumption 3.1, Assumption 3.2, Assumption 3.3, and Assumption 3.4,*

$$\mathbf{C}_t \preceq \mathbb{E}\left[\left(\mathbf{I} - \gamma\frac{1}{B}\mathcal{Q}_d(\mathbf{X})^\top\mathcal{Q}_d(\mathbf{X})\right)\mathbf{C}_{t-1}\left(\mathbf{I} - \gamma\frac{1}{B}\mathcal{Q}_d(\mathbf{X})^\top\mathcal{Q}_d(\mathbf{X})\right)\right] + \gamma^2\sigma_G^{(q)2}\mathbf{H}^{(q)},$$

$$\mathbf{B}_t = \mathbb{E}\left[\left(\mathbf{I} - \gamma\frac{1}{B}\mathcal{Q}_d(\mathbf{X})^\top\mathcal{Q}_d(\mathbf{X})\right)\mathbf{B}_{t-1}\left(\mathbf{I} - \gamma\frac{1}{B}\mathcal{Q}_d(\mathbf{X})^\top\mathcal{Q}_d(\mathbf{X})\right)\right],$$

*where*

$$\sigma_G^{(q)2} = \frac{\sup_t\left\{\left\|\mathbb{E}\left[\boldsymbol{\epsilon}_t^{(o)}\boldsymbol{\epsilon}_t^{(o)\top}|\mathbf{o}_t\right] + \mathbb{E}\left[\boldsymbol{\epsilon}_t^{(a)}\boldsymbol{\epsilon}_t^{(a)\top}|\mathbf{a}_t\right]\right\|\right\}}{B}$$

$$+ \alpha_B\sup_t\mathbb{E}_{\mathbf{w}_{t-1}}\left[\text{tr}\left(\mathbf{H}^{(q)}\mathbb{E}\left[\boldsymbol{\epsilon}_{t-1}^{(p)}\boldsymbol{\epsilon}_{t-1}^{(p)\top}|\mathbf{w}_{t-1}\right]\right)\right] + \frac{\sigma^2}{B},$$

*with $\mathbf{a}_t = \mathcal{Q}_d(\mathbf{X}_t)\mathcal{Q}_p(\mathbf{w}_{t-1})$, $\mathbf{o}_t = \mathcal{Q}_l(\mathbf{y}_t) - \mathcal{Q}_a\left(\mathcal{Q}_d(\mathbf{X}_t)\mathcal{Q}_p(\mathbf{w}_{t-1})\right)$ and $\|\cdot\|$ denoting the spectral norm.*

*Proof.* We cope with each term in $\boldsymbol{\Sigma}_t$ to provide an upper bound. For $\boldsymbol{\Sigma}_t^{\epsilon^{(p)}}$,

$$
\begin{aligned}
\boldsymbol{\Sigma}_t^{\epsilon^{(p)}} =& \frac{1}{B^2} \mathbb{E}\left[ \mathcal{Q}_d(\mathbf{X}_t)^\top \mathcal{Q}_d(\mathbf{X}_t) \boldsymbol{\epsilon}_{t-1}^{(p)} {\boldsymbol{\epsilon}_{t-1}^{(p)}}^\top \mathcal{Q}_d(\mathbf{X}_t)^\top \mathcal{Q}_d(\mathbf{X}_t) \right] \\
\preceq& \alpha_B \sup_t \mathbb{E}_{\mathbf{w}_{t-1}}\left[ \operatorname{tr}\left( \mathbf{H}^{(q)} \mathbb{E}\left[ \boldsymbol{\epsilon}_{t-1}^{(p)} {\boldsymbol{\epsilon}_{t-1}^{(p)}}^\top \big| \mathbf{w}_{t-1} \right] \right) \right] \mathbf{H}^{(q)},
\end{aligned}
$$

where the inequality holds by Assumption 3.3. For $\boldsymbol{\Sigma}_t^\xi$,

$$
\begin{aligned}
\boldsymbol{\Sigma}_t^\xi =& \frac{1}{B^2} \mathbb{E}\left[ \mathcal{Q}_d(\mathbf{X}_t)^\top \boldsymbol{\xi}_t \boldsymbol{\xi}_t^\top \mathcal{Q}_d(\mathbf{X}_t) \right] \\
=& \frac{1}{B^2} \mathbb{E}\left[ \sum_{i=1}^B \sum_{j=1}^B \mathcal{Q}_d(\mathbf{X}_t)^{i\,\top} \boldsymbol{\xi}_t^i \left( \mathcal{Q}_d(\mathbf{X}_t)^{j\,\top} \boldsymbol{\xi}_t^j \right)^\top \right] \\
=& \frac{1}{B^2} \sum_{i=1}^B \mathbb{E}\left[ \mathcal{Q}_d(\mathbf{X}_t)^{i\,\top} \boldsymbol{\xi}_t^i \left( \mathcal{Q}_d(\mathbf{X}_t)^{i\,\top} \boldsymbol{\xi}_t^i \right)^\top \right] \\
=& \frac{1}{B} \cdot \mathbb{E}\left[ \mathcal{Q}_d(\mathbf{x}) \xi \left( \mathcal{Q}_d(\mathbf{x}) \xi \right)^\top \right] \\
=& \frac{1}{B} \cdot \mathbb{E}\left[ \xi^2 \mathcal{Q}_d(\mathbf{x}) \mathcal{Q}_d(\mathbf{x})^\top \right] \\
\preceq& \frac{\sigma^2}{B} \cdot \mathbf{H}^{(q)},
\end{aligned}
\tag{C.8}
$$

where the third equality holds as samples are independent and data quantization is applied to each sample independently, the inequality holds by Assumption 3.4. For $\boldsymbol{\Sigma}_t^{\epsilon^{(o)}} + \boldsymbol{\Sigma}_t^{\epsilon^{(a)}}$,

$$
\begin{aligned}
\boldsymbol{\Sigma}_t^{\epsilon^{(o)}} + \boldsymbol{\Sigma}_t^{\epsilon^{(a)}} =& \frac{1}{B^2} \mathbb{E}\left[ \mathcal{Q}_d(\mathbf{X}_t)^\top (\boldsymbol{\epsilon}_t^{(o)} {\boldsymbol{\epsilon}_t^{(o)}}^\top + \boldsymbol{\epsilon}_t^{(a)} {\boldsymbol{\epsilon}_t^{(a)}}^\top) \mathcal{Q}_d(\mathbf{X}_t) \right] \\
=& \frac{1}{B^2} \mathbb{E}\left[ \mathcal{Q}_d(\mathbf{X}_t)^\top \left( \mathbb{E}\left[ \boldsymbol{\epsilon}_t^{(o)} {\boldsymbol{\epsilon}_t^{(o)}}^\top | \mathbf{o}_t \right] + \mathbb{E}\left[ \boldsymbol{\epsilon}_t^{(a)} {\boldsymbol{\epsilon}_t^{(a)}}^\top | \mathbf{a}_t \right] \right) \mathcal{Q}_d(\mathbf{X}_t) \right] \\
\preceq& \frac{1}{B^2} \mathbb{E}\left[ \left( \left\| \mathbb{E}\left[ \boldsymbol{\epsilon}_t^{(o)} {\boldsymbol{\epsilon}_t^{(o)}}^\top | \mathbf{o}_t \right] + \mathbb{E}\left[ \boldsymbol{\epsilon}_t^{(a)} {\boldsymbol{\epsilon}_t^{(a)}}^\top | \mathbf{a}_t \right] \right\| \right) \mathcal{Q}_d(\mathbf{X}_t)^\top \mathcal{Q}_d(\mathbf{X}_t) \right] \\
\preceq& \frac{1}{B^2} \sup_t \left[ \left\| \mathbb{E}\left[ \boldsymbol{\epsilon}_t^{(o)} {\boldsymbol{\epsilon}_t^{(o)}}^\top | \mathbf{o}_t \right] + \mathbb{E}\left[ \boldsymbol{\epsilon}_t^{(a)} {\boldsymbol{\epsilon}_t^{(a)}}^\top | \mathbf{a}_t \right] \right\| \right] \mathbb{E}\left[ \mathcal{Q}_d(\mathbf{X}_t)^\top \mathcal{Q}_d(\mathbf{X}_t) \right] \\
=& \frac{1}{B} \sup_t \left[ \left\| \mathbb{E}\left[ \boldsymbol{\epsilon}_t^{(o)} {\boldsymbol{\epsilon}_t^{(o)}}^\top | \mathbf{o}_t \right] + \mathbb{E}\left[ \boldsymbol{\epsilon}_t^{(a)} {\boldsymbol{\epsilon}_t^{(a)}}^\top | \mathbf{a}_t \right] \right\| \right] \mathbf{H}^{(q)},
\end{aligned}
$$

where $\| \cdot \|$ represents the matrix spectral norm.

Combining the upper bounds for $\boldsymbol{\Sigma}_t^{\epsilon^{(p)}}$, $\boldsymbol{\Sigma}_t^\xi$, $\boldsymbol{\Sigma}_t^{\epsilon^{(o)}} + \boldsymbol{\Sigma}_t^{\epsilon^{(a)}}$, (C.6) and (C.7) immediately completes the proof. $\qquad\square$

For multiplicative quantization, the explicit dependence of the conditional expectations on $\mathbf{w}_t$ renders Lemma C.2 inapplicable to the update rule for $\mathbb{E}[\boldsymbol{\eta}_t \otimes \boldsymbol{\eta}_t]$. We thus propose the following alternative update rule.

**Lemma C.3** (Update rule under multiplicative quantization). *If there exist $\epsilon_d, \epsilon_l, \epsilon_p, \epsilon_a$ and $\epsilon_o$ such that for any $i \in \{d, l, p, a, o\}$, quantization $\mathcal{Q}_i$ is $\epsilon_i$-multiplicative, then under Assumption 3.1, Assumption 3.2, Assumption 3.3, and Assumption 3.4, it holds*

$$
\begin{aligned}
\mathbf{C}_t \preceq& \mathbb{E}\left[ \left( \mathbf{I} - \frac{1}{B} \gamma \mathcal{Q}_d(\mathbf{X})^\top \mathcal{Q}_d(\mathbf{X}) \right) \mathbf{C}_{t-1} \left( \mathbf{I} - \frac{1}{B} \gamma \mathcal{Q}_d(\mathbf{X})^\top \mathcal{Q}_d(\mathbf{X}) \right) \right] \\
&+ \tilde{\epsilon} \mathbb{E}\left[ \frac{\gamma}{B} \mathcal{Q}_d(\mathbf{X})^\top \mathcal{Q}_d(\mathbf{X}) (\mathbf{B}_{t-1} + \mathbf{C}_{t-1}) \frac{\gamma}{B} \mathcal{Q}_d(\mathbf{X})^\top \mathcal{Q}_d(\mathbf{X}) \right] + \gamma^2 {\sigma_M^{(q)}}^2 \mathbf{H}^{(q)}, \\
\mathbf{B}_t =& \mathbb{E}\left[ \left( \mathbf{I} - \frac{1}{B} \gamma \mathcal{Q}_d(\mathbf{X})^\top \mathcal{Q}_d(\mathbf{X}) \right) \mathbf{B}_{t-1} \left( \mathbf{I} - \frac{1}{B} \gamma \mathcal{Q}_d(\mathbf{X})^\top \mathcal{Q}_d(\mathbf{X}) \right) \right],
\end{aligned}
$$

*where*

$$\tilde{\epsilon} = 8\epsilon_o(1 + \epsilon_p)(1 + \epsilon_a) + 4\epsilon_p + 4\epsilon_a(1 + \epsilon_p),$$

$$\sigma_M^{(q)^2} = \frac{(1 + 4\epsilon_o)\sigma^2}{B} + \frac{\|\mathbf{w}^*\|_{\mathbf{H}}^2}{1 + \epsilon_d}\alpha_B\left(4\epsilon_o[(1 + \epsilon_p)(1 + \epsilon_a) + 1] + 2\epsilon_a(1 + \epsilon_p) + 2\epsilon_p\right).$$

*Proof.* To complete the proof, we merely need to derive the upper bound for $\boldsymbol{\Sigma}_t = \boldsymbol{\Sigma}_t^{\xi} + \boldsymbol{\Sigma}_t^{\epsilon^{(a)}} + \boldsymbol{\Sigma}_t^{\epsilon^{(o)}} + \boldsymbol{\Sigma}_t^{\epsilon^{(p)}}$. Regarding $\boldsymbol{\Sigma}_t^{\xi}$, by the computation in the proof of Lemma C.2, i.e., (C.8),

$$\boldsymbol{\Sigma}_t^{\xi} \preceq \frac{\sigma^2}{B}\mathbf{H}^{(q)}. \tag{C.9}$$

Regarding $\boldsymbol{\Sigma}_t^{\epsilon^{(p)}}$,

$$
\begin{aligned}
\boldsymbol{\Sigma}_t^{\epsilon^{(p)}} =& \frac{1}{B^2}\mathbb{E}\left[\mathcal{Q}_d(\mathbf{X}_t)^{\top}\mathcal{Q}_d(\mathbf{X}_t)\mathbb{E}\left[\boldsymbol{\epsilon}_{t-1}^{(p)}\boldsymbol{\epsilon}_{t-1}^{(p)\top}\big|\mathbf{w}_{t-1}\right]\mathcal{Q}_d(\mathbf{X}_t)^{\top}\mathcal{Q}_d(\mathbf{X}_t)\right] \\
=& \frac{\epsilon_p}{B^2}\mathbb{E}\left[\mathcal{Q}_d(\mathbf{X}_t)^{\top}\mathcal{Q}_d(\mathbf{X}_t)\mathbf{w}_{t-1}\mathbf{w}_{t-1}^{\top}\mathcal{Q}_d(\mathbf{X}_t)^{\top}\mathcal{Q}_d(\mathbf{X}_t)\right] \\
\preceq& \frac{2\epsilon_p}{B^2}\mathbb{E}\left[\mathcal{Q}_d(\mathbf{X}_t)^{\top}\mathcal{Q}_d(\mathbf{X}_t)\boldsymbol{\eta}_{t-1}\boldsymbol{\eta}_{t-1}^{\top}\mathcal{Q}_d(\mathbf{X}_t)^{\top}\mathcal{Q}_d(\mathbf{X}_t)\right] \\
&+ \frac{2\epsilon_p}{B^2}\mathbb{E}\left[\mathcal{Q}_d(\mathbf{X}_t)^{\top}\mathcal{Q}_d(\mathbf{X}_t)\mathbf{w}^{(q)*}\mathbf{w}^{(q)*\top}\mathcal{Q}_d(\mathbf{X}_t)^{\top}\mathcal{Q}_d(\mathbf{X}_t)\right].
\end{aligned}
\tag{C.10}
$$

Regarding $\boldsymbol{\Sigma}_t^{\epsilon^{(a)}}$,

$$
\begin{aligned}
\boldsymbol{\Sigma}_t^{\epsilon^{(a)}} =& \frac{1}{B^2}\mathbb{E}\left[\mathcal{Q}_d(\mathbf{X}_t)^{\top}\boldsymbol{\epsilon}_t^{(a)}\boldsymbol{\epsilon}_t^{(a)\top}\mathcal{Q}_d(\mathbf{X}_t)\right] \\
=& \frac{\epsilon_a}{B^2}\mathbb{E}\left[\mathcal{Q}_d(\mathbf{X}_t)^{\top}\mathcal{Q}_d(\mathbf{X}_t)\mathbf{w}_{t-1}^{(q)}\mathbf{w}_{t-1}^{(q)\top}\mathcal{Q}_d(\mathbf{X}_t)^{\top}\mathcal{Q}_d(\mathbf{X}_t)\right] \\
=& \frac{(1 + \epsilon_p)\epsilon_a}{B^2}\mathbb{E}\left[\mathcal{Q}_d(\mathbf{X}_t)^{\top}\mathcal{Q}_d(\mathbf{X}_t)\mathbf{w}_{t-1}\mathbf{w}_{t-1}^{\top}\mathcal{Q}_d(\mathbf{X}_t)^{\top}\mathcal{Q}_d(\mathbf{X}_t)\right] \\
\preceq& \frac{2(1 + \epsilon_p)\epsilon_a}{B^2}\mathbb{E}\left[\mathcal{Q}_d(\mathbf{X}_t)^{\top}\mathcal{Q}_d(\mathbf{X}_t)\boldsymbol{\eta}_{t-1}\boldsymbol{\eta}_{t-1}^{\top}\mathcal{Q}_d(\mathbf{X}_t)^{\top}\mathcal{Q}_d(\mathbf{X}_t)\right] \\
&+ \frac{2(1 + \epsilon_p)\epsilon_a}{B^2}\mathbb{E}\left[\mathcal{Q}_d(\mathbf{X}_t)^{\top}\mathcal{Q}_d(\mathbf{X}_t)\mathbf{w}^{(q)*}\mathbf{w}^{(q)*\top}\mathcal{Q}_d(\mathbf{X}_t)^{\top}\mathcal{Q}_d(\mathbf{X}_t)\right].
\end{aligned}
\tag{C.11}
$$

Regarding $\boldsymbol{\Sigma}_t^{\epsilon^{(o)}}$, similar to $\boldsymbol{\Sigma}_t^{\epsilon^{(a)}}$, it holds

$$
\begin{aligned}
\boldsymbol{\Sigma}_t^{\epsilon^{(o)}} =& \frac{1}{B^2}\mathbb{E}\left[\mathcal{Q}_d(\mathbf{X}_t)^{\top}\boldsymbol{\epsilon}_t^{(o)}\boldsymbol{\epsilon}_t^{(o)\top}\mathcal{Q}_d(\mathbf{X}_t)\right] \\
=& \frac{\epsilon_o}{B^2}\mathbb{E}\left[\mathcal{Q}_d(\mathbf{X}_t)^{\top}\mathbf{o}_t\mathbf{o}_t^{\top}\mathcal{Q}_d(\mathbf{X}_t)\right] \\
\preceq& \frac{2\epsilon_o}{B^2}\mathbb{E}\left[\mathcal{Q}_d(\mathbf{X}_t)^{\top}\mathcal{Q}_l(\mathbf{y}_t)\mathcal{Q}_l(\mathbf{y}_t)^{\top}\mathcal{Q}_d(\mathbf{X}_t)\right] + \frac{2\epsilon_o}{B^2}\mathbb{E}\left[\mathcal{Q}_d(\mathbf{X}_t)^{\top}\mathcal{Q}_a(\mathbf{a}_t)\mathcal{Q}_a(\mathbf{a}_t)^{\top}\mathcal{Q}_d(\mathbf{X}_t)\right].
\end{aligned}
$$

For the second term,

$$
\begin{aligned}
& \frac{2\epsilon_o}{B^2}\mathbb{E}\left[\mathcal{Q}_d(\mathbf{X}_t)^{\top}\mathcal{Q}_a(\mathbf{a}_t)\mathcal{Q}_a(\mathbf{a}_t)^{\top}\mathcal{Q}_d(\mathbf{X}_t)\right] \\
\preceq& \frac{2(1 + \epsilon_a)\epsilon_o}{B^2}\mathbb{E}\left[\mathcal{Q}_d(\mathbf{X}_t)^{\top}\mathbf{a}_t\mathbf{a}_t^{\top}\mathcal{Q}_d(\mathbf{X}_t)\right] \\
\preceq& \frac{4(1 + \epsilon_p)(1 + \epsilon_a)\epsilon_o}{B^2}\mathbb{E}\left[\mathcal{Q}_d(\mathbf{X}_t)^{\top}\mathcal{Q}_d(\mathbf{X}_t)\boldsymbol{\eta}_{t-1}\boldsymbol{\eta}_{t-1}^{\top}\mathcal{Q}_d(\mathbf{X}_t)^{\top}\mathcal{Q}_d(\mathbf{X}_t)\right] \\
&+ \frac{4(1 + \epsilon_p)(1 + \epsilon_a)\epsilon_o}{B^2}\mathbb{E}\left[\mathcal{Q}_d(\mathbf{X}_t)^{\top}\mathcal{Q}_d(\mathbf{X}_t)\mathbf{w}^{(q)*}\mathbf{w}^{(q)*\top}\mathcal{Q}_d(\mathbf{X}_t)^{\top}\mathcal{Q}_d(\mathbf{X}_t)\right].
\end{aligned}
$$

For the first term,

$$\frac{2\epsilon_o}{B^2}\mathbb{E}\left[\mathcal{Q}_d(\mathbf{X}_t)^\top\mathcal{Q}_l(\mathbf{y}_t)\mathcal{Q}_l(\mathbf{y}_t)^\top\mathcal{Q}_d(\mathbf{X}_t)\right] \preceq \frac{4\epsilon_o}{B^2}\mathbb{E}\left[\mathcal{Q}_d(\mathbf{X}_t)^\top\boldsymbol{\xi}_t\boldsymbol{\xi}_t^\top\mathcal{Q}_d(\mathbf{X}_t)\right]$$
$$+\frac{4\epsilon_o}{B^2}\mathbb{E}\left[\mathcal{Q}_d(\mathbf{X}_t)^\top\mathcal{Q}_d(\mathbf{X}_t)\mathbf{w}^{(q)^*}\mathbf{w}^{(q)^{*\top}}\mathcal{Q}_d(\mathbf{X}_t)^\top\mathcal{Q}_d(\mathbf{X}_t)\right],$$

where we use $\boldsymbol{\xi}_t = \mathcal{Q}_l(\mathbf{y}_t) - \mathcal{Q}_d(\mathbf{X}_t)\mathbf{w}^{(q)^*}$. Further, by the bound for $\boldsymbol{\Sigma}_t^\xi$ (C.8), we have

$$\frac{1}{B^2}\mathbb{E}\left[\mathcal{Q}_d(\mathbf{X}_t)^\top\boldsymbol{\xi}_t\boldsymbol{\xi}_t^\top\mathcal{Q}_d(\mathbf{X}_t)\right] \preceq \frac{\sigma^2}{B}\mathbf{H}^{(q)},$$

it follows that

$$\boldsymbol{\Sigma}_t^{\epsilon^{(o)}} \preceq \frac{4\epsilon_o\sigma^2}{B}\mathbf{H}^{(q)} + \frac{4(1+\epsilon_p)(1+\epsilon_a)\epsilon_o}{B^2}\mathbb{E}\left[\mathcal{Q}_d(\mathbf{X}_t)^\top\mathcal{Q}_d(\mathbf{X}_t)\boldsymbol{\eta}_{t-1}\boldsymbol{\eta}_{t-1}^\top\mathcal{Q}_d(\mathbf{X}_t)^\top\mathcal{Q}_d(\mathbf{X}_t)\right]$$
$$+\frac{4\epsilon_o[(1+\epsilon_p)(1+\epsilon_a)+1]}{B^2}\mathbb{E}\left[\mathcal{Q}_d(\mathbf{X}_t)^\top\mathcal{Q}_d(\mathbf{X}_t)\mathbf{w}^{(q)^*}\mathbf{w}^{(q)^{*\top}}\mathcal{Q}_d(\mathbf{X}_t)^\top\mathcal{Q}_d(\mathbf{X}_t)\right].$$
$$(C.12)$$

Further, by Assumption 3.3, it holds

$$\frac{1}{B^2}\mathbb{E}\left[\mathcal{Q}_d(\mathbf{X}_t)^\top\mathcal{Q}_d(\mathbf{X}_t)\mathbf{w}^{(q)^*}\mathbf{w}^{(q)^{*\top}}\mathcal{Q}_d(\mathbf{X}_t)^\top\mathcal{Q}_d(\mathbf{X}_t)\right] \preceq \alpha_B \mathrm{tr}\left(\mathbf{H}^{(q)}\mathbf{w}^{(q)^*}\mathbf{w}^{(q)^{*\top}}\right)\mathbf{H}^{(q)},$$

then together with (C.9), (C.10), (C.11) and (C.12) it holds

$$\boldsymbol{\Sigma}_t \preceq \frac{(1+4\epsilon_o)\sigma^2}{B}\mathbf{H}^{(q)} + \alpha_B\left(4\epsilon_o[(1+\epsilon_p)(1+\epsilon_a)+1]+2\epsilon_a(1+\epsilon_p)+2\epsilon_p\right)\mathrm{tr}\left(\mathbf{H}^{(q)}\mathbf{w}^{(q)^*}\mathbf{w}^{(q)^{*\top}}\right)\mathbf{H}^{(q)}$$
$$+\frac{4\epsilon_o(1+\epsilon_p)(1+\epsilon_a)+2\epsilon_p+2\epsilon_a(1+\epsilon_p)}{B^2}\mathbb{E}\left[\mathcal{Q}_d(\mathbf{X}_t)^\top\mathcal{Q}_d(\mathbf{X}_t)\boldsymbol{\eta}_{t-1}\boldsymbol{\eta}_{t-1}^\top\mathcal{Q}_d(\mathbf{X}_t)^\top\mathcal{Q}_d(\mathbf{X}_t)\right].$$

Note that by the definition of multiplicative quantization,

$$\mathrm{tr}\left(\mathbf{H}^{(q)}\mathbf{w}^{(q)^*}\mathbf{w}^{(q)^{*\top}}\right) = \frac{\|\mathbf{w}^*\|_\mathbf{H}^2}{1+\epsilon_d},$$

then

$$\boldsymbol{\Sigma}_t \preceq \left[\frac{(1+4\epsilon_o)\sigma^2}{B} + \frac{\|\mathbf{w}^*\|_\mathbf{H}^2}{1+\epsilon_d}\alpha_B\left(4\epsilon_o[(1+\epsilon_p)(1+\epsilon_a)+1]+2\epsilon_a(1+\epsilon_p)+2\epsilon_p\right)\right]\mathbf{H}^{(q)}$$
$$+\frac{4\epsilon_o(1+\epsilon_p)(1+\epsilon_a)+2\epsilon_p+2\epsilon_a(1+\epsilon_p)}{B^2}\mathbb{E}\left[\mathcal{Q}_d(\mathbf{X}_t)^\top\mathcal{Q}_d(\mathbf{X}_t)\boldsymbol{\eta}_{t-1}\boldsymbol{\eta}_{t-1}^\top\mathcal{Q}_d(\mathbf{X}_t)^\top\mathcal{Q}_d(\mathbf{X}_t)\right].$$
$$(C.13)$$

Hence, by (C.13), (C.7) and $\mathbb{E}\left[\boldsymbol{\eta}_t \otimes \boldsymbol{\eta}_t\right] \preceq 2(\mathbf{B}_t + \mathbf{C}_t)$, we have

$$\mathbf{C}_t \preceq \mathbb{E}\left[\left(\mathbf{I} - \frac{1}{B}\gamma\mathcal{Q}_d(\mathbf{X})^\top\mathcal{Q}_d(\mathbf{X})\right)\mathbf{C}_{t-1}\left(\mathbf{I} - \frac{1}{B}\gamma\mathcal{Q}_d(\mathbf{X})^\top\mathcal{Q}_d(\mathbf{X})\right)\right]$$
$$+ [8\epsilon_o(1+\epsilon_p)(1+\epsilon_a)+4\epsilon_p+4\epsilon_a(1+\epsilon_p)]\mathbb{E}\left[\frac{\gamma}{B}\mathcal{Q}_d(\mathbf{X})^\top\mathcal{Q}_d(\mathbf{X})(\mathbf{B}_{t-1}+\mathbf{C}_{t-1})\frac{\gamma}{B}\mathcal{Q}_d(\mathbf{X})^\top\mathcal{Q}_d(\mathbf{X})\right]$$
$$+\gamma^2\left[\frac{(1+4\epsilon_o)\sigma^2}{B} + \frac{\|\mathbf{w}^*\|_\mathbf{H}^2}{1+\epsilon_d}\alpha_B\left(4\epsilon_o[(1+\epsilon_p)(1+\epsilon_a)+1]+2\epsilon_a(1+\epsilon_p)+2\epsilon_p\right)\right]\mathbf{H}^{(q)}.$$

$\square$

Equipped with Lemma C.1, Lemma C.2 and Lemma C.3, we are ready to derive bounds for $R_N^{(0)}$. As shown in Zou et al. (2023), we first perform bias-variance decomposition.

## C.3 BIAS-VARIANCE DECOMPOSITION

As in Zou et al. (2023), we perform bias-variance for excess risk, which is summarized as the following lemma. Here we slightly abuse the notations of $\mathbf{B}_t$ and $\mathbf{C}_t$.

**Lemma C.4** (Bias-variance decomposition under general quantization). *Under Assumption 3.1, Assumption 3.2, Assumption 3.3, and Assumption 3.4,*

$$R_N^{(0)}/2 \le \underbrace{\frac{1}{N^2} \cdot \sum_{t=0}^{N-1} \sum_{k=t}^{N-1} \left\langle (\mathbf{I} - \gamma \mathbf{H}^{(q)})^{k-t} \mathbf{H}^{(q)}, \mathbf{B}_t \right\rangle}_{\text{bias}} + \underbrace{\frac{1}{N^2} \cdot \sum_{t=0}^{N-1} \sum_{k=t}^{N-1} \left\langle (\mathbf{I} - \gamma \mathbf{H}^{(q)})^{k-t} \mathbf{H}^{(q)}, \mathbf{C}_t \right\rangle}_{\text{variance}},$$

*where*

$$\mathbf{B}_t := (\mathcal{I} - \gamma \mathcal{T}_B^{(q)})^t \circ \mathbf{B}_0, \quad \mathbf{B}_0 = \mathbb{E}\left[\boldsymbol{\eta}_0 \otimes \boldsymbol{\eta}_0\right].$$

$$\mathbf{C}_t := (\mathcal{I} - \gamma \mathcal{T}_B^{(q)}) \circ \mathbf{C}_{t-1} + \gamma^2 {\sigma_G^{(q)}}^2 \mathbf{H}^{(q)}, \quad \mathbf{C}_0 = \mathbf{0}.$$

*Proof.* By Lemma C.1,

$$R_N^{(0)} \le \frac{1}{N^2} \cdot \sum_{t=0}^{N-1} \sum_{k=t}^{N-1} \left\langle (\mathbf{I} - \gamma \mathbf{H}^{(q)})^{k-t} \mathbf{H}^{(q)}, \mathbb{E}[\boldsymbol{\eta}_t \otimes \boldsymbol{\eta}_t] \right\rangle.$$

The proof is immediately completed by Lemma C.2 and $\mathbb{E}[\boldsymbol{\eta}_t \otimes \boldsymbol{\eta}_t] \preceq 2(\mathbf{B}_t + \mathbf{C}_t)$. $\qquad \square$

For multiplicative quantization, we can directly deduce from Lemma C.4 by the update rule under multiplicative quantization (Lemma C.3).

**Lemma C.5** (Bias-variance decomposition under multiplicative quantization). *Under Assumption 3.1, Assumption 3.2, Assumption 3.3, and Assumption 3.4, if there exist $\epsilon_d, \epsilon_l, \epsilon_p, \epsilon_a$ and $\epsilon_o$ such that for any $i \in \{d, l, p, a, o\}$, quantization $\mathcal{Q}_i$ is $\epsilon_i$-multiplicative, then*

$$R_N^{(0)}/2 \le \underbrace{\frac{1}{N^2} \cdot \sum_{t=0}^{N-1} \sum_{k=t}^{N-1} \left\langle (\mathbf{I} - \gamma \mathbf{H}^{(q)})^{k-t} \mathbf{H}^{(q)}, \mathbf{B}_t^{(M)} \right\rangle}_{\text{bias}} + \underbrace{\frac{1}{N^2} \cdot \sum_{t=0}^{N-1} \sum_{k=t}^{N-1} \left\langle (\mathbf{I} - \gamma \mathbf{H}^{(q)})^{k-t} \mathbf{H}^{(q)}, \mathbf{C}_t^{(M)} \right\rangle}_{\text{variance}},$$

*where*

$$\mathbf{B}_t^{(M)} := (\mathcal{I} - \gamma \mathcal{T}_B^{(q)} + \tilde{\epsilon} \gamma^2 \mathcal{M}_B^{(q)})^t \circ \mathbf{B}_0^{(M)}, \quad \mathbf{B}_0^{(M)} = \mathbb{E}\left[\boldsymbol{\eta}_0 \otimes \boldsymbol{\eta}_0\right].$$

$$\mathbf{C}_t^{(M)} := (\mathcal{I} - \gamma \mathcal{T}_B^{(q)} + \tilde{\epsilon} \gamma^2 \mathcal{M}_B^{(q)}) \circ \mathbf{C}_{t-1}^{(M)} + \gamma^2 {\sigma_M^{(q)}}^2 \mathbf{H}^{(q)}, \quad \mathbf{C}_0^{(M)} = 0.$$

*Proof.* By Lemma C.3,

$$\begin{aligned} \mathbf{C}_t \preceq & \mathbb{E}\left[ \left( \mathbf{I} - \frac{1}{B} \gamma \mathcal{Q}_d(\mathbf{X})^\top \mathcal{Q}_d(\mathbf{X}) \right) \mathbf{C}_{t-1} \left( \mathbf{I} - \frac{1}{B} \gamma \mathcal{Q}_d(\mathbf{X})^\top \mathcal{Q}_d(\mathbf{X}) \right) \right] \\ & + \tilde{\epsilon} \mathbb{E}\left[ \frac{\gamma}{B} \mathcal{Q}_d(\mathbf{X})^\top \mathcal{Q}_d(\mathbf{X}) (\mathbf{B}_{t-1} + \mathbf{C}_{t-1}) \frac{\gamma}{B} \mathcal{Q}_d(\mathbf{X})^\top \mathcal{Q}_d(\mathbf{X}) \right] + \gamma^2 {\sigma_M^{(q)}}^2 \mathbf{H}^{(q)}, \\ \mathbf{B}_t = & \mathbb{E}\left[ \left( \mathbf{I} - \frac{1}{B} \gamma \mathcal{Q}_d(\mathbf{X})^\top \mathcal{Q}_d(\mathbf{X}) \right) \mathbf{B}_{t-1} \left( \mathbf{I} - \frac{1}{B} \gamma \mathcal{Q}_d(\mathbf{X})^\top \mathcal{Q}_d(\mathbf{X}) \right) \right]. \end{aligned}$$

Hence,

$$\begin{aligned} \mathbb{E}\left[\boldsymbol{\eta}_t \otimes \boldsymbol{\eta}_t\right] \preceq & 2(\mathbf{B}_t + \mathbf{C}_t) \\ \preceq & 2 \left[ (\mathcal{I} - \gamma \mathcal{T}_B^{(q)} + \tilde{\epsilon} \gamma^2 \mathcal{M}_B^{(q)}) \circ (\mathbf{B}_{t-1} + \mathbf{C}_{t-1}) + \gamma^2 {\sigma_M^{(q)}}^2 \mathbf{H}^{(q)} \right] \\ \preceq & 2 \left( \mathbf{B}_t^{(M)} + \mathbf{C}_t^{(M)} \right). \end{aligned}$$

Applying Lemma C.1 completes the proof. $\qquad \square$

### C.4 BOUNDING THE BIAS ERROR

By Lemma C.4,

$$
\begin{aligned}
\text{bias} &= \frac{1}{N^2} \sum_{t=0}^{N-1} \sum_{k=t}^{N-1} \left\langle (\mathbf{I} - \gamma \mathbf{H}^{(q)})^{k-t} \mathbf{H}^{(q)}, \mathbf{B}_t \right\rangle \\
&= \frac{1}{\gamma N^2} \sum_{t=0}^{N-1} \left\langle \mathbf{I} - (\mathbf{I} - \gamma \mathbf{H}^{(q)})^{N-t}, \mathbf{B}_t \right\rangle \qquad\qquad \text{(C.14)} \\
&\le \frac{1}{\gamma N^2} \langle \mathbf{I} - (\mathbf{I} - \gamma \mathbf{H}^{(q)})^N, \sum_{t=0}^{N-1} \mathbf{B}_t \rangle.
\end{aligned}
$$

For $1 \le n \le N$, let $\mathbf{S}_n = \sum_{t=0}^{n-1} \mathbf{B}_t$, $\mathbf{S}_n^{(M)} = \sum_{t=0}^{n-1} \mathbf{B}_t^{(M)}$, then we only need to bound $\mathbf{S}_N$ and $\mathbf{S}_N^{(M)}$ to bound bias term under general quantization and multiplicative quantization, respectively. We first derive the update rule for $\mathbf{S}_t$ and $\mathbf{S}_t^{(M)}$.

**Lemma C.6** (Initial study of $\mathbf{S}_t$). *For* $1 \le t \le N$,

$$
\mathbf{S}_t \preceq (\mathcal{I} - \gamma \tilde{\mathcal{T}}^{(q)}) \circ \mathbf{S}_{t-1} + \gamma^2 \mathcal{M}_B^{(q)} \circ \mathbf{S}_N + \mathbf{B}_0.
$$

*Proof.* By definition,

$$
\begin{aligned}
\mathbf{S}_t &= \sum_{k=0}^{t-1} (\mathcal{I} - \gamma \mathcal{T}_B^{(q)})^k \circ \mathbf{B}_0 \\
&= (\mathcal{I} - \gamma \mathcal{T}_B^{(q)}) \circ \left( \sum_{k=1}^{t-1} (\mathcal{I} - \gamma \mathcal{T}_B^{(q)})^{k-1} \circ \mathbf{B}_0 \right) + \mathbf{B}_0 \qquad \text{(C.15)} \\
&= (\mathcal{I} - \gamma \mathcal{T}_B^{(q)}) \circ \mathbf{S}_{t-1} + \mathbf{B}_0.
\end{aligned}
$$

Then we convert $\mathcal{T}_B^{(q)}$ to $\tilde{\mathcal{T}}^{(q)}$. By (C.15),

$$
\begin{aligned}
\mathbf{S}_t &= (\mathcal{I} - \gamma \mathcal{T}_B^{(q)}) \circ \mathbf{S}_{t-1} + \mathbf{B}_0 \\
&= (\mathcal{I} - \gamma \tilde{\mathcal{T}}^{(q)}) \circ \mathbf{S}_{t-1} + \gamma (\tilde{\mathcal{T}}^{(q)} - \mathcal{T}_B^{(q)}) \circ \mathbf{S}_{t-1} + \mathbf{B}_0 \\
&= (\mathcal{I} - \gamma \tilde{\mathcal{T}}^{(q)}) \circ \mathbf{S}_{t-1} + \gamma^2 (\mathcal{M}_B^{(q)} - \widetilde{\mathcal{M}}^{(q)}) \circ \mathbf{S}_{t-1} + \mathbf{B}_0 \\
&\preceq (\mathcal{I} - \gamma \tilde{\mathcal{T}}^{(q)}) \circ \mathbf{S}_{t-1} + \gamma^2 \mathcal{M}_B^{(q)} \circ \mathbf{S}_N + \mathbf{B}_0,
\end{aligned}
$$

where the third equality holds by the definition of linear operators. $\qquad\square$

**Lemma C.7** (Initial study of $\mathbf{S}_t^{(M)}$). *For* $1 \le t \le N$,

$$
\mathbf{S}_t^{(M)} \preceq (\mathcal{I} - \gamma \tilde{\mathcal{T}}^{(q)}) \circ \mathbf{S}_{t-1}^{(M)} + (1 + \tilde{\epsilon}) \gamma^2 \mathcal{M}_B^{(q)} \circ \mathbf{S}_N^{(M)} + \mathbf{B}_0.
$$

*Proof.* The proof is similar to the proof for Lemma C.6.

$$
\begin{aligned}
\mathbf{S}_t^{(M)} &= (\mathcal{I} - \gamma \mathcal{T}_B^{(q)} + \tilde{\epsilon} \gamma^2 \mathcal{M}_B^{(q)}) \circ \mathbf{S}_{t-1}^{(M)} + \mathbf{B}_0 \\
&= (\mathcal{I} - \gamma \tilde{\mathcal{T}}^{(q)}) \circ \mathbf{S}_{t-1}^{(M)} + \gamma (\tilde{\mathcal{T}}^{(q)} - \mathcal{T}_B^{(q)}) \circ \mathbf{S}_{t-1}^{(M)} + \tilde{\epsilon} \gamma^2 \mathcal{M}_B^{(q)} \circ \mathbf{S}_{t-1}^{(M)} + \mathbf{B}_0 \\
&= (\mathcal{I} - \gamma \tilde{\mathcal{T}}^{(q)}) \circ \mathbf{S}_{t-1}^{(M)} + \gamma^2 ((1 + \tilde{\epsilon}) \mathcal{M}_B^{(q)} - \widetilde{\mathcal{M}}^{(q)}) \circ \mathbf{S}_{t-1}^{(M)} + \mathbf{B}_0 \\
&\preceq (\mathcal{I} - \gamma \tilde{\mathcal{T}}^{(q)}) \circ \mathbf{S}_{t-1}^{(M)} + (1 + \tilde{\epsilon}) \gamma^2 \mathcal{M}_B^{(q)} \circ \mathbf{S}_N^{(M)} + \mathbf{B}_0.
\end{aligned}
$$

$\qquad\square$

**Lemma C.8** (A bound for $\mathcal{M}_B^{(q)} \circ \mathbf{S}_t$). *For* $1 \le t \le N$, *under Assumption 3.1, Assumption 3.2, Assumption 3.3, and Assumption 3.4, if* $\gamma < \frac{1}{\alpha_B \mathrm{tr}(\mathbf{H}^{(q)})}$, *then*

$$
\mathcal{M}_B^{(q)} \circ \mathbf{S}_t \preceq \frac{\alpha_B \cdot \mathrm{tr}\left( \left[ \mathcal{I} - (\mathcal{I} - \gamma \tilde{\mathcal{T}}^{(q)})^t \right] \circ \mathbf{B}_0 \right)}{\gamma (1 - \gamma \alpha_B \, \mathrm{tr}(\mathbf{H}^{(q)}))} \cdot \mathbf{H}^{(q)}.
$$

*Proof.* We prove by deriving a crude bound for $\mathbf{S}_t$ and applying $\mathcal{M}_B^{(q)}$ to this crude bound. Take summation via the update rule, we have

$$\mathbf{S}_t = \sum_{k=0}^{t-1} (\mathcal{I} - \gamma \mathcal{T}_B^{(q)})^k \circ \mathbf{B}_0 = \gamma^{-1} \mathcal{T}_B^{(q)^{-1}} \circ \left[ \mathcal{I} - (\mathcal{I} - \gamma \mathcal{T}_B^{(q)})^t \right] \circ \mathbf{B}_0.$$

Note that

$$\mathcal{I} - \gamma \widetilde{\mathcal{T}}^{(q)} \preceq \mathcal{I} - \gamma \mathcal{T}_B^{(q)}, \quad (\mathcal{I} - (\mathcal{I} - \gamma \mathcal{T}_B^{(q)})^t) \preceq (\mathcal{I} - (\mathcal{I} - \gamma \widetilde{\mathcal{T}}^{(q)})^t),$$

and further note that $\mathcal{T}_B^{(q)^{-1}}$ is a PSD mapping [4], and $[\mathcal{I} - (\mathcal{I} - \gamma \widetilde{\mathcal{T}}^{(q)})^t] \circ \mathbf{B}_0$ is a PSD matrix, we obtain

$$\mathbf{S}_t \preceq \gamma^{-1} \mathcal{T}_B^{(q)^{-1}} \circ (\mathcal{I} - (\mathcal{I} - \gamma \widetilde{\mathcal{T}}^{(q)})^t) \circ \mathbf{B}_0.$$

For simplicity, we denote $\mathbf{A} := (\mathcal{I} - (\mathcal{I} - \gamma \widetilde{\mathcal{T}}^{(q)})^t) \circ \mathbf{B}_0$. We then tackle $\mathcal{T}_B^{(q)^{-1}} \circ \mathbf{A}$. To be specific, we apply $\widetilde{\mathcal{T}}^{(q)}$.

$$\widetilde{\mathcal{T}}^{(q)} \circ \mathcal{T}_B^{(q)^{-1}} \circ \mathbf{A} \preceq \gamma \mathcal{M}_B^{(q)} \circ \mathcal{T}_B^{(q)^{-1}} \circ \mathbf{A} + \mathbf{A}.$$

Therefore,

$$\mathcal{T}_B^{(q)^{-1}} \circ \mathbf{A} \preceq \gamma (\widetilde{\mathcal{T}}^{(q)})^{-1} \circ \mathcal{M}_B^{(q)} \circ \mathcal{T}_B^{(q)^{-1}} \circ \mathbf{A} + (\widetilde{\mathcal{T}}^{(q)})^{-1} \circ \mathbf{A}.$$

Then we apply $\mathcal{M}_B^{(q)}$ on both sides.

$$\mathcal{M}_B^{(q)} \circ (\mathcal{T}_B^{(q)^{-1}} \circ \mathbf{A}) \preceq \mathcal{M}_B^{(q)} \circ \gamma (\widetilde{\mathcal{T}}^{(q)})^{-1} \circ \mathcal{M}_B^{(q)} \circ \mathcal{T}_B^{(q)^{-1}} \circ \mathbf{A} + \mathcal{M}_B^{(q)} \circ (\widetilde{\mathcal{T}}^{(q)})^{-1} \circ \mathbf{A}$$

$$\preceq \sum_{t=0}^{\infty} (\gamma \mathcal{M}_B^{(q)} \circ (\widetilde{\mathcal{T}}^{(q)})^{-1})^t \circ (\mathcal{M}_B^{(q)} \circ (\widetilde{\mathcal{T}}^{(q)})^{-1} \circ \mathbf{A}) \text{ (By recursion).}$$

(C.16)

By Assumption 3.3,

$$\mathcal{M}_B^{(q)} \circ (\widetilde{\mathcal{T}}^{(q)})^{-1} \circ \mathbf{A} \preceq \alpha_B \operatorname{tr}(\mathbf{H}^{(q)} (\widetilde{\mathcal{T}}^{(q)})^{-1} \circ \mathbf{A}) \mathbf{H}^{(q)}$$

$$= \alpha_B \gamma \operatorname{tr} \left( \sum_{t=0}^{\infty} \mathbf{H}^{(q)} (\mathbf{I} - \gamma \mathbf{H}^{(q)})^t \mathbf{A} (\mathbf{I} - \gamma \mathbf{H}^{(q)})^t \right) \mathbf{H}^{(q)}$$

$$= \alpha_B \operatorname{tr} \left( \mathbf{H}^{(q)} (2 \mathbf{H}^{(q)} - \gamma (\mathbf{H}^{(q)})^2)^{-1} \mathbf{A} \right) \mathbf{H}^{(q)}$$

$$\preceq \alpha_B \operatorname{tr}(\mathbf{A}) \mathbf{H}^{(q)},$$

where the first equality holds by the definition of $\widetilde{\mathcal{T}}^{(q)}$ and the last inequality requires the condition that $\gamma < \frac{1}{\alpha_B \operatorname{tr}(\mathbf{H}^{(q)})}$. Hence, by (C.16), and further by $(\widetilde{\mathcal{T}}^{(q)})^{-1} \mathbf{H}^{(q)} \preceq \mathbf{I}$ and $\mathcal{M}_B^{(q)} \circ \mathbf{I} \preceq \alpha_B \operatorname{tr}(\mathbf{H}^{(q)}) \mathbf{H}^{(q)}$, we obtain

$$\mathcal{M}_B^{(q)} \circ (\mathcal{T}_B^{(q)^{-1}} \circ \mathbf{A}) \preceq \sum_{t=0}^{\infty} (\gamma \mathcal{M}_B^{(q)} \circ (\widetilde{\mathcal{T}}^{(q)})^{-1})^t \circ (\mathcal{M}_B^{(q)} \circ (\widetilde{\mathcal{T}}^{(q)})^{-1} \circ \mathbf{A})$$

$$\preceq \alpha_B \operatorname{tr}(\mathbf{A}) \sum_{t=0}^{\infty} (\gamma \alpha_B \operatorname{tr}(\mathbf{H}^{(q)}))^t \mathbf{H}^{(q)}$$

$$\preceq \frac{\alpha_B \operatorname{tr}(\mathbf{A})}{1 - \gamma \alpha_B \operatorname{tr}(\mathbf{H}^{(q)})} \cdot \mathbf{H}^{(q)}.$$

Therefore,

$$\mathcal{M}_B^{(q)} \circ \mathbf{S}_t \preceq \gamma^{-1} \frac{\alpha_B \operatorname{tr}(\mathbf{A})}{1 - \gamma \alpha_B \operatorname{tr}(\mathbf{H}^{(q)})} \cdot \mathbf{H}^{(q)} = \frac{\alpha_B \cdot \operatorname{tr} \left( \left[ \mathcal{I} - (\mathcal{I} - \gamma \widetilde{\mathcal{T}}^{(q)t}) \right] \circ \mathbf{B}_0 \right)}{\gamma (1 - \gamma \alpha_B \operatorname{tr}(\mathbf{H}^{(q)}))} \cdot \mathbf{H}^{(q)}.$$

$\square$

---

[4] $\mathcal{T}_B^{(q)^{-1}}$ is a PSD mapping under the condition that $\gamma < \frac{1}{\alpha_B \operatorname{tr}(\mathbf{H}^{(q)})}$, which can be directly deduced by Lemma B.1 in Zou et al. (2023). We omit the proof here for simplicity.

**Lemma C.9** (A bound for $\mathcal{M}_B^{(q)} \circ \mathbf{S}_t^{(M)}$). *For $1 \leq t \leq N$, under Assumption 3.1, Assumption 3.2, Assumption 3.3, and Assumption 3.4, if $\gamma < \frac{1}{(1+\tilde{\epsilon})\alpha_B \mathrm{tr}(\mathbf{H}^{(q)})}$,*

$$\mathcal{M}_B^{(q)} \circ \mathbf{S}_t^{(M)} \preceq \frac{\alpha_B \cdot \mathrm{tr}\left(\left[\mathcal{I} - (\mathcal{I} - \gamma\widetilde{\mathcal{T}}^{(q)})^t\right] \circ \mathbf{B}_0\right)}{\gamma(1 - (1+\tilde{\epsilon})\alpha_B \mathrm{tr}(\mathbf{H}^{(q)}))} \cdot \mathbf{H}^{(q)}.$$

*Proof.* The first step is to derive a crude bound for $\mathbf{S}_t^{(M)}$. Take summation via the update rule, we have [5]

$$\mathbf{S}_t^{(M)} = \sum_{k=0}^{t-1}(\mathcal{I} - \gamma\mathcal{T}_B^{(q)} + \tilde{\epsilon}\gamma^2\mathcal{M}_B^{(q)})^k \circ \mathbf{B}_0 = \gamma^{-1}(\mathcal{T}_B^{(q)} - \tilde{\epsilon}\gamma\mathcal{M}_B^{(q)})^{-1} \circ \left[\mathcal{I} - (\mathcal{I} - \gamma\mathcal{T}_B^{(q)} + \tilde{\epsilon}\gamma^2\mathcal{M}_B^{(q)})^t\right] \circ \mathbf{B}_0.$$

Note that

$$\mathcal{I} - \gamma\widetilde{\mathcal{T}}^{(q)} \preceq \mathcal{I} - \gamma\mathcal{T}_B^{(q)}, \quad (\mathcal{I} - (\mathcal{I} - \gamma\mathcal{T}_B^{(q)} + \tilde{\epsilon}\gamma^2\mathcal{M}_B^{(q)})^t) \preceq (\mathcal{I} - (\mathcal{I} - \gamma\widetilde{\mathcal{T}}^{(q)} + \tilde{\epsilon}\gamma^2\mathcal{M}_B^{(q)})^t),$$

we obtain

$$\mathbf{S}_t^{(M)} \preceq \gamma^{-1}(\mathcal{T}_B^{(q)} - \tilde{\epsilon}\gamma\mathcal{M}_B^{(q)})^{-1} \circ (\mathcal{I} - (\mathcal{I} - \gamma\widetilde{\mathcal{T}}^{(q)} + \tilde{\epsilon}\gamma^2\mathcal{M}_B^{(q)})^t) \circ \mathbf{B}_0.$$

Denote $\mathbf{A} := (\mathcal{I} - (\mathcal{I} - \gamma\widetilde{\mathcal{T}}^{(q)} + \tilde{\epsilon}\gamma^2\mathcal{M}_B^{(q)})^t) \circ \mathbf{B}_0$, then applying $\widetilde{\mathcal{T}}^{(q)}$

$$\widetilde{\mathcal{T}}^{(q)} \circ (\mathcal{T}_B^{(q)} - \tilde{\epsilon}\gamma\mathcal{M}_B^{(q)})^{-1} \circ \mathbf{A} \preceq (1 + \tilde{\epsilon})\gamma\mathcal{M}_B^{(q)} \circ (\mathcal{T}_B^{(q)} - \tilde{\epsilon}\gamma\mathcal{M}_B^{(q)})^{-1} \circ \mathbf{A} + \mathbf{A}.$$

Therefore

$$(\mathcal{T}_B^{(q)} - \tilde{\epsilon}\gamma\mathcal{M}_B^{(q)})^{-1} \circ \mathbf{A} \preceq (1 + \tilde{\epsilon})\gamma(\widetilde{\mathcal{T}}^{(q)})^{-1} \circ \mathcal{M}_B^{(q)} \circ (\mathcal{T}_B^{(q)} - \tilde{\epsilon}\gamma\mathcal{M}_B^{(q)})^{-1} \circ \mathbf{A} + (\widetilde{\mathcal{T}}^{(q)})^{-1} \circ \mathbf{A}.$$

Then we undertake the second step, applying $\mathcal{M}_B^{(q)}$ on both sides.

$$\mathcal{M}_B^{(q)} \circ (\mathcal{T}_B^{(q)} - \tilde{\epsilon}\gamma\mathcal{M}_B^{(q)})^{-1} \circ \mathbf{A} \preceq \sum_{t=0}^{\infty}((1+\tilde{\epsilon})\gamma\mathcal{M}_B^{(q)} \circ (\widetilde{\mathcal{T}}^{(q)})^{-1})^t \circ (\mathcal{M}_B^{(q)} \circ (\widetilde{\mathcal{T}}^{(q)})^{-1} \circ \mathbf{A}). \quad \text{(C.17)}$$

By Assumption 3.3,

$$\begin{aligned}
\mathcal{M}_B^{(q)} \circ (\widetilde{\mathcal{T}}^{(q)})^{-1} \circ \mathbf{A} &\preceq \alpha_B \mathrm{tr}(\mathbf{H}^{(q)}(\widetilde{\mathcal{T}}^{(q)})^{-1} \circ \mathbf{A})\mathbf{H}^{(q)} \\
&= \alpha_B\gamma \mathrm{tr}\left(\sum_{t=0}^{\infty}\mathbf{H}^{(q)}(\mathbf{I} - \gamma\mathbf{H}^{(q)})^t\mathbf{A}(\mathbf{I} - \gamma\mathbf{H}^{(q)})^t\right)\mathbf{H}^{(q)} \\
&= \alpha_B\mathrm{tr}\left(\mathbf{H}^{(q)}(2\mathbf{H}^{(q)} - \gamma(\mathbf{H}^{(q)})^2)^{-1}\mathbf{A}\right)\mathbf{H}^{(q)} \\
&\preceq \alpha_B\mathrm{tr}(\mathbf{A})\mathbf{H}^{(q)},
\end{aligned} \quad \text{(C.18)}$$

where the last inequality requires the condition that $\gamma < \frac{1}{\alpha_B\mathrm{tr}(\mathbf{H}^{(q)})}$. Hence, by (C.17), (C.18), and further by $(\widetilde{\mathcal{T}}^{(q)})^{-1}\mathbf{H}^{(q)} \preceq \mathbf{I}$ and $\mathcal{M}_B^{(q)} \circ \mathbf{I} \preceq \alpha_B \mathrm{tr}(\mathbf{H}^{(q)})\mathbf{H}^{(q)}$, we obtain

$$\begin{aligned}
\mathcal{M}_B^{(q)} \circ ((\mathcal{T}_B^{(q)} - \tilde{\epsilon}\gamma\mathcal{M}_B^{(q)})^{-1} \circ \mathbf{A}) &\preceq \sum_{t=0}^{\infty}((1+\tilde{\epsilon})\gamma\mathcal{M}_B^{(q)} \circ (\widetilde{\mathcal{T}}^{(q)})^{-1})^t \circ (\mathcal{M}_B^{(q)} \circ (\widetilde{\mathcal{T}}^{(q)})^{-1} \circ \mathbf{A}) \\
&\preceq \alpha_B \mathrm{tr}(\mathbf{A}) \sum_{t=0}^{\infty}((1+\tilde{\epsilon})\gamma\alpha_B \mathrm{tr}(\mathbf{H}^{(q)}))^t\mathbf{H}^{(q)} \\
&\preceq \frac{\alpha_B \mathrm{tr}(\mathbf{A})}{1 - (1+\tilde{\epsilon})\gamma\alpha_B \mathrm{tr}(\mathbf{H}^{(q)})} \cdot \mathbf{H}^{(q)}.
\end{aligned}$$

Therefore,

$$\mathcal{M}_B^{(q)} \circ \mathbf{S}_t^{(M)} \preceq \gamma^{-1}\frac{\alpha_B \mathrm{tr}(\mathbf{A})}{1 - (1+\tilde{\epsilon})\gamma\alpha_B \mathrm{tr}(\mathbf{H}^{(q)})} \cdot \mathbf{H}^{(q)} \preceq \frac{\alpha_B \cdot \mathrm{tr}\left(\left[\mathcal{I} - (\mathcal{I} - \gamma\widetilde{\mathcal{T}}^{(q)})^t\right] \circ \mathbf{B}_0\right)}{\gamma(1 - (1+\tilde{\epsilon})\gamma\alpha_B \mathrm{tr}(\mathbf{H}^{(q)}))} \cdot \mathbf{H}^{(q)}.$$

$\square$

---

[5]$(\mathcal{T}_B^{(q)} - \tilde{\epsilon}\gamma\mathcal{M}_B^{(q)})^{-1}$ is a PSD mapping under the condition that $\gamma < \frac{1}{(1+\tilde{\epsilon})\alpha_B\mathrm{tr}(\mathbf{H}^{(q)})}$, which can be directly deduced by Lemma B.1 in Zou et al. (2023). We omit the proof here for simplicity.

By Lemma C.6, Lemma C.7, Lemma C.8 and Lemma C.9, we can provide a refined bound for $\mathbf{S}_t$ and $\mathbf{S}_t^{(M)}$. Then we are ready to bound the bias error.

**Lemma C.10** (A bound for bias under general quantization). *Under Assumption 3.1, Assumption 3.2, Assumption 3.3, and Assumption 3.4, if the stepsize satisfies $\gamma < \frac{1}{\alpha_B \operatorname{tr}(\mathbf{H}^{(q)})}$, then*

$$
\begin{aligned}
\text{bias} \leq & \frac{2\alpha_B \left( \|\mathbf{w}_0 - \mathbf{w}^{(q)*}\|_{\mathbf{I}_{0:k^*}^{(q)}}^2 + N\gamma\|\mathbf{w}_0 - \mathbf{w}^{(q)*}\|_{\mathbf{H}_{k^*:\infty}^{(q)}}^2 \right)}{N\gamma(1 - \gamma\alpha_B \operatorname{tr}(\mathbf{H}^{(q)}))} \cdot \left( \frac{k^*}{N} + N\gamma^2 \sum_{i>k^*} (\lambda_i^{(q)})^2 \right) \\
& + \frac{1}{\gamma^2 N^2} \cdot \|\mathbf{w}_0 - \mathbf{w}^{(q)*}\|_{(\mathbf{H}_{0:k^*}^{(q)})^{-1}}^2 + \|\mathbf{w}_0 - \mathbf{w}^{(q)*}\|_{\mathbf{H}_{k^*:\infty}^{(q)}}^2 .
\end{aligned}
$$

*Proof.* Recalling Lemma C.6, we can derive a refined upper bound for $\mathbf{S}_t$ by Lemma C.8:

$$
\begin{aligned}
\mathbf{S}_t \preceq & (\mathcal{I} - \gamma\tilde{\mathcal{T}}^{(q)}) \circ \mathbf{S}_{t-1} + \gamma^2 \mathcal{M}_B^{(q)} \circ \mathbf{S}_N + \mathbf{B}_0 \\
\preceq & (\mathcal{I} - \gamma\tilde{\mathcal{T}}^{(q)}) \circ \mathbf{S}_{t-1} + \frac{\gamma\alpha_B \cdot \operatorname{tr}\left( \left[ \mathcal{I} - (\mathcal{I} - \gamma\tilde{\mathcal{T}}^{(q)})^N \right] \circ \mathbf{B}_0 \right)}{(1 - \gamma\alpha_B \operatorname{tr}(\mathbf{H}^{(q)}))} \cdot \mathbf{H}^{(q)} + \mathbf{B}_0 \\
\preceq & \sum_{k=0}^{t-1} (\mathcal{I} - \gamma\tilde{\mathcal{T}}^{(q)})^k \left( \frac{\gamma\alpha_B \cdot \operatorname{tr}\left( \left[ \mathcal{I} - (\mathcal{I} - \gamma\tilde{\mathcal{T}}^{(q)})^N \right] \circ \mathbf{B}_0 \right)}{(1 - \gamma\alpha_B \operatorname{tr}(\mathbf{H}^{(q)}))} \cdot \mathbf{H}^{(q)} + \mathbf{B}_0 \right) \\
= & \sum_{k=0}^{t-1} (\mathbf{I} - \gamma\mathbf{H}^{(q)})^k \left( \frac{\gamma\alpha_B \cdot \operatorname{tr}\left( \mathbf{B}_0 - (\mathbf{I} - \gamma\mathbf{H}^{(q)})^N \mathbf{B}_0 (\mathbf{I} - \gamma\mathbf{H}^{(q)})^N \right)}{(1 - \gamma\alpha_B \operatorname{tr}(\mathbf{H}^{(q)}))} \cdot \mathbf{H}^{(q)} + \mathbf{B}_0 \right) (\mathbf{I} - \gamma\mathbf{H}^{(q)})^k .
\end{aligned}
$$

$$(C.19)$$

Before providing our upper bound for the bias error, we denote

$$
\mathbf{B}_{a,b} := \mathbf{B}_a - (\mathbf{I} - \gamma\mathbf{H}^{(q)})^{b-a} \mathbf{B}_a (\mathbf{I} - \gamma\mathbf{H}^{(q)})^{b-a} .
$$

Then by (C.14) and (C.19),

$$
\begin{aligned}
\text{bias} \leq & \frac{1}{\gamma N^2} \langle \mathbf{I} - (\mathbf{I} - \gamma\mathbf{H}^{(q)})^N, \sum_{t=0}^{N-1} \mathbf{B}_t \rangle \\
\leq & \frac{1}{\gamma N^2} \sum_{k=0}^{N-1} \left\langle \mathbf{I} - (\mathbf{I} - \gamma\mathbf{H}^{(q)})^N, (\mathbf{I} - \gamma\mathbf{H}^{(q)})^k \left( \frac{\gamma\alpha_B \cdot \operatorname{tr}(\mathbf{B}_{0,N})}{1 - \gamma\alpha_B \operatorname{tr}(\mathbf{H}^{(q)})} \cdot \mathbf{H}^{(q)} + \mathbf{B}_0 \right) (\mathbf{I} - \gamma\mathbf{H}^{(q)})^k \right\rangle \\
= & \frac{1}{\gamma N^2} \sum_{k=0}^{N-1} \left\langle (\mathbf{I} - \gamma\mathbf{H}^{(q)})^{2k} - (\mathbf{I} - \gamma\mathbf{H}^{(q)})^{N+2k}, \left( \frac{\gamma\alpha_B \cdot \operatorname{tr}(\mathbf{B}_{0,N})}{1 - \gamma\alpha_B \operatorname{tr}(\mathbf{H}^{(q)})} \cdot \mathbf{H}^{(q)} + \mathbf{B}_0 \right) \right\rangle .
\end{aligned}
$$

Note that

$$
\begin{aligned}
(\mathbf{I} - \gamma\mathbf{H}^{(q)})^{2k} - (\mathbf{I} - \gamma\mathbf{H}^{(q)})^{N+2k} = & \left( \mathbf{I} - \gamma\mathbf{H}^{(q)} \right)^k \left( \left( \mathbf{I} - \gamma\mathbf{H}^{(q)} \right)^k - \left( \mathbf{I} - \gamma\mathbf{H}^{(q)} \right)^{N+k} \right) \\
& \preceq (\mathbf{I} - \gamma\mathbf{H}^{(q)})^k - (\mathbf{I} - \gamma\mathbf{H}^{(q)})^{N+k},
\end{aligned}
$$

we obtain

$$
\text{bias} \leq \frac{1}{\gamma N^2} \sum_{k=0}^{N-1} \left\langle (\mathbf{I} - \gamma\mathbf{H}^{(q)})^k - (\mathbf{I} - \gamma\mathbf{H}^{(q)})^{N+k}, \frac{\gamma\alpha_B \cdot \operatorname{tr}(\mathbf{B}_{0,N})}{1 - \gamma\alpha_B \operatorname{tr}(\mathbf{H}^{(q)})} \cdot \mathbf{H}^{(q)} + \mathbf{B}_0 \right\rangle .
$$

Therefore, it suffices to upper bound the following two terms

$$
I_1 = \frac{\alpha_B \operatorname{tr}(\mathbf{B}_{0,N})}{N^2(1 - \gamma\alpha \operatorname{tr}(\mathbf{H}^{(q)}))} \sum_{k=0}^{N-1} \left\langle (\mathbf{I} - \gamma\mathbf{H}^{(q)})^k - (\mathbf{I} - \gamma\mathbf{H}^{(q)})^{N+k}, \mathbf{H}^{(q)} \right\rangle ,
$$

$$
I_2 = \frac{1}{\gamma N^2} \sum_{k=0}^{N-1} \left\langle (\mathbf{I} - \gamma\mathbf{H}^{(q)})^k - (\mathbf{I} - \gamma\mathbf{H}^{(q)})^{N+k}, \mathbf{B}_0 \right\rangle .
$$

Regarding $I_1$, since $\mathbf{H}^{(q)}$ and $\mathbf{I} - \gamma\mathbf{H}^{(q)}$ can be diagonalized simultaneously,

$$
\begin{aligned}
I_1 &= \frac{\alpha_B \operatorname{tr}(\mathbf{B}_{0,N})}{N^2(1 - \gamma\alpha_B \operatorname{tr}(\mathbf{H}^{(q)}))} \sum_{k=0}^{N-1} \sum_i \left[ (1 - \gamma\lambda_i^{(q)})^k - (1 - \gamma\lambda_i^{(q)})^{N+k} \right] \lambda_i^{(q)} \\
&= \frac{\alpha_B \operatorname{tr}(\mathbf{B}_{0,N})}{\gamma N^2(1 - \gamma\alpha_B \operatorname{tr}(\mathbf{H}^{(q)}))} \sum_i \left[ 1 - (1 - \gamma\lambda_i^{(q)})^N \right]^2 \\
&\leq \frac{\alpha_B \operatorname{tr}(\mathbf{B}_{0,N})}{\gamma N^2(1 - \gamma\alpha_B \operatorname{tr}(\mathbf{H}^{(q)}))} \sum_i \min\left\{ 1, \gamma^2 N^2(\lambda_i^{(q)})^2 \right\} \\
&\leq \frac{\alpha_B \operatorname{tr}(\mathbf{B}_{0,N})}{\gamma(1 - \gamma\alpha_B \operatorname{tr}(\mathbf{H}^{(q)}))} \cdot \left( \frac{k^*}{N^2} + \gamma^2 \sum_{i > k^*} (\lambda_i^{(q)})^2 \right),
\end{aligned}
$$

where $k^* = \max\{k : \lambda_k^{(q)} \geq \frac{1}{N\gamma}\}$. Then we tackle $\operatorname{tr}(\mathbf{B}_{0,N})$.

$$
\begin{aligned}
\operatorname{tr}(\mathbf{B}_{0,N}) &= \operatorname{tr}\left( \mathbf{B}_0 - (\mathbf{I} - \gamma\mathbf{H}^{(q)})^N \mathbf{B}_0 (\mathbf{I} - \gamma\mathbf{H}^{(q)})^N \right) \\
&= \sum_i \left( 1 - (1 - \gamma\lambda_i^{(q)})^{2N} \right) \cdot \left( \langle \mathbf{w}_0 - \mathbf{w}^{(q)*}, \mathbf{v}_i^{(q)} \rangle \right)^2 \\
&\leq 2 \sum_i \min\{1, N\gamma\lambda_i^{(q)}\} \left( \langle \mathbf{w}_0 - \mathbf{w}^{(q)*}, \mathbf{v}_i^{(q)} \rangle \right)^2 \\
&\leq 2 \left( \|\mathbf{w}_0 - \mathbf{w}^{(q)*}\|_{\mathbf{I}_{0:k^*}}^2 + N\gamma \|\mathbf{w}_0 - \mathbf{w}^{(q)*}\|_{\mathbf{H}_{k^*:\infty}}^2 \right).
\end{aligned}
\tag{C.20}
$$

Hence,

$$
I_1 \leq \frac{2\alpha_B \left( \|\mathbf{w}_0 - \mathbf{w}^{(q)*}\|_{\mathbf{I}_{0:k^*}^{(q)}}^2 + N\gamma \|\mathbf{w}_0 - \mathbf{w}^{(q)*}\|_{\mathbf{H}_{k^*:\infty}^{(q)}}^2 \right)}{N\gamma(1 - \gamma\alpha_B \operatorname{tr}(\mathbf{H}^{(q)}))} \cdot \left( \frac{k^*}{N} + N\gamma^2 \sum_{i > k^*} (\lambda_i^{(q)})^2 \right).
$$

Regarding $I_2$, decompose $\mathbf{H}^{(q)} = \mathbf{V}^{(q)} \boldsymbol{\Lambda}^{(q)} \mathbf{V}^{(q)\top}$, then

$$
I_2 = \frac{1}{\gamma N^2} \sum_{k=0}^{N-1} \langle (\mathbf{I} - \gamma\boldsymbol{\Lambda}^{(q)})^k - (\mathbf{I} - \gamma\boldsymbol{\Lambda}^{(q)})^{N+k}, \mathbf{V}^{(q)\top} \mathbf{B}_0 \mathbf{V}^{(q)} \rangle.
$$

Note that $\mathbf{B}_0 = \boldsymbol{\eta}_0 \boldsymbol{\eta}_0^\top$, it can be shown that the diagonal entries of $\mathbf{V}^{(q)\top} \mathbf{B}_0 \mathbf{V}^{(q)}$ are $\omega_1^2, \dots,$ where $\omega_i = \mathbf{v}_i^{(q)\top} \boldsymbol{\eta}_0 = \mathbf{v}_i^{(q)\top} (\mathbf{w}_0 - \mathbf{w}^{(q)*})$. Hence,

$$
\begin{aligned}
I_2 &= \frac{1}{\gamma N^2} \sum_{k=0}^{N-1} \sum_i \left[ (1 - \gamma\lambda_i^{(q)})^k - (1 - \gamma\lambda_i^{(q)})^{N+k} \right] \omega_i^2 \\
&= \frac{1}{\gamma^2 N^2} \sum_i \frac{\omega_i^2}{\lambda_i^{(q)}} \left[ 1 - (1 - \gamma\lambda_i^{(q)})^N \right]^2 \\
&\leq \frac{1}{\gamma^2 N^2} \sum_i \frac{\omega_i^2}{\lambda_i^{(q)}} \min\left\{ 1, \gamma^2 N^2(\lambda_i^{(q)})^2 \right\} \\
&\leq \frac{1}{\gamma^2 N^2} \cdot \sum_{i \leq k^*} \frac{\omega_i^2}{\lambda_i^{(q)}} + \sum_{i > k^*} \lambda_i^{(q)} \omega_i^2 \\
&= \frac{1}{\gamma^2 N^2} \cdot \|\mathbf{w}_0 - \mathbf{w}^{(q)*}\|_{(\mathbf{H}_{0:k^*}^{(q)})^{-1}}^2 + \|\mathbf{w}_0 - \mathbf{w}^{(q)*}\|_{\mathbf{H}_{k^*:\infty}^{(q)}}^2.
\end{aligned}
$$

In conclusion, if the stepsize satisfies $\gamma < \frac{1}{\alpha_B \operatorname{tr}(\mathbf{H}^{(q)})}$,

$$
\begin{aligned}
\text{bias} \leq & I_1 + I_2 \\
\leq & \frac{2\alpha_B \left( \|\mathbf{w}_0 - \mathbf{w}^{(q)^*}\|_{\mathbf{I}_{0:k^*}^{(q)}}^2 + N\gamma\|\mathbf{w}_0 - \mathbf{w}^{(q)^*}\|_{\mathbf{H}_{k^*:\infty}^{(q)}}^2 \right)}{N\gamma(1 - \gamma\alpha_B \operatorname{tr}(\mathbf{H}^{(q)}))} \cdot \left( \frac{k^*}{N} + N\gamma^2 \sum_{i > k^*} (\lambda_i^{(q)})^2 \right) \\
& + \frac{1}{\gamma^2 N^2} \cdot \|\mathbf{w}_0 - \mathbf{w}^{(q)^*}\|_{(\mathbf{H}_{0:k^*}^{(q)})^{-1}}^2 + \|\mathbf{w}_0 - \mathbf{w}^{(q)^*}\|_{\mathbf{H}_{k^*:\infty}^{(q)}}^2 \,.
\end{aligned}
$$

$\square$

**Lemma C.11** (A bound for bias under multiplicative quantization). *Under Assumption 3.1, Assumption 3.2, Assumption 3.3, and Assumption 3.4, if the stepsize satisfies $\gamma < \frac{1}{(1+\tilde{\epsilon})\alpha_B \operatorname{tr}(\mathbf{H}^{(q)})}$, if there exist $\epsilon_d, \epsilon_l, \epsilon_p, \epsilon_a$ and $\epsilon_o$ such that for any $i \in \{d, l, p, a, o\}$, quantization $\mathcal{Q}_i$ is $\epsilon_i$-multiplicative, then*

$$
\begin{aligned}
\text{bias} \leq & \frac{2(1+\tilde{\epsilon})\alpha_B \left( \|\mathbf{w}_0 - \mathbf{w}^{(q)^*}\|_{\mathbf{I}_{0:k^*}^{(q)}}^2 + N\gamma\|\mathbf{w}_0 - \mathbf{w}^{(q)^*}\|_{\mathbf{H}_{k^*:\infty}^{(q)}}^2 \right)}{N\gamma(1 - (1+\tilde{\epsilon})\gamma\alpha_B \operatorname{tr}(\mathbf{H}^{(q)}))} \cdot \left( \frac{k^*}{N} + N\gamma^2 \sum_{i > k^*} (\lambda_i^{(q)})^2 \right) \\
& + \frac{1}{\gamma^2 N^2} \cdot \|\mathbf{w}_0 - \mathbf{w}^{(q)^*}\|_{(\mathbf{H}_{0:k^*}^{(q)})^{-1}}^2 + \|\mathbf{w}_0 - \mathbf{w}^{(q)^*}\|_{\mathbf{H}_{k^*:\infty}^{(q)}}^2 \,.
\end{aligned}
$$

*Proof.* Recalling Lemma C.7, we can derive an upper bound for $\mathbf{S}_t$ by Lemma C.9:

$$
\begin{aligned}
\mathbf{S}_t \preceq & (\mathcal{I} - \gamma\tilde{\mathcal{T}}^{(q)}) \circ \mathbf{S}_{t-1} + (1+\tilde{\epsilon})\gamma^2 \mathcal{M}_B^{(q)} \circ \mathbf{S}_N + \mathbf{B}_0 \\
\preceq & (\mathcal{I} - \gamma\tilde{\mathcal{T}}^{(q)}) \circ \mathbf{S}_{t-1} + \frac{(1+\tilde{\epsilon})\gamma\alpha_B \cdot \operatorname{tr}\left( \left[ \mathcal{I} - (\mathcal{I} - \gamma\tilde{\mathcal{T}}^{(q)})^N \right] \circ \mathbf{B}_0 \right)}{(1 - (1+\tilde{\epsilon})\gamma\alpha_B \operatorname{tr}(\mathbf{H}^{(q)}))} \cdot \mathbf{H}^{(q)} + \mathbf{B}_0 \\
\preceq & \sum_{k=0}^{t-1} (\mathcal{I} - \gamma\tilde{\mathcal{T}}^{(q)})^k \left( \frac{(1+\tilde{\epsilon})\gamma\alpha_B \cdot \operatorname{tr}\left( \left[ \mathcal{I} - (\mathcal{I} - \gamma\tilde{\mathcal{T}}^{(q)})^N \right] \circ \mathbf{B}_0 \right)}{(1 - (1+\tilde{\epsilon})\gamma\alpha_B \operatorname{tr}(\mathbf{H}^{(q)}))} \cdot \mathbf{H}^{(q)} + \mathbf{B}_0 \right) \\
= & \sum_{k=0}^{t-1} (\mathbf{I} - \gamma\mathbf{H}^{(q)})^k \left( \frac{(1+\tilde{\epsilon})\gamma\alpha_B \cdot \operatorname{tr}\left( \mathbf{B}_0 - (\mathbf{I} - \gamma\mathbf{H}^{(q)})^N \mathbf{B}_0 (\mathbf{I} - \gamma\mathbf{H}^{(q)})^N \right)}{(1 - (1+\tilde{\epsilon})\gamma\alpha_B \operatorname{tr}(\mathbf{H}^{(q)}))} \cdot \mathbf{H}^{(q)} + \mathbf{B}_0 \right) (\mathbf{I} - \gamma\mathbf{H}^{(q)})^k .
\end{aligned}
$$

Repeat the same computation in the proof of Lemma C.10, we obtain

$$
\text{bias} \leq \frac{1}{\gamma N^2} \sum_{k=0}^{N-1} \left\langle (\mathbf{I} - \gamma\mathbf{H}^{(q)})^k - (\mathbf{I} - \gamma\mathbf{H}^{(q)})^{N+k}, \frac{(1+\tilde{\epsilon})\gamma\alpha_B \cdot \operatorname{tr}(\mathbf{B}_{0,N})}{1 - (1+\tilde{\epsilon})\gamma\alpha_B \operatorname{tr}(\mathbf{H}^{(q)})} \cdot \mathbf{H}^{(q)} + \mathbf{B}_0 \right\rangle .
$$

(C.21)

Therefore, it suffices to upper bound the following two terms

$$
\begin{aligned}
I_1 &= \frac{(1+\tilde{\epsilon})\alpha_B \operatorname{tr}(\mathbf{B}_{0,N})}{N^2(1 - (1+\tilde{\epsilon})\gamma\alpha \operatorname{tr}(\mathbf{H}^{(q)}))} \sum_{k=0}^{N-1} \left\langle (\mathbf{I} - \gamma\mathbf{H}^{(q)})^k - (\mathbf{I} - \gamma\mathbf{H}^{(q)})^{N+k}, \mathbf{H}^{(q)} \right\rangle , \\
I_2 &= \frac{1}{\gamma N^2} \sum_{k=0}^{N-1} \left\langle (\mathbf{I} - \gamma\mathbf{H}^{(q)})^k - (\mathbf{I} - \gamma\mathbf{H}^{(q)})^{N+k}, \mathbf{B}_0 \right\rangle .
\end{aligned}
$$

Repeating the computation in the proof of Lemma C.10,

$$
I_1 \leq \frac{2(1+\tilde{\epsilon})\alpha_B \left( \|\mathbf{w}_0 - \mathbf{w}^{(q)^*}\|_{\mathbf{I}_{0:k^*}^{(q)}}^2 + N\gamma\|\mathbf{w}_0 - \mathbf{w}^{(q)^*}\|_{\mathbf{H}_{k^*:\infty}^{(q)}}^2 \right)}{N\gamma(1 - (1+\tilde{\epsilon})\gamma\alpha_B \operatorname{tr}(\mathbf{H}^{(q)}))} \cdot \left( \frac{k^*}{N} + N\gamma^2 \sum_{i > k^*} (\lambda_i^{(q)})^2 \right) .
$$

$$
I_2 \leq \frac{1}{\gamma^2 N^2} \cdot \|\mathbf{w}_0 - \mathbf{w}^{(q)^*}\|_{(\mathbf{H}_{0:k^*}^{(q)})^{-1}}^2 + \|\mathbf{w}_0 - \mathbf{w}^{(q)^*}\|_{\mathbf{H}_{k^*:\infty}^{(q)}}^2 .
$$

In conclusion, if the stepsize satisfies $\gamma < \frac{1}{(1+\tilde{\epsilon})\alpha_B \mathrm{tr}(\mathbf{H}^{(q)})}$,

$$
\begin{aligned}
\text{bias} \leq &\frac{2(1+\tilde{\epsilon})\alpha_B \left( \|\mathbf{w}_0 - \mathbf{w}^{(q)*}\|^2_{\mathbf{I}^{(q)}_{0:k^*}} + N\gamma\|\mathbf{w}_0 - \mathbf{w}^{(q)*}\|^2_{\mathbf{H}^{(q)}_{k^*:\infty}} \right)}{N\gamma(1 - (1+\tilde{\epsilon})\gamma\alpha_B \mathrm{tr}(\mathbf{H}^{(q)}))} \cdot \left( \frac{k^*}{N} + N\gamma^2 \sum_{i>k^*}(\lambda_i^{(q)})^2 \right) \\
&+ \frac{1}{\gamma^2 N^2} \cdot \|\mathbf{w}_0 - \mathbf{w}^{(q)*}\|^2_{(\mathbf{H}^{(q)}_{0:k^*})^{-1}} + \|\mathbf{w}_0 - \mathbf{w}^{(q)*}\|^2_{\mathbf{H}^{(q)}_{k^*:\infty}}.
\end{aligned}
$$

$\square$

## C.5 BOUNDING THE VARIANCE ERROR

Recalling Lemma C.4 and Lemma C.5, the key part of bounding the variance error is to derive an upper bound for $\mathbf{C}_t$ and $\mathbf{C}_t^{(M)}$, where

$$
\mathbf{C}_t := (\mathcal{I} - \gamma\mathcal{T}_B^{(q)})\mathbf{C}_{t-1} + \gamma^2\sigma_G^{(q)2}\mathbf{H}^{(q)}, \quad \mathbf{C}_0 = \mathbf{0}.
$$

$$
\mathbf{C}_t^{(M)} := (\mathcal{I} - \gamma\mathcal{T}_B^{(q)} + \tilde{\epsilon}\gamma^2\mathcal{M}_B^{(q)})\mathbf{C}_{t-1}^{(M)} + \gamma^2\sigma_M^{(q)2}\mathbf{H}^{(q)}, \quad \mathbf{C}_0^{(M)} = 0.
$$

We first estimate $\mathbf{C}_t$ by converting $\mathcal{T}_B^{(q)}$ to $\widetilde{\mathcal{T}}^{(q)}$.

$$
\begin{aligned}
\mathbf{C}_t =&(\mathcal{I} - \gamma\mathcal{T}_B^{(q)}) \circ \mathbf{C}_{t-1} + \gamma^2\sigma_G^{(q)2}\mathbf{H}^{(q)} \\
=&(\mathcal{I} - \gamma\widetilde{\mathcal{T}}^{(q)}) \circ \mathbf{C}_{t-1} + \gamma(\widetilde{\mathcal{T}}^{(q)} - \mathcal{T}_B^{(q)}) \circ \mathbf{C}_{t-1} + \gamma^2\sigma_G^{(q)2}\mathbf{H}^{(q)} \\
=&(\mathcal{I} - \gamma\widetilde{\mathcal{T}}^{(q)}) \circ \mathbf{C}_{t-1} + \gamma^2(\mathcal{M}_B^{(q)} - \widetilde{\mathcal{M}}^{(q)}) \circ \mathbf{C}_{t-1} + \gamma^2\sigma_G^{(q)2}\mathbf{H}^{(q)} \\
\preceq&(\mathcal{I} - \gamma\widetilde{\mathcal{T}}^{(q)}) \circ \mathbf{C}_{t-1} + \gamma^2\mathcal{M}_B^{(q)} \circ \mathbf{C}_{t-1} + \gamma^2\sigma_G^{(q)2}\mathbf{H}^{(q)}.
\end{aligned} \tag{C.22}
$$

Similarly,

$$
\begin{aligned}
\mathbf{C}_t^{(M)} =&(\mathcal{I} - \gamma\mathcal{T}_B^{(q)} + \tilde{\epsilon}\gamma^2\mathcal{M}_B^{(q)})\mathbf{C}_{t-1}^{(M)} + \gamma^2\sigma_M^{(q)2}\mathbf{H}^{(q)} \\
=&(\mathcal{I} - \gamma\widetilde{\mathcal{T}}^{(q)}) \circ \mathbf{C}_{t-1}^{(M)} + \gamma(\widetilde{\mathcal{T}}^{(q)} - \mathcal{T}_B^{(q)} + \tilde{\epsilon}\gamma\mathcal{M}_B^{(q)}) \circ \mathbf{C}_{t-1}^{(M)} + \gamma^2\sigma_M^{(q)2}\mathbf{H}^{(q)} \\
=&(\mathcal{I} - \gamma\widetilde{\mathcal{T}}^{(q)}) \circ \mathbf{C}_{t-1}^{(M)} + \gamma^2(\mathcal{M}_B^{(q)} - \widetilde{\mathcal{M}}^{(q)} + \tilde{\epsilon}\gamma\mathcal{M}_B^{(q)}) \circ \mathbf{C}_{t-1}^{(M)} + \gamma^2\sigma_M^{(q)2}\mathbf{H}^{(q)} \\
\preceq&(\mathcal{I} - \gamma\widetilde{\mathcal{T}}^{(q)}) \circ \mathbf{C}_{t-1}^{(M)} + \gamma^2(1 + \tilde{\epsilon})\mathcal{M}_B^{(q)} \circ \mathbf{C}_{t-1}^{(M)} + \gamma^2\sigma_M^{(q)2}\mathbf{H}^{(q)}.
\end{aligned} \tag{C.23}
$$

The following two lemmas provide upper bounds for $\mathcal{M}_B^{(q)} \circ \mathbf{C}_t$ and $\mathcal{M}_B^{(q)} \circ \mathbf{C}_t^{(M)}$.

**Lemma C.12** (A bound for $\mathcal{M}_B^{(q)} \circ \mathbf{C}_t$). *For $t \geq 1$, under Assumption 3.1, Assumption 3.2, Assumption 3.3, and Assumption 3.4, if the stepsize $\gamma \leq \frac{1}{\alpha_B \mathrm{tr}(\mathbf{H}^{(q)})}$, then*

$$
\mathcal{M}_B^{(q)} \circ \mathbf{C}_t \preceq \frac{\alpha_B \mathrm{tr}(\mathbf{H}^{(q)})\gamma\sigma_G^{(q)2}}{1 - \gamma\alpha_B \mathrm{tr}(\mathbf{H}^{(q)})}\mathbf{H}^{(q)}.
$$

*Proof.* The main goal is to derive a crude upper bound for $\mathbf{C}_t$. Denote $\mathbf{\Sigma} = \sigma_G^{(q)2}\mathbf{H}^{(q)}$.

**Step 1: $\mathbf{C}_t$ is increasing.** By definition,

$$
\begin{aligned}
\mathbf{C}_t &= (\mathcal{I} - \gamma\mathcal{T}_B^{(q)}) \circ \mathbf{C}_{t-1} + \gamma^2\mathbf{\Sigma} \\
&= \gamma^2 \sum_{k=0}^{t-1}(\mathcal{I} - \gamma\mathcal{T}_B^{(q)})^k \circ \mathbf{\Sigma} \quad \text{(solving the recursion)} \\
&= \mathbf{C}_{t-1} + \gamma^2(\mathcal{I} - \gamma\mathcal{T}_B^{(q)})^{t-1} \circ \mathbf{\Sigma} \\
&\succeq \mathbf{C}_{t-1}. \quad \text{(since $\mathcal{I} - \gamma\mathcal{T}_B^{(q)}$ is a PSD mapping).}
\end{aligned}
$$

**Step 2: $\mathbf{C}_\infty$ exists.** It suffices to show that $\mathrm{tr}(\mathbf{C}_t)$ is uniformly upper bounded. To be specific, for any $t \geq 1$,

$$\mathbf{C}_t = \gamma^2 \sum_{k=0}^{t-1} (\mathcal{I} - \gamma \mathcal{T}_B^{(q)})^k \circ \mathbf{\Sigma} \preceq \gamma^2 \sum_{t=0}^{\infty} (\mathcal{I} - \gamma \mathcal{T}_B^{(q)})^t \circ \mathbf{\Sigma}.$$

Then

$$\mathrm{tr}(\mathbf{C}_t) \leq \gamma^2 \sum_{t=0}^{\infty} \mathrm{tr}\left( (\mathcal{I} - \gamma \mathcal{T}_B^{(q)})^t \circ \mathbf{\Sigma} \right) := \gamma^2 \sum_{t=0}^{\infty} \mathrm{tr}(\mathbf{E}_t) \leq \frac{\gamma \, \mathrm{tr}(\mathbf{\Sigma})}{\lambda_d^{(q)}} < \infty,$$

where the second inequality holds by the iteration:

$$\mathrm{tr}(\mathbf{E}_t) = \mathrm{tr}(\mathbf{E}_{t-1}) - 2\gamma \mathrm{tr}(\mathbf{H}^{(q)} \mathbf{E}_{t-1}) + \gamma^2 \mathrm{tr}\left( \mathbf{E}_{t-1} \mathbb{E}\left[ \frac{1}{B^2} \mathbf{X}^{(q)\top} \mathbf{X}^{(q)} \mathbf{X}^{(q)\top} \mathbf{X}^{(q)} \right] \right)$$

$$\leq \mathrm{tr}(\mathbf{E}_{t-1}) - (2\gamma - \gamma^2 \alpha_B \mathrm{tr}(\mathbf{H}^{(q)})) \, \mathrm{tr}(\mathbf{H}^{(q)} \mathbf{E}_{t-1})$$

$$\leq \mathrm{tr}\left( (\mathbf{I} - \gamma \mathbf{H}^{(q)}) \mathbf{E}_{t-1} \right)$$

$$\leq (1 - \gamma \lambda_d^{(q)}) \, \mathrm{tr}(\mathbf{E}_{t-1}),$$

where the first inequality holds by Assumption 3.3 and the second inequality holds if $\gamma \leq \frac{1}{\alpha_B \mathrm{tr}(\mathbf{H}^{(q)})}$.

**Step 3: upper bound $\mathbf{C}_\infty$.** By the update rule for $\mathbf{C}_t$,

$$\mathbf{C}_\infty = (\mathcal{I} - \gamma \mathcal{T}_B^{(q)}) \circ \mathbf{C}_\infty + \gamma^2 \mathbf{\Sigma},$$

which immediately implies

$$\mathbf{C}_\infty = \gamma \mathcal{T}_B^{(q)-1} \circ \mathbf{\Sigma}. \tag{C.24}$$

We provide the upper bound by applying $\widetilde{\mathcal{T}}^{(q)}$.

$$\widetilde{\mathcal{T}}^{(q)} \circ \mathbf{C}_\infty = \mathcal{T}_B^{(q)} \circ \mathbf{C}_\infty + \gamma \mathcal{M}_B^{(q)} \circ \mathbf{C}_\infty - \gamma \widetilde{\mathcal{M}}^{(q)} \circ \mathbf{C}_\infty$$

$$= \gamma \mathbf{\Sigma} + \gamma \mathcal{M}_B^{(q)} \circ \mathbf{C}_\infty - \gamma \widetilde{\mathcal{M}}^{(q)} \circ \mathbf{C}_\infty$$

$$\preceq \gamma \mathbf{\Sigma} + \gamma \mathcal{M}_B^{(q)} \circ \mathbf{C}_\infty,$$

where the first equality holds by the definition of $\mathcal{T}_B^{(q)}$ and $\widetilde{\mathcal{T}}^{(q)}$ and the second equality holds by (C.24). Hence,

$$\widetilde{\mathcal{T}}^{(q)} \circ \mathbf{C}_\infty \preceq \gamma \sigma_G^{(q)2} \mathbf{H}^{(q)} + \gamma \mathcal{M}_B^{(q)} \circ \mathbf{C}_\infty.$$

Therefore, by applying $(\widetilde{\mathcal{T}}^{(q)})^{-1}$ we have

$$\mathbf{C}_\infty \preceq \gamma \sigma_G^{(q)2} \cdot (\widetilde{\mathcal{T}}^{(q)})^{-1} \circ \mathbf{H}^{(q)} + \gamma (\widetilde{\mathcal{T}}^{(q)})^{-1} \circ \mathcal{M}_B^{(q)} \circ \mathbf{C}_\infty$$

$$\preceq \gamma \sigma_G^{(q)2} \cdot \sum_{t=0}^{\infty} \left( \gamma (\widetilde{\mathcal{T}}^{(q)})^{-1} \circ \mathcal{M}_B^{(q)} \right)^t \circ (\widetilde{\mathcal{T}}^{(q)})^{-1} \circ \mathbf{H}^{(q)}. \quad \text{(solving the recursion)} \tag{C.25}$$

We first deal with $(\widetilde{\mathcal{T}}^{(q)})^{-1} \circ \mathbf{H}^{(q)}$.

$$(\widetilde{\mathcal{T}}^{(q)})^{-1} \circ \mathbf{H}^{(q)} = \gamma \sum_{t=0}^{\infty} (\mathcal{I} - \gamma \widetilde{\mathcal{T}}^{(q)})^t \circ \mathbf{H}^{(q)}$$

$$= \gamma \sum_{t=0}^{\infty} (\mathbf{I} - \gamma \mathbf{H}^{(q)})^t \mathbf{H}^{(q)} (\mathbf{I} - \gamma \mathbf{H}^{(q)})^t \tag{C.26}$$

$$\preceq \gamma \sum_{t=0}^{\infty} (\mathbf{I} - \gamma \mathbf{H}^{(q)})^t \mathbf{H}^{(q)}$$

$$= \mathbf{I},$$

where the second equality uses the definition of $\widetilde{\mathcal{T}}^{(q)}$. Hence, by (C.25) and (C.26),

$$
\begin{aligned}
\mathbf{C}_\infty &\preceq \gamma \sigma_G^{(q)2} \cdot \sum_{t=0}^\infty (\gamma (\widetilde{\mathcal{T}}^{(q)})^{-1} \circ \mathcal{M}_B^{(q)})^t \circ \mathbf{I} \\
&= \gamma \sigma_G^{(q)2} \cdot \sum_{t=0}^\infty (\gamma (\widetilde{\mathcal{T}}^{(q)})^{-1} \circ \mathcal{M}_B^{(q)})^{t-1} \gamma (\widetilde{\mathcal{T}}^{(q)})^{-1} \circ \mathcal{M}_B^{(q)} \circ \mathbf{I} \\
&\preceq \gamma \sigma_G^{(q)2} \cdot \sum_{t=0}^\infty (\gamma (\widetilde{\mathcal{T}}^{(q)})^{-1} \circ \mathcal{M}_B^{(q)})^{t-1} \circ \gamma \alpha_B \mathrm{tr}(\mathbf{H}^{(q)}) \mathbf{I} & \text{(C.27)} \\
&\preceq \gamma \sigma_G^{(q)2} \cdot \sum_{t=0}^\infty \left( \gamma \alpha_B \mathrm{tr}(\mathbf{H}^{(q)}) \right)^t \mathbf{I} \\
&= \frac{\gamma \sigma_G^{(q)2}}{1 - \gamma \alpha_B \mathrm{tr}(\mathbf{H}^{(q)})} \mathbf{I},
\end{aligned}
$$

where the second inequality holds by the fact that $\mathcal{M}_B^{(q)} \circ \mathbf{I} \preceq \alpha_B \mathrm{tr}(\mathbf{H}^{(q)}) \mathbf{H}^{(q)}$.

Here we complete deriving a crude upper bound for $\mathbf{C}_t$:

$$
\mathbf{C}_t \preceq \mathbf{C}_\infty \preceq \frac{\gamma \sigma_G^{(q)2}}{1 - \gamma \alpha_B \mathrm{tr}(\mathbf{H}^{(q)})} \mathbf{I}.
$$

Then by $\mathcal{M}_B^{(q)} \circ \mathbf{I} \preceq \alpha_B \mathrm{tr}(\mathbf{H}^{(q)}) \mathbf{H}^{(q)}$ again,

$$
\mathcal{M}_B^{(q)} \circ \mathbf{C}_t \preceq \frac{\alpha_B \mathrm{tr}(\mathbf{H}^{(q)}) \gamma \sigma_G^{(q)2}}{1 - \gamma \alpha_B \mathrm{tr}(\mathbf{H}^{(q)})} \mathbf{H}^{(q)}.
$$

$\square$

**Lemma C.13** (A bound for $\mathcal{M}_B^{(q)} \circ \mathbf{C}_t^{(M)}$). *For $t \geq 1$, under Assumption 3.1, Assumption 3.2, Assumption 3.3, and Assumption 3.4, if the stepsize $\gamma \leq \frac{1}{(1+\tilde{\epsilon})\alpha_B \mathrm{tr}(\mathbf{H}^{(q)})}$, then*

$$
\mathcal{M}_B^{(q)} \circ \mathbf{C}_t^{(M)} \preceq \frac{\alpha_B \mathrm{tr}(\mathbf{H}^{(q)}) \gamma \sigma_M^{(q)2}}{1 - (1+\tilde{\epsilon})\gamma \alpha_B \mathrm{tr}(\mathbf{H}^{(q)})} \mathbf{H}^{(q)}.
$$

*Proof.* The proof idea is similar to the proof of Lemma C.12 while the main goal is to derive a crude upper bound for $\mathbf{C}_t^{(M)}$. We deduce from the proof of Lemma C.12 that [6]

$$
\mathbf{C}_t^{(M)} \preceq \mathbf{C}_\infty^{(M)} = \gamma (\mathcal{T}_B^{(q)} - \tilde{\epsilon} \gamma \mathcal{M}_B^{(q)})^{-1} \circ \sigma_M^{(q)2} \mathbf{H}^{(q)}. \tag{C.28}
$$

We provide the upper bound for $\mathbf{C}_\infty^{(M)}$ by applying $\widetilde{\mathcal{T}}^{(q)}$.

$$
\begin{aligned}
\widetilde{\mathcal{T}}^{(q)} \circ \mathbf{C}_\infty^{(M)} &= (\mathcal{T}_B^{(q)} - \tilde{\epsilon} \gamma \mathcal{M}_B^{(q)}) \circ \mathbf{C}_\infty^{(M)} + (1+\tilde{\epsilon})\gamma \mathcal{M}_B^{(q)} \circ \mathbf{C}_\infty^{(M)} - \gamma \widetilde{\mathcal{M}}^{(q)} \circ \mathbf{C}_\infty^{(M)} \\
&= \gamma \sigma_M^{(q)2} \mathbf{H}^{(q)} + (1+\tilde{\epsilon})\gamma \mathcal{M}_B^{(q)} \circ \mathbf{C}_\infty^{(M)} - \gamma \widetilde{\mathcal{M}}^{(q)} \circ \mathbf{C}_\infty^{(M)} \\
&\preceq \gamma \sigma_M^{(q)2} \mathbf{H}^{(q)} + (1+\tilde{\epsilon})\gamma \mathcal{M}_B^{(q)} \circ \mathbf{C}_\infty^{(M)},
\end{aligned}
$$

where the first equality holds by the definition of $\widetilde{\mathcal{T}}^{(q)}$ and the second equality holds by the definition of $\mathbf{C}_\infty^{(M)}$ (C.28). Therefore, applying $(\widetilde{\mathcal{T}}^{(q)})^{-1}$ we have

$$
\begin{aligned}
\mathbf{C}_\infty^{(M)} &\preceq \gamma \sigma_M^{(q)2} \cdot (\widetilde{\mathcal{T}}^{(q)})^{-1} \circ \mathbf{H}^{(q)} + (1+\tilde{\epsilon})\gamma (\widetilde{\mathcal{T}}^{(q)})^{-1} \circ \mathcal{M}_B^{(q)} \circ \mathbf{C}_\infty^{(M)} \\
&\preceq \gamma \sigma_M^{(q)2} \cdot \sum_{t=0}^\infty ((1+\tilde{\epsilon})\gamma (\widetilde{\mathcal{T}}^{(q)})^{-1} \circ \mathcal{M}_B^{(q)})^t \circ (\widetilde{\mathcal{T}}^{(q)})^{-1} \circ \mathbf{H}^{(q)}. \quad \text{(solving the recursion)}
\end{aligned}
$$

$$(\text{C.29})$$

---

[6] $(\mathcal{T}_B^{(q)} - \tilde{\epsilon}\gamma \mathcal{M}_B^{(q)})^{-1}$ exists under the condition that $\gamma < \frac{1}{(1+\tilde{\epsilon})\alpha_B \mathrm{tr}(\mathbf{H}^{(q)})}$, which can be directly deduced by Lemma B.1 in Zou et al. (2023). We omit the proof here for simplicity.

By the computation (C.26) in the proof for Lemma C.12,

$$(\widetilde{\mathcal{T}}^{(q)})^{-1} \circ \mathbf{H}^{(q)} \preceq \mathbf{I}. \tag{C.30}$$

Hence, by (C.29) and (C.30),

$$\begin{aligned}
\mathbf{C}_\infty^{(M)} &\preceq \gamma \sigma_M^{(q)2} \cdot \sum_{t=0}^\infty ((1+\tilde{\epsilon})\gamma(\widetilde{\mathcal{T}}^{(q)})^{-1} \circ \mathcal{M}_B^{(q)})^t \circ \mathbf{I} \\
&= \gamma \sigma_M^{(q)2} \cdot \sum_{t=0}^\infty ((1+\tilde{\epsilon})\gamma(\widetilde{\mathcal{T}}^{(q)})^{-1} \circ \mathcal{M}_B^{(q)})^{t-1} (1+\tilde{\epsilon})\gamma(\widetilde{\mathcal{T}}^{(q)})^{-1} \circ \mathcal{M}_B^{(q)} \circ \mathbf{I} \\
&\preceq \gamma \sigma_M^{(q)2} \cdot \sum_{t=0}^\infty ((1+\tilde{\epsilon})\gamma(\widetilde{\mathcal{T}}^{(q)})^{-1} \circ \mathcal{M}_B^{(q)})^{t-1} \circ (1+\tilde{\epsilon})\gamma\alpha_B \mathrm{tr}(\mathbf{H}^{(q)})\mathbf{I} \\
&\preceq \gamma \sigma_M^{(q)2} \cdot \sum_{t=0}^\infty \left((1+\tilde{\epsilon})\gamma\alpha_B \mathrm{tr}(\mathbf{H}^{(q)})\right)^t \mathbf{I} \\
&= \frac{\gamma \sigma_M^{(q)2}}{1-(1+\tilde{\epsilon})\gamma\alpha_B \mathrm{tr}(\mathbf{H}^{(q)})}\mathbf{I},
\end{aligned}$$

where the second inequality holds by the fact that $\mathcal{M}_B^{(q)} \circ \mathbf{I} \preceq \alpha_B\, \mathrm{tr}(\mathbf{H}^{(q)})\mathbf{H}^{(q)}$.

Therefore, we complete the proof by

$$\mathcal{M}_B^{(q)} \circ \mathbf{C}_t^{(M)} \preceq \frac{\alpha_B \mathrm{tr}(\mathbf{H}^{(q)})\gamma \sigma_M^{(q)2}}{1-(1+\tilde{\epsilon})\gamma\alpha_B \mathrm{tr}(\mathbf{H}^{(q)})}\mathbf{H}^{(q)}.$$

$\square$

By (C.22), (C.23), Lemma C.12 and Lemma C.13, we can provide a refined bound for $\mathbf{C}_t$ and $\mathbf{C}_t^{(M)}$. Then we are ready to bound the variance error.

**Lemma C.14** (A bound for variance under general quantization). *Under Assumption 3.1, Assumption 3.2, Assumption 3.3, and Assumption 3.4, if the stepsize satisfies $\gamma < \frac{1}{\alpha_B \mathrm{tr}(\mathbf{H}^{(q)})}$, then*

$$\mathrm{variance} \leq \frac{\sigma_G^{(q)2}}{1-\gamma\alpha_B \mathrm{tr}(\mathbf{H}^{(q)})}\left(\frac{k^*}{N} + N\gamma^2 \cdot \sum_{i>k^*}(\lambda_i^{(q)})^2\right).$$

*Proof.* We first provide a refined upper bound for $\mathbf{C}_t$. By (C.22),

$$\begin{aligned}
\mathbf{C}_t &\preceq (\mathcal{I}-\gamma\widetilde{\mathcal{T}}^{(q)}) \circ \mathbf{C}_{t-1} + \gamma^2 \mathcal{M}_B^{(q)} \circ \mathbf{C}_{t-1} + \gamma^2 \sigma_G^{(q)2}\mathbf{H}^{(q)} \\
&\preceq (\mathcal{I}-\gamma\widetilde{\mathcal{T}}^{(q)}) \circ \mathbf{C}_{t-1} + \frac{\gamma^2 \alpha_B \mathrm{tr}(\mathbf{H}^{(q)})\gamma \sigma_G^{(q)2}}{1-\gamma\alpha_B \mathrm{tr}(\mathbf{H}^{(q)})}\mathbf{H}^{(q)} + \gamma^2 \sigma_G^{(q)2}\mathbf{H}^{(q)} \\
&= (\mathcal{I}-\gamma\widetilde{\mathcal{T}}^{(q)}) \circ \mathbf{C}_{t-1} + \frac{\gamma^2 \sigma_G^{(q)2}}{1-\gamma\alpha_B \mathrm{tr}(\mathbf{H}^{(q)})}\mathbf{H}^{(q)} \\
&\preceq \frac{\gamma^2 \sigma_G^{(q)2}}{1-\gamma\alpha_B \mathrm{tr}(\mathbf{H}^{(q)})} \cdot \sum_{k=0}^{t-1}(\mathcal{I}-\gamma\widetilde{\mathcal{T}}^{(q)})^k \circ \mathbf{H}^{(q)} \quad \text{(solving the recursion)} \\
&= \frac{\gamma^2 \sigma_G^{(q)2}}{1-\gamma\alpha_B \mathrm{tr}(\mathbf{H}^{(q)})} \cdot \sum_{k=0}^{t-1}(\mathbf{I}-\gamma\mathbf{H}^{(q)})^k \mathbf{H}^{(q)}(\mathbf{I}-\gamma\mathbf{H}^{(q)})^k \\
&\preceq \frac{\gamma^2 \sigma_G^{(q)2}}{1-\gamma\alpha_B \mathrm{tr}(\mathbf{H}^{(q)})} \cdot \sum_{k=0}^{t-1}(\mathbf{I}-\gamma\mathbf{H}^{(q)})^k \mathbf{H}^{(q)} \\
&= \frac{\gamma \sigma_G^{(q)2}}{1-\gamma\alpha_B \mathrm{tr}(\mathbf{H}^{(q)})} \cdot \left(\mathbf{I}-(\mathbf{I}-\gamma\mathbf{H}^{(q)})^t\right),
\end{aligned} \tag{C.31}$$

where the second inequality holds by Lemma C.12 and the second equality holds by the definition of $\widetilde{\mathcal{T}}^{(q)}$.

After providing a refined bound for $\mathbf{C}_t$, we are ready to bound the variance. By Lemma C.4,

$$
\begin{aligned}
\text{variance} =& \frac{1}{N^2} \cdot \sum_{t=0}^{N-1} \sum_{k=t}^{N-1} \left\langle (\mathbf{I} - \gamma\mathbf{H}^{(q)})^{k-t}\mathbf{H}^{(q)}, \mathbf{C}_t \right\rangle \\
=& \frac{1}{\gamma N^2} \sum_{t=0}^{N-1} \left\langle \mathbf{I} - (\mathbf{I} - \gamma\mathbf{H}^{(q)})^{N-t}, \mathbf{C}_t \right\rangle \\
\leq& \frac{1}{\gamma^2 N^2} \frac{\gamma^2 \sigma_G^{(q)2}}{1 - \gamma\alpha_B\text{tr}(\mathbf{H}^{(q)})} \sum_{t=0}^{N-1} \left\langle \mathbf{I} - (\mathbf{I} - \gamma\mathbf{H}^{(q)})^{N-t}, \mathbf{I} - (\mathbf{I} - \gamma\mathbf{H}^{(q)})^t \right\rangle \\
=& \frac{1}{\gamma^2 N^2} \frac{\gamma^2 \sigma_G^{(q)2}}{1 - \gamma\alpha_B\text{tr}(\mathbf{H}^{(q)})} \sum_i \sum_{t=0}^{N-1} \left[1 - (1 - \gamma\lambda_i^{(q)})^{N-t}\right] \left[1 - (1 - \gamma\lambda_i^{(q)})^t\right] \\
\leq& \frac{1}{\gamma^2 N^2} \frac{\gamma^2 \sigma_G^{(q)2}}{1 - \gamma\alpha_B\text{tr}(\mathbf{H}^{(q)})} \sum_i \sum_{t=0}^{N-1} \left[1 - (1 - \gamma\lambda_i^{(q)})^{N}\right] \left[1 - (1 - \gamma\lambda_i^{(q)})^{N}\right] \quad \text{(C.32)} \\
=& \frac{1}{\gamma^2 N} \frac{\gamma^2 \sigma_G^{(q)2}}{1 - \gamma\alpha_B\text{tr}(\mathbf{H}^{(q)})} \sum_i \left[1 - (1 - \gamma\lambda_i^{(q)})^{N}\right]^2 \\
\leq& \frac{1}{\gamma^2 N} \frac{\gamma^2 \sigma_G^{(q)2}}{1 - \gamma\alpha_B\text{tr}(\mathbf{H}^{(q)})} \sum_i \min\left\{1, \gamma^2 N^2 (\lambda_i^{(q)})^2\right\} \\
\leq& \frac{1}{\gamma^2 N} \frac{\gamma^2 \sigma_G^{(q)2}}{1 - \gamma\alpha_B\text{tr}(\mathbf{H}^{(q)})} \left(k^* + N^2\gamma^2 \cdot \sum_{i>k^*} (\lambda_i^{(q)})^2\right) \\
=& \frac{\sigma_G^{(q)2}}{1 - \gamma\alpha_B\text{tr}(\mathbf{H}^{(q)})} \left(\frac{k^*}{N} + N\gamma^2 \cdot \sum_{i>k^*} (\lambda_i^{(q)})^2\right),
\end{aligned}
$$

where the first inequality holds by (C.31) and the last inequality holds by the definition of $k^* = \max\left\{k : \lambda_k^{(q)} \geq \frac{1}{N\gamma}\right\}$. This immediately completes the proof. $\qquad\square$

**Lemma C.15** (A bound for variance under multiplicative quantization). *Under Assumption 3.1, Assumption 3.2, Assumption 3.3, and Assumption 3.4, if the stepsize satisfies $\gamma < \frac{1}{(1+\tilde{\epsilon})\alpha_B\text{tr}(\mathbf{H}^{(q)})}$, if there exist $\epsilon_d, \epsilon_l, \epsilon_p, \epsilon_a$ and $\epsilon_o$ such that for any $i \in \{d,l,p,a,o\}$, quantization $\mathcal{Q}_i$ is $\epsilon_i$-multiplicative, then*

$$
\text{variance} \leq \frac{\sigma_M^{(q)2}}{1 - (1+\tilde{\epsilon})\gamma\alpha_B\text{tr}(\mathbf{H}^{(q)})} \left(\frac{k^*}{N} + N\gamma^2 \cdot \sum_{i>k^*} (\lambda_i^{(q)})^2\right).
$$

*Proof.* Applying (C.23), and repeating the computation in the proof of Lemma C.14,

$$
\begin{aligned}
\mathbf{C}_t^{(M)} \preceq& (\mathcal{I} - \gamma\widetilde{\mathcal{T}}^{(q)}) \circ \mathbf{C}_{t-1}^{(M)} + \gamma^2(1+\tilde{\epsilon})\mathcal{M}_B^{(q)} \circ \mathbf{C}_{t-1}^{(M)} + \gamma^2\sigma_M^{(q)2}\mathbf{H}^{(q)} \\
\preceq& (\mathcal{I} - \gamma\widetilde{\mathcal{T}}^{(q)}) \circ \mathbf{C}_{t-1}^{(M)} + \gamma^2(1+\tilde{\epsilon})\frac{\alpha_B\text{tr}(\mathbf{H}^{(q)})\gamma\sigma_M^{(q)2}}{1 - (1+\tilde{\epsilon})\gamma\alpha_B\text{tr}(\mathbf{H}^{(q)})}\mathbf{H}^{(q)} + \gamma^2\sigma_M^{(q)2}\mathbf{H}^{(q)} \\
=& (\mathcal{I} - \gamma\widetilde{\mathcal{T}}^{(q)}) \circ \mathbf{C}_{t-1}^{(M)} + \frac{\gamma^2\sigma_M^{(q)2}\mathbf{H}^{(q)}}{1 - (1+\tilde{\epsilon})\gamma\alpha_B\text{tr}(\mathbf{H}^{(q)})}\mathbf{H}^{(q)} \\
\preceq& \frac{\gamma\sigma_M^{(q)2}}{1 - (1+\tilde{\epsilon})\gamma\alpha_B\text{tr}(\mathbf{H}^{(q)})} \cdot \left(\mathbf{I} - (\mathbf{I} - \gamma\mathbf{H}^{(q)})^t\right),
\end{aligned}
$$

where the second inequality holds by Lemma C.13 and the last inequality repeats the proof in (C.31).

Therefore, repeating the procedure in the proof for Lemma C.14, we directly deduce that

$$\text{variance} \leq \frac{\sigma_M^{(q)^2}}{1 - (1 + \tilde{\epsilon})\gamma\alpha_B \text{tr}(\mathbf{H}^{(q)})} \left( \frac{k^*}{N} + N\gamma^2 \cdot \sum_{i > k^*} (\lambda_i^{(q)})^2 \right),$$

which immediately completes the proof. □

**Lemma C.16** (A bound for $R_N^{(0)}$ under general quantization). *Under Assumption 3.1, Assumption 3.2, Assumption 3.3, and Assumption 3.4, if the stepsize satisfies $\gamma < \frac{1}{\alpha_B \text{tr}(\mathbf{H}^{(q)})}$, then*

$$R_N^{(0)}/2 \leq \frac{2\alpha_B \left( \|\mathbf{w}_0 - \mathbf{w}^{(q)^*}\|_{\mathbf{I}_{0:k^*}^{(q)}}^2 + N\gamma \|\mathbf{w}_0 - \mathbf{w}^{(q)^*}\|_{\mathbf{H}_{k^*:\infty}^{(q)}}^2 \right)}{N\gamma(1 - \gamma\alpha_B \text{tr}(\mathbf{H}^{(q)}))} \cdot \left( \frac{k^*}{N} + N\gamma^2 \sum_{i > k^*} (\lambda_i^{(q)})^2 \right)$$

$$+ \frac{1}{\gamma^2 N^2} \cdot \|\mathbf{w}_0 - \mathbf{w}^{(q)^*}\|_{(\mathbf{H}_{0:k^*}^{(q)})^{-1}}^2 + \|\mathbf{w}_0 - \mathbf{w}^{(q)^*}\|_{\mathbf{H}_{k^*:\infty}^{(q)}}^2$$

$$+ \frac{\sigma_G^{(q)^2}}{1 - \gamma\alpha_B \text{tr}(\mathbf{H}^{(q)})} \left( \frac{k^*}{N} + N\gamma^2 \cdot \sum_{i > k^*} (\lambda_i^{(q)})^2 \right),$$

*where $k^* = \max\left\{ k : \lambda_k^{(q)} \geq \frac{1}{N\gamma} \right\}$ and*

$$\sigma_G^{(q)^2} = \frac{\sup_t \left\{ \left\| \mathbb{E}\left[ \boldsymbol{\epsilon}_t^{(o)} \boldsymbol{\epsilon}_t^{(o)^\top} | \mathbf{o}_t \right] + \mathbb{E}\left[ \boldsymbol{\epsilon}_t^{(a)} \boldsymbol{\epsilon}_t^{(a)^\top} | \mathbf{a}_t \right] \right\| \right\}}{B}$$

$$+ \alpha_B \sup_t \mathbb{E}_{\mathbf{w}_{t-1}} \left[ \text{tr}\left( \mathbf{H}^{(q)} \mathbb{E}\left[ \boldsymbol{\epsilon}_{t-1}^{(p)} \boldsymbol{\epsilon}_{t-1}^{(p)^\top} | \mathbf{w}_{t-1} \right] \right) \right] + \frac{\sigma^2}{B}.$$

*Proof.* The proof is immediately completed by Lemma C.4, Lemma C.10 and Lemma C.14. □

**Lemma C.17** (A bound for $R_N^{(0)}$ under multiplicative quantization). *Under Assumption 3.1, Assumption 3.2, Assumption 3.3, and Assumption 3.4, if the stepsize satisfies $\gamma < \frac{1}{(1+\tilde{\epsilon})\alpha_B \text{tr}(\mathbf{H}^{(q)})}$, if there exist $\epsilon_d, \epsilon_l, \epsilon_p, \epsilon_a$ and $\epsilon_o$ such that for any $i \in \{d, l, p, a, o\}$, quantization $\mathcal{Q}_i$ is $\epsilon_i$-multiplicative, then*

$$R_N^{(0)}/2 \leq \frac{2(1 + \tilde{\epsilon})\alpha_B \left( \|\mathbf{w}_0 - \mathbf{w}^{(q)^*}\|_{\mathbf{I}_{0:k^*}^{(q)}}^2 + N\gamma \|\mathbf{w}_0 - \mathbf{w}^{(q)^*}\|_{\mathbf{H}_{k^*:\infty}^{(q)}}^2 \right)}{N\gamma(1 - (1 + \tilde{\epsilon})\gamma\alpha_B \text{tr}(\mathbf{H}^{(q)}))} \cdot \left( \frac{k^*}{N} + N\gamma^2 \sum_{i > k^*} (\lambda_i^{(q)})^2 \right)$$

$$+ \frac{1}{\gamma^2 N^2} \cdot \|\mathbf{w}_0 - \mathbf{w}^{(q)^*}\|_{(\mathbf{H}_{0:k^*}^{(q)})^{-1}}^2 + \|\mathbf{w}_0 - \mathbf{w}^{(q)^*}\|_{\mathbf{H}_{k^*:\infty}^{(q)}}^2$$

$$+ \frac{\sigma_M^{(q)^2}}{1 - (1 + \tilde{\epsilon})\gamma\alpha_B \text{tr}(\mathbf{H}^{(q)})} \left( \frac{k^*}{N} + N\gamma^2 \cdot \sum_{i > k^*} (\lambda_i^{(q)})^2 \right),$$

*where $k^* = \max\left\{ k : \lambda_k^{(q)} \geq \frac{1}{N\gamma} \right\}$ and*

$$\tilde{\epsilon} = 8\epsilon_o(1 + \epsilon_p)(1 + \epsilon_a) + 4\epsilon_p + 4\epsilon_a(1 + \epsilon_p),$$

$$\sigma_M^{(q)^2} = \frac{(1 + 4\epsilon_o)\sigma^2}{B} + \frac{\|\mathbf{w}^*\|_{\mathbf{H}}^2}{1 + \epsilon_d}\alpha_B \left( 4\epsilon_o[(1 + \epsilon_p)(1 + \epsilon_a) + 1] + 2\epsilon_a(1 + \epsilon_p) + 2\epsilon_p \right).$$

*Proof.* The proof is immediately completed by Lemma C.5, Lemma C.11 and Lemma C.15. □

# D DEFERRING PROOFS

## D.1 PROOF FOR THEOREM 4.1

*Proof.* By Lemma A.3, (B.1), (B.2), (C.1) and Lemma C.16, we have

$$\mathbb{E}[\mathcal{E}(\overline{\mathbf{w}}_N)] \leq 2\text{VarErr} + 2\text{BiasErr} + \text{ApproxErr},$$

where

$$\text{VarErr} = \frac{2\alpha_B \left( \frac{\|\mathbf{w}_0 - \mathbf{w}^{(q)*}\|^2_{\mathbf{I}^{(q)}_{0:k^*}}}{N\gamma} + \|\mathbf{w}_0 - \mathbf{w}^{(q)*}\|^2_{\mathbf{H}^{(q)}_{k^*:\infty}} \right) + \sigma_G^{(q)2}}{1 - \gamma\alpha_B \text{tr}(\mathbf{H}^{(q)})} \left( \frac{k^*}{N} + N\gamma^2 \cdot \sum_{i>k^*} (\lambda_i^{(q)})^2 \right),$$

$$\text{BiasErr} = \frac{1}{\gamma^2 N^2} \cdot \|\mathbf{w}_0 - \mathbf{w}^{(q)*}\|^2_{(\mathbf{H}^{(q)}_{0:k^*})^{-1}} + \|\mathbf{w}_0 - \mathbf{w}^{(q)*}\|^2_{\mathbf{H}^{(q)}_{k^*:\infty}},$$

$$\text{ApproxErr} = \|\mathbf{w}^*\|^2_{\mathbf{D}_2^{\mathbf{H}}} + \frac{1}{2}\|\mathbf{w}^*\|^2_{\mathbf{D}_1^{\mathbf{H}}},$$

with $\mathbf{D} = \mathbf{H}^{(q)} - \mathbf{H}$, $k^* = \max\left\{ k : \lambda_k^{(q)} \geq \frac{1}{N\gamma} \right\}$, and

$$\mathbf{D}_2^{\mathbf{H}} = \mathbf{H}(\mathbf{H}^{(q)})^{-1} \frac{1}{N\gamma} \left( \mathbf{I} - (\mathbf{I} - \gamma\mathbf{H}^{(q)})^N \right) (\mathbf{H}^{(q)})^{-1} \mathbf{D}(\mathbf{H}^{(q)})^{-1}\mathbf{H}, \quad \mathbf{D}_1^{\mathbf{H}} = \mathbf{D}(\mathbf{H}^{(q)})^{-1}\mathbf{H}(\mathbf{H}^{(q)})^{-1}\mathbf{D},$$

$$\sigma_G^{(q)2} = \frac{\sup_t \left\{ \left\| \mathbb{E}\left[ \boldsymbol{\epsilon}_t^{(o)} \boldsymbol{\epsilon}_t^{(o)\top} | \mathbf{o}_t \right] + \mathbb{E}\left[ \boldsymbol{\epsilon}_t^{(a)} \boldsymbol{\epsilon}_t^{(a)\top} | \mathbf{a}_t \right] \right\| \right\}}{B}$$

$$+ \alpha_B \sup_t \mathbb{E}_{\mathbf{w}_{t-1}} \left[ \text{tr}\left( \mathbf{H}^{(q)} \mathbb{E}\left[ \boldsymbol{\epsilon}_{t-1}^{(p)} \boldsymbol{\epsilon}_{t-1}^{(p)\top} | \mathbf{w}_{t-1} \right] \right) \right] + \frac{\sigma^2}{B}.$$

Let the initialization $\mathbf{w}_0 = 0$ completes the proof. $\qquad\square$

## D.2 PROOF FOR THEOREM 4.2

We prove a tighter excess risk bound under multiplicative quantization in this subsection:

**Theorem D.1** (Multiplicative quantization). *Under Assumption 3.1, 3.2, 3.3 and 3.4, if there exist $\epsilon_d, \epsilon_l, \epsilon_p, \epsilon_a$ and $\epsilon_o$ such that for any $i \in \{d, l, p, a, o\}$, quantization $\mathcal{Q}_i$ is $\epsilon_i$-multiplicative, and the stepsize satisfies $\gamma < \frac{1}{\alpha_B(1+\epsilon_o)[1+\epsilon_p+\epsilon_a(1+\epsilon_p)](1+\epsilon_d)\text{tr}(\mathbf{H})}$, then the excess risk can be upper bounded as follows.*

$$\mathbb{E}[\mathcal{E}(\overline{\mathbf{w}}_N)] \lesssim \text{ApproxErr} + \text{VarErr} + \text{BiasErr},$$

*where*

$$\text{ApproxErr} \lesssim \frac{\epsilon_d}{1 + \epsilon_d}\|\mathbf{w}^*\|^2_{\mathbf{H}}, \quad \text{BiasErr} \lesssim \frac{1}{\gamma^2 N^2} \cdot \|\mathbf{w}^{(q)*}\|^2_{(\mathbf{H}^{(q)}_{0:k^*})^{-1}} + \|\mathbf{w}^{(q)*}\|^2_{\mathbf{H}^{(q)}_{k^*:\infty}},$$

$$\text{VarErr} \lesssim \left( \frac{k^*}{N} + N\gamma^2(1+\epsilon_d)^2 \sum_{i>k^*} \lambda_i^2 \right) \frac{\frac{(1+\epsilon_o)\sigma^2}{B} + \alpha_B \sigma_M^2}{1 - \gamma\alpha_B(1+\epsilon_o)[1+\epsilon_p+\epsilon_a(1+\epsilon_p)](1+\epsilon_d)\text{tr}(\mathbf{H})},$$

*with*

$$\sigma_M^2 = [\epsilon_o + (1+\epsilon_o)(\epsilon_p + \epsilon_a(1+\epsilon_p))]\|\mathbf{w}^*\|^2_{\mathbf{H}} + (1+\epsilon_o)[1+\epsilon_p+\epsilon_a(1+\epsilon_p)] \left( \frac{\|\mathbf{w}^{(q)*}\|^2_{\mathbf{I}^{(q)}_{0:k^*}}}{N\gamma} + \|\mathbf{w}^{(q)*}\|^2_{\mathbf{H}^{(q)}_{k^*:\infty}} \right).$$

*Proof.* By Lemma A.3, Lemma B.1, (C.1) and Lemma C.17, we have

$$\mathbb{E}[\mathcal{E}(\overline{\mathbf{w}}_N)] \leq 2\text{VarErr} + 2\text{BiasErr} + \text{ApproxErr},$$

where

$$\text{VarErr} = \frac{\sigma_M^{(q)2} + 2(1+\tilde{\epsilon})\alpha_B \left( \frac{\|\mathbf{w}_0 - \mathbf{w}^{(q)*}\|^2_{\mathbf{I}_{0:k^*}^{(q)}}}{N\gamma} + \|\mathbf{w}_0 - \mathbf{w}^{(q)*}\|^2_{\mathbf{H}_{k^*:\infty}^{(q)}} \right)}{1 - (1+\tilde{\epsilon})\gamma\alpha_B \text{tr}(\mathbf{H}^{(q)})} \left( \frac{k^*}{N} + N\gamma^2 \cdot \sum_{i>k^*} (\lambda_i^{(q)})^2 \right),$$

$$\text{BiasErr} = \frac{1}{\gamma^2 N^2} \cdot \|\mathbf{w}_0 - \mathbf{w}^{(q)*}\|^2_{(\mathbf{H}_{0:k^*}^{(q)})^{-1}} + \|\mathbf{w}_0 - \mathbf{w}^{(q)*}\|^2_{\mathbf{H}_{k^*:\infty}^{(q)}},$$

$$\text{ApproxErr} = \frac{\epsilon_d^2}{2(1+\epsilon_d)^2} \|\mathbf{w}^*\|^2_{\mathbf{H}} + \frac{\epsilon_d}{(1+\epsilon_d)^2} \|\mathbf{w}^*\|^2_{\mathbf{H}} \lesssim \frac{\epsilon_d}{1+\epsilon_d} \|\mathbf{w}^*\|^2_{\mathbf{H}},$$

with $k^* = \max \left\{ k : \lambda_k^{(q)} \geq \frac{1}{N\gamma} \right\}$ and

$$\tilde{\epsilon} = 8\epsilon_o(1+\epsilon_p)(1+\epsilon_a) + 4\epsilon_p + 4\epsilon_a(1+\epsilon_p) \lesssim \epsilon_o + (1+\epsilon_o)(\epsilon_p + \epsilon_a(1+\epsilon_p)),$$

$$\sigma_M^{(q)2} = \frac{(1+4\epsilon_o)\sigma^2}{B} + \frac{\|\mathbf{w}^*\|^2_{\mathbf{H}}}{1+\epsilon_d}\alpha_B \left(4\epsilon_o[(1+\epsilon_p)(1+\epsilon_a)+1] + 2\epsilon_a(1+\epsilon_p) + 2\epsilon_p \right)$$

$$\lesssim \frac{(1+\epsilon_o)\sigma^2}{B} + \|\mathbf{w}^*\|^2_{\mathbf{H}}\alpha_B \left(\epsilon_o + (1+\epsilon_o)(\epsilon_p + \epsilon_a(1+\epsilon_p)) \right).$$

Let initialization $\mathbf{w}_0 = 0$. Regarding VarErr, noticing that $1 + \tilde{\epsilon} \lesssim (1+\epsilon_o)[1 + \epsilon_p + \epsilon_a(1+\epsilon_p)]$, we have

$$\sigma_M^{(q)2} + 2(1+\tilde{\epsilon})\alpha_B \left( \frac{\|\mathbf{w}^{(q)*}\|^2_{\mathbf{I}_{0:k^*}^{(q)}}}{N\gamma} + \|\mathbf{w}^{(q)*}\|^2_{\mathbf{H}_{k^*:\infty}^{(q)}} \right)$$

$$\lesssim \frac{(1+\epsilon_o)\sigma^2}{B} + \alpha_B \|\mathbf{w}^*\|^2_{\mathbf{H}} \left( \epsilon_o + (1+\epsilon_o)(\epsilon_p + \epsilon_a(1+\epsilon_p)) \right)$$

$$+ \alpha_B(1+\epsilon_o)[1 + \epsilon_p + \epsilon_a(1+\epsilon_p)] \left( \frac{\|\mathbf{w}^{(q)*}\|^2_{\mathbf{I}_{0:k^*}^{(q)}}}{N\gamma} + \|\mathbf{w}^{(q)*}\|^2_{\mathbf{H}_{k^*:\infty}^{(q)}} \right).$$

Then the proof is completed by $\text{tr}(\mathbf{H}^{(q)}) = (1+\epsilon_d)\text{tr}(\mathbf{H})$ and $\lambda_i^{(q)} = (1+\epsilon_d)\lambda_i$. $\qquad\square$

Theorem 4.2 can be deduced from Theorem D.1 by noticing that

$$\sigma_M^2 \lesssim (1+\epsilon_o)[1 + \epsilon_p + \epsilon_a(1+\epsilon_p)] \|\mathbf{w}^*\|^2_{\mathbf{H}},$$

where we use

$$\frac{\|\mathbf{w}^{(q)*}\|^2_{\mathbf{I}_{0:k^*}^{(q)}}}{N\gamma} + \|\mathbf{w}^{(q)*}\|^2_{\mathbf{H}_{k^*:\infty}^{(q)}} \leq \|\mathbf{w}^{(q)*}\|^2_{\mathbf{H}^{(q)}} = \frac{\|\mathbf{w}^*\|^2_{\mathbf{H}}}{1+\epsilon_d} \leq \|\mathbf{w}^*\|^2_{\mathbf{H}}.$$

### D.3 Proof for Corollary 4.1

We provide a tighter excess risk bound under additive quantization in this subsection:

**Corollary D.1** (Additive quantization). *Under Assumption 3.1, 3.2, 3.3 and 3.4, if there exist $\epsilon_d, \epsilon_l, \epsilon_p, \epsilon_a$ and $\epsilon_o$ such that for any $i \in \{d, l, p, a, o\}$, quantization $\mathcal{Q}_i$ is $\epsilon_i$-additive, and the stepsize satisfies $\gamma < \frac{1}{\alpha_B[\text{tr}(\mathbf{H})+d\epsilon_d]}$, then*

$$\mathbb{E}[\mathcal{E}(\overline{\mathbf{w}}_N)] \lesssim \text{ApproxErr} + \text{VarErr} + \text{BiasErr},$$

*where*

$$\text{ApproxErr} \lesssim \frac{\epsilon_d}{\lambda_d + \epsilon_d} \|\mathbf{w}^*\|^2_{\mathbf{H}}, \quad \text{BiasErr} \lesssim \frac{1}{\gamma^2 N^2} \cdot \|\mathbf{w}^{(q)*}\|^2_{(\mathbf{H}_{0:k^*}^{(q)})^{-1}} + \|\mathbf{w}^{(q)*}\|^2_{\mathbf{H}_{k^*:\infty}^{(q)}},$$

$$\text{VarErr} \lesssim \frac{\alpha_B \left( \frac{\|\mathbf{w}^{(q)*}\|^2_{\mathbf{I}_{0:k^*}^{(q)}}}{N\gamma} + \|\mathbf{w}^{(q)*}\|^2_{\mathbf{H}_{k^*:\infty}^{(q)}} \right) + \frac{\sigma^2 + \epsilon_o + \epsilon_a}{B} + \alpha_B\epsilon_p[\text{tr}(\mathbf{H}) + d\epsilon_d]}{1 - \gamma\alpha_B[\text{tr}(\mathbf{H}) + d\epsilon_d]} \left( \frac{k^*}{N} + N\gamma^2 \cdot \sum_{i>k^*} (\lambda_i + \epsilon_d)^2 \right).$$

*Proof.* By Theorem 4.1,

$$\mathbb{E}[\mathcal{E}(\overline{\mathbf{w}}_N)] \leq 2\text{VarErr} + 2\text{BiasErr} + \text{ApproxErr},$$

where

$$\text{VarErr} \leq \frac{2\alpha_B \left( \frac{\|\mathbf{w}^{(q)*}\|^2_{\mathbf{I}_{0:k^*}^{(q)}}}{N\gamma} + \|\mathbf{w}^{(q)*}\|^2_{\mathbf{H}_{k^*:\infty}^{(q)}} \right) + \sigma_G^{(q)2}}{1 - \gamma\alpha_B \text{tr}(\mathbf{H}^{(q)})} \left( \frac{k^*}{N} + N\gamma^2 \cdot \sum_{i>k^*} (\lambda_i^{(q)})^2 \right),$$

$$\text{BiasErr} \leq \frac{1}{\gamma^2 N^2} \cdot \|\mathbf{w}^{(q)*}\|^2_{(\mathbf{H}_{0:k^*}^{(q)})^{-1}} + \|\mathbf{w}^{(q)*}\|^2_{\mathbf{H}_{k^*:\infty}^{(q)}},$$

$$\text{ApproxErr} \leq \|\mathbf{w}^*\|^2_{\mathbf{D}(\mathbf{H}+\mathbf{D})^{-1}\mathbf{H}(\mathbf{H}+\mathbf{D})^{-1}\mathbf{D}} + \|\mathbf{w}^*\|^2_{\mathbf{D_H}},$$

with $\sigma_G^{(q)2} = \frac{\sigma^2 + \sup_t \left\{ \left\| \mathbb{E}\left[ \boldsymbol{\epsilon}_t^{(o)} \boldsymbol{\epsilon}_t^{(o)\top} | \mathbf{o}_t \right] + \mathbb{E}\left[ \boldsymbol{\epsilon}_t^{(a)} \boldsymbol{\epsilon}_t^{(a)\top} | \mathbf{a}_t \right] \right\| \right\}}{B} + \alpha_B \sup_t \mathbb{E}\left[ \text{tr}\left( \mathbf{H}^{(q)} \boldsymbol{\epsilon}_{t-1}^{(p)} \boldsymbol{\epsilon}_{t-1}^{(p)\top} \right) \right]$ and

$\mathbf{D_H} = \mathbf{H}(\mathbf{H}^{(q)})^{-1} \frac{1}{N\gamma} \left( \mathbf{I} - (\mathbf{I} - \gamma\mathbf{H}^{(q)})^N \right) (\mathbf{H}^{(q)})^{-1} \mathbf{D}(\mathbf{H}^{(q)})^{-1}\mathbf{H}.$

Under additive quantization, it holds

$$\text{tr}(\mathbf{H}^{(q)}) = \text{tr}(\mathbf{H}) + d\epsilon_d, \quad \sum_{i>k^*} (\lambda_i^{(q)})^2 = \sum_{i>k^*} (\lambda_i + \epsilon_d)^2,$$

and

$$\sigma_G^{(q)2} = \frac{\sigma^2 + \epsilon_o + \epsilon_a}{B} + \alpha_B \epsilon_p [\text{tr}(\mathbf{H}) + d\epsilon_d].$$

Then we have

$$\text{VarErr} \lesssim \frac{\alpha_B \left( \frac{\|\mathbf{w}^{(q)*}\|^2_{\mathbf{I}_{0:k^*}^{(q)}}}{N\gamma} + \|\mathbf{w}^{(q)*}\|^2_{\mathbf{H}_{k^*:\infty}^{(q)}} \right) + \frac{\sigma^2 + \epsilon_o + \epsilon_a}{B} + \alpha_B \epsilon_p [\text{tr}(\mathbf{H}) + d\epsilon_d]}{1 - \gamma\alpha_B [\text{tr}(\mathbf{H}) + d\epsilon_d]} \left( \frac{k^*}{N} + N\gamma^2 \cdot \sum_{i>k^*} (\lambda_i + \epsilon_d)^2 \right).$$

The proof is completed by Lemma B.2:

$$\text{ApproxErr} \leq \frac{\epsilon_d^2}{2(\lambda_d + \epsilon_d)^2} \|\mathbf{w}^*\|^2_{\mathbf{H}} + \frac{\lambda_1 \epsilon_d}{(\lambda_d + \epsilon_d)(\lambda_1 + \epsilon_d)} \|\mathbf{w}^*\|^2_{\mathbf{H}} \lesssim \frac{\epsilon_d}{\lambda_d + \epsilon_d} \|\mathbf{w}^*\|^2_{\mathbf{H}}.$$

$\square$

Corollary 4.1 can be deduced from Corollary D.1 by noticing that

$$\frac{\|\mathbf{w}^{(q)*}\|^2_{\mathbf{I}_{0:k^*}^{(q)}}}{N\gamma} + \|\mathbf{w}^{(q)*}\|^2_{\mathbf{H}_{k^*:\infty}^{(q)}} \leq \|\mathbf{w}^{(q)*}\|^2_{\mathbf{H}^{(q)}} = \mathbf{w}^{*\top} \mathbf{H}(\mathbf{H}^{(q)})^{-1}\mathbf{H}\mathbf{w}^* \leq \|\mathbf{w}^*\|^2_{\mathbf{H}}.$$

### D.4 PROOF FOR THE MULTIPLICATIVE STATEMENT IN COROLLARY 4.2

*Proof.* Recall that $k_0^* = \max\{k : \lambda_k \geq \frac{1}{N\gamma}\}$,

$$R_0 = \underbrace{\left( \frac{k_0^*}{N} + N\gamma^2 \cdot \sum_{i>k_0^*} \lambda_i^2 \right) \frac{\alpha_B \left( \frac{1}{N\gamma} \|\mathbf{w}^*\|^2_{\mathbf{I}_{0:k_0^*}} + \|\mathbf{w}^*\|^2_{\mathbf{H}_{k_0^*:\infty}} \right) + \frac{\sigma^2}{B}}{1 - \gamma\alpha_B \text{tr}(\mathbf{H})}}_{\text{EffectiveVar}}$$

$$+ \underbrace{\frac{1}{\gamma^2 N^2} \cdot \|\mathbf{w}^*\|^2_{(\mathbf{H}_{0:k_0^*})^{-1}} + \|\mathbf{w}^*\|^2_{\mathbf{H}_{k_0^*:\infty}}}_{\text{EffectiveBias}},$$

and by Theorem D.1,

$$\mathbb{E}[\mathcal{E}(\overline{\mathbf{w}}_N)] \lesssim \text{ApproxErr} + \text{VarErr} + \text{BiasErr},$$

where

$$\text{ApproxErr} \lesssim \frac{\epsilon_d}{1+\epsilon_d} \|\mathbf{w}^*\|_{\mathbf{H}}^2, \quad \text{BiasErr} \leq \frac{1}{\gamma^2 N^2} \cdot \|\mathbf{w}^{(q)*}\|_{(\mathbf{H}_{0:k^*}^{(q)})^{-1}}^2 + \|\mathbf{w}^{(q)*}\|_{\mathbf{H}_{k^*:\infty}^{(q)}}^2,$$

$$\text{VarErr} \lesssim \left( \frac{k^*}{N} + N\gamma^2(1+\epsilon_d)^2 \sum_{i>k^*} \lambda_i^2 \right) \frac{\frac{(1+\epsilon_o)\sigma^2}{B} + \alpha_B \sigma_M^2}{1 - \gamma\alpha_B(1+\epsilon_o)[1+\epsilon_p+\epsilon_a(1+\epsilon_p)](1+\epsilon_d)\text{tr}(\mathbf{H})},$$

with

$$\sigma_M^2 = [\epsilon_o+(1+\epsilon_o)(\epsilon_p+\epsilon_a(1+\epsilon_p))] \|\mathbf{w}^*\|_{\mathbf{H}}^2 + (1+\epsilon_o)[1+\epsilon_p+\epsilon_a(1+\epsilon_p)] \left( \frac{\|\mathbf{w}^{(q)*}\|_{\mathbf{I}_{0:k^*}^{(q)}}^2}{N\gamma} + \|\mathbf{w}^{(q)*}\|_{\mathbf{H}_{k^*:\infty}^{(q)}}^2 \right).$$

We then compare the upper bound of $\mathbb{E}[\mathcal{E}(\overline{\mathbf{w}}_N)]$ with $R_0$. Regarding VarErr, we first analyze $\frac{k^*}{N} + N\gamma^2(1+\epsilon_d)^2 \sum_{i>k^*} \lambda_i^2$. Note that for $k_0^* < i \leq k^*$, $\frac{1}{N\gamma(1+\epsilon_d)} \leq \lambda_i < \frac{1}{N\gamma}$, we have

$$\frac{k^*}{N} + N\gamma^2(1+\epsilon_d)^2 \cdot \sum_{i>k^*} \lambda_i^2$$

$$= \frac{k_0^*}{N} + \frac{k^*-k_0^*}{N} - N\gamma^2(1+\epsilon_d)^2 \cdot \sum_{k_0^*<i\leq k^*} \lambda_i^2 + N\gamma^2(1+\epsilon_d)^2 \cdot \sum_{i>k_0^*} \lambda_i^2$$

$$\leq \frac{k_0^*}{N} + \frac{k^*-k_0^*}{N} - N\gamma^2(1+\epsilon_d)^2(k^*-k_0^*)\frac{1}{N^2\gamma^2(1+\epsilon_d)^2} + N\gamma^2(1+\epsilon_d)^2 \cdot \sum_{i>k_0^*} \lambda_i^2$$

$$= \frac{k_0^*}{N} + N\gamma^2(1+\epsilon_d)^2 \cdot \sum_{i>k_0^*} \lambda_i^2.$$

We then analyze $\frac{\|\mathbf{w}^{(q)*}\|_{\mathbf{I}_{0:k^*}^{(q)}}^2}{N\gamma} + \|\mathbf{w}^{(q)*}\|_{\mathbf{H}_{k^*:\infty}^{(q)}}^2$. Similarly,

$$\frac{\|\mathbf{w}^{(q)*}\|_{\mathbf{I}_{0:k^*}^{(q)}}^2}{N\gamma} + \|\mathbf{w}^{(q)*}\|_{\mathbf{H}_{k^*:\infty}^{(q)}}^2$$

$$= \frac{\|\mathbf{w}^{(q)*}\|_{\mathbf{I}_{0:k_0^*}^{(q)}}^2}{N\gamma} + \frac{\|\mathbf{w}^{(q)*}\|_{\mathbf{I}_{k_0^*:k^*}^{(q)}}^2}{N\gamma} - \|\mathbf{w}^{(q)*}\|_{\mathbf{H}_{k_0^*:k^*}^{(q)}}^2 + \|\mathbf{w}^{(q)*}\|_{\mathbf{H}_{k_0^*:\infty}^{(q)}}^2$$

$$\leq \frac{\|\mathbf{w}^{(q)*}\|_{\mathbf{I}_{0:k_0^*}^{(q)}}^2}{N\gamma} + \|\mathbf{w}^{(q)*}\|_{\mathbf{H}_{k_0^*:\infty}^{(q)}}^2$$

$$\leq \frac{\|\mathbf{w}^*\|_{\mathbf{I}_{0:k_0^*}}^2}{N\gamma} + \|\mathbf{w}^*\|_{\mathbf{H}_{k_0^*:\infty}}^2.$$

Therefore, the sufficient conditions for $\text{VarErr} \lesssim \text{EffectiveVar}$ are

$$\epsilon_d \lesssim 1, \quad \epsilon_o, \epsilon_a, \epsilon_p \lesssim \left( \frac{\sigma^2}{B\alpha_B\|\mathbf{w}^*\|_{\mathbf{H}}^2} + \frac{\frac{1}{N\gamma}\|\mathbf{w}^*\|_{\mathbf{I}_{0:k_0^*}}^2 + \|\mathbf{w}^*\|_{\mathbf{H}_{k_0^*:\infty}}^2}{\|\mathbf{w}^*\|_{\mathbf{H}}^2} \right) \wedge 1,$$

Secondly, we analyze BiasErr. Similarly,

$$\frac{1}{\gamma^2 N^2} \cdot \|\mathbf{w}^{(q)*}\|_{(\mathbf{H}_{0:k^*}^{(q)})^{-1}}^2 + \|\mathbf{w}^{(q)*}\|_{\mathbf{H}_{k^*:\infty}^{(q)}}^2$$

$$= \frac{1}{\gamma^2 N^2} \cdot \left( \|\mathbf{w}^{(q)*}\|_{(\mathbf{H}_{0:k_0^*}^{(q)})^{-1}}^2 + \|\mathbf{w}^{(q)*}\|_{(\mathbf{H}_{k_0^*:k^*}^{(q)})^{-1}}^2 \right) - \|\mathbf{w}^{(q)*}\|_{\mathbf{H}_{k_0^*:k^*}^{(q)}}^2 + \|\mathbf{w}^{(q)*}\|_{\mathbf{H}_{k_0^*:\infty}^{(q)}}^2 \qquad \text{(D.1)}$$

$$\leq \frac{1}{\gamma^2 N^2} \cdot \|\mathbf{w}^{(q)*}\|_{(\mathbf{H}_{0:k_0^*}^{(q)})^{-1}}^2 + \|\mathbf{w}^{(q)*}\|_{\mathbf{H}_{k_0^*:\infty}^{(q)}}^2$$

$$\leq \frac{1}{\gamma^2 N^2} \cdot \|\mathbf{w}^*\|_{(\mathbf{H}_{0:k_0^*})^{-1}}^2 + \|\mathbf{w}^*\|_{\mathbf{H}_{k_0^*:\infty}}^2 = \text{EffectiveBias}.$$

Thirdly, the sufficient condition for ApproxErr $\lesssim R_0$ is $\epsilon_d \lesssim \frac{R_0}{\|\mathbf{w}^*\|_{\mathbf{H}}^2}$. Overall, we require

$$\epsilon_d \lesssim 1 \wedge \frac{R_0}{\|\mathbf{w}^*\|_{\mathbf{H}}^2}, \quad \epsilon_o, \epsilon_a, \epsilon_p \lesssim \left( \frac{\sigma^2}{B\alpha_B\|\mathbf{w}^*\|_{\mathbf{H}}^2} + \frac{\frac{1}{N\gamma}\|\mathbf{w}^*\|_{\mathbf{I}_{0:k_0^*}}^2 + \|\mathbf{w}^*\|_{\mathbf{H}_{k_0^*:\infty}}^2}{\|\mathbf{w}^*\|_{\mathbf{H}}^2} \right) \wedge 1.$$

$\square$

### D.5 PROOF FOR THE ADDITIVE STATEMENT IN COROLLARY 4.2

*Proof.* Recall that $k_0^* = \max\{k : \lambda_k \geq \frac{1}{N\gamma}\}$,

$$R_0 = \underbrace{\left( \frac{k_0^*}{N} + N\gamma^2 \cdot \sum_{i>k_0^*} \lambda_i^2 \right) \frac{\alpha_B \left( \frac{1}{N\gamma}\|\mathbf{w}^*\|_{\mathbf{I}_{0:k_0^*}}^2 + \|\mathbf{w}^*\|_{\mathbf{H}_{k_0^*:\infty}}^2 \right) + \frac{\sigma^2}{B}}{1 - \gamma\alpha_B\mathrm{tr}\,(\mathbf{H})}}_{\text{EffectiveVar}}$$

$$+ \underbrace{\frac{1}{\gamma^2 N^2} \cdot \|\mathbf{w}^*\|_{(\mathbf{H}_{0:k_0^*})^{-1}}^2 + \|\mathbf{w}^*\|_{\mathbf{H}_{k_0^*:\infty}}^2}_{\text{EffectiveBias}},$$

and by Corollary D.1,

$$\mathbb{E}[\mathcal{E}(\overline{\mathbf{w}}_N)] \lesssim \text{ApproxErr} + \text{VarErr} + \text{BiasErr},$$

where

$$\text{ApproxErr} \lesssim \frac{\epsilon_d}{\lambda_d + \epsilon_d} \|\mathbf{w}^*\|_{\mathbf{H}}^2, \quad \text{BiasErr} \lesssim \frac{1}{\gamma^2 N^2} \cdot \|\mathbf{w}^{(q)*}\|_{(\mathbf{H}_{0:k^*}^{(q)})^{-1}}^2 + \|\mathbf{w}^{(q)*}\|_{\mathbf{H}_{k^*:\infty}^{(q)}}^2,$$

$$\text{VarErr} \lesssim \frac{\alpha_B \left( \frac{\|\mathbf{w}^{(q)*}\|_{\mathbf{I}_{0:k^*}^{(q)}}^2}{N\gamma} + \|\mathbf{w}^{(q)*}\|_{\mathbf{H}_{k^*:\infty}^{(q)}}^2 \right) + \frac{\sigma^2+\epsilon_o+\epsilon_a}{B} + \alpha_B\epsilon_p[\mathrm{tr}(\mathbf{H}) + d\epsilon_d]}{1 - \gamma\alpha_B[\mathrm{tr}(\mathbf{H}) + d\epsilon_d]} \left( \frac{k^*}{N} + N\gamma^2 \cdot \sum_{i>k^*}(\lambda_i + \epsilon_d)^2 \right).$$

We then compare the upper bound of $\mathbb{E}[\mathcal{E}(\overline{\mathbf{w}}_N)]$ with $R_0$. Regarding VarErr, we first analyze $\frac{k^*}{N} + N\gamma^2 \cdot \sum_{i>k^*}(\lambda_i + \epsilon_d)^2$. Recall that for $k_0^* < i \leq k^*$, $\frac{1}{N\gamma} - \epsilon_d \leq \lambda_i < \frac{1}{N\gamma}$,

$$\frac{k^*}{N} + N\gamma^2 \sum_{i>k^*}(\lambda_i + \epsilon_d)^2 = \frac{k_0^*}{N} + \frac{k^* - k_0^*}{N} - N\gamma^2 \sum_{k_0^*<i\leq k^*}(\lambda_i + \epsilon_d)^2 + N\gamma^2 \sum_{i>k_0^*}(\lambda_i + \epsilon_d)^2$$

$$\leq \frac{k_0^*}{N} + N\gamma^2 \sum_{i>k_0^*}(\lambda_i + \epsilon_d)^2.$$

We then analyze $\frac{\|\mathbf{w}^{(q)*}\|_{\mathbf{I}_{0:k^*}^{(q)}}^2}{N\gamma} + \|\mathbf{w}^{(q)*}\|_{\mathbf{H}_{k^*:\infty}^{(q)}}^2$. Similarly,

$$\frac{\|\mathbf{w}^{(q)*}\|_{\mathbf{I}_{0:k^*}^{(q)}}^2}{N\gamma} + \|\mathbf{w}^{(q)*}\|_{\mathbf{H}_{k^*:\infty}^{(q)}}^2 \leq \frac{\|\mathbf{w}^*\|_{\mathbf{I}_{0:k_0^*}}^2}{N\gamma} + \|\mathbf{w}^*\|_{\mathbf{H}_{k_0^*:\infty}}^2. \tag{D.2}$$

Therefore, the sufficient conditions for VarErr $\lesssim$ EffectiveVar are

$$\epsilon_d \lesssim \sqrt{\frac{\frac{k_0^*}{N} + N\gamma^2 \cdot \sum_{i>k_0^*} \lambda_i^2}{N\gamma^2(d - k_0^*)}}, \quad \epsilon_p \lesssim \frac{\sigma^2}{B\alpha_B[\mathrm{tr}(\mathbf{H}) + d\epsilon_d]} + \frac{\frac{\|\mathbf{w}^*\|_{\mathbf{I}_{0:k_0^*}}^2}{N\gamma} + \|\mathbf{w}^*\|_{\mathbf{H}_{k_0^*:\infty}}^2}{\mathrm{tr}(\mathbf{H}) + d\epsilon_d},$$

$$\epsilon_a, \epsilon_o \lesssim \sigma^2 + B\alpha_B \left( \frac{\|\mathbf{w}^*\|_{\mathbf{I}_{0:k_0^*}}^2}{N\gamma} + \|\mathbf{w}^*\|_{\mathbf{H}_{k_0^*:\infty}}^2 \right).$$

Secondly, we analyze BiasErr. Similarly,

$$
\begin{aligned}
&\frac{1}{\gamma^2 N^2} \cdot \|\mathbf{w}^{(q)^*}\|^2_{(\mathbf{H}^{(q)}_{0:k^*})^{-1}} + \|\mathbf{w}^{(q)^*}\|^2_{\mathbf{H}^{(q)}_{k^*:\infty}} \\
=&\frac{1}{\gamma^2 N^2} \cdot \left( \|\mathbf{w}^{(q)^*}\|^2_{(\mathbf{H}^{(q)}_{0:k_0^*})^{-1}} + \|\mathbf{w}^{(q)^*}\|^2_{(\mathbf{H}^{(q)}_{k_0^*:k^*})^{-1}} \right) - \|\mathbf{w}^{(q)^*}\|^2_{\mathbf{H}^{(q)}_{k_0^*:k^*}} + \|\mathbf{w}^{(q)^*}\|^2_{\mathbf{H}^{(q)}_{k_0^*:\infty}} \\
\leq&\frac{1}{\gamma^2 N^2} \cdot \|\mathbf{w}^{(q)^*}\|^2_{(\mathbf{H}^{(q)}_{0:k_0^*})^{-1}} + \|\mathbf{w}^{(q)^*}\|^2_{\mathbf{H}^{(q)}_{k_0^*:\infty}} \\
\leq&\frac{1}{\gamma^2 N^2} \cdot \|\mathbf{w}^*\|^2_{(\mathbf{H}_{0:k_0^*})^{-1}} + \|\mathbf{w}^*\|^2_{\mathbf{H}_{k_0^*:\infty}} = \text{EffectiveBias}.
\end{aligned}
$$

Thirdly, the sufficient condition for $\text{ApproxErr} \lesssim R_0$ is $\epsilon_d \lesssim \frac{R_0}{\|\mathbf{w}^*\|^2_{\mathbf{H}}} \lambda_d$. Overall, we require

$$
\epsilon_d \lesssim \sqrt{\frac{\frac{k_0^*}{N} + N\gamma^2 \cdot \sum_{i>k_0^*} \lambda_i^2}{N\gamma^2(d - k_0^*)}} \wedge \frac{R_0 \lambda_d}{\|\mathbf{w}^*\|^2_{\mathbf{H}}}, \quad \epsilon_a, \epsilon_o \lesssim \sigma^2 + B\alpha_B \left( \frac{\|\mathbf{w}^*\|^2_{\mathbf{I}_{0:k_0^*}}}{N\gamma} + \|\mathbf{w}^*\|^2_{\mathbf{H}_{k_0^*:\infty}} \right),
$$

$$
\epsilon_p \lesssim \frac{\sigma^2}{B\alpha_B[\text{tr}(\mathbf{H}) + d\epsilon_d]} + \frac{\frac{\|\mathbf{w}^*\|^2_{\mathbf{I}_{0:k_0^*}}}{N\gamma} + \|\mathbf{w}^*\|^2_{\mathbf{H}_{k_0^*:\infty}}}{\text{tr}(\mathbf{H}) + d\epsilon_d}.
$$

$\square$

## D.6 Proof for the Multiplicative Statement in Corollary 4.3

*Proof.* We prove by applying Theorem 4.2:

$$
\mathbb{E}[\mathcal{E}(\overline{\mathbf{w}}_N)] \lesssim \text{ApproxErr} + \text{VarErr} + \text{BiasErr},
$$

where

$$
\text{ApproxErr} \lesssim \frac{\epsilon_d}{1 + \epsilon_d} \|\mathbf{w}^*\|^2_{\mathbf{H}}, \quad \text{BiasErr} \lesssim \frac{1}{\gamma^2 N^2} \cdot \|\mathbf{w}^{(q)^*}\|^2_{(\mathbf{H}^{(q)}_{0:k^*})^{-1}} + \|\mathbf{w}^{(q)^*}\|^2_{\mathbf{H}^{(q)}_{k^*:\infty}},
$$

$$
\text{VarErr} \lesssim \left( \frac{k^*}{N} + N\gamma^2(1 + \epsilon_d)^2 \sum_{i>k^*} \lambda_i^2 \right) \frac{\frac{(1+\epsilon_o)\sigma^2}{B} + \alpha_B(1 + \epsilon_o)[1 + \epsilon_p + \epsilon_a(1 + \epsilon_p)]\|\mathbf{w}^*\|^2_{\mathbf{H}}}{1 - \gamma\alpha_B(1 + \epsilon_o)[1 + \epsilon_p + \epsilon_a(1 + \epsilon_p)](1 + \epsilon_d)\text{tr}(\mathbf{H})}.
$$

We first deal with VarErr under power-law spectrum Assumption 4.1. Under multiplicative quantization, we can estimate $k^*$ by

$$
(1 + \epsilon_d)k^{*-a} \approx \frac{1}{N\gamma},
$$

that is

$$
k^* \approx [N\gamma(1 + \epsilon_d)]^{\frac{1}{a}}. \tag{D.3}
$$

Further, the power-law Assumption 4.1 also implies that for any positive $k$,

$$
\sum_{i>k} i^{-a} \approx k^{1-a}. \tag{D.4}
$$

By (D.3) and (D.4),

$$
\frac{k^*}{N} + N\gamma^2(1 + \epsilon_d)^2 \sum_{i>k^*} \lambda_i^2 \lesssim \frac{\min\left\{ d, [N\gamma(1 + \epsilon_d)]^{\frac{1}{a}} + N\gamma^2(1 + \epsilon_d)^2 [N\gamma(1 + \epsilon_d)]^{\frac{1-2a}{a}} \right\}}{N}
$$

$$
\lesssim \frac{\min\left\{ d, [N\gamma(1 + \epsilon_d)]^{\frac{1}{a}} \right\}}{N}.
$$

Moreover, under polynomial spectrum Assumption 4.1,

$$
\text{tr}(\mathbf{H}) \asymp 1, \quad \mathbb{E}\|\mathbf{w}^*\|^2_{\mathbf{H}} \asymp 1.
$$

Therefore, under Assumption 4.1, by applying stepsize $\gamma < \frac{1}{2\alpha_B(1+\epsilon_o)[1+\epsilon_p+\epsilon_a(1+\epsilon_p)](1+\epsilon_d)\mathrm{tr}(\mathbf{H})}$ and taking expectation on $\mathbf{w}^*$, it holds that

$$\mathbb{E}_{\mathbf{w}^*}\mathrm{VarErr} \lesssim \frac{\min\left\{d, [N\gamma(1+\epsilon_d)]^{\frac{1}{a}}\right\}}{N}(1+\epsilon_o)[1+\epsilon_p+\epsilon_a(1+\epsilon_p)], \qquad (D.5)$$

where we use $\sigma^2 \lesssim 1$.

We secondly deal with BiasErr. Under Assumption 4.1, using (D.1),

$$\begin{aligned}
\mathbb{E}_{\mathbf{w}^*}\mathrm{BiasErr} &\leq \mathbb{E}_{\mathbf{w}^*}\left[\frac{1}{\gamma^2 N^2}\cdot\|\mathbf{w}^*\|_{(\mathbf{H}_{0:k_0^*})^{-1}}^2 + \|\mathbf{w}^*\|_{\mathbf{H}_{k_0^*:\infty}}^2\right] \\
&= \frac{1}{N^2\gamma^2}\sum_{i=1}^{k_0^*}\lambda_i^{-1} + \sum_{i>k_0^*}^d \lambda_i \\
&\leq \frac{k_0^*}{N\gamma} + \sum_{i>k_0^*}^d \lambda_i \\
&\asymp \frac{k_0^*}{N\gamma} + (k_0^*)^{1-a} \\
&\lesssim \max\left\{d^{1-a}, (N\gamma)^{1/a-1}\right\}.
\end{aligned} \qquad (D.6)$$

Therefore, together with (D.5) and (D.6), and taking expectation on $\mathbf{w}^*$, we have

$$\mathbb{E}[\mathcal{E}(\overline{\mathbf{w}}_N)] \lesssim \frac{\epsilon_d}{1+\epsilon_d} + \max\left\{d^{1-a}, (N\gamma)^{1/a-1}\right\} + \frac{\min\left\{d, [N\gamma(1+\epsilon_d)]^{\frac{1}{a}}\right\}}{N}(1+\epsilon_o)[1+\epsilon_p+\epsilon_a(1+\epsilon_p)].$$

Denote $R = \mathbb{E}[\mathcal{E}(\overline{\mathbf{w}}_N)] - \frac{\epsilon_d}{1+\epsilon_d}$.

- $d > [N\gamma(1+\epsilon_d)]^{\frac{1}{a}}$
  In this case,

$$\begin{aligned}
R &\lesssim (N\gamma)^{1/a-1} + \frac{[N\gamma(1+\epsilon_d)]^{\frac{1}{a}}}{N}(1+\epsilon_o)[1+\epsilon_p+\epsilon_a(1+\epsilon_p)] \\
&\lesssim N^{1/a-1}(1+\epsilon_o)[1+\epsilon_p+\epsilon_a(1+\epsilon_p)](1+\epsilon_d)^{1/a}.
\end{aligned}$$

- $(N\gamma)^{1/a} < d \leq [N\gamma(1+\epsilon_d)]^{\frac{1}{a}}$
  In this case,

$$\begin{aligned}
R &\lesssim (N\gamma)^{1/a-1} + (1+\epsilon_o)[1+\epsilon_p+\epsilon_a(1+\epsilon_p)]\frac{d}{N} \\
&\lesssim (N\gamma)^{1/a-1} + (1+\epsilon_o)[1+\epsilon_p+\epsilon_a(1+\epsilon_p)]\frac{[N\gamma(1+\epsilon_d)]^{\frac{1}{a}}}{N} \\
&\lesssim N^{1/a-1}(1+\epsilon_o)[1+\epsilon_p+\epsilon_a(1+\epsilon_p)](1+\epsilon_d)^{1/a}.
\end{aligned}$$

- $d \leq (N\gamma)^{1/a}$
  In this case,

$$\begin{aligned}
R &\lesssim d^{1-a} + (1+\epsilon_o)[1+\epsilon_p+\epsilon_a(1+\epsilon_p)]\frac{d}{N} \\
&\lesssim d^{1-a} + (1+\epsilon_o)[1+\epsilon_p+\epsilon_a(1+\epsilon_p)]N^{1/a-1}.
\end{aligned}$$

Overall,

$$\mathbb{E}\left[\mathcal{E}(\overline{\mathbf{w}}_N)\right] \lesssim \frac{\epsilon_d}{1+\epsilon_d} + d^{1-a} + N^{1/a-1}(1+\epsilon_o)[1+\epsilon_p+\epsilon_a(1+\epsilon_p)](1+\epsilon_d)^{1/a}.$$

$\square$

### D.7 PROOF FOR THE ADDITIVE STATEMENT IN COROLLARY 4.3

*Proof.* We prove by applying Corollary 4.1:

$$\mathbb{E}[\mathcal{E}(\overline{\mathbf{w}}_N)] \lesssim \text{ApproxErr} + \text{VarErr} + \text{BiasErr},$$

where

$$\text{ApproxErr} \lesssim \frac{\epsilon_d}{\lambda_d + \epsilon_d} \|\mathbf{w}^*\|_{\mathbf{H}}^2, \quad \text{BiasErr} \lesssim \frac{1}{\gamma^2 N^2} \cdot \|\mathbf{w}^{(q)*}\|_{(\mathbf{H}_{0:k^*}^{(q)})^{-1}}^2 + \|\mathbf{w}^{(q)*}\|_{\mathbf{H}_{k^*:\infty}^{(q)}}^2,$$

$$\text{VarErr} \lesssim \frac{\alpha_B \|\mathbf{w}^*\|_{\mathbf{H}}^2 + \frac{\sigma^2 + \epsilon_o + \epsilon_a}{B} + \alpha_B \epsilon_p [\text{tr}(\mathbf{H}) + p\epsilon_d]}{1 - \gamma \alpha_B [\text{tr}(\mathbf{H}) + p\epsilon_d]} \left( \frac{k^*}{N} + N\gamma^2 \cdot \sum_{i>k^*} (\lambda_i + \epsilon_d)^2 \right).$$

We first deal with BiasErr. Under Assumption 4.1, using (D.2),

$$\begin{aligned}
\mathbb{E}_{\mathbf{w}^*} \text{BiasErr} &\leq \mathbb{E}_{\mathbf{w}^*} \left[ \frac{1}{\gamma^2 N^2} \cdot \|\mathbf{w}^*\|_{(\mathbf{H}_{0:k_0^*})^{-1}}^2 + \|\mathbf{w}^*\|_{\mathbf{H}_{k_0^*:\infty}}^2 \right] \\
&= \frac{1}{N^2 \gamma^2} \sum_{i=1}^{k_0^*} \lambda_i^{-1} + \sum_{i>k_0^*}^{d} \lambda_i \\
&\leq \frac{k_0^*}{N\gamma} + \sum_{i>k_0^*}^{d} \lambda_i \\
&= \frac{k_0^*}{N\gamma} + (k_0^*)^{1-a} \\
&\lesssim \max\left\{ d^{1-a}, (N\gamma)^{1/a-1} \right\}.
\end{aligned} \tag{D.7}$$

We then analyze VarErr. If $\epsilon_d + d^{-a} \geq \frac{1}{N\gamma}$, then

$$\frac{k^*}{N} + N\gamma^2 \cdot \sum_{i>k^*} (\lambda_i + \epsilon_d)^2 = \frac{d}{N}.$$

Otherwise,

$$\begin{aligned}
&\frac{k^*}{N} + N\gamma^2 \cdot \sum_{i>k^*} (\lambda_i + \epsilon_d)^2 \\
&\lesssim \frac{\left( \frac{1}{N\gamma} - \epsilon_d \right)^{-1/a} + N^2\gamma^2 \left( \frac{1}{N\gamma} - \epsilon_d \right)^{-(1-2a)/a} + \epsilon_d^2 N^2\gamma^2 \left[ d - \left( \frac{1}{N\gamma} - \epsilon_d \right)^{-1/a} \right]}{N} \\
&\lesssim \frac{\left( \frac{1}{N\gamma} - \epsilon_d \right)^{-1/a} + \epsilon_d^2 N^2\gamma^2 \left[ d - \left( \frac{1}{N\gamma} - \epsilon_d \right)^{-1/a} \right]}{N}.
\end{aligned}$$

Denote $k_{\text{eff}} = \left[ d^{-a} \vee \left( \frac{1}{N\gamma} - \epsilon_d \right) \right]^{-1/a}$, it follows that

$$\frac{k^*}{N} + N\gamma^2 \cdot \sum_{i>k^*} (\lambda_i + \epsilon_d)^2 \lesssim \frac{k_{\text{eff}} + \epsilon_d^2 N^2\gamma^2 (d - k_{\text{eff}})}{N}.$$

Hence, under Assumption 4.1, taking expectation on $\mathbf{w}^*$ and applying stepsize $\gamma < \frac{1}{2\alpha_B [\text{tr}(\mathbf{H}) + p\epsilon_d]}$,

$$\mathbb{E}_{\mathbf{w}^*} \text{VarErr} \lesssim \left( 1 + \frac{\epsilon_o + \epsilon_a}{B} + \epsilon_p (1 + d\epsilon_d) \right) \frac{k_{\text{eff}} + \epsilon_d^2 N^2\gamma^2 (d - k_{\text{eff}})}{N}. \tag{D.8}$$

Therefore, together with (D.7) and (D.8), and taking expectation on $\mathbf{w}^*$, we have

$$\mathbb{E}[\mathcal{E}(\overline{\mathbf{w}}_N)] \lesssim \frac{\epsilon_d}{d^{-a} + \epsilon_d} + \max\left\{ d^{1-a}, (N\gamma)^{1/a-1} \right\} + \left( 1 + \frac{\epsilon_o + \epsilon_a}{B} + \epsilon_p (1 + d\epsilon_d) \right) \frac{k_{\text{eff}} + \epsilon_d^2 N^2\gamma^2 (d - k_{\text{eff}})}{N}.$$

Denote $R = \mathbb{E}[\mathcal{E}(\overline{\mathbf{w}}_N)] - \frac{\epsilon_d}{d^{-a} + \epsilon_d}$.

- $d^{-a} \le 1/(N\gamma) - \epsilon_d$

In this case, let $\epsilon'_d = d^a \epsilon_d$. Then $d^{-a} \le 1/(N\gamma) - d^{-a}\epsilon'_d$. That is, $d^{-a} \le \frac{1}{N\gamma(1+\epsilon'_d)}$.

$$R \lesssim (N\gamma)^{1/a-1} + \left(1 + \frac{\epsilon_o + \epsilon_a}{B} + \epsilon_p(1 + d\epsilon_d)\right) \frac{\left(\frac{1}{N\gamma} - \epsilon_d\right)^{-1/a} + \epsilon_d^2 N^2 \gamma^2 \left(d - \left(\frac{1}{N\gamma} - \epsilon_d\right)^{-1/a}\right)}{N}$$

$$= (N\gamma)^{1/a-1} + \left(1 + \frac{\epsilon_o + \epsilon_a}{B} + \epsilon_p(1 + d\epsilon_d)\right) \frac{\left(\frac{1}{N\gamma} - d^{-a}\epsilon'_d\right)^{-1/a}(1 - \epsilon_d^2 N^2 \gamma^2) + \epsilon_d^2 N^2 \gamma^2 d}{N}$$

$$\le (N\gamma)^{1/a-1} + \left(1 + \frac{\epsilon_o + \epsilon_a}{B} + \epsilon_p(1 + d\epsilon_d)\right) \frac{(1 + \epsilon'_d)^{1/a}(N\gamma)^{1/a}(1 - \epsilon_d^2 N^2 \gamma^2) + \epsilon_d^2 N^2 \gamma^2 d}{N}$$

$$\le (N\gamma)^{1/a-1} + \left(1 + \frac{\epsilon_o + \epsilon_a}{B} + \epsilon_p(1 + d\epsilon_d)\right) \left[\frac{(1 + \epsilon'_d)^{1/a}(N\gamma)^{1/a}}{N} + \epsilon_d^2 N\gamma^2 d\right].$$

We then focus on $\epsilon_d^2 N\gamma^2 d$. By $N\gamma \le \frac{d^a}{1+\epsilon'_d}$, we have

$$\epsilon_d^2 N\gamma^2 d \lesssim \frac{\epsilon_d^2}{1 + \epsilon'_d} d^{1+a} = \frac{(\epsilon'_d)^2}{1 + \epsilon'_d} d^{1-a}.$$

Therefore,

$$R \lesssim N^{1/a-1}(1 + d^a \epsilon_d)^{1/a} \left(1 + \frac{\epsilon_o + \epsilon_a}{B} + \epsilon_p(1 + d\epsilon_d)\right) + \left(1 + \frac{\epsilon_o + \epsilon_a}{B} + \epsilon_p(1 + d\epsilon_d)\right) \frac{(d^a \epsilon_d)^2}{1 + d^a \epsilon_d} d^{1-a}.$$

- $\frac{1}{N\gamma} - \epsilon_d < d^{-a} \le \frac{1}{N\gamma}$

In this case, $\frac{1}{N\gamma} - d^{-a}\epsilon'_d < d^{-a}$, that is, $d^a < N\gamma(1 + \epsilon'_d)$. Consequently,

$$R \lesssim (N\gamma)^{1/a-1} + \left(1 + \frac{\epsilon_o + \epsilon_a}{B} + \epsilon_p(1 + d\epsilon_d)\right) \frac{d}{N}$$

$$\lesssim (N\gamma)^{1/a-1} + \left(1 + \frac{\epsilon_o + \epsilon_a}{B} + \epsilon_p(1 + d\epsilon_d)\right) (1 + \epsilon'_d)^{1/a} N^{1/a-1}$$

$$\lesssim \left(1 + \frac{\epsilon_o + \epsilon_a}{B} + \epsilon_p(1 + d\epsilon_d)\right) (1 + d^a \epsilon_d)^{1/a} N^{1/a-1}.$$

- $d^{-a} > \frac{1}{N\gamma}$

In this case,

$$R \lesssim d^{1-a} + \left(1 + \frac{\epsilon_o + \epsilon_a}{B} + \epsilon_p(1 + d\epsilon_d)\right) \frac{d}{N}$$

$$\lesssim d^{1-a} + \left(1 + \frac{\epsilon_o + \epsilon_a}{B} + \epsilon_p(1 + d\epsilon_d)\right) N^{1/a-1}.$$

Overall,

$$\mathbb{E}[\mathcal{E}(\overline{\mathbf{w}}_N)] \lesssim \frac{d^a \epsilon_d}{1 + d^a \epsilon_d} + \left(1 + \frac{\epsilon_o + \epsilon_a}{B} + \epsilon_p(1 + d\epsilon_d)\right) \left[N^{\frac{1}{a}-1}(1 + d^a \epsilon_d)^{\frac{1}{a}} + d^{1-a}\left(1 + \frac{(d^a \epsilon_d)^2}{1 + d^a \epsilon_d}\right)\right].$$

$\square$

## E   DISCUSSION OF ASSUMPTIONS

In this section, we verify Assumption 3.3 and Assumption 3.4 under the standard fourth moment and noise assumptions made on the full-precision data (Zou et al., 2023).

**Assumption E.1.** *Assume there exists a positive constant $\alpha_0 > 0$, such that for any PSD matrix $\mathbf{A}$, it holds that*

$$\mathbb{E}\left[\mathbf{x}\mathbf{x}^\top \mathbf{A} \mathbf{x}\mathbf{x}^\top\right] \preceq \alpha_0 \operatorname{tr}(\mathbf{H}\mathbf{A})\mathbf{H}.$$

**Assumption E.2.** *Assume there exists a constant $\sigma_0^2$ such that*

$$\mathbb{E}\left[(y - \langle \mathbf{w}^*, \mathbf{x}\rangle)^2 \mathbf{x}\mathbf{x}^\top\right] \preceq \sigma_0^2 \mathbf{H}.$$

We consider specific quantization schemes.

**Example E.1** (**Strong multiplicative quantization**). *We consider a strong multiplicative quantization. In this case, there exist constants $\epsilon_d, \epsilon_d'$ such that*

$$\mathbb{E}\left[\boldsymbol{\epsilon}^{(d)}\boldsymbol{\epsilon}^{(d)\top}\Big|\mathbf{x}\right] = \epsilon_d \mathbf{x}\mathbf{x}^\top, \quad \mathbb{E}\left[\boldsymbol{\epsilon}^{(d)}\boldsymbol{\epsilon}^{(d)\top}\mathbf{A}\boldsymbol{\epsilon}^{(d)}\boldsymbol{\epsilon}^{(d)\top}\Big|\mathbf{x}\right] \preceq \epsilon_d' \mathbf{x}\mathbf{x}^\top \mathbf{A}\mathbf{x}\mathbf{x}^\top.$$

**Example E.2** (**Strong additive quantization**). *We consider a strong additive quantization. In this case, there exist constants $\epsilon_d, \epsilon_d'$ such that*

$$\mathbb{E}\left[\boldsymbol{\epsilon}^{(d)}\boldsymbol{\epsilon}^{(d)\top}\Big|\mathbf{x}\right] = \epsilon_d \mathbf{I}, \quad \mathbb{E}\left[\boldsymbol{\epsilon}^{(d)}\boldsymbol{\epsilon}^{(d)\top}\mathbf{A}\boldsymbol{\epsilon}^{(d)}\boldsymbol{\epsilon}^{(d)\top}\Big|\mathbf{x}\right] \preceq \epsilon_d' \mathrm{tr}(\mathbf{A})\mathbf{I}. \tag{E.1}$$

### E.1  Discussion of Assumption 3.3

Under Assumption E.1, we are ready to verify if Assumption 3.3 can be satisfied. We begin by:

$$
\begin{aligned}
\mathbb{E}\left[\mathbf{x}^{(q)}\mathbf{x}^{(q)\top}\mathbf{A}\mathbf{x}^{(q)}\mathbf{x}^{(q)\top}\right] =& \mathbb{E}\left[\left(\mathbf{x}^{(q)\top}\mathbf{A}\mathbf{x}^{(q)}\right)\mathbf{x}^{(q)}\mathbf{x}^{(q)\top}\right] \\
\preceq& 2\mathbb{E}\left[\left(\mathbf{x}^{(q)\top}\mathbf{A}\mathbf{x}^{(q)}\right)\left(\mathbf{x}\mathbf{x}^\top + \boldsymbol{\epsilon}^{(d)}\boldsymbol{\epsilon}^{(d)\top}\right)\right] \\
\preceq& 4\mathbb{E}\left[\left(\mathbf{x}^\top\mathbf{A}\mathbf{x} + \boldsymbol{\epsilon}^{(d)\top}\mathbf{A}\boldsymbol{\epsilon}^{(d)}\right)\left(\mathbf{x}\mathbf{x}^\top + \boldsymbol{\epsilon}^{(d)}\boldsymbol{\epsilon}^{(d)\top}\right)\right] \\
=& 4\mathbb{E}\left[\mathbf{x}\mathbf{x}^\top\mathbf{A}\mathbf{x}\mathbf{x}^\top\right] + 4\mathbb{E}\left[\boldsymbol{\epsilon}^{(d)}\boldsymbol{\epsilon}^{(d)\top}\mathbf{A}\boldsymbol{\epsilon}^{(d)}\boldsymbol{\epsilon}^{(d)\top}\right] \\
&+ 4\mathbb{E}\left[\left(\mathbf{x}^\top\mathbf{A}\mathbf{x}\right)\boldsymbol{\epsilon}^{(d)}\boldsymbol{\epsilon}^{(d)\top}\right] + 4\mathbb{E}\left[\left(\boldsymbol{\epsilon}^{(d)\top}\mathbf{A}\boldsymbol{\epsilon}^{(d)}\right)\mathbf{x}\mathbf{x}^\top\right].
\end{aligned}
\tag{E.2}
$$

**Lemma E.1.** *Under strong multiplicative quantization E.1 and Assumption E.1,*

$$\mathbb{E}\left[\mathbf{x}^{(q)}\mathbf{x}^{(q)\top}\mathbf{A}\mathbf{x}^{(q)}\mathbf{x}^{(q)\top}\right] \lesssim \alpha_0(1 + \epsilon_d + \epsilon_d')\mathrm{tr}(\mathbf{H}^{(q)}\mathbf{A})\mathbf{H}^{(q)}.$$

*Proof.* We proof by (E.2). From Assumption E.1,

$$\mathbb{E}\left[\mathbf{x}\mathbf{x}^\top\mathbf{A}\mathbf{x}\mathbf{x}^\top\right] \preceq \alpha_0 \, \mathrm{tr}(\mathbf{H}\mathbf{A})\mathbf{H}. \tag{E.3}$$

Under strong multiplicative quantization E.1, we have

$$\mathbb{E}\left[\left(\boldsymbol{\epsilon}^{(d)\top}\mathbf{A}\boldsymbol{\epsilon}^{(d)}\right)\mathbf{x}\mathbf{x}^\top\right] = \epsilon_d\mathbb{E}\left[\mathbf{x}\mathbf{x}^\top\mathbf{A}\mathbf{x}\mathbf{x}^\top\right] \preceq \epsilon_d\alpha_0 \, \mathrm{tr}(\mathbf{H}\mathbf{A})\mathbf{H}, \tag{E.4}$$

$$\mathbb{E}\left[\left(\mathbf{x}^\top\mathbf{A}\mathbf{x}\right)\boldsymbol{\epsilon}^{(d)}\boldsymbol{\epsilon}^{(d)\top}\right] = \epsilon_d\mathbb{E}\left[\mathbf{x}\mathbf{x}^\top\mathbf{A}\mathbf{x}\mathbf{x}^\top\right] \preceq \epsilon_d\alpha_0 \, \mathrm{tr}(\mathbf{H}\mathbf{A})\mathbf{H}, \tag{E.5}$$

and

$$\mathbb{E}\left[\boldsymbol{\epsilon}^{(d)}\boldsymbol{\epsilon}^{(d)\top}\mathbf{A}\boldsymbol{\epsilon}^{(d)}\boldsymbol{\epsilon}^{(d)\top}\right] \preceq \epsilon_d'\mathbb{E}\left[\mathbf{x}\mathbf{x}^\top\mathbf{A}\mathbf{x}\mathbf{x}^\top\right] \preceq \epsilon_d'\alpha_0 \, \mathrm{tr}(\mathbf{H}\mathbf{A})\mathbf{H}. \tag{E.6}$$

Therefore, together with (E.2), (E.3), (E.4), (E.5) and (E.6), we have

$$\mathbb{E}\left[\mathbf{x}^{(q)}\mathbf{x}^{(q)\top}\mathbf{A}\mathbf{x}^{(q)}\mathbf{x}^{(q)\top}\right] \lesssim \alpha_0(1 + \epsilon_d + \epsilon_d')\mathrm{tr}(\mathbf{H}\mathbf{A})\mathbf{H} \leq \alpha_0(1 + \epsilon_d + \epsilon_d')\mathrm{tr}(\mathbf{H}^{(q)}\mathbf{A})\mathbf{H}^{(q)}.$$

That is, under strong multiplicative quantization Example E.1 and fourth moment Assumption E.1 on full-precision data, Assumption 3.3 is verified. $\square$

**Lemma E.2.** *Under strong additive quantization E.2 and Assumption E.1,*

$$\mathbb{E}\left[\mathbf{x}^{(q)}\mathbf{x}^{(q)\top}\mathbf{A}\mathbf{x}^{(q)}\mathbf{x}^{(q)\top}\right] \lesssim (1 + \alpha_0)\left(1 + \frac{\epsilon_d'}{\epsilon_d^2}\right)\mathrm{tr}(\mathbf{H}^{(q)}\mathbf{A})\mathbf{H}^{(q)}.$$

*Proof.* We proof by (E.2). Under strong additive quantization E.2,

$$\mathbb{E}\left[\left(\boldsymbol{\epsilon}^{(d)\top}\mathbf{A}\boldsymbol{\epsilon}^{(d)}\right)\mathbf{x}\mathbf{x}^{\top}\right] \preceq \epsilon_d \operatorname{tr}(\mathbf{A})\mathbf{H}, \tag{E.7}$$

$$\mathbb{E}\left[\left(\mathbf{x}^{\top}\mathbf{A}\mathbf{x}\right)\boldsymbol{\epsilon}^{(d)}\boldsymbol{\epsilon}^{(d)\top}\right] \preceq \epsilon_d \operatorname{tr}(\mathbf{H}\mathbf{A})\mathbf{I}, \tag{E.8}$$

and

$$\mathbb{E}\left[\boldsymbol{\epsilon}^{(d)}\boldsymbol{\epsilon}^{(d)\top}\mathbf{A}\boldsymbol{\epsilon}^{(d)}\boldsymbol{\epsilon}^{(d)\top}\right] \preceq \epsilon_d' \operatorname{tr}(\mathbf{A})\mathbf{I}. \tag{E.9}$$

Therefore, together with (E.2), (E.7), (E.8) and (E.9), we have

$$\mathbb{E}\left[\mathbf{x}^{(q)}\mathbf{x}^{(q)\top}\mathbf{A}\mathbf{x}^{(q)}\mathbf{x}^{(q)\top}\right] \lesssim (1+\alpha_0)\left(1+\frac{\epsilon_d'}{\epsilon_d^2}\right)\operatorname{tr}(\mathbf{H}^{(q)}\mathbf{A})\mathbf{H}^{(q)}.$$

That is, under strong additive quantization Example E.2 and fourth moment Assumption E.1 on full-precision data, Assumption 3.3 is verified. □

## E.2 DISCUSSION OF ASSUMPTION 3.4

Under Assumption E.2, we are ready to verify if Assumption 3.4 can be satisfied. We begin by:

$$\begin{aligned}
&\mathbb{E}\left[(y^{(q)} - \langle\mathbf{w}^{(q)*},\mathbf{x}^{(q)}\rangle)^2\mathbf{x}^{(q)}\mathbf{x}^{(q)\top}\right] \\
&=\mathbb{E}\left[(y^{(q)} - y + y - \langle\mathbf{w}^*,\mathbf{x}\rangle + \langle\mathbf{w}^*,\mathbf{x}\rangle - \langle\mathbf{w}^{(q)*},\mathbf{x}^{(q)}\rangle)^2\mathbf{x}^{(q)}\mathbf{x}^{(q)\top}\right] \\
&\preceq 3\mathbb{E}\left[(y^{(q)} - y)^2\mathbf{x}^{(q)}\mathbf{x}^{(q)\top}\right] + 3\mathbb{E}\left[(y - \langle\mathbf{w}^*,\mathbf{x}\rangle)^2\mathbf{x}^{(q)}\mathbf{x}^{(q)\top}\right] \\
&\quad +3\mathbb{E}\left[(\langle\mathbf{w}^*,\mathbf{x}\rangle - \langle\mathbf{w}^{(q)*},\mathbf{x}^{(q)}\rangle)^2\mathbf{x}^{(q)}\mathbf{x}^{(q)\top}\right] \\
&\preceq 3\mathbb{E}\left[(y^{(q)} - y)^2\mathbf{x}^{(q)}\mathbf{x}^{(q)\top}\right] + 3\mathbb{E}\left[(y - \langle\mathbf{w}^*,\mathbf{x}\rangle)^2\mathbf{x}^{(q)}\mathbf{x}^{(q)\top}\right] \\
&\quad +6\mathbb{E}\left[\langle\mathbf{w}^{(q)*} - \mathbf{w}^*,\mathbf{x}\rangle^2\mathbf{x}^{(q)}\mathbf{x}^{(q)\top}\right] + 6\mathbb{E}\left[\langle\mathbf{w}^{(q)*},\boldsymbol{\epsilon}^{(d)}\rangle^2\mathbf{x}^{(q)}\mathbf{x}^{(q)\top}\right].
\end{aligned} \tag{E.10}$$

**Lemma E.3.** *Under strong multiplicative quantization E.1, Assumption E.1, and Assumption E.2,*

$$\mathbb{E}\left[(y^{(q)} - \langle\mathbf{w}^{(q)*},\mathbf{x}^{(q)}\rangle)^2\mathbf{x}^{(q)}\mathbf{x}^{(q)\top}\right] \precsim \left(\sigma_0^2 + \epsilon_l + \frac{1+\epsilon_d+\epsilon_d'}{1+\epsilon_d}\alpha_0\|\mathbf{w}^*\|_{\mathbf{H}}^2\right)\mathbf{H}^{(q)}.$$

*Proof.* Regarding $\mathbb{E}\left[(y - \langle\mathbf{w}^*,\mathbf{x}\rangle)^2\mathbf{x}^{(q)}\mathbf{x}^{(q)\top}\right]$,

$$\begin{aligned}
\mathbb{E}\left[(y - \langle\mathbf{w}^*,\mathbf{x}\rangle)^2\mathbf{x}^{(q)}\mathbf{x}^{(q)\top}\right] &\preceq 2\mathbb{E}\left[(y - \langle\mathbf{w}^*,\mathbf{x}\rangle)^2\mathbf{x}\mathbf{x}^{\top}\right] + 2\mathbb{E}\left[(y - \langle\mathbf{w}^*,\mathbf{x}\rangle)^2\boldsymbol{\epsilon}^{(d)}\boldsymbol{\epsilon}^{(d)\top}\right] \\
&\preceq 2(1+\epsilon_d)\mathbb{E}\left[(y - \langle\mathbf{w}^*,\mathbf{x}\rangle)^2\mathbf{x}\mathbf{x}^{\top}\right] \\
&\preceq 2(1+\epsilon_d)\sigma_0^2\mathbf{H},
\end{aligned} \tag{E.11}$$

where the second inequality holds by the definition of Example E.1 and the last inequality holds by Assumption E.2. Regarding $\mathbb{E}\left[\langle\mathbf{w}^{(q)*},\boldsymbol{\epsilon}^{(d)}\rangle^2\mathbf{x}^{(q)}\mathbf{x}^{(q)\top}\right]$,

$$\begin{aligned}
&\mathbb{E}\left[\langle\mathbf{w}^{(q)*},\boldsymbol{\epsilon}^{(d)}\rangle^2\mathbf{x}^{(q)}\mathbf{x}^{(q)\top}\right] \\
&=\mathbb{E}\left[\boldsymbol{\epsilon}^{(d)\top}\mathbf{w}^{(q)*}\mathbf{w}^{(q)*\top}\boldsymbol{\epsilon}^{(d)}\mathbf{x}^{(q)}\mathbf{x}^{(q)\top}\right] \\
&\preceq 2\mathbb{E}\left[\boldsymbol{\epsilon}^{(d)\top}\mathbf{w}^{(q)*}\mathbf{w}^{(q)*\top}\boldsymbol{\epsilon}^{(d)}\mathbf{x}\mathbf{x}^{\top}\right] + 2\mathbb{E}\left[\boldsymbol{\epsilon}^{(d)\top}\mathbf{w}^{(q)*}\mathbf{w}^{(q)*\top}\boldsymbol{\epsilon}^{(d)}\boldsymbol{\epsilon}^{(d)}\boldsymbol{\epsilon}^{(d)\top}\right] \\
&\preceq 2\epsilon_d\alpha_0\operatorname{tr}(\mathbf{w}^{(q)*}\mathbf{w}^{(q)*\top}\mathbf{H})\mathbf{H} + 2\epsilon_d'\alpha_0\operatorname{tr}(\mathbf{w}^{(q)*}\mathbf{w}^{(q)*\top}\mathbf{H})\mathbf{H},
\end{aligned} \tag{E.12}$$

where the last inequality holds by the definition of Example E.1 and Assumption E.1. Regarding the term

$$\mathbb{E}\left[\langle \mathbf{w}^{(q)^*} - \mathbf{w}^*, \mathbf{x}\rangle^2 \mathbf{x}^{(q)}\mathbf{x}^{(q)^\top}\right] \preceq 2\mathbb{E}\left[\langle \mathbf{w}^{(q)^*} - \mathbf{w}^*, \mathbf{x}\rangle^2 \mathbf{x}\mathbf{x}^\top\right] + 2\mathbb{E}\left[\langle \mathbf{w}^{(q)^*} - \mathbf{w}^*, \mathbf{x}\rangle^2 \boldsymbol{\epsilon}^{(d)}\boldsymbol{\epsilon}^{(d)^\top}\right]$$

$$\preceq 2(1 + \epsilon_d)\alpha_0 \text{tr}\left((\mathbf{w}^{(q)^*} - \mathbf{w}^*)(\mathbf{w}^{(q)^*} - \mathbf{w}^*)^\top \mathbf{H}\right)\mathbf{H},$$

(E.13)

where the last inequality holds by the definition of Example E.1 and Assumption E.1. Regarding $\mathbb{E}\left[(y^{(q)} - y)^2 \mathbf{x}^{(q)}\mathbf{x}^{(q)^\top}\right]$, if we further assume that there exists a constant $C$ such that $\mathbb{E}\left[y^2\mathbf{x}\mathbf{x}^\top\right] \preceq C\mathbf{H}$, then

$$\mathbb{E}\left[(y^{(q)} - y)^2 \mathbf{x}^{(q)}\mathbf{x}^{(q)^\top}\right] \preceq 2\mathbb{E}\left[(y^{(q)} - y)^2 \mathbf{x}\mathbf{x}^\top\right] + 2\mathbb{E}\left[(y^{(q)} - y)^2 \boldsymbol{\epsilon}^{(d)}\boldsymbol{\epsilon}^{(d)^\top}\right]$$

$$\preceq 2(1 + \epsilon_d)\mathbb{E}\left[(y^{(q)} - y)^2 \mathbf{x}\mathbf{x}^\top\right]$$

$$= 2(1 + \epsilon_d)\epsilon_l \mathbb{E}[y^2\mathbf{x}\mathbf{x}^\top]$$

$$\preceq 2(1 + \epsilon_d)\epsilon_l C\mathbf{H}.$$

(E.14)

Therefore, together with (E.10), (E.11), (E.12), (E.13) and (E.14), we have

$$\mathbb{E}\left[(y^{(q)} - \langle \mathbf{w}^{(q)^*}, \mathbf{x}^{(q)}\rangle)^2 \mathbf{x}^{(q)}\mathbf{x}^{(q)^\top}\right]$$

$$\precsim (1 + \epsilon_d)\sigma_0^2\mathbf{H} + (\epsilon_d + \epsilon_d')\alpha_0 \left\|\mathbf{w}^{(q)^*}\right\|_{\mathbf{H}}^2 \mathbf{H} + (1 + \epsilon_d)\alpha_0 \left\|\mathbf{w}^{(q)^*} - \mathbf{w}^*\right\|_{\mathbf{H}}^2 \mathbf{H} + (1 + \epsilon_d)\epsilon_l\mathbf{H}$$

$$\precsim \left[(1 + \epsilon_d)(\sigma_0^2 + \epsilon_l) + (1 + \epsilon_d + \epsilon_d')\alpha_0\|\mathbf{w}^*\|_{\mathbf{H}}^2\right]\mathbf{H}$$

$$= \left(\sigma_0^2 + \epsilon_l + \frac{1 + \epsilon_d + \epsilon_d'}{1 + \epsilon_d}\alpha_0\|\mathbf{w}^*\|_{\mathbf{H}}^2\right)\mathbf{H}^{(q)}.$$

That is, under strong multiplicative quantization Example E.1 and fourth moment Assumption E.2 on full-precision data, Assumption 3.4 is verified. $\square$

**Lemma E.4.** *Under strong additive quantization E.2, Assumption E.1 and Assumption E.2,*

$$\mathbb{E}\left[(y^{(q)} - \langle \mathbf{w}^{(q)^*}, \mathbf{x}^{(q)}\rangle)^2 \mathbf{x}^{(q)}\mathbf{x}^{(q)^\top}\right] \precsim \left[\sigma_0^2 + \epsilon_l + \epsilon_d(1 + \frac{\epsilon_d'}{\epsilon_d^2})\|\mathbf{w}^*\|^2 + (1 + \alpha_0)\|\mathbf{w}^*\|_{\mathbf{H}}^2\right]\mathbf{H}^{(q)}.$$

*Proof.* Regarding $\mathbb{E}\left[(y - \langle \mathbf{w}^*, \mathbf{x}\rangle)^2 \mathbf{x}^{(q)}\mathbf{x}^{(q)^\top}\right]$, if we further assume that $\mathbb{E}\left[(y - \langle \mathbf{w}^*, \mathbf{x}\rangle)^2\right] \leq \sigma_0^2$, then

$$\mathbb{E}\left[(y - \langle \mathbf{w}^*, \mathbf{x}\rangle)^2 \mathbf{x}^{(q)}\mathbf{x}^{(q)^\top}\right] \preceq 2\mathbb{E}\left[(y - \langle \mathbf{w}^*, \mathbf{x}\rangle)^2 \mathbf{x}\mathbf{x}^\top\right] + 2\mathbb{E}\left[(y - \langle \mathbf{w}^*, \mathbf{x}\rangle)^2 \boldsymbol{\epsilon}^{(d)}\boldsymbol{\epsilon}^{(d)^\top}\right]$$

$$\preceq 2\sigma_0^2\mathbf{H} + 2\epsilon_d\sigma_0^2\mathbf{I}.$$

(E.15)

Regarding $\mathbb{E}\left[(y^{(q)} - y)^2 \mathbf{x}^{(q)}\mathbf{x}^{(q)^\top}\right]$,

$$\mathbb{E}\left[(y^{(q)} - y)^2 \mathbf{x}^{(q)}\mathbf{x}^{(q)^\top}\right] \leq \epsilon_d\epsilon_l\mathbf{I}.$$

(E.16)

Regarding $\mathbb{E}\left[\langle \mathbf{w}^{(q)^*}, \boldsymbol{\epsilon}^{(d)}\rangle^2 \mathbf{x}^{(q)}\mathbf{x}^{(q)^\top}\right]$,

$$\mathbb{E}\left[\langle \mathbf{w}^{(q)^*}, \boldsymbol{\epsilon}^{(d)}\rangle^2 \mathbf{x}^{(q)}\mathbf{x}^{(q)^\top}\right]$$

$$= \mathbb{E}\left[\boldsymbol{\epsilon}^{(d)^\top}\mathbf{w}^{(q)^*}\mathbf{w}^{(q)^*^\top}\boldsymbol{\epsilon}^{(d)}\mathbf{x}^{(q)}\mathbf{x}^{(q)^\top}\right]$$

$$\preceq 2\mathbb{E}\left[\boldsymbol{\epsilon}^{(d)^\top}\mathbf{w}^{(q)^*}\mathbf{w}^{(q)^*^\top}\boldsymbol{\epsilon}^{(d)}\mathbf{x}\mathbf{x}^\top\right] + 2\mathbb{E}\left[\boldsymbol{\epsilon}^{(d)^\top}\mathbf{w}^{(q)^*}\mathbf{w}^{(q)^*^\top}\boldsymbol{\epsilon}^{(d)}\boldsymbol{\epsilon}^{(d)}\boldsymbol{\epsilon}^{(d)^\top}\right]$$

$$\preceq 2\epsilon_d\text{tr}(\mathbf{w}^{(q)^*}\mathbf{w}^{(q)^*^\top})\mathbf{H} + 2\epsilon_d'\text{tr}(\mathbf{w}^{(q)^*}\mathbf{w}^{(q)^*^\top})\mathbf{I}.$$

(E.17)

Regarding $\mathbb{E}\left[\langle \mathbf{w}^{(q)^*} - \mathbf{w}^*, \mathbf{x}\rangle^2 \mathbf{x}^{(q)} \mathbf{x}^{(q)^\top}\right]$,

$$
\mathbb{E}\left[\langle \mathbf{w}^{(q)^*} - \mathbf{w}^*, \mathbf{x}\rangle^2 \mathbf{x}^{(q)} \mathbf{x}^{(q)^\top}\right]
$$

$$
=\mathbb{E}\left[\mathbf{x}^\top (\mathbf{w}^{(q)^*} - \mathbf{w}^*)(\mathbf{w}^{(q)^*} - \mathbf{w}^*)^\top \mathbf{x}\mathbf{x}^{(q)}\mathbf{x}^{(q)^\top}\right]
$$

$$
\preceq 2\mathbb{E}\left[\mathbf{x}^\top (\mathbf{w}^{(q)^*} - \mathbf{w}^*)(\mathbf{w}^{(q)^*} - \mathbf{w}^*)^\top \mathbf{x}\mathbf{x}\mathbf{x}^\top\right] + 2\mathbb{E}\left[\mathbf{x}^\top (\mathbf{w}^{(q)^*} - \mathbf{w}^*)(\mathbf{w}^{(q)^*} - \mathbf{w}^*)^\top \mathbf{x}\boldsymbol{\epsilon}^{(d)}\boldsymbol{\epsilon}^{(d)^\top}\right]
$$

$$
\preceq 2\alpha_0 \mathrm{tr}\left((\mathbf{w}^{(q)^*} - \mathbf{w}^*)(\mathbf{w}^{(q)^*} - \mathbf{w}^*)^\top \mathbf{H}\right)\mathbf{H} + 2\epsilon_d \mathrm{tr}\left((\mathbf{w}^{(q)^*} - \mathbf{w}^*)(\mathbf{w}^{(q)^*} - \mathbf{w}^*)^\top \mathbf{H}\right)\mathbf{I}. \tag{E.18}
$$

Therefore, together with (E.10), (E.15), (E.16), (E.17) and (E.18), we have

$$
\mathbb{E}\left[(y^{(q)} - \langle \mathbf{w}^{(q)^*}, \mathbf{x}^{(q)}\rangle)^2 \mathbf{x}^{(q)}\mathbf{x}^{(q)^\top}\right]
$$

$$
\precsim (\sigma_0^2 + \epsilon_l)\mathbf{H}^{(q)} + \epsilon_d(1 + \frac{\epsilon_d'}{\epsilon_d^2})\left\|\mathbf{w}^{(q)^*}\right\|^2 \mathbf{H}^{(q)} + (1 + \alpha_0)\left\|\mathbf{w}^{(q)^*} - \mathbf{w}^*\right\|_{\mathbf{H}}^2 \mathbf{H}^{(q)}
$$

$$
\leq \left[\sigma_0^2 + \epsilon_l + \epsilon_d(1 + \frac{\epsilon_d'}{\epsilon_d^2})\|\mathbf{w}^*\|^2 + (1 + \alpha_0)\|\mathbf{w}^*\|_{\mathbf{H}}^2\right]\mathbf{H}^{(q)}.
$$

That is, under strong additive quantization Example E.2 and noise Assumption E.2 on full-precision data, Assumption 3.4 is verified. □

## F EXTENSION TO QUANTIZED MASTER WEIGHTS

In the quantized SGD algorithm (quantized SGD), the master weight maintains full precision. In this section, we demonstrate that our theoretical framework can naturally extend to the setting where the master weight is also quantized. For simplicity, we only discuss the bounds for $R_N^{(0)}$. The theoretical bounds are presented in Theorem F.1, Theorem F.2 and Theorem F.3 for general quantization, additive quantization and multiplicative quantization, respectively. These results demonstrate that when the master weights are quantized, quantized SGD requires stricter conditions on the step size to ensure convergence. Furthermore, the final excess risk bounds incorporate additional error terms, which degrades generalization performance.

We first present the algorithm and the propagation of $\mathbb{E}\left[\boldsymbol{\eta}_t \boldsymbol{\eta}_t^\top\right]$. Specifically, we consider

$$
\mathbf{w}_t = \mathcal{Q}_p(\mathbf{w}_{t-1}) + \gamma\frac{1}{B}\mathcal{Q}_d(\mathbf{X}_t)^\top \mathcal{Q}_o\left(\mathcal{Q}_l(\mathbf{y}_t) - \mathcal{Q}_a\left(\mathcal{Q}_d(\mathbf{X}_t)\mathcal{Q}_p(\mathbf{w}_{t-1})\right)\right), \quad t = 1, ..., N. \tag{F.1}
$$

When master weight is quantized, Lemma A.1 changes to

$$
\boldsymbol{\eta}_t = \left(\mathbf{I} - \frac{1}{B}\gamma\mathcal{Q}_d(\mathbf{X}_t)^\top \mathcal{Q}_d(\mathbf{X}_t)\right)\boldsymbol{\eta}_{t-1} + \gamma\frac{1}{B}\mathcal{Q}_d(\mathbf{X}_t)^\top\left[\boldsymbol{\xi}_t + \boldsymbol{\epsilon}_t^{(o)} - \boldsymbol{\epsilon}_t^{(a)} - \mathcal{Q}_d(\mathbf{X}_t)\boldsymbol{\epsilon}_{t-1}^{(p)}\right] + \boldsymbol{\epsilon}_{t-1}^{(p)}. \tag{F.2}
$$

In particular, the coefficient for parameter quantization error $\boldsymbol{\epsilon}_{t-1}^{(p)}$ changes from $-\frac{1}{B}\gamma\mathcal{Q}_d(\mathbf{X}_t)^\top \mathcal{Q}_d(\mathbf{X}_t)$ to $\mathbf{I} - \frac{1}{B}\gamma\mathcal{Q}_d(\mathbf{X}_t)^\top \mathcal{Q}_d(\mathbf{X}_t)$. Therefore, we can rewrite Lemma C.2 and Lemma C.3 as follows.

**Lemma F.1** (Update rule under general quantization with quantized master weight). *Under Assumption 3.1, Assumption 3.2, Assumption 3.3, and Assumption 3.4,*

$$
\mathbb{E}\left[\boldsymbol{\eta}_t \boldsymbol{\eta}_t^\top\right] \preceq 2(\mathbf{B}_t + \mathbf{C}_t),
$$

*where*

$$
\mathbf{C}_t \preceq \mathbb{E}\left[\left(\mathbf{I} - \gamma\frac{1}{B}\mathcal{Q}_d(\mathbf{X})^\top \mathcal{Q}_d(\mathbf{X})\right)\mathbf{C}_{t-1}\left(\mathbf{I} - \gamma\frac{1}{B}\mathcal{Q}_d(\mathbf{X})^\top \mathcal{Q}_d(\mathbf{X})\right)\right] + \gamma^2 {\sigma_G^{(q)}}^2 \mathbf{H}^{(q)} + 2\mathbb{E}\left[\boldsymbol{\epsilon}_{t-1}^{(p)}\boldsymbol{\epsilon}_{t-1}^{(p)^\top}\right],
$$

$$
\mathbf{B}_t = \mathbb{E}\left[\left(\mathbf{I} - \gamma\frac{1}{B}\mathcal{Q}_d(\mathbf{X})^\top \mathcal{Q}_d(\mathbf{X})\right)\mathbf{B}_{t-1}\left(\mathbf{I} - \gamma\frac{1}{B}\mathcal{Q}_d(\mathbf{X})^\top \mathcal{Q}_d(\mathbf{X})\right)\right],
$$

*with* $\mathbf{C}_0 = \mathbf{0}$, $\mathbf{B}_0 = \mathbb{E}\left[\boldsymbol{\eta}_0 \boldsymbol{\eta}_0^\top\right]$ *and*

$$\sigma_G^{(q)^2} = \frac{\sup_t \left\{ \left\| \mathbb{E}\left[\boldsymbol{\epsilon}_t^{(o)} \boldsymbol{\epsilon}_t^{(o)^\top} | \mathbf{o}_t\right] + \mathbb{E}\left[\boldsymbol{\epsilon}_t^{(a)} \boldsymbol{\epsilon}_t^{(a)^\top} | \mathbf{a}_t\right] \right\| \right\}}{B}$$
$$+ 2\alpha_B \sup_t \mathbb{E}_{\mathbf{w}_{t-1}}\left[ \mathrm{tr}\left( \mathbf{H}^{(q)} \mathbb{E}\left[\boldsymbol{\epsilon}_{t-1}^{(p)} \boldsymbol{\epsilon}_{t-1}^{(p)^\top} | \mathbf{w}_{t-1}\right] \right) \right] + \frac{\sigma^2}{B},$$

*Proof.* Noticing that

$$\mathbb{E}\left[ \left(\mathbf{I} - \frac{1}{B}\gamma \mathcal{Q}_d(\mathbf{X}_t)^\top \mathcal{Q}_d(\mathbf{X}_t)\right) \boldsymbol{\epsilon}_{t-1}^{(p)} \boldsymbol{\epsilon}_{t-1}^{(p)^\top} \left(\mathbf{I} - \frac{1}{B}\gamma \mathcal{Q}_d(\mathbf{X}_t)^\top \mathcal{Q}_d(\mathbf{X}_t)\right) \right]$$
$$\preceq 2\frac{\gamma^2}{B^2} \mathbb{E}\left[ \mathcal{Q}_d(\mathbf{X}_t)^\top \mathcal{Q}_d(\mathbf{X}_t) \boldsymbol{\epsilon}_{t-1}^{(p)} \boldsymbol{\epsilon}_{t-1}^{(p)^\top} \mathcal{Q}_d(\mathbf{X}_t)^\top \mathcal{Q}_d(\mathbf{X}_t) \right] + 2\mathbb{E}\left[ \boldsymbol{\epsilon}_{t-1}^{(p)} \boldsymbol{\epsilon}_{t-1}^{(p)^\top} \right], \tag{F.3}$$

together with (F.2) and Lemma C.2, we complete the proof. $\qquad\square$

**Lemma F.2** (Update rule under multiplicative quantization with quantized master weight). *If there exist* $\epsilon_d, \epsilon_l, \epsilon_p, \epsilon_a$ *and* $\epsilon_o$ *such that for any* $i \in \{d, l, p, a, o\}$, *quantization* $\mathcal{Q}_i$ *is* $\epsilon_i$-*multiplicative, then under Assumption 3.1, Assumption 3.2, Assumption 3.3, and Assumption 3.4, it holds*

$$\mathbb{E}\left[\boldsymbol{\eta}_t \boldsymbol{\eta}_t^\top\right] \preceq 2(\mathbf{B}_t + \mathbf{C}_t),$$

*where*

$$\mathbf{C}_t \preceq \mathbb{E}\left[ \left(\mathbf{I} - \frac{1}{B}\gamma \mathcal{Q}_d(\mathbf{X})^\top \mathcal{Q}_d(\mathbf{X})\right) \mathbf{C}_{t-1} \left(\mathbf{I} - \frac{1}{B}\gamma \mathcal{Q}_d(\mathbf{X})^\top \mathcal{Q}_d(\mathbf{X})\right) \right] + 8\epsilon_p \left(\mathbf{C}_{t-1} + \mathbf{B}_{t-1}\right)$$
$$+ \tilde{\epsilon} \mathbb{E}\left[ \frac{\gamma}{B} \mathcal{Q}_d(\mathbf{X})^\top \mathcal{Q}_d(\mathbf{X}) (\mathbf{B}_{t-1} + \mathbf{C}_{t-1}) \frac{\gamma}{B} \mathcal{Q}_d(\mathbf{X})^\top \mathcal{Q}_d(\mathbf{X}) \right] + \gamma^2 \sigma_M^{(q)^2} \mathbf{H}^{(q)} + 4\epsilon_p \mathbf{w}^{(q)^*}(\mathbf{w}^{(q)^*})^\top,$$
$$\mathbf{B}_t = \mathbb{E}\left[ \left(\mathbf{I} - \frac{1}{B}\gamma \mathcal{Q}_d(\mathbf{X})^\top \mathcal{Q}_d(\mathbf{X})\right) \mathbf{B}_{t-1} \left(\mathbf{I} - \frac{1}{B}\gamma \mathcal{Q}_d(\mathbf{X})^\top \mathcal{Q}_d(\mathbf{X})\right) \right],$$

*with*

$$\tilde{\epsilon} = 8\epsilon_o(1 + \epsilon_p)(1 + \epsilon_a) + 8\epsilon_p + 4\epsilon_a(1 + \epsilon_p),$$
$$\sigma_M^{(q)^2} = \frac{(1 + 4\epsilon_o)\sigma^2}{B} + \frac{\|\mathbf{w}^*\|_{\mathbf{H}}^2}{1 + \epsilon_d}\alpha_B \left(4\epsilon_o[(1 + \epsilon_p)(1 + \epsilon_a) + 1] + 2\epsilon_a(1 + \epsilon_p) + 4\epsilon_p\right).$$

*Proof.* Under multiplicative quantization,

$$\mathbb{E}\left[\boldsymbol{\epsilon}_{t-1}^{(p)} \boldsymbol{\epsilon}_{t-1}^{(p)^\top}\right] = \epsilon_p \mathbb{E}\left[\mathbf{w}_{t-1} \mathbf{w}_{t-1}^\top\right] \preceq 2\epsilon_p \mathbb{E}\left[\boldsymbol{\eta}_{t-1} \boldsymbol{\eta}_{t-1}^\top\right] + 2\epsilon_p \mathbf{w}^{(q)^*}(\mathbf{w}^{(q)^*})^\top. \tag{F.4}$$

By (F.2), (F.3), (F.4) and Lemma C.3, the proof is immediately completed. $\qquad\square$

### F.1 GENERAL QUANTIZATION

In this section, we derive upper bounds for $R_N^{(0)}$ under general quantization. We first perform bias-variance decomposition under general quantization. By Lemma F.1, we have

$$\mathbb{E}\left[\boldsymbol{\eta}_t \boldsymbol{\eta}_t^\top\right] \preceq 2\left(\mathbf{B}_t + \mathbf{C}_t\right),$$

where

$$\mathbf{B}_t := (\mathcal{I} - \gamma \mathcal{T}_B^{(q)})^t \circ \mathbf{B}_0, \quad \mathbf{B}_0 = \mathbb{E}\left[\boldsymbol{\eta}_0 \boldsymbol{\eta}_0^\top\right],$$

and

$$\mathbf{C}_t := (\mathcal{I} - \gamma \mathcal{T}_B^{(q)}) \circ \mathbf{C}_{t-1} + \gamma^2 \sigma_G^{(q)^2} \mathbf{H}^{(q)} + 2\mathbb{E}\left[\boldsymbol{\epsilon}_{t-1}^{(p)} \boldsymbol{\epsilon}_{t-1}^{(p)^\top}\right], \quad \mathbf{C}_0 = \mathbf{0}.$$

Then by Lemma C.1,

$$R_N^{(0)}/2 \leq \underbrace{\frac{1}{N^2} \cdot \sum_{t=0}^{N-1} \sum_{k=t}^{N-1} \left\langle (\mathbf{I} - \gamma \mathbf{H}^{(q)})^{k-t} \mathbf{H}^{(q)}, \mathbf{B}_t \right\rangle}_{\text{bias}} + \underbrace{\frac{1}{N^2} \cdot \sum_{t=0}^{N-1} \sum_{k=t}^{N-1} \left\langle (\mathbf{I} - \gamma \mathbf{H}^{(q)})^{k-t} \mathbf{H}^{(q)}, \mathbf{C}_t \right\rangle}_{\text{variance}}.$$

(F.5)

Noticing that the bias error when master weight is quantized is the same as the bias error when master weight maintains full precision, we then only need to derive bounds for variance error. Similar to (C.22),

$$\mathbf{C}_t \preceq (\mathcal{I} - \gamma \widetilde{\mathcal{T}}^{(q)}) \circ \mathbf{C}_{t-1} + \gamma^2 \mathcal{M}_B^{(q)} \circ \mathbf{C}_{t-1} + \gamma^2 \sigma_G^{(q)^2} \mathbf{H}^{(q)} + 2\mathbb{E}\left[ \boldsymbol{\epsilon}_{t-1}^{(p)} \boldsymbol{\epsilon}_{t-1}^{(p)\top} \right]. \quad (\text{F.6})$$

**Lemma F.3** (A bound for $\mathcal{M}_B^{(q)} \circ \mathbf{C}_t$ with quantized master weight). *For $t \geq 1$, under Assumption 3.1, Assumption 3.2, Assumption 3.3, and Assumption 3.4, if the stepsize $\gamma \leq \frac{1}{\alpha_B \text{tr}(\mathbf{H}^{(q)})}$, then*

$$\mathcal{M}_B^{(q)} \circ \mathbf{C}_t \preceq \frac{\alpha_B \text{tr}(\mathbf{H}^{(q)}) \left( \gamma \sigma_G^{(q)^2} + \frac{2d\mu}{\gamma \text{tr}(\mathbf{H}^{(q)})} \right)}{1 - \gamma \alpha_B \text{tr}(\mathbf{H}^{(q)})} \mathbf{H}^{(q)}.$$

*Proof.* We first derive a crude bound for $\mathbf{C}_t$. Denote $\boldsymbol{\Sigma} = \sigma_G^{(q)^2} \mathbf{H}^{(q)} + \frac{2}{\gamma^2} \sup_t \mathbb{E}\left[ \boldsymbol{\epsilon}_{t-1}^{(p)} \boldsymbol{\epsilon}_{t-1}^{(p)\top} \right]$. By (C.24),

$$\mathbf{C}_t \preceq \mathbf{C}_\infty = \gamma \mathcal{T}_B^{(q)^{-1}} \circ \boldsymbol{\Sigma}.$$

Applying $\widetilde{\mathcal{T}}^{(q)}$, we have

$$\begin{aligned}
\widetilde{\mathcal{T}}^{(q)} \circ \mathbf{C}_\infty &= \mathcal{T}_B^{(q)} \circ \mathbf{C}_\infty + \gamma \mathcal{M}_B^{(q)} \circ \mathbf{C}_\infty - \gamma \widetilde{\mathcal{M}}^{(q)} \circ \mathbf{C}_\infty \\
&= \gamma \boldsymbol{\Sigma} + \gamma \mathcal{M}_B^{(q)} \circ \mathbf{C}_\infty - \gamma \widetilde{\mathcal{M}}^{(q)} \circ \mathbf{C}_\infty \\
&\preceq \gamma \boldsymbol{\Sigma} + \gamma \mathcal{M}_B^{(q)} \circ \mathbf{C}_\infty \\
&= \gamma \sigma_G^{(q)^2} \mathbf{H}^{(q)} + \frac{2}{\gamma} \sup_t \mathbb{E}\left[ \boldsymbol{\epsilon}_{t-1}^{(p)} \boldsymbol{\epsilon}_{t-1}^{(p)\top} \right] + \gamma \mathcal{M}_B^{(q)} \circ \mathbf{C}_\infty.
\end{aligned}$$

Applying $(\widetilde{\mathcal{T}}^{(q)})^{-1}$ and by solving recursion, we have

$$\begin{aligned}
\mathbf{C}_\infty &\preceq \gamma \sigma_G^{(q)^2} (\widetilde{\mathcal{T}}^{(q)})^{-1} \circ \mathbf{H}^{(q)} + \frac{2}{\gamma} (\widetilde{\mathcal{T}}^{(q)})^{-1} \circ \sup_t \mathbb{E}\left[ \boldsymbol{\epsilon}_{t-1}^{(p)} \boldsymbol{\epsilon}_{t-1}^{(p)\top} \right] + \gamma (\widetilde{\mathcal{T}}^{(q)})^{-1} \circ \mathcal{M}_B^{(q)} \circ \mathbf{C}_\infty \\
&\preceq \sum_{t=0}^{\infty} \left( \gamma (\widetilde{\mathcal{T}}^{(q)})^{-1} \circ \mathcal{M}_B^{(q)} \right)^t \circ \left( \gamma \sigma_G^{(q)^2} (\widetilde{\mathcal{T}}^{(q)})^{-1} \circ \mathbf{H}^{(q)} + \frac{2}{\gamma} (\widetilde{\mathcal{T}}^{(q)})^{-1} \circ \sup_t \mathbb{E}\left[ \boldsymbol{\epsilon}_{t-1}^{(p)} \boldsymbol{\epsilon}_{t-1}^{(p)\top} \right] \right).
\end{aligned}$$

(F.7)

By (C.27),

$$\sum_{t=0}^{\infty} \left( \gamma (\widetilde{\mathcal{T}}^{(q)})^{-1} \circ \mathcal{M}_B^{(q)} \right)^t \circ \gamma \sigma_G^{(q)^2} (\widetilde{\mathcal{T}}^{(q)})^{-1} \circ \mathbf{H}^{(q)} \preceq \frac{\gamma \sigma_G^{(q)^2}}{1 - \gamma \alpha_B \text{tr}(\mathbf{H}^{(q)})} \mathbf{I}. \quad (\text{F.8})$$

Noticing that

$$\begin{aligned}
(\widetilde{\mathcal{T}}^{(q)})^{-1} \circ \sup_t \mathbb{E}\left[ \boldsymbol{\epsilon}_{t-1}^{(p)} \boldsymbol{\epsilon}_{t-1}^{(p)\top} \right] &= \gamma \sum_{t=0}^{\infty} (\mathcal{I} - \gamma \widetilde{\mathcal{T}}^{(q)})^t \circ \sup_t \mathbb{E}\left[ \boldsymbol{\epsilon}_{t-1}^{(p)} \boldsymbol{\epsilon}_{t-1}^{(p)\top} \right] \\
&= \gamma \sum_{t=0}^{\infty} (\mathbf{I} - \gamma \mathbf{H}^{(q)})^t \sup_t \mathbb{E}\left[ \boldsymbol{\epsilon}_{t-1}^{(p)} \boldsymbol{\epsilon}_{t-1}^{(p)\top} \right] (\mathbf{I} - \gamma \mathbf{H}^{(q)})^t \\
&\preceq \sup_t \left\| \mathbb{E}\left[ \boldsymbol{\epsilon}_{t-1}^{(p)} \boldsymbol{\epsilon}_{t-1}^{(p)\top} \right] \right\| (\mathbf{H}^{(q)})^{-1} := \mu \cdot (\mathbf{H}^{(q)})^{-1},
\end{aligned}$$

we have

$$
\sum_{t=0}^{\infty} \left( \gamma (\widetilde{\mathcal{T}}^{(q)})^{-1} \circ \mathcal{M}_B^{(q)} \right)^t \circ \frac{2}{\gamma} (\widetilde{\mathcal{T}}^{(q)})^{-1} \circ \mathbb{E} \left[ \boldsymbol{\epsilon}_{t-1}^{(p)} \boldsymbol{\epsilon}_{t-1}^{(p)\top} \right]
$$

$$
= \sum_{t=0}^{\infty} \left( \gamma (\widetilde{\mathcal{T}}^{(q)})^{-1} \circ \mathcal{M}_B^{(q)} \right)^t \circ \frac{2\mu}{\gamma} (\mathbf{H}^{(q)})^{-1}
$$

$$
= \sum_{t=0}^{\infty} \left( \gamma (\widetilde{\mathcal{T}}^{(q)})^{-1} \circ \mathcal{M}_B^{(q)} \right)^{t-1} \gamma (\widetilde{\mathcal{T}}^{(q)})^{-1} \circ \mathcal{M}_B^{(q)} \circ \frac{2\mu}{\gamma} (\mathbf{H}^{(q)})^{-1}
$$

$$
\preceq \sum_{t=0}^{\infty} \left( \gamma (\widetilde{\mathcal{T}}^{(q)})^{-1} \circ \mathcal{M}_B^{(q)} \right)^{t-1} \gamma (\widetilde{\mathcal{T}}^{(q)})^{-1} \circ \alpha_B \mathrm{tr}(\mathbf{I}) \frac{2\mu}{\gamma} \mathbf{H}^{(q)} \tag{F.9}
$$

$$
\preceq \sum_{t=0}^{\infty} \left( \gamma (\widetilde{\mathcal{T}}^{(q)})^{-1} \circ \mathcal{M}_B^{(q)} \right)^{t-1} 2\mu \alpha_B d \cdot \mathbf{I}
$$

$$
\preceq 2d\mu\alpha_B \cdot \sum_{t=0}^{\infty} \left( \gamma \alpha_B \mathrm{tr}(\mathbf{H}^{(q)}) \right)^{t-1} \mathbf{I}
$$

$$
= \frac{2d\mu}{\gamma \mathrm{tr}(\mathbf{H}^{(q)})} \cdot \sum_{t=0}^{\infty} \left( \gamma \alpha_B \mathrm{tr}(\mathbf{H}^{(q)}) \right)^t \mathbf{I}
$$

$$
= \frac{2d\mu}{\gamma \mathrm{tr}(\mathbf{H}^{(q)})} \frac{1}{1 - \gamma \alpha_B \mathrm{tr}\left(\mathbf{H}^{(q)}\right)} \mathbf{I}.
$$

Therefore, together with (F.7), (F.8) and (F.9), we have

$$
\mathbf{C}_t \preceq \mathbf{C}_\infty \preceq \frac{\gamma \sigma_G^{(q)^2} + \frac{2d\mu}{\gamma \mathrm{tr}(\mathbf{H}^{(q)})}}{1 - \gamma \alpha_B \mathrm{tr}(\mathbf{H}^{(q)})} \mathbf{I}.
$$

It follows that

$$
\mathcal{M}_B^{(q)} \circ \mathbf{C}_t \preceq \frac{\alpha_B \mathrm{tr}(\mathbf{H}^{(q)}) \left( \gamma \sigma_G^{(q)^2} + \frac{2d\mu}{\gamma \mathrm{tr}(\mathbf{H}^{(q)})} \right)}{1 - \gamma \alpha_B \mathrm{tr}(\mathbf{H}^{(q)})} \mathbf{H}^{(q)}.
$$

$\square$

Together with (F.6), we can provide a refined bound for $\mathbf{C}_t$ and we are now ready to bound the variance error.

**Lemma F.4** (A bound for variance under general quantization with quantized master weight). *Under Assumption 3.1, Assumption 3.2, Assumption 3.3, and Assumption 3.4, if the stepsize satisfies $\gamma < \frac{1}{\alpha_B \mathrm{tr}(\mathbf{H}^{(q)})}$, then*

$$
\text{variance} \leq \frac{\sigma_G^{(q)^2} + \frac{2\alpha_B d\mu}{\gamma}}{1 - \gamma \alpha_B \mathrm{tr}(\mathbf{H}^{(q)})} \left( \frac{k^*}{N} + N\gamma^2 \cdot \sum_{i>k^*} (\lambda_i^{(q)})^2 \right) + \frac{2\mu}{\gamma} \left( \sum_{i \leq k^*} \frac{1}{N\gamma\lambda_i^{(q)}} + N\gamma \sum_{i>k^*} \lambda_i^{(q)} \right),
$$

*where $\mu = \sup_t \left\| \mathbb{E} \left[ \boldsymbol{\epsilon}_{t-1}^{(p)} \boldsymbol{\epsilon}_{t-1}^{(p)\top} \right] \right\|$.*

*Proof.* We first provide a refined upper bound for $\mathbf{C}_t$. By (F.6) and Lemma F.3,

$$\mathbf{C}_t \preceq (\mathcal{I} - \gamma\widetilde{\mathcal{T}}^{(q)}) \circ \mathbf{C}_{t-1} + \gamma^2 \mathcal{M}_B^{(q)} \circ \mathbf{C}_{t-1} + \gamma^2 {\sigma_G^{(q)}}^2 \mathbf{H}^{(q)} + 2\mathbb{E}\left[\boldsymbol{\epsilon}_{t-1}^{(p)} \boldsymbol{\epsilon}_{t-1}^{(p)}{}^\top\right]$$

$$\preceq (\mathcal{I} - \gamma\widetilde{\mathcal{T}}^{(q)}) \circ \mathbf{C}_{t-1} + \frac{\gamma^2 \alpha_B \mathrm{tr}(\mathbf{H}^{(q)})\left(\gamma{\sigma_G^{(q)}}^2 + \frac{2d\mu}{\gamma\mathrm{tr}(\mathbf{H}^{(q)})}\right)}{1 - \gamma\alpha_B\mathrm{tr}(\mathbf{H}^{(q)})}\mathbf{H}^{(q)} + \gamma^2{\sigma_G^{(q)}}^2\mathbf{H}^{(q)} + 2\mu\mathbf{I}$$

$$= (\mathcal{I} - \gamma\widetilde{\mathcal{T}}^{(q)}) \circ \mathbf{C}_{t-1} + \frac{\gamma^2 {\sigma_G^{(q)}}^2 + 2\gamma\alpha_B d\mu}{1 - \gamma\alpha_B\mathrm{tr}(\mathbf{H}^{(q)})}\mathbf{H}^{(q)} + 2\mu\mathbf{I}$$

$$= \frac{\gamma^2{\sigma_G^{(q)}}^2 + 2\gamma\alpha_B d\mu}{1 - \gamma\alpha_B\mathrm{tr}(\mathbf{H}^{(q)})} \cdot \sum_{k=0}^{t-1}(\mathcal{I} - \gamma\widetilde{\mathcal{T}}^{(q)})^k \circ \mathbf{H}^{(q)} + 2\mu\sum_{k=0}^{t-1}(\mathcal{I} - \gamma\widetilde{\mathcal{T}}^{(q)})^k \circ \mathbf{I}$$

$$= \frac{\gamma^2{\sigma_G^{(q)}}^2 + 2\gamma\alpha_B d\mu}{1 - \gamma\alpha_B\mathrm{tr}(\mathbf{H}^{(q)})} \cdot \sum_{k=0}^{t-1}(\mathbf{I} - \gamma\mathbf{H}^{(q)})^k\mathbf{H}^{(q)}(\mathbf{I} - \gamma\mathbf{H}^{(q)})^k + 2\mu\sum_{k=0}^{t-1}(\mathbf{I} - \gamma\mathbf{H}^{(q)})^{2k}$$

$$\preceq \frac{\gamma^2{\sigma_G^{(q)}}^2 + 2\gamma\alpha_B d\mu}{1 - \gamma\alpha_B\mathrm{tr}(\mathbf{H}^{(q)})} \cdot \sum_{k=0}^{t-1}(\mathbf{I} - \gamma\mathbf{H}^{(q)})^k\mathbf{H}^{(q)} + 2\mu\sum_{k=0}^{t-1}(\mathbf{I} - \gamma\mathbf{H}^{(q)})^k$$

$$= \underbrace{\frac{\gamma{\sigma_G^{(q)}}^2 + 2\alpha_B d\mu}{1 - \gamma\alpha_B\mathrm{tr}(\mathbf{H}^{(q)})} \cdot \left(\mathbf{I} - (\mathbf{I} - \gamma\mathbf{H}^{(q)})^t\right)}_{\mathbf{V}_1} + \underbrace{\frac{2\mu}{\gamma}\left(\mathbf{I} - (\mathbf{I} - \gamma\mathbf{H}^{(q)})^t\right)(\mathbf{H}^{(q)})^{-1}}_{\mathbf{V}_2}.$$

By (C.32),

$$\frac{1}{N^2} \cdot \sum_{t=0}^{N-1}\sum_{k=t}^{N-1}\left\langle(\mathbf{I} - \gamma\mathbf{H}^{(q)})^{k-t}\mathbf{H}^{(q)}, \mathbf{V}_1\right\rangle \le \frac{{\sigma_G^{(q)}}^2 + \frac{2\alpha_B d\mu}{\gamma}}{1 - \gamma\alpha_B\mathrm{tr}(\mathbf{H}^{(q)})}\left(\frac{k^*}{N} + N\gamma^2 \cdot \sum_{i>k^*}(\lambda_i^{(q)})^2\right). \tag{F.10}$$

Regarding $\mathbf{V}_2$,

$$\frac{1}{N^2} \cdot \sum_{t=0}^{N-1}\sum_{k=t}^{N-1}\left\langle(\mathbf{I} - \gamma\mathbf{H}^{(q)})^{k-t}\mathbf{H}^{(q)}, \mathbf{V}_2\right\rangle$$

$$= \frac{2\mu}{\gamma^2 N^2} \cdot \sum_{t=0}^{N-1}\left\langle\mathbf{I} - (\mathbf{I} - \gamma\mathbf{H}^{(q)})^{N-t}, \left(\mathbf{I} - (\mathbf{I} - \gamma\mathbf{H}^{(q)})^t\right)(\mathbf{H}^{(q)})^{-1}\right\rangle$$

$$= \frac{2\mu}{\gamma^2 N^2}\sum_i\sum_{t=0}^{N-1}\left[1 - (1 - \gamma\lambda_i^{(q)})^{N-t}\right]\left[1 - (1 - \gamma\lambda_i^{(q)})^t\right]\frac{1}{\lambda_i^{(q)}} \tag{F.11}$$

$$\le \frac{2\mu}{\gamma^2 N}\sum_i\min\left\{\frac{1}{\lambda_i^{(q)}}, \gamma^2 N^2\lambda_i^{(q)}\right\}$$

$$\le \frac{2\mu}{\gamma}\left(\sum_{i\le k^*}\frac{1}{N\gamma\lambda_i^{(q)}} + N\gamma\sum_{i>k^*}\lambda_i^{(q)}\right).$$

Together with (F.5), (F.10) and (F.11), we have

$$\text{variance} \le \frac{{\sigma_G^{(q)}}^2 + \frac{2\alpha_B d\mu}{\gamma}}{1 - \gamma\alpha_B\mathrm{tr}(\mathbf{H}^{(q)})}\left(\frac{k^*}{N} + N\gamma^2 \cdot \sum_{i>k^*}(\lambda_i^{(q)})^2\right) + 2\frac{\mu}{\gamma}\left(\sum_{i\le k^*}\frac{1}{N\gamma\lambda_i^{(q)}} + N\gamma\sum_{i>k^*}\lambda_i^{(q)}\right),$$

where $\mu = \sup_t \left\|\mathbb{E}\left[\boldsymbol{\epsilon}_{t-1}^{(p)}\boldsymbol{\epsilon}_{t-1}^{(p)}{}^\top\right]\right\|$. $\qquad\square$

**Theorem F.1** (A bound for $R_N^{(0)}$ under general quantization with quantized master weight). *Under Assumption 3.1, Assumption 3.2, Assumption 3.3, and Assumption 3.4, if the stepsize satisfies $\gamma <$*

$\frac{1}{\alpha_B \text{tr}(\mathbf{H}^{(q)})}$, *then*

$$R_N^{(0)}/2 \leq \frac{2\alpha_B \left( \|\mathbf{w}_0 - \mathbf{w}^{(q)^*}\|_{\mathbf{I}_{0:k^*}^{(q)}}^2 + N\gamma \|\mathbf{w}_0 - \mathbf{w}^{(q)^*}\|_{\mathbf{H}_{k^*:\infty}^{(q)}}^2 \right)}{N\gamma(1 - \gamma\alpha_B \text{tr}(\mathbf{H}^{(q)}))} \cdot \left( \frac{k^*}{N} + N\gamma^2 \sum_{i>k^*} (\lambda_i^{(q)})^2 \right)$$

$$+ \frac{1}{\gamma^2 N^2} \cdot \|\mathbf{w}_0 - \mathbf{w}^{(q)^*}\|_{(\mathbf{H}_{0:k^*}^{(q)})^{-1}}^2 + \|\mathbf{w}_0 - \mathbf{w}^{(q)^*}\|_{\mathbf{H}_{k^*:\infty}^{(q)}}^2$$

$$+ \frac{\sigma_G^{(q)^2} + \frac{2\alpha_B d\mu}{\gamma}}{1 - \gamma\alpha_B \text{tr}(\mathbf{H}^{(q)})} \left( \frac{k^*}{N} + N\gamma^2 \cdot \sum_{i>k^*} (\lambda_i^{(q)})^2 \right) + \frac{2\mu}{\gamma} \left( \sum_{i \leq k^*} \frac{1}{N\gamma\lambda_i^{(q)}} + N\gamma \sum_{i>k^*} \lambda_i^{(q)} \right),$$

*where* $k^* = \max\left\{ k : \lambda_k^{(q)} \geq \frac{1}{N\gamma} \right\}$, $\mu = \sup_t \left\| \mathbb{E}\left[ \boldsymbol{\epsilon}_{t-1}^{(p)} \boldsymbol{\epsilon}_{t-1}^{(p)^\top} \right] \right\|$ *and*

$$\sigma_G^{(q)^2} = \frac{\sup_t \left\{ \left\| \mathbb{E}\left[ \boldsymbol{\epsilon}_t^{(o)} \boldsymbol{\epsilon}_t^{(o)^\top} | \mathbf{o}_t \right] + \mathbb{E}\left[ \boldsymbol{\epsilon}_t^{(a)} \boldsymbol{\epsilon}_t^{(a)^\top} | \mathbf{a}_t \right] \right\| \right\}}{B}$$

$$+ 2\alpha_B \sup_t \mathbb{E}_{\mathbf{w}_{t-1}} \left[ \text{tr}\left( \mathbf{H}^{(q)} \mathbb{E}\left[ \boldsymbol{\epsilon}_{t-1}^{(p)} \boldsymbol{\epsilon}_{t-1}^{(p)^\top} | \mathbf{w}_{t-1} \right] \right) \right] + \frac{\sigma^2}{B}.$$

*Proof.* The proof is completed by (F.5), Lemma C.10 and Lemma F.4. $\qquad\square$

Next, we deduce an upper bound for $R_N^{(0)}$ under additive quantization from Theorem F.1.

## F.2 ADDITIVE QUANTIZATION

**Theorem F.2** (A bound for $R_N^{(0)}$ under additive quantization with quantized master weight)**.** *Under Assumption 3.1, Assumption 3.2, Assumption 3.3, and Assumption 3.4, if there exist $\epsilon_d, \epsilon_l, \epsilon_p, \epsilon_a$ and $\epsilon_o$ such that for any $i \in \{d, l, p, a, o\}$, quantization $\mathcal{Q}_i$ is $\epsilon_i$-additive, if the stepsize satisfies $\gamma < \frac{1}{\alpha_B[\text{tr}(\mathbf{H}) + d\epsilon_d]}$, then*

$$R_N^{(0)}/2 \leq \frac{2\alpha_B \left( \|\mathbf{w}_0 - \mathbf{w}^{(q)^*}\|_{\mathbf{I}_{0:k^*}^{(q)}}^2 + N\gamma \|\mathbf{w}_0 - \mathbf{w}^{(q)^*}\|_{\mathbf{H}_{k^*:\infty}^{(q)}}^2 \right)}{N\gamma(1 - \gamma\alpha_B[\text{tr}(\mathbf{H}) + d\epsilon_d])} \cdot \left( \frac{k^*}{N} + N\gamma^2 \sum_{i>k^*} (\lambda_i + \epsilon_d)^2 \right)$$

$$+ \frac{1}{\gamma^2 N^2} \cdot \|\mathbf{w}_0 - \mathbf{w}^{(q)^*}\|_{(\mathbf{H}_{0:k^*}^{(q)})^{-1}}^2 + \|\mathbf{w}_0 - \mathbf{w}^{(q)^*}\|_{\mathbf{H}_{k^*:\infty}^{(q)}}^2$$

$$+ \frac{\frac{\sigma^2 + \epsilon_o + \epsilon_a}{B} + 2\epsilon_p \alpha_B \left( \frac{d}{\gamma} + \text{tr}(\mathbf{H}) + d\epsilon_d \right)}{1 - \gamma\alpha_B[\text{tr}(\mathbf{H}) + d\epsilon_d]} \left( \frac{k^*}{N} + N\gamma^2 \cdot \sum_{i>k^*} (\lambda_i + \epsilon_d)^2 \right)$$

$$+ \frac{2\epsilon_p}{\gamma} \left( \sum_{i \leq k^*} \frac{1}{N\gamma(\lambda_i + \epsilon_d)} + N\gamma \sum_{i>k^*} (\lambda_i + \epsilon_d) \right).$$

*Proof.* The proof is completed by Theorem F.1. $\qquad\square$

## F.3 MULTIPLICATIVE QUANTIZATION

In this section, we derive upper bounds for $R_N^{(0)}$ under multiplicative quantization, i.e., there exist $\epsilon_d, \epsilon_l, \epsilon_p, \epsilon_a$ and $\epsilon_o$ such that for any $i \in \{d, l, p, a, o\}$, quantization $\mathcal{Q}_i$ is $\epsilon_i$-multiplicative. We first perform bias-variance decomposition under multiplicative quantization. Denote

$$\mathbf{B}_t^{(M)} := ((1 + 8\epsilon_p)\mathcal{I} - \gamma\mathcal{T}_B^{(q)} + \tilde{\epsilon}\gamma^2 \mathcal{M}_B^{(q)})^t \circ \mathbf{B}_0^{(M)}, \quad \mathbf{B}_0^{(M)} = \mathbb{E}\left[ \boldsymbol{\eta}_0 \boldsymbol{\eta}_0^\top \right].$$

$$\mathbf{C}_t^{(M)} := ((1 + 8\epsilon_p)\mathcal{I} - \gamma\mathcal{T}_B^{(q)} + \tilde{\epsilon}\gamma^2 \mathcal{M}_B^{(q)}) \circ \mathbf{C}_{t-1}^{(M)} + \gamma^2 \sigma_M^{(q)^2} \mathbf{H}^{(q)} + 4\epsilon_p \mathbf{w}^{(q)^*} (\mathbf{w}^{(q)^*})^\top, \quad \mathbf{C}_0^{(M)} = \mathbf{0}.$$

By Lemma F.2, we have
$$\mathbb{E}\left[\boldsymbol{\eta}_t \boldsymbol{\eta}_t^\top\right] \preceq 2\left(\mathbf{B}_t^{(M)} + \mathbf{C}_t^{(M)}\right).$$

Then by Lemma C.1,

$$R_N^{(0)}/2 \le \underbrace{\frac{1}{N^2} \cdot \sum_{t=0}^{N-1}\sum_{k=t}^{N-1}\left\langle(\mathbf{I}-\gamma\mathbf{H}^{(q)})^{k-t}\mathbf{H}^{(q)}, \mathbf{B}_t^{(M)}\right\rangle}_{\text{bias}} + \underbrace{\frac{1}{N^2}\cdot\sum_{t=0}^{N-1}\sum_{k=t}^{N-1}\left\langle(\mathbf{I}-\gamma\mathbf{H}^{(q)})^{k-t}\mathbf{H}^{(q)},\mathbf{C}_t^{(M)}\right\rangle}_{\text{variance}}.$$

$$\text{(F.12)}$$

### F.3.1 ANALYSIS OF VARIANCE ERROR

Similar to (C.23),

$$\begin{aligned}
\mathbf{C}_t^{(M)} &= ((1+8\epsilon_p)\mathcal{I} - \gamma\mathcal{T}_B^{(q)} + \tilde{\epsilon}\gamma^2\mathcal{M}_B^{(q)})\mathbf{C}_{t-1}^{(M)} + \gamma^2\sigma_M^{(q)^2}\mathbf{H}^{(q)} + 4\epsilon_p\mathbf{w}^{(q)^*}(\mathbf{w}^{(q)^*})^\top\\
&\preceq ((1+8\epsilon_p)\mathcal{I} - \gamma\widetilde{\mathcal{T}}^{(q)}) \circ \mathbf{C}_{t-1}^{(M)} + \gamma^2(1+\tilde{\epsilon})\mathcal{M}_B^{(q)} \circ \mathbf{C}_{t-1}^{(M)}\\
&\quad + \gamma^2\sigma_M^{(q)^2}\mathbf{H}^{(q)} + 4\epsilon_p\mathbf{w}^{(q)^*}(\mathbf{w}^{(q)^*})^\top.
\end{aligned}$$

Recall that $\widetilde{\mathcal{T}}^{(q)} = \mathbf{H}^{(q)}\otimes\mathbf{I} + \mathbf{I}\otimes\mathbf{H}^{(q)} - \gamma\mathbf{H}^{(q)}\otimes\mathbf{H}^{(q)}$. Denote $\widetilde{\mathcal{T}}_2^{(q)} = \mathbf{H}^{(q)}\otimes\mathbf{I} + \mathbf{I}\otimes\mathbf{H}^{(q)} - \gamma/2\mathbf{H}^{(q)}\otimes\mathbf{H}^{(q)}$. Before proceeding, we find conditions for step size such that $8\epsilon_p\mathcal{I} \preceq \gamma\widetilde{\mathcal{T}}^{(q)} - \frac{\gamma}{2}\widetilde{\mathcal{T}}_2^{(q)}$. It suffices to restrict:

$$\gamma < \frac{2}{3\lambda_1^{(q)}}, \quad 8\epsilon_p \le \gamma\lambda_d^{(q)} - \frac{3}{4}\gamma^2\lambda_d^{(q)^2}. \tag{F.13}$$

Equipped with (F.13),

$$\mathbf{C}_t^{(M)} \preceq (\mathcal{I} - \frac{\gamma}{2}\widetilde{\mathcal{T}}_2^{(q)}) \circ \mathbf{C}_{t-1}^{(M)} + \gamma^2(1+\tilde{\epsilon})\mathcal{M}_B^{(q)} \circ \mathbf{C}_{t-1}^{(M)} + \gamma^2\sigma_M^{(q)^2}\mathbf{H}^{(q)} + 4\epsilon_p\mathbf{w}^{(q)^*}(\mathbf{w}^{(q)^*})^\top.$$

We would like to remark that, in the analysis of variance, to simplify $\mathbf{w}^{(q)^*}(\mathbf{w}^{(q)^*})^\top$, we assume the parameter prior
$$\mathbb{E}\left[\mathbf{w}^*\mathbf{w}^{*\top}\right] = \mathbf{I},$$

and take expectation over $\mathbf{w}^*$ on variance error. It follows that [7]

$$\mathbf{C}_t^{(M)} \preceq (\mathcal{I} - \frac{\gamma}{2}\widetilde{\mathcal{T}}_2^{(q)}) \circ \mathbf{C}_{t-1}^{(M)} + \gamma^2(1+\tilde{\epsilon})\mathcal{M}_B^{(q)} \circ \mathbf{C}_{t-1}^{(M)} + \gamma^2\sigma_M^{(q)^2}\mathbf{H}^{(q)} + 4\epsilon_p\mathbf{I}. \tag{F.14}$$

In subsequent analysis, we first derive a crude bound for $\mathbf{C}_t^{(M)}$, then establish a refined bound for $\mathbf{C}_t^{(M)}$ via (F.14).

**Lemma F.5** (A bound for $\mathcal{M}_B^{(q)} \circ \mathbf{C}_t^{(M)}$ with quantized master weight). *For $t \ge 1$, under Assumption 3.1, Assumption 3.2, Assumption 3.3, and Assumption 3.4, if the step size satisfies*

$$1 > 2(1+\tilde{\epsilon})\gamma\alpha_B\text{tr}(\mathbf{H}^{(q)}), \quad 8\epsilon_p \le \gamma\lambda_d^{(q)} - \frac{3}{4}\gamma^2\lambda_d^{(q)^2},$$

*then*

$$\mathcal{M}_B^{(q)} \circ \mathbf{C}_t^{(M)} \preceq \frac{2\gamma\sigma_M^{(q)^2} + \frac{8d\epsilon_p}{\gamma\text{tr}(\mathbf{H}^{(q)})}}{1 - 2(1+\tilde{\epsilon})\gamma\alpha_B\text{tr}(\mathbf{H}^{(q)})}\alpha_B\text{tr}(\mathbf{H}^{(q)})\mathbf{H}^{(q)}.$$

*Proof.* By (F.14),

$$\mathbf{C}_t^{(M)} \preceq \mathbf{C}_\infty^{(M)} \preceq \gamma\left(\frac{1}{2}\widetilde{\mathcal{T}}_2^{(q)} - \gamma(1+\tilde{\epsilon})\mathcal{M}_B^{(q)}\right)^{-1} \circ \underbrace{\left(\sigma_M^{(q)^2}\mathbf{H}^{(q)} + 4\frac{\epsilon_p}{\gamma^2}\mathbf{I}\right)}_{\boldsymbol{\Sigma}}.$$

---

[7]Actually, (F.14) holds under the expectation of $\mathbf{w}^*$. Slightly abusing notations, we omit $\mathbb{E}_{\mathbf{w}^*}$.

Applying $\widetilde{\mathcal{T}}_2^{(q)}$,

$$\widetilde{\mathcal{T}}_2^{(q)} \circ \mathbf{C}_\infty^{(M)} = (\widetilde{\mathcal{T}}_2^{(q)} - 2(1+\tilde{\epsilon})\gamma \mathcal{M}_B^{(q)}) \circ \mathbf{C}_\infty^{(M)} + 2(1+\tilde{\epsilon})\gamma \mathcal{M}_B^{(q)} \circ \mathbf{C}_\infty^{(M)}$$
$$\preceq 2\gamma\boldsymbol{\Sigma} + 2(1+\tilde{\epsilon})\gamma \mathcal{M}_B^{(q)} \circ \mathbf{C}_\infty^{(M)}.$$

Applying $(\widetilde{\mathcal{T}}_2^{(q)})^{-1}$ we have

$$\mathbf{C}_\infty^{(M)} \preceq 2\gamma\sigma_M^{(q)^2} \cdot (\widetilde{\mathcal{T}}_2^{(q)})^{-1} \circ \mathbf{H}^{(q)} + \frac{8\epsilon_p}{\gamma}(\widetilde{\mathcal{T}}_2^{(q)})^{-1} \circ \mathbf{I} + 2(1+\tilde{\epsilon})\gamma(\widetilde{\mathcal{T}}_2^{(q)})^{-1} \circ \mathcal{M}_B^{(q)} \circ \mathbf{C}_\infty^{(M)}$$
$$\preceq \sum_{t=0}^\infty \left(2(1+\tilde{\epsilon})\gamma(\widetilde{\mathcal{T}}_2^{(q)})^{-1} \circ \mathcal{M}_B^{(q)}\right)^t \circ (\widetilde{\mathcal{T}}_2^{(q)})^{-1} \circ \left(2\gamma\sigma_M^{(q)^2}\mathbf{H}^{(q)} + \frac{8\epsilon_p}{\gamma}\mathbf{I}\right).$$

Noticing that $(\widetilde{\mathcal{T}}_2^{(q)})^{-1} \circ \mathbf{H}^{(q)} \preceq \mathbf{I}$ and $(\widetilde{\mathcal{T}}_2^{(q)})^{-1} \circ \mathbf{I} \preceq (\mathbf{H}^{(q)})^{-1}$, we have

$$\mathbf{C}_\infty^{(M)} \preceq \sum_{t=0}^\infty \left(2(1+\tilde{\epsilon})\gamma(\widetilde{\mathcal{T}}_2^{(q)})^{-1} \circ \mathcal{M}_B^{(q)}\right)^t \circ (\widetilde{\mathcal{T}}_2^{(q)})^{-1} \circ \left(2\gamma\sigma_M^{(q)^2}\mathbf{H}^{(q)} + \frac{8\epsilon_p}{\gamma}\mathbf{I}\right)$$
$$\preceq \sum_{t=0}^\infty \left(2(1+\tilde{\epsilon})\gamma(\widetilde{\mathcal{T}}_2^{(q)})^{-1} \circ \mathcal{M}_B^{(q)}\right)^t \circ \left(2\gamma\sigma_M^{(q)^2}\mathbf{I} + \frac{8\epsilon_p}{\gamma}(\mathbf{H}^{(q)})^{-1}\right). \tag{F.15}$$

Firstly,

$$2(1+\tilde{\epsilon})\gamma(\widetilde{\mathcal{T}}_2^{(q)})^{-1} \circ \mathcal{M}_B^{(q)} \circ \mathbf{I} \preceq 2(1+\tilde{\epsilon})\gamma\alpha_B \text{tr}(\mathbf{H}^{(q)})\mathbf{I}. \tag{F.16}$$

Secondly,

$$2(1+\tilde{\epsilon})\gamma(\widetilde{\mathcal{T}}_2^{(q)})^{-1} \circ \mathcal{M}_B^{(q)} \circ (\mathbf{H}^{(q)})^{-1} \preceq 2(1+\tilde{\epsilon})\gamma\alpha_B \text{tr}(\mathbf{I})\mathbf{I}. \tag{F.17}$$

Therefore, together with (F.15), (F.16) and (F.17), we have

$$\mathbf{C}_\infty^{(M)} \preceq \sum_{t=0}^\infty \left(2(1+\tilde{\epsilon})\gamma(\widetilde{\mathcal{T}}_2^{(q)})^{-1} \circ \mathcal{M}_B^{(q)}\right)^{t-1} \circ \mathbf{I} \cdot \left[4(1+\tilde{\epsilon})\gamma^2\sigma_M^{(q)^2}\alpha_B \text{tr}(\mathbf{H}^{(q)}) + 16(1+\tilde{\epsilon})\alpha_B d\epsilon_p\right]$$
$$= \left(2\gamma\sigma_M^{(q)^2} + \frac{8d\epsilon_p}{\gamma\text{tr}(\mathbf{H}^{(q)})}\right) \sum_{t=0}^\infty \left(2(1+\tilde{\epsilon})\gamma(\widetilde{\mathcal{T}}_2^{(q)})^{-1} \circ \mathcal{M}_B^{(q)}\right)^{t-1} \circ \mathbf{I} \cdot 2(1+\tilde{\epsilon})\gamma\alpha_B \text{tr}(\mathbf{H}^{(q)})$$
$$\preceq \left(2\gamma\sigma_M^{(q)^2} + \frac{8d\epsilon_p}{\gamma\text{tr}(\mathbf{H}^{(q)})}\right) \sum_{t=0}^\infty \left[2(1+\tilde{\epsilon})\gamma\alpha_B \text{tr}(\mathbf{H}^{(q)})\right]^t \mathbf{I}$$
$$= \frac{2\gamma\sigma_M^{(q)^2} + \frac{8d\epsilon_p}{\gamma\text{tr}(\mathbf{H}^{(q)})}}{1 - 2(1+\tilde{\epsilon})\gamma\alpha_B \text{tr}(\mathbf{H}^{(q)})}\mathbf{I},$$

where the second inequality uses (F.16). At last,

$$\mathcal{M}_B^{(q)} \circ \mathbf{C}_t^{(M)} \preceq \frac{2\gamma\sigma_M^{(q)^2} + \frac{8d\epsilon_p}{\gamma\text{tr}(\mathbf{H}^{(q)})}}{1 - 2(1+\tilde{\epsilon})\gamma\alpha_B \text{tr}(\mathbf{H}^{(q)})}\alpha_B \text{tr}(\mathbf{H}^{(q)})\mathbf{H}^{(q)}.$$

$\square$

**Lemma F.6** (A bound for variance under multiplicative quantization with quantized master weight).
*Under Assumption 3.1, Assumption 3.2, Assumption 3.3, and Assumption 3.4, if the step size satisfies*

$$1 > 2(1+\tilde{\epsilon})\gamma\alpha_B \text{tr}(\mathbf{H}^{(q)}), \quad 8\epsilon_p \leq \gamma\lambda_d^{(q)} - \frac{3}{4}\gamma^2\lambda_d^{(q)^2},$$

*then*

$$\mathbb{E}_{\mathbf{w}^*}\text{variance} \leq \frac{4\sigma_M^{(q)^2} + 32(1+\tilde{\epsilon})d\epsilon_p\alpha_B/\gamma}{1 - 2(1+\tilde{\epsilon})\gamma\alpha_B \text{tr}(\mathbf{H}^{(q)})}\left(\frac{k^*}{N} + N\gamma^2 \cdot \sum_{i>k^*}(\lambda_i^{(q)})^2\right)$$
$$+ \frac{16\epsilon_p}{\gamma}\left(\sum_{i\leq k^*}\frac{1}{N\gamma\lambda_i^{(q)}} + N\gamma\sum_{i>k^*}\lambda_i^{(q)}\right).$$

*Proof.* We first provide a refined upper bound for $\mathbf{C}_t$. By (F.14) and Lemma F.5,

$$
\begin{aligned}
\mathbf{C}_t^{(M)} \preceq & (\mathcal{I} - \frac{\gamma}{2}\widetilde{\mathcal{T}}_2^{(q)}) \circ \mathbf{C}_{t-1}^{(M)} + \gamma^2(1+\tilde{\epsilon})\mathcal{M}_B^{(q)} \circ \mathbf{C}_{t-1}^{(M)} + \gamma^2\sigma_M^{(q)2}\mathbf{H}^{(q)} + 4\epsilon_p\mathbf{I} \\
\preceq & (\mathcal{I} - \frac{\gamma}{2}\widetilde{\mathcal{T}}_2^{(q)}) \circ \mathbf{C}_{t-1}^{(M)} + \gamma^2\sigma_M^{(q)2}\mathbf{H}^{(q)} + 4\epsilon_p\mathbf{I} \\
& + \gamma^2(1+\tilde{\epsilon})\frac{2\gamma\sigma_M^{(q)2} + \frac{8d\epsilon_p}{\gamma\mathrm{tr}(\mathbf{H}^{(q)})}}{1 - 2(1+\tilde{\epsilon})\gamma\alpha_B\mathrm{tr}(\mathbf{H}^{(q)})}\alpha_B\mathrm{tr}(\mathbf{H}^{(q)})\mathbf{H}^{(q)} \\
= & (\mathcal{I} - \frac{\gamma}{2}\widetilde{\mathcal{T}}_2^{(q)}) \circ \mathbf{C}_{t-1}^{(M)} + \frac{\gamma^2\sigma_M^{(q)2} + 8\gamma(1+\tilde{\epsilon})d\epsilon_p\alpha_B}{1 - 2(1+\tilde{\epsilon})\gamma\alpha_B\mathrm{tr}(\mathbf{H}^{(q)})}\mathbf{H}^{(q)} + 4\epsilon_p\mathbf{I} \\
= & \frac{\gamma^2\sigma_M^{(q)2} + 8\gamma(1+\tilde{\epsilon})d\epsilon_p\alpha_B}{1 - 2(1+\tilde{\epsilon})\gamma\alpha_B\mathrm{tr}(\mathbf{H}^{(q)})}\sum_{k=0}^{t-1}(\mathcal{I} - \frac{\gamma}{2}\widetilde{\mathcal{T}}_2^{(q)})^k \circ \mathbf{H}^{(q)} + 4\epsilon_p\sum_{k=0}^{t-1}(\mathcal{I} - \frac{\gamma}{2}\widetilde{\mathcal{T}}_2^{(q)})^k \circ \mathbf{I} \\
\preceq & \frac{\gamma^2\sigma_M^{(q)2} + 8\gamma(1+\tilde{\epsilon})d\epsilon_p\alpha_B}{1 - 2(1+\tilde{\epsilon})\gamma\alpha_B\mathrm{tr}(\mathbf{H}^{(q)})}\sum_{k=0}^{t-1}(\mathbf{I} - \frac{\gamma}{2}\mathbf{H}^{(q)})^k\mathbf{H}^{(q)} + 4\epsilon_p\sum_{k=0}^{t-1}(\mathbf{I} - \frac{\gamma}{2}\mathbf{H}^{(q)})^k \\
= & \underbrace{\frac{2\gamma\sigma_M^{(q)2} + 16(1+\tilde{\epsilon})d\epsilon_p\alpha_B}{1 - 2(1+\tilde{\epsilon})\gamma\alpha_B\mathrm{tr}(\mathbf{H}^{(q)})}(\mathbf{I} - (\mathbf{I} - \frac{\gamma}{2}\mathbf{H}^{(q)})^t)}_{\mathbf{V}_1} + \underbrace{\frac{8\epsilon_p}{\gamma}(\mathbf{I} - (\mathbf{I} - \frac{\gamma}{2}\mathbf{H}^{(q)})^t)(\mathbf{H}^{(q)})^{-1}}_{\mathbf{V}_2}.
\end{aligned}
$$
(F.18)

Recall (F.12),

$$
\mathbb{E}_{\mathbf{w}^*}\mathrm{variance} \leq \frac{1}{N^2} \cdot \sum_{t=0}^{N-1}\sum_{k=t}^{N-1}\left\langle(\mathbf{I} - \gamma\mathbf{H}^{(q)})^{k-t}\mathbf{H}^{(q)}, \mathbf{V}_1 + \mathbf{V}_2\right\rangle.
$$

Regarding $\mathbf{V}_1$,

$$
\begin{aligned}
& \frac{1}{N^2} \cdot \sum_{t=0}^{N-1}\sum_{k=t}^{N-1}\left\langle(\mathbf{I} - \gamma\mathbf{H}^{(q)})^{k-t}\mathbf{H}^{(q)}, \mathbf{V}_1\right\rangle \\
\leq & \frac{1}{N^2} \cdot \sum_{t=0}^{N-1}\sum_{k=t}^{N-1}\left\langle(\mathbf{I} - \frac{\gamma}{2}\mathbf{H}^{(q)})^{k-t}\mathbf{H}^{(q)}, \mathbf{V}_1\right\rangle \\
\leq & \frac{2}{N^2\gamma}\frac{2\gamma\sigma_M^{(q)2} + 16(1+\tilde{\epsilon})d\epsilon_p\alpha_B}{1 - 2(1+\tilde{\epsilon})\gamma\alpha_B\mathrm{tr}(\mathbf{H}^{(q)})}\sum_{t=0}^{N-1}\left\langle\mathbf{I} - (\mathbf{I} - \frac{\gamma}{2}\mathbf{H}^{(q)})^{N-t}, \mathbf{I} - (\mathbf{I} - \frac{\gamma}{2}\mathbf{H}^{(q)})^t\right\rangle \\
= & \frac{2}{N^2\gamma}\frac{2\gamma\sigma_M^{(q)2} + 16(1+\tilde{\epsilon})d\epsilon_p\alpha_B}{1 - 2(1+\tilde{\epsilon})\gamma\alpha_B\mathrm{tr}(\mathbf{H}^{(q)})}\sum_{t=0}^{N-1}\sum_i(1 - (1 - \frac{\gamma}{2}\lambda_i^{(q)})^{N-t})(1 - (1 - \frac{\gamma}{2}\lambda_i^{(q)})^t) \\
\leq & \frac{2}{N^2\gamma}\frac{2\gamma\sigma_M^{(q)2} + 16(1+\tilde{\epsilon})d\epsilon_p\alpha_B}{1 - 2(1+\tilde{\epsilon})\gamma\alpha_B\mathrm{tr}(\mathbf{H}^{(q)})}\sum_{t=0}^{N-1}\sum_i\left[1 - (1 - \frac{\gamma}{2}\lambda_i^{(q)})^N\right]^2 \\
\leq & \frac{2}{N\gamma}\frac{2\gamma\sigma_M^{(q)2} + 16(1+\tilde{\epsilon})d\epsilon_p\alpha_B}{1 - 2(1+\tilde{\epsilon})\gamma\alpha_B\mathrm{tr}(\mathbf{H}^{(q)})}\sum_i\min\left\{1, \frac{\gamma^2N^2}{4}\lambda_i^{(q)2}\right\} \\
\leq & \frac{2}{N\gamma}\frac{2\gamma\sigma_M^{(q)2} + 16(1+\tilde{\epsilon})d\epsilon_p\alpha_B}{1 - 2(1+\tilde{\epsilon})\gamma\alpha_B\mathrm{tr}(\mathbf{H}^{(q)})}\sum_i\min\left\{1, \gamma^2N^2\lambda_i^{(q)2}\right\} \\
= & \frac{4\sigma_M^{(q)2} + 32(1+\tilde{\epsilon})d\epsilon_p\alpha_B/\gamma}{1 - 2(1+\tilde{\epsilon})\gamma\alpha_B\mathrm{tr}(\mathbf{H}^{(q)})}\left(\frac{k^*}{N} + N\gamma^2 \cdot \sum_{i>k^*}(\lambda_i^{(q)})^2\right).
\end{aligned}
$$
(F.19)

Regarding $\mathbf{V}_2$,

$$
\begin{aligned}
&\frac{1}{N^2} \cdot \sum_{t=0}^{N-1} \sum_{k=t}^{N-1} \left\langle (\mathbf{I} - \gamma \mathbf{H}^{(q)})^{k-t} \mathbf{H}^{(q)}, \mathbf{V}_2 \right\rangle \\
\leq& \frac{2}{N^2\gamma} \cdot \sum_{t=0}^{N-1} \left\langle \mathbf{I} - (\mathbf{I} - \frac{\gamma}{2}\mathbf{H}^{(q)})^{N-t}, \mathbf{V}_2 \right\rangle \\
=& \frac{8\epsilon_p}{\gamma} \frac{2}{N^2\gamma} \cdot \sum_{t=0}^{N-1} \left\langle \mathbf{I} - (\mathbf{I} - \frac{\gamma}{2}\mathbf{H}^{(q)})^{N-t}, (\mathbf{I} - (\mathbf{I} - \frac{\gamma}{2}\mathbf{H}^{(q)})^{t})(\mathbf{H}^{(q)})^{-1} \right\rangle \\
=& \frac{8\epsilon_p}{\gamma} \frac{2}{N^2\gamma} \cdot \sum_{t=0}^{N-1} \sum_i \left[ 1 - (1 - \frac{\gamma}{2}\lambda_i^{(q)})^{N-t} \right] \left[ 1 - (1 - \frac{\gamma}{2}\lambda_i^{(q)})^{t} \right] \frac{1}{\lambda_i^{(q)}} \\
\leq& \frac{16\epsilon_p}{N\gamma^2} \sum_i \min\left\{ \frac{1}{\lambda_i^{(q)}}, N^2\gamma^2\lambda_i^{(q)} \right\} \\
=& \frac{16\epsilon_p}{\gamma} \left( \sum_{i \leq k^*} \frac{1}{N\gamma\lambda_i^{(q)}} + N\gamma \sum_{i > k^*} \lambda_i^{(q)} \right).
\end{aligned} \tag{F.20}
$$

Together with (F.12), (F.18), (F.19) and (F.20),

$$
\mathbb{E}_{\mathbf{w}^*} \text{variance} \leq \frac{4\sigma_M^{(q)^2} + 32(1+\tilde{\epsilon})d\epsilon_p\alpha_B/\gamma}{1 - 2(1+\tilde{\epsilon})\gamma\alpha_B\text{tr}(\mathbf{H}^{(q)})} \left( \frac{k^*}{N} + N\gamma^2 \cdot \sum_{i > k^*} (\lambda_i^{(q)})^2 \right) + \frac{16\epsilon_p}{\gamma} \left( \sum_{i \leq k^*} \frac{1}{N\gamma\lambda_i^{(q)}} + N\gamma \sum_{i > k^*} \lambda_i^{(q)} \right).
$$

$\square$

### F.3.2 ANALYSIS OF BIAS ERROR

Recall that

$$
\mathbf{B}_t^{(M)} := ((1 + 8\epsilon_p)\mathcal{I} - \gamma\mathcal{T}_B^{(q)} + \tilde{\epsilon}\gamma^2\mathcal{M}_B^{(q)})^t \circ \mathbf{B}_0^{(M)}, \quad \mathbf{B}_0^{(M)} = \mathbf{B}_0 = \mathbb{E}\left[ \boldsymbol{\eta}_0\boldsymbol{\eta}_0^\top \right],
$$

From (C.14) we deduce that

$$
\text{bias} \leq \frac{1}{\gamma N^2} \left\langle \mathbf{I} - (\mathbf{I} - \gamma\mathbf{H}^{(q)})^N, \sum_{t=0}^{N-1} \mathbf{B}_t^{(M)} \right\rangle. \tag{F.21}
$$

Denote $\mathbf{S}_n^{(M)} = \sum_{t=0}^{n-1} \mathbf{B}_t^{(M)}$. Motivated by Lemma C.7,

$$
\begin{aligned}
\mathbf{S}_t^{(M)} =& ((1 + 8\epsilon_p)\mathcal{I} - \gamma\mathcal{T}_B^{(q)} + \tilde{\epsilon}\gamma^2\mathcal{M}_B^{(q)}) \circ \mathbf{S}_{t-1}^{(M)} + \mathbf{B}_0 \\
=& ((1 + 8\epsilon_p)\mathcal{I} - \gamma\widetilde{\mathcal{T}}^{(q)}) \circ \mathbf{S}_{t-1}^{(M)} + \gamma(\widetilde{\mathcal{T}}^{(q)} - \mathcal{T}_B^{(q)}) \circ \mathbf{S}_{t-1}^{(M)} + \tilde{\epsilon}\gamma^2\mathcal{M}_B^{(q)} \circ \mathbf{S}_{t-1}^{(M)} + \mathbf{B}_0 \\
=& ((1 + 8\epsilon_p)\mathcal{I} - \gamma\widetilde{\mathcal{T}}^{(q)}) \circ \mathbf{S}_{t-1}^{(M)} + \gamma^2((1+\tilde{\epsilon})\mathcal{M}_B^{(q)} - \widetilde{\mathcal{M}}^{(q)}) \circ \mathbf{S}_{t-1}^{(M)} + \mathbf{B}_0 \\
\preceq& ((1 + 8\epsilon_p)\mathcal{I} - \gamma\tilde{\mathcal{T}}^{(q)}) \circ \mathbf{S}_{t-1}^{(M)} + (1+\tilde{\epsilon})\gamma^2\mathcal{M}_B^{(q)} \circ \mathbf{S}_{t-1}^{(M)} + \mathbf{B}_0.
\end{aligned}
$$

Similar to the analysis of variance error, we consider

$$
\gamma < \frac{2}{3\lambda_1^{(q)}}, \quad 8\epsilon_p \leq \gamma\lambda_d^{(q)} - \frac{3}{4}\gamma^2\lambda_d^{(q)^2}.
$$

It follows that $8\epsilon_p\mathcal{I} \preceq \gamma\widetilde{\mathcal{T}}^{(q)} - \frac{\gamma}{2}\widetilde{\mathcal{T}}_2^{(q)}$ and hence

$$
\mathbf{S}_t^{(M)} \preceq (\mathcal{I} - \frac{\gamma}{2}\widetilde{\mathcal{T}}_2^{(q)}) \circ \mathbf{S}_{t-1}^{(M)} + (1+\tilde{\epsilon})\gamma^2\mathcal{M}_B^{(q)} \circ \mathbf{S}_{t-1}^{(M)} + \mathbf{B}_0. \tag{F.22}
$$

**Lemma F.7** (A bound for $\mathcal{M}_B^{(q)} \circ \mathbf{S}_t^{(M)}$ with quantized master weight). *For $1 \le t \le N$, under Assumption 3.1, Assumption 3.2, Assumption 3.3, and Assumption 3.4, if the step size satisfies*

$$1 > 2(1 + \tilde{\epsilon})\gamma\alpha_B \mathrm{tr}(\mathbf{H}^{(q)}), \quad 8\epsilon_p \le \gamma\lambda_d^{(q)} - \frac{3}{4}\gamma^2 \lambda_d^{(q)\,2},$$

*then*

$$\mathcal{M}_B^{(q)} \circ \mathbf{S}_t^{(M)} \preceq \frac{2\alpha_B \cdot \mathrm{tr}\left(\left[\mathcal{I} - (\mathcal{I} - \frac{\gamma}{2}\widetilde{\mathcal{T}}_2^{(q)})^t\right] \circ \mathbf{B}_0\right)}{\gamma(1 - 2(1 + \tilde{\epsilon})\gamma\alpha_B \mathrm{tr}(\mathbf{H}^{(q)}))} \cdot \mathbf{H}^{(q)}.$$

*Proof.* From (F.22),

$$\mathbf{S}_t^{(M)} \preceq \sum_{k=0}^{t-1} \left(\mathcal{I} - \frac{\gamma}{2}\widetilde{\mathcal{T}}_2^{(q)} + (1 + \tilde{\epsilon})\gamma^2 \mathcal{M}_B^{(q)}\right)^k \circ \mathbf{B}_0$$

$$= \frac{1}{\gamma}\left(\frac{1}{2}\widetilde{\mathcal{T}}_2^{(q)} - (1 + \tilde{\epsilon})\gamma\mathcal{M}_B^{(q)}\right)^{-1} \circ \underbrace{\left[\mathcal{I} - \left(\mathcal{I} - \frac{\gamma}{2}\widetilde{\mathcal{T}}_2^{(q)} + (1 + \tilde{\epsilon})\gamma^2 \mathcal{M}_B^{(q)}\right)^t\right] \circ \mathbf{B}_0}_{\mathbf{A}}.$$

Applying $\widetilde{\mathcal{T}}_2^{(q)}$, we have

$$\widetilde{\mathcal{T}}_2^{(q)} \circ \left(\frac{1}{2}\widetilde{\mathcal{T}}_2^{(q)} - (1 + \tilde{\epsilon})\gamma\mathcal{M}_B^{(q)}\right)^{-1} \circ \mathbf{A}$$

$$= 2\mathbf{A} + 2(1 + \tilde{\epsilon})\gamma\mathcal{M}_B^{(q)} \circ \left(\frac{1}{2}\widetilde{\mathcal{T}}_2^{(q)} - (1 + \tilde{\epsilon})\gamma\mathcal{M}_B^{(q)}\right)^{-1} \circ \mathbf{A}.$$

Applying $(\widetilde{\mathcal{T}}_2^{(q)})^{-1}$, we have

$$\left(\frac{1}{2}\widetilde{\mathcal{T}}_2^{(q)} - (1 + \tilde{\epsilon})\gamma\mathcal{M}_B^{(q)}\right)^{-1} \circ \mathbf{A}$$

$$= 2(\widetilde{\mathcal{T}}_2^{(q)})^{-1} \circ \mathbf{A} + 2(\widetilde{\mathcal{T}}_2^{(q)})^{-1} \circ (1 + \tilde{\epsilon})\gamma\mathcal{M}_B^{(q)} \circ \left(\frac{1}{2}\widetilde{\mathcal{T}}_2^{(q)} - (1 + \tilde{\epsilon})\gamma\mathcal{M}_B^{(q)}\right)^{-1} \circ \mathbf{A}.$$

Applying $\mathcal{M}_B^{(q)}$, we have

$$\mathcal{M}_B^{(q)} \circ \left(\frac{1}{2}\widetilde{\mathcal{T}}_2^{(q)} - (1 + \tilde{\epsilon})\gamma\mathcal{M}_B^{(q)}\right)^{-1} \circ \mathbf{A}$$

$$= 2\mathcal{M}_B^{(q)} \circ (\widetilde{\mathcal{T}}_2^{(q)})^{-1} \circ (1 + \tilde{\epsilon})\gamma\mathcal{M}_B^{(q)} \circ \left(\frac{1}{2}\widetilde{\mathcal{T}}_2^{(q)} - (1 + \tilde{\epsilon})\gamma\mathcal{M}_B^{(q)}\right)^{-1} \circ \mathbf{A} + 2\mathcal{M}_B^{(q)} \circ (\widetilde{\mathcal{T}}_2^{(q)})^{-1} \circ \mathbf{A}.$$

It follows that

$$\mathcal{M}_B^{(q)} \circ \left(\frac{1}{2}\widetilde{\mathcal{T}}_2^{(q)} - (1 + \tilde{\epsilon})\gamma\mathcal{M}_B^{(q)}\right)^{-1} \circ \mathbf{A}$$

$$= \sum_{t=0}^{\infty} \left(2(1 + \tilde{\epsilon})\gamma\mathcal{M}_B^{(q)} \circ (\widetilde{\mathcal{T}}_2^{(q)})^{-1}\right)^t \circ 2\mathcal{M}_B^{(q)} \circ (\widetilde{\mathcal{T}}_2^{(q)})^{-1} \circ \mathbf{A}.$$

(F.23)

By Assumption 3.3,

$$\mathcal{M}_B^{(q)} \circ (\widetilde{\mathcal{T}}_2^{(q)})^{-1} \circ \mathbf{A} \preceq \alpha_B \, \mathrm{tr}(\mathbf{H}^{(q)}(\widetilde{\mathcal{T}}_2^{(q)})^{-1} \circ \mathbf{A})\mathbf{H}^{(q)}$$

$$= \alpha_B \frac{\gamma}{2} \, \mathrm{tr}\left(\sum_{t=0}^{\infty} \mathbf{H}^{(q)}(\mathbf{I} - \frac{\gamma}{2}\mathbf{H}^{(q)})^t \mathbf{A}(\mathbf{I} - \frac{\gamma}{2}\mathbf{H}^{(q)})^t\right) \mathbf{H}^{(q)}$$

$$= \alpha_B \mathrm{tr}\left(\mathbf{H}^{(q)}(2\mathbf{H}^{(q)} - \frac{\gamma}{2}(\mathbf{H}^{(q)})^2)^{-1}\mathbf{A}\right) \mathbf{H}^{(q)}$$

$$\preceq \alpha_B \mathrm{tr}(\mathbf{A})\mathbf{H}^{(q)},$$

together with (F.23), $(\widetilde{\mathcal{T}}_2^{(q)})^{-1} \circ \mathbf{H}^{(q)} \preceq \mathbf{I}$ and $\mathcal{M}_B^{(q)} \circ \mathbf{I} \preceq \alpha_B \operatorname{tr}(\mathbf{H}^{(q)})\mathbf{H}^{(q)}$, we have

$$\mathcal{M}_B^{(q)} \circ \left(\frac{1}{2}\widetilde{\mathcal{T}}_2^{(q)} - (1+\widetilde{\epsilon})\gamma \mathcal{M}_B^{(q)}\right)^{-1} \circ \mathbf{A}$$

$$\preceq \sum_{t=0}^{\infty} \left(2(1+\widetilde{\epsilon})\gamma \mathcal{M}_B^{(q)} \circ (\widetilde{\mathcal{T}}_2^{(q)})^{-1}\right)^t \circ 2\alpha_B \operatorname{tr}(\mathbf{A})\mathbf{H}^{(q)}$$

$$= 2\alpha_B \operatorname{tr}(\mathbf{A}) \sum_{t=0}^{\infty} \left(2(1+\widetilde{\epsilon})\gamma \mathcal{M}_B^{(q)} \circ (\widetilde{\mathcal{T}}_2^{(q)})^{-1}\right)^t \circ \mathbf{H}^{(q)}$$

$$\preceq 2\alpha_B \operatorname{tr}(\mathbf{A}) \sum_{t=0}^{\infty} \left(2(1+\widetilde{\epsilon})\gamma \alpha_B \operatorname{tr}(\mathbf{H}^{(q)})\right)^t \mathbf{H}^{(q)}$$

$$= \frac{2\alpha_B \operatorname{tr}(\mathbf{A})}{1 - 2(1+\widetilde{\epsilon})\gamma \alpha_B \operatorname{tr}(\mathbf{H}^{(q)})} \mathbf{H}^{(q)}.$$

Therefore, recall that

$$\mathbf{A} = \left[\mathcal{I} - \left(\mathcal{I} - \frac{\gamma}{2}\widetilde{\mathcal{T}}_2^{(q)} + (1+\widetilde{\epsilon})\gamma^2 \mathcal{M}_B^{(q)}\right)^t\right] \circ \mathbf{B}_0 \preceq \left[\mathcal{I} - \left(\mathcal{I} - \frac{\gamma}{2}\widetilde{\mathcal{T}}_2^{(q)}\right)^t\right] \circ \mathbf{B}_0,$$

we have

$$\mathcal{M}_B^{(q)} \circ \mathbf{S}_t^{(M)} \preceq \gamma^{-1} \frac{2\alpha_B \operatorname{tr}(\mathbf{A})}{1 - 2(1+\widetilde{\epsilon})\gamma \alpha_B \operatorname{tr}(\mathbf{H}^{(q)})} \cdot \mathbf{H}^{(q)} \preceq \frac{2\alpha_B \cdot \operatorname{tr}\left(\left[\mathcal{I} - (\mathcal{I} - \frac{\gamma}{2}\widetilde{\mathcal{T}}_2^{(q)})^t\right] \circ \mathbf{B}_0\right)}{\gamma(1 - 2(1+\widetilde{\epsilon})\gamma \alpha_B \operatorname{tr}(\mathbf{H}^{(q)}))} \cdot \mathbf{H}^{(q)}.$$

$\square$

Together with (F.22) and Lemma F.7, we are now ready to bound the bias error with quantized master weight.

**Lemma F.8** (A bound for bias under multiplicative quantization with quantized master weight). *Under Assumption 3.1, Assumption 3.2, Assumption 3.3, and Assumption 3.4, if the stepsize satisfies*

$$1 > 2(1+\widetilde{\epsilon})\gamma \alpha_B \operatorname{tr}(\mathbf{H}^{(q)}), \quad 8\epsilon_p \leq \gamma \lambda_d^{(q)} - \frac{3}{4}\gamma^2 \lambda_d^{(q)2},$$

*then*

$$\operatorname{bias} \leq \frac{8(1+\widetilde{\epsilon})\alpha_B \cdot \left(\frac{\|\mathbf{w}_0 - \mathbf{w}^{(q)*}\|_{\mathbf{I}_{0:k^*}}^2}{N\gamma} + \|\mathbf{w}_0 - \mathbf{w}^{(q)*}\|_{\mathbf{H}_{k^*:\infty}}^2\right)}{1 - 2(1+\widetilde{\epsilon})\gamma \alpha_B \operatorname{tr}(\mathbf{H}^{(q)})} \left(\frac{k^*}{N} + N\gamma^2 \sum_{i>k^*} \lambda_i^{(q)2}\right)$$

$$+ \frac{4}{\gamma^2 N^2} \cdot \|\mathbf{w}_0 - \mathbf{w}^{(q)*}\|_{(\mathbf{H}_{0:k^*}^{(q)})^{-1}}^2 + 4\|\mathbf{w}_0 - \mathbf{w}^{(q)*}\|_{\mathbf{H}_{k^*:\infty}^{(q)}}^2.$$

*Proof.* By (F.22) and Lemma F.7,

$$\mathbf{S}_t^{(M)} \preceq (\mathcal{I} - \frac{\gamma}{2}\widetilde{\mathcal{T}}_2^{(q)}) \circ \mathbf{S}_{t-1}^{(M)} + (1+\widetilde{\epsilon})\gamma^2 \mathcal{M}_B^{(q)} \circ \mathbf{S}_{t-1}^{(M)} + \mathbf{B}_0$$

$$\preceq (\mathcal{I} - \frac{\gamma}{2}\widetilde{\mathcal{T}}_2^{(q)}) \circ \mathbf{S}_{t-1}^{(M)} + (1+\widetilde{\epsilon})\gamma^2 \frac{2\alpha_B \cdot \operatorname{tr}\left(\left[\mathcal{I} - (\mathcal{I} - \frac{\gamma}{2}\widetilde{\mathcal{T}}_2^{(q)})^N\right] \circ \mathbf{B}_0\right)}{\gamma(1 - 2(1+\widetilde{\epsilon})\gamma \alpha_B \operatorname{tr}(\mathbf{H}^{(q)}))} \cdot \mathbf{H}^{(q)} + \mathbf{B}_0$$

$$= \sum_{k=0}^{t-1} (\mathcal{I} - \frac{\gamma}{2}\widetilde{\mathcal{T}}_2^{(q)})^k \circ \left((1+\widetilde{\epsilon})\gamma^2 \frac{2\alpha_B \cdot \operatorname{tr}\left(\left[\mathcal{I} - (\mathcal{I} - \frac{\gamma}{2}\widetilde{\mathcal{T}}_2^{(q)})^N\right] \circ \mathbf{B}_0\right)}{\gamma(1 - 2(1+\widetilde{\epsilon})\gamma \alpha_B \operatorname{tr}(\mathbf{H}^{(q)}))} \cdot \mathbf{H}^{(q)} + \mathbf{B}_0\right)$$

$$= \sum_{k=0}^{t-1} (\mathbf{I} - \frac{\gamma}{2}\mathbf{H}^{(q)})^k \left((1+\widetilde{\epsilon})\gamma^2 \frac{2\alpha_B \cdot \operatorname{tr}\left(\left[\mathcal{I} - (\mathcal{I} - \frac{\gamma}{2}\widetilde{\mathcal{T}}_2^{(q)})^N\right] \circ \mathbf{B}_0\right)}{\gamma(1 - 2(1+\widetilde{\epsilon})\gamma \alpha_B \operatorname{tr}(\mathbf{H}^{(q)}))} \cdot \mathbf{H}^{(q)} + \mathbf{B}_0\right)(\mathbf{I} - \frac{\gamma}{2}\mathbf{H}^{(q)})^k.$$

Before providing our upper bound for the bias error, we denote

$$\mathbf{B}_{a,b} := \mathbf{B}_a - (\mathbf{I} - \frac{\gamma}{2}\mathbf{H}^{(q)})^{b-a}\mathbf{B}_a(\mathbf{I} - \frac{\gamma}{2}\mathbf{H}^{(q)})^{b-a}.$$

Recall from (F.21) that

$$\text{bias} \leq \frac{1}{\gamma N^2}\left\langle \mathbf{I} - (\mathbf{I} - \gamma\mathbf{H}^{(q)})^N, \sum_{t=0}^{N-1}\mathbf{B}_t^{(M)}\right\rangle,$$

we have

$$\text{bias} \leq \frac{2}{\gamma N^2}\left\langle \mathbf{I} - (\mathbf{I} - \frac{\gamma}{2}\mathbf{H}^{(q)})^N, \sum_{t=0}^{N-1}\mathbf{B}_t^{(M)}\right\rangle$$

$$\leq \frac{2}{\gamma N^2}\left\langle \mathbf{I} - (\mathbf{I} - \frac{\gamma}{2}\mathbf{H}^{(q)})^N, \sum_{k=0}^{N-1}(\mathbf{I} - \frac{\gamma}{2}\mathbf{H}^{(q)})^k\left(\frac{2(1+\tilde{\epsilon})\gamma\alpha_B \cdot \text{tr}\,(\mathbf{B}_{0,N})}{1 - 2(1+\tilde{\epsilon})\gamma\alpha_B\,\text{tr}(\mathbf{H}^{(q)})}\cdot\mathbf{H}^{(q)} + \mathbf{B}_0\right)(\mathbf{I} - \frac{\gamma}{2}\mathbf{H}^{(q)})^k\right\rangle$$

$$= \frac{2}{\gamma N^2}\sum_{k=0}^{N-1}\left\langle (\mathbf{I} - \frac{\gamma}{2}\mathbf{H}^{(q)})^{2k} - (\mathbf{I} - \frac{\gamma}{2}\mathbf{H}^{(q)})^{N+2k}, \frac{2(1+\tilde{\epsilon})\gamma\alpha_B \cdot \text{tr}\,(\mathbf{B}_{0,N})}{1 - 2(1+\tilde{\epsilon})\gamma\alpha_B\,\text{tr}(\mathbf{H}^{(q)})}\cdot\mathbf{H}^{(q)} + \mathbf{B}_0\right\rangle.$$

Note that

$$(\mathbf{I} - \frac{\gamma}{2}\mathbf{H}^{(q)})^{2k} - (\mathbf{I} - \frac{\gamma}{2}\mathbf{H}^{(q)})^{N+2k} \preceq (\mathbf{I} - \frac{\gamma}{2}\mathbf{H}^{(q)})^k - (\mathbf{I} - \frac{\gamma}{2}\mathbf{H}^{(q)})^{N+k},$$

we have

$$\text{bias} \leq \frac{2}{\gamma N^2}\sum_{k=0}^{N-1}\left\langle (\mathbf{I} - \frac{\gamma}{2}\mathbf{H}^{(q)})^k - (\mathbf{I} - \frac{\gamma}{2}\mathbf{H}^{(q)})^{N+k}, \frac{2(1+\tilde{\epsilon})\gamma\alpha_B \cdot \text{tr}\,(\mathbf{B}_{0,N})}{1 - 2(1+\tilde{\epsilon})\gamma\alpha_B\,\text{tr}(\mathbf{H}^{(q)})}\cdot\mathbf{H}^{(q)} + \mathbf{B}_0\right\rangle.$$

Therefore it suffices to bound the following two terms:

$$I_1 = \frac{2}{\gamma N^2}\sum_{k=0}^{N-1}\left\langle (\mathbf{I} - \frac{\gamma}{2}\mathbf{H}^{(q)})^k - (\mathbf{I} - \frac{\gamma}{2}\mathbf{H}^{(q)})^{N+k}, \frac{2(1+\tilde{\epsilon})\gamma\alpha_B \cdot \text{tr}\,(\mathbf{B}_{0,N})}{1 - 2(1+\tilde{\epsilon})\gamma\alpha_B\,\text{tr}(\mathbf{H}^{(q)})}\cdot\mathbf{H}^{(q)}\right\rangle,$$

$$I_2 = \frac{2}{\gamma N^2}\sum_{k=0}^{N-1}\left\langle (\mathbf{I} - \frac{\gamma}{2}\mathbf{H}^{(q)})^k - (\mathbf{I} - \frac{\gamma}{2}\mathbf{H}^{(q)})^{N+k}, \mathbf{B}_0\right\rangle.$$

Regarding $I_1$,

$$\sum_{k=0}^{N-1}\left\langle (\mathbf{I} - \frac{\gamma}{2}\mathbf{H}^{(q)})^k - (\mathbf{I} - \frac{\gamma}{2}\mathbf{H}^{(q)})^{N+k}, \mathbf{H}^{(q)}\right\rangle$$

$$= \sum_{k=0}^{N-1}\sum_i\left[(1 - \frac{\gamma}{2}\lambda_i^{(q)})^k - (1 - \frac{\gamma}{2}\lambda_i^{(q)})^{N+k}\right]\lambda_i^{(q)}$$

$$= \frac{2}{\gamma}\sum_i\left[[1 - (1 - \frac{\gamma}{2}\lambda_i^{(q)})^N] - [(1 - \frac{\gamma}{2}\lambda_i^{(q)})^N - (1 - \frac{\gamma}{2}\lambda_i^{(q)})^{2N}]\right]$$

$$= \frac{2}{\gamma}\sum_i\left(1 - (1 - \frac{\gamma}{2}\lambda_i^{(q)})^N\right)^2$$

$$\leq \frac{2}{\gamma}\sum_i\min\left\{1, N^2\gamma^2\lambda_i^{(q)^2}\right\}$$

$$\leq \frac{2}{\gamma}\left(k^* + N^2\gamma^2\sum_{i>k^*}\lambda_i^{(q)^2}\right),$$

Hence,

$$I_1 \leq \frac{4}{\gamma^2 N^2}\frac{2(1+\tilde{\epsilon})\gamma\alpha_B \cdot \text{tr}\,(\mathbf{B}_{0,N})}{1 - 2(1+\tilde{\epsilon})\gamma\alpha_B\,\text{tr}(\mathbf{H}^{(q)})}\left(k^* + N^2\gamma^2\sum_{i>k^*}\lambda_i^{(q)^2}\right).$$

Then we tackle $\text{tr}(\mathbf{B}_{0,N})$.

$$
\begin{aligned}
\text{tr}(\mathbf{B}_{0,N}) &= \text{tr}\left(\mathbf{B}_0 - (\mathbf{I} - \frac{\gamma}{2}\mathbf{H}^{(q)})^N \mathbf{B}_0 (\mathbf{I} - \frac{\gamma}{2}\mathbf{H}^{(q)})^N\right) \\
&= \sum_i \left(1 - (1 - \frac{\gamma}{2}\lambda_i^{(q)})^{2N}\right) \cdot \left(\langle \mathbf{w}_0 - \mathbf{w}^{(q)^*}, \mathbf{v}_i^{(q)}\rangle\right)^2 \\
&\leq \sum_i \min\{1, N\gamma\lambda_i^{(q)}\}\left(\langle \mathbf{w}_0 - \mathbf{w}^{(q)^*}, \mathbf{v}_i^{(q)}\rangle\right)^2 \\
&\leq \|\mathbf{w}_0 - \mathbf{w}^{(q)^*}\|^2_{\mathbf{I}_{0:k^*}} + N\gamma\|\mathbf{w}_0 - \mathbf{w}^{(q)^*}\|^2_{\mathbf{H}_{k^*:\infty}}.
\end{aligned}
\tag{F.24}
$$

Therefore,

$$
I_1 \leq \frac{4}{\gamma^2 N^2} \frac{2(1+\tilde{\epsilon})\gamma\alpha_B \cdot \left(\|\mathbf{w}_0 - \mathbf{w}^{(q)^*}\|^2_{\mathbf{I}_{0:k^*}} + N\gamma\|\mathbf{w}_0 - \mathbf{w}^{(q)^*}\|^2_{\mathbf{H}_{k^*:\infty}}\right)}{1 - 2(1+\tilde{\epsilon})\gamma\alpha_B\,\text{tr}(\mathbf{H}^{(q)})}\left(k^* + N^2\gamma^2\sum_{i>k^*}\lambda_i^{(q)^2}\right).
$$

Regarding $I_2$,

$$
\begin{aligned}
I_2 &= \frac{2}{\gamma N^2}\sum_{k=0}^{N-1}\left\langle (\mathbf{I} - \frac{\gamma}{2}\mathbf{H}^{(q)})^k - (\mathbf{I} - \frac{\gamma}{2}\mathbf{H}^{(q)})^{N+k}, \mathbf{B}_0\right\rangle \\
&= \frac{2}{\gamma N^2}\sum_{k=0}^{N-1}\sum_i\left[(1 - \frac{\gamma}{2}\lambda_i^{(q)})^k - (1 - \frac{\gamma}{2}\lambda_i^{(q)})^{N+k}\right]\omega_i^2 \\
&= \frac{4}{\gamma^2 N^2}\sum_i\left[1 - (1 - \frac{\gamma}{2}\lambda_i^{(q)})^N\right]^2\frac{\omega_i^2}{\lambda_i^{(q)}} \\
&\leq \frac{4}{\gamma^2 N^2}\sum_i\frac{\omega_i^2}{\lambda_i^{(q)}}\min\left\{1, \gamma^2 N^2(\lambda_i^{(q)})^2\right\} \\
&= \frac{4}{\gamma^2 N^2}\cdot\|\mathbf{w}_0 - \mathbf{w}^{(q)^*}\|^2_{(\mathbf{H}_{0:k^*}^{(q)})^{-1}} + 4\|\mathbf{w}_0 - \mathbf{w}^{(q)^*}\|^2_{\mathbf{H}_{k^*:\infty}^{(q)}}.
\end{aligned}
$$

Overall,

$$
\begin{aligned}
\text{bias} &\leq I_1 + I_2 \\
&\leq \frac{8(1+\tilde{\epsilon})\alpha_B \cdot \left(\frac{\|\mathbf{w}_0 - \mathbf{w}^{(q)^*}\|^2_{\mathbf{I}_{0:k^*}}}{N\gamma} + \|\mathbf{w}_0 - \mathbf{w}^{(q)^*}\|^2_{\mathbf{H}_{k^*:\infty}}\right)}{1 - 2(1+\tilde{\epsilon})\gamma\alpha_B\,\text{tr}(\mathbf{H}^{(q)})}\left(\frac{k^*}{N} + N\gamma^2\sum_{i>k^*}\lambda_i^{(q)^2}\right) \\
&\quad + \frac{4}{\gamma^2 N^2}\cdot\|\mathbf{w}_0 - \mathbf{w}^{(q)^*}\|^2_{(\mathbf{H}_{0:k^*}^{(q)})^{-1}} + 4\|\mathbf{w}_0 - \mathbf{w}^{(q)^*}\|^2_{\mathbf{H}_{k^*:\infty}^{(q)}}.
\end{aligned}
$$

$\square$

Based on the analysis for bias and variance error, we are ready to present the bounds for $R_N^{(0)}$ under multiplicative quantization with quantized master weight.

**Theorem F.3** (A bound for $R_N^{(0)}$ under multiplicative quantization with quantized master weight). *Suppose there exist $\epsilon_d, \epsilon_l, \epsilon_p, \epsilon_a$ and $\epsilon_o$ such that for any $i \in \{d, l, p, a, o\}$, quantization $\mathcal{Q}_i$ is $\epsilon_i$-multiplicative. Under Assumption 3.1, Assumption 3.2, Assumption 3.3, and Assumption 3.4, if the step size satisfies*

$$
1 > 2(1+\tilde{\epsilon})\gamma\alpha_B\text{tr}(\mathbf{H}^{(q)}), \quad 8\epsilon_p \leq \gamma\lambda_d^{(q)} - \frac{3}{4}\gamma^2\lambda_d^{(q)^2},
$$

*then taking expectation over* $\mathbf{w}^*$ *on* variance [8],

$$R_N^{(0)}/2 \leq \frac{4\sigma_M^{(q)2} + 32(1+\tilde{\epsilon})d\epsilon_p\alpha_B/\gamma}{1 - 2(1+\tilde{\epsilon})\gamma\alpha_B\mathrm{tr}(\mathbf{H}^{(q)})}\left(\frac{k^*}{N} + N\gamma^2 \cdot \sum_{i>k^*}(\lambda_i^{(q)})^2\right)$$

$$+ \frac{16\epsilon_p}{\gamma}\left(\sum_{i\leq k^*}\frac{1}{N\gamma\lambda_i^{(q)}} + N\gamma\sum_{i>k^*}\lambda_i^{(q)}\right)$$

$$+ \frac{8(1+\tilde{\epsilon})\alpha_B \cdot \left(\frac{\|\mathbf{w}_0 - \mathbf{w}^{(q)*}\|_{\mathbf{I}_{0:k^*}}^2}{N\gamma} + \|\mathbf{w}_0 - \mathbf{w}^{(q)*}\|_{\mathbf{H}_{k^*:\infty}}^2\right)}{1 - 2(1+\tilde{\epsilon})\gamma\alpha_B\,\mathrm{tr}(\mathbf{H}^{(q)})}\left(\frac{k^*}{N} + N\gamma^2\sum_{i>k^*}\lambda_i^{(q)2}\right)$$

$$+ \frac{4}{\gamma^2 N^2} \cdot \|\mathbf{w}_0 - \mathbf{w}^{(q)*}\|_{(\mathbf{H}_{0:k^*}^{(q)})^{-1}}^2 + 4\|\mathbf{w}_0 - \mathbf{w}^{(q)*}\|_{\mathbf{H}_{k^*:\infty}^{(q)}}^2,$$

*where*

$$\tilde{\epsilon} = 8\epsilon_o(1+\epsilon_p)(1+\epsilon_a) + 8\epsilon_p + 4\epsilon_a(1+\epsilon_p),$$

$$\sigma_M^{(q)2} = \frac{(1+4\epsilon_o)\sigma^2}{B} + \frac{\mathbb{E}_{\mathbf{w}^*}\|\mathbf{w}^*\|_{\mathbf{H}}^2}{1+\epsilon_d}\alpha_B\left(4\epsilon_o[(1+\epsilon_p)(1+\epsilon_a)+1] + 2\epsilon_a(1+\epsilon_p) + 4\epsilon_p\right).$$

*Proof.* The proof is completed by (F.12), Lemma F.6 and Lemma F.8. $\square$

# G   DETAILS OF ADDITIONAL EXPERIMENTS

## G.1   ADDITIONAL DATASETS

For the supplementary experiments, we consider both synthetic and real-world datasets.

**Synthetic dataset.**   We construct a synthetic regression dataset whose covariance spectrum follows an exponential decay. Specifically, the eigenvalues are given by

$$\lambda_i = e^{-i}, \qquad i = 1, 2, \ldots, d.$$

This allows us to examine the behavior of our method under rapidly decaying spectral structures, complementing the polynomial-decay setting used in the main paper.

**Real-world dataset: `Communities and Crime`.**   We additionally evaluate on the publicly available `Communities and Crime` dataset, which contains community-level statistics from across the United States. The features integrate socio-economic indicators from the 1990 U.S. Census, law-enforcement statistics from the 1990 LEMAS survey, and crime records from the 1995 FBI Uniform Crime Reporting (UCR) program. The task is a standard regression problem: predicting the *per-capita violent crime rate* from community attributes. The dataset contains about 2000 instances with 122 features.

## G.2   EXPERIMENTAL SETTINGS AND RESULTS

We describe below the protocol for each additional experiment and corresponding results. In both real-world dataset and synthetic datasets, we examine how do additive vs. multiplicative quantization affect learning (generalization) performance.

- **Real-world regression (Communities and Crime).** We apply both *additive* and *multiplicative* quantization schemes (with fixed quantization error level $\varepsilon = 0.01$) to the regression task on `Communities and Crime` dataset. For each quantization method, we evaluate the resulting population risk $\mathbb{E}_{\mathbf{x},y}\left[(y - \langle\mathbf{w},\mathbf{x}\rangle)^2\right]$. As illustrated in Figure 2(a), the results demonstrate that, unlike additive quantization, the multiplicative scheme successfully maintains the performance of full-precision SGD. This aligns with our theoretical finding that multiplicative quantization exhibits greater tolerance to quantization error level.

---

[8]Here we assume that $\mathbb{E}\mathbf{w}^*\mathbf{w}^{*\top} = \mathbf{I}$.

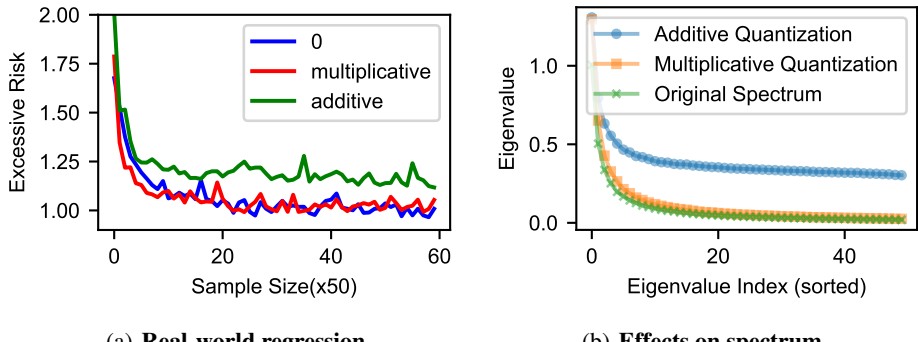

(a) **Real-world regression**          (b) **Effects on spectrum**

Figure 2: **Comparison between multiplicative quantization and additive quantization**. (a): Real-world regression (Communities and Crime). (b): Effect of quantization on data spectrum.

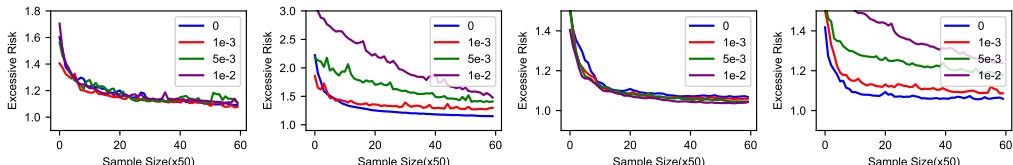

(a) **Multiplicative** (FP-like) (b) **Additive** (INT-like) (c) **Multiplicative** (FP-like) (d) **Additive** (INT-like)

Figure 3: **Generalization under quantization.** Test risk for SGD with iterate averaging under multiplicative (FP-like) vs. additive (INT-like) quantization. (a) and (b): vary the quantization level at fixed $B = 10$. (c) and (d): vary the quantization level under exponential decay.

- **Effect of quantization on data spectrum.** Using the same settings on `Communities and Crime` dataset, we record the resulting empirical covariance spectra to study how each quantization type perturbs the underlying eigenvalue structure. Results are shown in Figure 2(b). It is shown that additive errors errors dramatically distort the spectrum of effective data covariance while multiplicative quantization errors largely preserve the spectral structure. This visualization corroborates the specific mechanism by which additive and multiplicative quantization lead to distinct generalization behaviors.

- **Sensitivity analysis on batch size and spectral decay.** To demonstrate the robustness of our findings, we conduct additional experiments varying the batch size and data spectrum. First, we extend the batch size to $B = 10$ (with $d = 200$) and vary the quantization error level $\varepsilon \in \{0.001, 0.005, 0.01\}$. Second, we replace the polynomial-decay spectrum with an exponential-decay synthetic dataset while keeping other settings identical. The results, shown in Figures 3(a)–3(b) (batch size) and Figures 3(c)–3(d) (spectral decay), consistently mirror the findings in the main paper: multiplicative quantization preserves the generalization performance of full-precision SGD across various quantization error levels, whereas additive quantization suffers from performance degradation as the error level increases.

## H    THE USE OF LLMS

The use of large language models (LLMs) in this work was limited to linguistic polishing of the text (e.g., grammar, clarity, and readability) and was not involved in any research phases, from conceptualization and proofing to experimentation and interpretation.

