# OpenReview forum: "Learning under Quantization for High-Dimensional Linear Regression"
_ICLR.cc/2026/Conference — ICLR 2026 Poster_

### Official Review · Reviewer_MJwg · 2025-10-21

**Soundness:** 4
**Presentation:** 4
**Contribution:** 4
**Rating:** 8
**Confidence:** 4

**Summary:**

The paper presents a systematic theoretical analysis of the generalization performance of quantized Stochastic Gradient Descent (SGD) in the context of high-dimensional linear regression. It studies the effects of quantizing five key components: data, labels, parameters, activations, and output gradients. The main result is a precise, decomposed bound on the excess risk, which clearly separates the approximation error, bias error, variance error, and the unique quantization error. The authors contrast two major quantization error models: additive (related to INT) and multiplicative (related to FP). They use their bounds to provide conditions under which quantized SGD achieves performance comparable to its full-precision counterpart.

**Strengths:**

The paper tackles an important and novel theoretical problem with significant practical implications. The systematic analysis of five quantization targets is comprehensive. The derived generalization bounds for quantized SGD are the first of their kind to my knowledge. The comparison between additive and multiplicative quantization, particularly the insight into spectral distortion, is a major contribution. The paper is well-written, and the theoretical results are intuitively explained and justified.

**Weaknesses:**

1. The analysis is restricted to high-dimensional linear regression, leaving the extension to non-linear models (like neural networks) as a critical open question.
2. More significantly, the paper provides only an upper bound on the excess risk. As noted, this bound, especially for additive quantization (scaling with d and N), appears non-tight and is not compared directly against the numerical simulations, undermining the conclusiveness of the theoretical comparison between additive and multiplicative models. Proving a matching lower bound or providing a tighter analysis for the additive case is necessary to solidify the theoretical claims.

Overall, the paper makes a foundational contribution to the theoretical understanding of quantization's impact on learning performance. The novel bounds and the systematic approach to different quantization targets (data, parameters, etc.) provide invaluable, actionable theoretical guidance for the design of efficient learning systems. While the lack of a tighter lower bound and the restriction to the linear model are limitations, the complexity and significance of the problem tackled, combined with the depth of the current analysis, make this work a major theoretical advance worthy of publication. The authors should be encouraged to pursue a tighter analysis of the additive and multiplicative quantization errors in the revision.

**Questions:**

1.	Can the author comment on what they think would be the effect of the quantization error in nonlinear models and other algorithm designs, such as multipass SGD and momentum? Do they think that the implication of the result regarding multiplicative quantization will also carry in these settings?
2.	What does the notation $≲$ mean?
3.	Lines 355-361: Please indicate where the fourth term appears. It would be helpful to provide a proof sketch that explains all the pieces in the proof. I understand the analysis is similar to Zou et al (2023). However, it would be nice to have a summary here.
4.	Line 336: Is it clear that these errors remain O(1)? Please provide more intuition on why, rather than just directing to that appendix, as this is central to the analysis.
5.	Lines 393-397: I am not sure I understand this. The errors are not completely decoupled, as $\epsilon_a$, $\epsilon_o$, depend on $\epsilon_d$, $\epsilon_p$.
6.	Line 463: It would be nice to provide a clearer explanation and maybe plot the Hessian spectrum as well in both cases, to illustrate the effect.
Minor:
1.	Line 293 remove either “plays” or “relies”
2.	Line 394 \epsilon_0 should be \epsilon _o.

---

> ### Author Response · Authors · 2025-11-23
> **Response to Reviewer MJwg's comment (1/6)**
>
> Thank you for your positive feedback and we appreciate your strong support for the completeness of our work. We address your concerns in the following response:
>
> **Q1.1:** Can the author comment on what they think would be the effect of the quantization error in nonlinear models? Do they think that the implication of the result regarding multiplicative quantization will also carry in these settings?
>
> **A1.1:** Thanks for your comment. In nonlinear models, **the implication that multiplicative data quantization error does not scale with data dimension $d$ remains to be true**, as the approximation error originates from the algorithm-independent excess risk decomposition (Lemma A.2). We also believe **the key quantization effect** that multiplicative data quantization alleviates the data spectrum while additive data quantization does not, and other effects of quantization such as the noise amplification of parameters, activations and gradients quantization, **still hold true in several nonlinear cases** discussed as follows.
>
> - single neuron (ReLU / Leaky ReLU)
>
> 	Current analysis for linear regression **can be extended to (Leaky) ReLU regression (single neuron)** by combining our theoretical framework and existing techniques for ReLU regression under full-precision SGD [3]. Specifically, quantized SGD in ReLU regression is formulated as $$\mathbf{w} _ {t}=\mathbf{w} _ {t-1}+\gamma\mathcal{Q} _ o\left(\mathcal{Q} _ l(y _ {t})-\mathrm{ReLU}(\mathcal{Q} _ a(\mathcal{Q} _ d(\mathbf{x} _ {t})^{\top}\mathcal{Q} _ p(\mathbf{w} _ {t-1})))\right)\cdot\mathbf{1}[\mathcal{Q} _ d(\mathbf{x} _ {t})^{\top}\mathcal{Q} _ p(\mathbf{w} _ {t-1})>0]\mathcal{Q} _ d(\mathbf{x} _ {t}).$$
>
> 	Technically, it suffices to analyze a simplified Generalized Linear Model Perceptron (GLM-tron) algorithm which ignores the derivative of ReLU(·) in its updates [3]: $$\mathbf{w} _ {t}=\mathbf{w} _ {t-1}+\gamma\mathcal{Q} _ o\left(\mathcal{Q} _ l(y _ {t})-\mathrm{ReLU}(\mathcal{Q} _ a(\mathcal{Q} _ d(\mathbf{x} _ {t})^{\top}\mathcal{Q} _ p(\mathbf{w} _ {t-1})))\right)\mathcal{Q} _ d(\mathbf{x} _ {t}).$$ Similar to our Lemma A.1, deviation error propagation in ReLU regression can be formulated as $$\begin{aligned}&\mathbf{w} _ {t}-{\mathbf{w}^{(q)}}^* =\underbrace{\left(\mathbf{I}-\gamma\mathbf{1}[Q _ a(\mathcal{Q} _ d(\mathbf{x} _ {t})^{\top}\mathcal{Q} _ p(\mathbf{w} _ {t-1}))>0]\mathcal{Q} _ d(\mathbf{x} _ {t})\mathcal{Q} _ d(\mathbf{x} _ {t})^\top\right)(\mathbf{w} _ {t-1}-{\mathbf{w}^{(q)}}^* )} _ {\mathbf{c}}\newline+&\underbrace{\gamma \left(\mathbf{1}[Q _ a(\mathcal{Q} _ d(\mathbf{x} _ {t})^{\top}\mathcal{Q} _ p({\mathbf{w}^{(q)}}^* ))>0]-\mathbf{1}[Q _ a(\mathcal{Q} _ d(\mathbf{x} _ {t})^{\top}\mathcal{Q} _ p(\mathbf{w} _ {t-1}))>0]\right) \mathcal{Q} _ d(\mathbf{x} _ {t})\mathcal{Q} _ d(\mathbf{x} _ {t})^{\top}{\mathbf{w}^{(q)}}^* } _ {\mathbf{f}}\newline+&\underbrace{\gamma\left[\pmb{\epsilon} _ t^{(o)}-(\pmb{\epsilon} _ t^{(a)}+\mathcal{Q} _ d(\mathbf{x} _ {t})^{\top}\pmb{\epsilon} _ {t-1}^{(p)})\mathbf{1}[Q _ a(\mathcal{Q} _ d(\mathbf{x} _ {t})^{\top}\mathcal{Q} _ p(\mathbf{w} _ {t-1}))>0]+\pmb{\xi} _ t\right]\mathcal{Q} _ d(\mathbf{x} _ {t})} _ {\mathbf{n}}\end{aligned},$$ where the three parts can be understood as a contraction term ($\mathbf{c}$), a fluctuation term ($\mathbf{f}$) and a noise term ($\mathbf{n}$). It is worth noting that $\mathbf{c}$ and $\mathbf{n}$ correspond to the propagation in linear regression while $\mathbf{f}$ is an extra error. Utilizing mild assumptions in [3] and our techniques in Lemma D.2 and Lemma D.3, **the (diagonal) propagation of** $\mathbf{A} _ t:=\mathbb{E}\left[(\mathbf{w} _ {t}-{\mathbf{w}^{(q)}}^* )\otimes (\mathbf{w} _ {t}-{\mathbf{w}^{(q)}}^* )\right]$ **can be upper bounded by a linear update rule**: $$\mathrm{diag}(\mathbf{A} _ t)\preceq (\mathbf{I}-\frac{\gamma}{2}\mathbf{H}^{(q)})\mathrm{diag}(\mathbf{A} _ {t-1})+2\gamma^2C\mathrm{diag}(\mathcal{M}^{(q)}\circ \mathbf{A} _ {t-1})+C(\sigma^{(q)},\Vert{\mathbf{w}^{(q)}}^* \Vert),\tag{* }$$ where $C=1$ for additive quantization, $C=1+\tilde{\epsilon}$ for multiplicative quantization and $C(\sigma^{(q)},\Vert{\mathbf{w}^{(q)}}^* \Vert)$ is a constant dependent of quantization-amplifed noise $\sigma^{(q)}$ and $\Vert{\mathbf{w}^{(q)}}^* \Vert$. The subsequent analysis can direcly apply our current analytic framework. Since $(*)$ has the same structure as the propagation rule in linear regression (Lemma D.2 and Lemma D.3), the impact of quantization in ReLU regression is consistent with that in linear regression.

---

> ### Author Response · Authors · 2025-11-23
> **Response to Reviewer MJwg's comment (2/6)**
>
> - multiple neurons (laze training / NTK regime)
>
> 	Our results can also be extended to the finite-width neural network, let $f(\mathbf{x};\mathbf{W})$ be the neural network function with model weights $\mathbf{W}$, starting from random initialization $\mathbf{W}^{(0)}$, the neural network function during the training can be approximated by $f(\mathbf{x};\mathbf{W})\approx f(\mathbf{x};\mathbf{W}^{(0)})+\langle \nabla f(\mathbf{x};\mathbf{W}^{(0)}),\mathbf{W}-\mathbf{W}^{(0)}\rangle$. Then, when the neural network is sufficiently wide, the training dynamics can be approximated by a random feature model, i.e., a linear function on the neural tangent feature $\phi(\mathbf{x})=\nabla f(\mathbf{x};\mathbf{W}^{(0)})$. Therefore, in NTK regime, the error propagation becomes: $$ \mathbf{w} _ t=\mathbf{w} _ {t-1}+\gamma \mathcal{Q} _ o\Big(\mathcal{Q} _ l({y} _ t)-\mathcal{Q} _ a\big(\mathcal{Q} _ g\left(\phi(\mathcal{Q} _ d(\mathbf{x} _ t))\right)^\top\mathcal{Q} _ p(\mathbf{w} _ {t-1})\big)\Big)\mathcal{Q} _ g\left(\phi(\mathcal{Q} _ d(\mathbf{x} _ t))\right).$$ **This update rule in NTK regime resembles that in linear regression, with a key shift from $\mathcal{Q}_d(\mathbf{x}_t)$ to $\mathcal{Q}_g\left(\phi(\mathcal{Q}_d(\mathbf{x}_t))\right)$**. On the technical level, the subsequent analysis can be completed by applying our current theoretical framework, by replacing $\mathbf{H}^{(q)}:=\mathbb{E}[\mathcal{Q}_d(\mathbf{x})\mathcal{Q}_d(\mathbf{x})^\top]$ with $\widetilde{\mathbf{H}}^{(q)}:=\mathbb{E}[\mathcal{Q}_g\left(\phi(\mathcal{Q}_d(\mathbf{x}_t))\right)\mathcal{Q}_g\left(\phi(\mathcal{Q}_d(\mathbf{x}_t))\right)^\top]$. Hence, effects of quantization in NTK regime are consistent with that in linear regression: data quantization (and gradient quantization $\mathcal{Q}_g$) distorts effective data spectrum, parameters, activations and output gradients quantization amplify noise.
>
> **Q1.2:** Can the author comment on what they think would be the effect of the quantization error in other algorithm designs, such as multipass SGD and momentum? Do they think that the implication of the result regarding multiplicative quantization will also carry in these settings?
>
> **A1.2:** Thanks for your comment. In other algorithm designs, **the implication that multiplicative data quantization error does not scale with data dimension $d$ remains to be true**, as the approximation error originates from the algorithm-independent excess risk decomposition (Lemma A.2). We believe the key quantization effect that multiplicative data quantization alleviates the data spectrum while additive data quantization does not, and other **effects of quantization still hold true** under other algorithm designs. We discuss multi-pass SGD and SGD with momentum respectively.
>
> - multi-pass SGD
>
> 	We believe under multi-pass SGD, our analytic framework can be used to analyze the algorithm-dependent component $R=\frac{1}{2}\langle \mathbf{H}^{(q)},\mathbb{E}[\overline{\pmb{\eta}} _ N\otimes\overline{\pmb{\eta}} _ N]\rangle$ based on existing techniques for full-precision multi-pass SGD [1]. Concretely, $R$ can be split into three components (similar to Eq. (5.3) in [1], for simplicity we consider batchsize $B=1$ and last iterate risk):$$ R=\underbrace{\frac{1}{2}\Vert\widehat{\mathbf{w}} _ {t}-{\mathbf{w}^{(q)}}^{* }\Vert _ {\mathbf{H}^{(q)}}^{2}} _ {\mathrm{quantized \ GD\ error}}+\underbrace{\frac{\eta^{2}}{2}\sum _ {k=0}^{t-1}\left\langle\mathcal{G}^{t-1-k}\circ(C _ i\mathcal{M}-\widetilde{\mathcal{M}})\circ\mathbf{E} _ {k},\mathbf{H}^{(q)}\right\rangle} _ {\text{quantized Fluctuation error}}+\mathrm{ExtraError}, \tag{** }$$where:
>
> 	- the quantizied multi-pass SGD iterate $\mathbf{w}_{t}$ is defined by $$\mathbf{w} _ {t}=\mathbf{w} _ {t-1}+\eta\mathcal{Q}(\mathbf{x} _ {i _ t})\mathcal{Q} _ o\left(\mathcal{Q} _ l(y _ {i _ t})-\mathcal{Q} _ a\left(\mathcal{Q} _ d(\mathbf{x} _ {i _ t})^\top \mathcal{Q} _ p(\mathbf{w} _ {t-1})\right)\right),$$ with $\mathbf{E} _ {t}=\mathbb{E}\left[(\mathbf{w} _ {t}-\widehat{\mathbf{w}})(\mathbf{w} _ {t}-\widehat{\mathbf{w}})^\top\right]$ denoting the second moment of the deviation from the quantized minimum norm interpolator $\widehat{\mathbf{w}}=\mathcal{Q} _ d(\mathbf{X})^\top [\mathcal{Q} _ d(\mathbf{X})\mathcal{Q} _ d(\mathbf{X})^\top]^{-1}\mathcal{Q} _ l(\mathbf{y}).$
>
> 	- the quantized GD iterate $\widehat{\mathbf{w}} _ {t}$ is defined by $$\widehat{\mathbf{w}} _ {t}=\widehat{\mathbf{w}} _ {t-1}+\eta\frac{1}{n}\sum _ {i=1}^n \mathcal{Q}(\mathbf{x} _ {i})\mathcal{Q} _ o\left(\mathcal{Q} _ l(y _ {i})-\mathcal{Q} _ a\left(\mathcal{Q} _ d(\mathbf{x} _ {i})^\top \mathcal{Q} _ p(\widehat{\mathbf{w}} _ {t-1})\right)\right).$$
>
> 	- $\mathrm{ExtraError}$ arises from a critical discrepancy: $$\mathbb{E} _ {\rm SGD}\left[\mathbf{x} _ {i _ t}^{(q)}{\mathbf{x} _ {i _ t}^{(q)}}^\top\right]\neq \frac{1}{n}\sum _ {i=1}^n \mathbf{x} _ {i}^{(q)}{\mathbf{x} _ {i}^{(q)}}^\top,$$ as quantization operator is non-linear.

---

> ### Author Response · Authors · 2025-11-23
> **Response to Reviewer MJwg's comment (3/6)**
>
> Following the decomposition $\mathbf{(**)}$ we established, the analysis hinges on bounding the quantized GD error, the Fluctuation error and $\mathrm{ExtraError}$. **Given the strong structural similarity between quantized GD and quantized one-pass SGD, we can directly apply our technical framework to bound the quantized GD error**. Regarding the quantized Fluctuation error, we aim to establish an upper bound for the following expression: $${\mathbf{x} _ i^{(q)}}^\top(\mathbf{I}-\eta\mathbf{\Sigma}^{(q)})^k\mathbf{H}^{(q)}(\mathbf{I}-\eta\mathbf{\Sigma}^{(q)})^k\mathbf{x} _ i^{(q)}=\underbrace{{\mathbf{x} _ i^{(q)}}^\top(\mathbf{I}-\eta\mathbf{\Sigma}^{(q)})^k\mathbf{\Sigma}^{(q)}(\mathbf{I}-\eta\mathbf{\Sigma}^{(q)})^k\mathbf{x} _ i^{(q)}} _ {\Theta_1}+\underbrace{{\mathbf{x} _ i^{(q)}}^\top(\mathbf{I}-\eta\mathbf{\Sigma}^{(q)})^k(\mathbf{H}^{(q)}-\mathbf{\Sigma}^{(q)})(\mathbf{I}-\eta\mathbf{\Sigma}^{(q)})^k\mathbf{x} _ i^{(q)}} _ {\Theta_2}.$$ A technical barrier to analyze $\Theta _ 1$ and $\Theta _ 2$ is that existing concentration techniques in random matrix analysis can not be directly applied to derive concentration bounds for quantized data i.e., $\mathbf{\Sigma}^{(q)}$. We defer the comprehensive study of these quantized random matrix analysis and the discrepancy in $\mathrm{ExtraError}$ to future work. Overall, as the simplicity between quantized GD and quantized one-pass SGD, we believe our implication of the results still carry in multi-pass SGD.
>
> - SGD with momentum
>
> 	We can extend our theory to quantized SGD with momentum by combining our technical framework for quantized SGD with existing techniques for full-precision SGD with momentum [2]. In particular, the accelerated SGD (ASGD) scheme in [2] maintains three sequences $\mathbf{u} _ t, \mathbf{v} _ t, \mathbf{w} _ t$ which are updated as $$\mathbf{u} _ {t-1}=\alpha\mathbf{w} _ {t-1}+(1-\alpha)\mathbf{v} _ {t-1},\quad \mathbf{w} _ {t}=\mathbf{u} _ {t-1}-\delta\widehat{\nabla}L(\mathbf{u} _ {t-1}),\quad \mathbf{v} _ t=\beta\mathbf{u} _ {t-1}+(1-\beta)\mathbf{v} _ {t-1}-\gamma\widehat{\nabla}L(\mathbf{u} _ {t-1}).$$ Define the centered ASGD iterate $\pmb{\eta}_t:=\begin{bmatrix}\mathbf{w} _ t-\mathbf{w}^* \newline\mathbf{u} _ t-\mathbf{w}^* \end{bmatrix}.$ Then it holds $$\pmb{\eta} _ t=\widehat{\mathbf{A}} _ t\pmb{\eta} _ {t-1}+\pmb{\zeta} _ t,\quad\mathrm{where}\quad\widehat{\mathbf{A}} _ t:=\begin{bmatrix}\mathbf{0} & \mathbf{I}-\delta\mathbf{x} _ t\mathbf{x} _ t^\top \newline -c\mathbf{I} & (1+c)\mathbf{I}-q\mathbf{x} _ t\mathbf{x} _ t^\top\end{bmatrix},\quad\pmb{\zeta} _ t:=\begin{bmatrix}\delta\cdot\epsilon _ t\mathbf{x} _ t \newline q\cdot\epsilon _ t\mathbf{x} _ t\end{bmatrix}.$$ This propagation is similar to the update rule in one-pass SGD. Under quantization, by our analytic technique, $\mathbf{x} _ t$ will be replaced with its quantized counterpart $\mathbf{x} _ t^q$ and noise term $\epsilon _ t$ will be amplified by quantization errors. Together with techniques in [2], our theoretical framework can be extended to the ASGD case, i.e., SGD with momentum. Typically, as the difference between quantized SGD and quantized SGD with momentum is independent of the quantization effects, we believe our implication of the results still carry in SGD with momentum.
>
> **Q2:** Proving a matching lower bound or providing a tighter analysis.
>
> **A2:** Thanks for your comment. We are able to provide a lower bound for additive and multiplicative quantization. Typically, **these lower bounds are larger than standard matching excess risk bound in full-precision SGD (with an additional approximation error)**. Specifically, under additive quantization, quantization effects of parameters, activations and output gradients still hold in the lower bound. We provide a proof sketch and summarize our lower bounds as follows.

---

> ### Author Response · Authors · 2025-11-23
> **Response to Reviewer MJwg's comment (4/6)**
>
> Generally, we start by the excess risk decomposition in Lemma A.2: $$
>         \mathbb{E}[\mathcal{E}(\overline{\mathbf{w}} _ N)]=\frac{1}{2}\langle\mathbf{H}^{(q)},\mathbb{E}[\overline{\pmb{\eta}} _ N\otimes\overline{\pmb{\eta}} _ N]\rangle+\frac{1}{2}\Vert\mathbf{w}^* \Vert _ {\mathbf{D}(\mathbf{H}+\mathbf{D})^{-1}\mathbf{H}(\mathbf{H}+\mathbf{D})^{-1}\mathbf{D}}^2+\frac{1}{2}({\mathbf{w}^{(q)}}^* )^\top \mathbf{D}{\mathbf{w}^{(q)}}^* -\frac{1}{2}\mathbb{E}\left[\overline{\mathbf{w}} _ N^\top \mathbf{D} \overline{\mathbf{w}} _ N\right].$$ By this decomposition, **the major technical challenge is to tackle the negative term** $-\frac{1}{2}\mathbb{E}\left[\overline{\mathbf{w}} _ N^\top \mathbf{D} \overline{\mathbf{w}} _ N\right].$ Intuitively, this term is merged to the main algorithm-dependent term $\frac{1}{2}\langle\mathbf{H}^{(q)},\mathbb{E}[\overline{\pmb{\eta}} _ N\otimes\overline{\pmb{\eta}} _ N]\rangle$ and one of the algorithm-independent terms $\frac{1}{2}({\mathbf{w}^{(q)}}^* )^\top \mathbf{D} {\mathbf{w}^{(q)}}^* $. Specifically, $$\mathbb{E}\left[\overline{\mathbf{w}} _ N^\top \mathbf{D} \overline{\mathbf{w}} _ N\right]= ({\mathbf{w}^{(q)}}^* )^\top \mathbf{D} {\mathbf{w}^{(q)}}^* + 2({\mathbf{w}^{(q)}}^* )^\top \mathbf{D} \mathbb{E}[\overline{\pmb{\eta}} _ N] + \mathbb{E}[\overline{\pmb{\eta}} _ N^\top \mathbf{D} \overline{\pmb{\eta}} _ N].$$ Therefore, we have $$\mathbb{E}[\mathcal{E}(\overline{\mathbf{w}} _ N)] = \frac{1}{2}\Vert \mathbf{w}^* \Vert _ {\mathbf{D}(\mathbf{H}+\mathbf{D})^{-1}\mathbf{H}(\mathbf{H}+\mathbf{D})^{-1}\mathbf{D}}^2 + \frac{1}{2}\mathbb{E}[\overline{\pmb{\eta}} _ N^\top \mathbf{H} \overline{\pmb{\eta}} _ N] - ({\mathbf{w}^{(q)}}^* )^\top \mathbf{D} \mathbb{E}[\overline{\pmb{\eta}} _ N].$$ Note that by the unbiased quantization assumption, $\mathbb{E}\left[\pmb{\eta} _ t|\pmb{\eta} _ {t-1}\right]=\left(\mathbf{I}-\gamma \mathbf{H}^{(q)}\right)\pmb{\eta} _ {t-1},$ we have $$\mathbb{E}[\overline{\pmb{\eta}} _ N] = \frac{1}{N}\sum _ {t=0}^{N-1} (\mathbf{I}-\gamma \mathbf{H}^{(q)})^t \pmb{\eta} _ 0=- \frac{1}{N\gamma} {\mathbf{H}^{(q)}}^{-1} \left( \mathbf{I} - (\mathbf{I}-\gamma \mathbf{H}^{(q)})^N \right) {\mathbf{w}^{(q)}}^* .$$ Thus $$\mathbb{E}[\mathcal{E}(\overline{\mathbf{w}} _ N)] = \frac{1}{2}\Vert\mathbf{w}^* \Vert _ {\mathbf{D}(\mathbf{H}+\mathbf{D})^{-1}\mathbf{H}(\mathbf{H}+\mathbf{D})^{-1}\mathbf{D}}^2 + \frac{1}{2}\mathbb{E}[\overline{\pmb{\eta}} _ N^\top \mathbf{H} \overline{\pmb{\eta}} _ N]+({\mathbf{w}^{(q)}}^* )^\top \mathbf{D}\frac{1}{N\gamma} {\mathbf{H}^{(q)}}^{-1} \left( \mathbf{I} - (\mathbf{I}-\gamma \mathbf{H}^{(q)})^N \right) {\mathbf{w}^{(q)}}^* .\tag{\\#}$$
>
> Then **the second technical barrier is to tackle the algorithm-dependent term** $\frac{1}{2}\mathbb{E}[\overline{\pmb{\eta}} _ N^\top \mathbf{H} \overline{\pmb{\eta}} _ N].$ In particular, there are two challenges: (i) a lower bound for the update rule of $\mathbb{E}[\pmb{\eta} _ t \otimes \pmb{\eta} _ t],$ (ii) the gap between $\mathbf{H}$ and $\mathbf{H}^{(q)}$ in $\frac{1}{2}\mathbb{E}[\overline{\pmb{\eta}} _ N^\top \mathbf{H} \overline{\pmb{\eta}} _ N].$
>
> For (i), we consider the well-specificed model as [4]. Under additive quantization, **the update rule owns a lower bound matching its upper bound**, as the second moment of quantization error is constant. In contrast, **multiplicative quantization may merely admit a lower bound discarding parameters, activations and output gradients quantization error**: $$\mathbb{E}[\pmb{\eta} _ t\otimes\pmb{\eta} _ t]=\mathbb{E}\left[\left(\mathbf{I}-\gamma\frac{1}{B} {\mathcal{Q} _ d(\mathbf{X})}^\top \mathcal{Q} _ d(\mathbf{X}) \right)\mathbb{E}[\pmb{\eta} _ {t-1}\otimes\pmb{\eta} _ {t-1}]\left(\mathbf{I}-\gamma\frac{1}{B} {\mathcal{Q} _ d(\mathbf{X})}^\top \mathcal{Q} _ d(\mathbf{X}) \right)\right]+\gamma^2 \pmb{\sigma}^2\mathbf{H}^{(q)}.$$ This is because the extra $\mathbf{w} _ {t-1}$ induced by parameters, activations and output gradients might be not able to convert to linear terms $\pmb{\eta} _ {t-1}$ as the inequality $(\mathbf{u}+\mathbf{v})(\mathbf{u}+\mathbf{v})^\top \preceq 2\mathbf{u}\mathbf{u}^\top+2\mathbf{v}\mathbf{v}^\top$ does not hold in case of lower bound.
>
> For (ii), by simply **utilizing the monotonicity that $\mathbf{H}\preceq \mathbf{H}^{(q)}$ under both multiplicative and additive quantization**, $\frac{1}{2}\mathbb{E}[\overline{\pmb{\eta}} _ N^\top \mathbf{H} \overline{\pmb{\eta}} _ N]$ can be lower bounded by directly replacing the spectrum of $\mathbf{H}^{(q)}$ with the spectrum of $\mathbf{H}$.

---

> ### Author Response · Authors · 2025-11-23
> **Response to Reviewer MJwg's comment (5/6)**
>
> To conclude, together with excess risk decomposition (#), lower bounds of $\mathbb{E}[\pmb{\eta}_t \otimes \pmb{\eta}_t],$ and the monotonicity that $\mathbf{H}\preceq \mathbf{H}^{(q)}$, we are able to derive lower bounds for quantized SGD:
>
> - multiplicative quantization
>
> 	$$\mathbb{E}[\mathcal{E}(\overline{\mathbf{w}} _ N)] \gtrsim \mathrm{ApproxErr}+\mathrm{VarErr}+\mathrm{BiasErr},$$ where $$\begin{aligned}
>             &\mathrm{ApproxErr}\gtrsim \left\Vert\mathbf{w}^* \right\Vert _ \mathbf{H}^2 \left(\frac{\epsilon _ d^2}{(1+\epsilon _ d)^2}+\frac{\epsilon _ d}{(1+\epsilon_d)^3} \frac{1 - (1-\gamma (1+\epsilon _ d)\lambda _ {\min})^N}{N\gamma}\right),\newline &\mathrm{BiasErr}\gtrsim\frac{1}{\gamma^2N^2}\Vert{\mathbf{w}}^* \Vert _ {\mathbf{H} _ {0:k^* }^{-1}}^2+\Vert{\mathbf{w}}^*  \Vert _ {\mathbf{H} _ {k^* :\infty}}^2,\newline
>             &\mathrm{VarErr}\gtrsim\left(\frac{k^* }{N}+N\gamma^2\sum _ {i>k^* }\lambda _ i^2\right) \left(\frac{\frac{\sigma^2}{B}+\beta \left(\frac{\Vert{\mathbf{w}}^* \Vert _ {\mathbf{I} _ {0:k^* }}^2}{N\gamma}+\Vert {\mathbf{w}}^* \Vert _ {\mathbf{H} _ {k^* :\infty}}^2\right)}{1-\gamma\beta C_\epsilon\mathrm{tr}\left(\mathbf{H}\right)}\right).
>         \end{aligned}$$
>
> - additive quantization
>
> 	$$\mathbb{E}[\mathcal{E}(\overline{\mathbf{w}} _ N)] \gtrsim \mathrm{ApproxErr}+\mathrm{VarErr}+\mathrm{BiasErr},$$ where $$\begin{aligned}&\mathrm{ApproxErr}\gtrsim  \epsilon _ d^2 \sum _ {i=1}^d \frac{\lambda _ i}{(\lambda _ i+\epsilon _ d)^2}+\frac{1}{N\gamma} \epsilon _ d \sum _ {i=1}^d \frac{\lambda _ i^2}{(\lambda _ i+\epsilon _ d)^3}\left[ 1-(1-\gamma(\lambda _ i+\epsilon _ d))^N \right],\newline
> 		&\mathrm{BiasErr}\gtrsim\frac{1}{\gamma^2N^2}\Vert{\mathbf{w}}^* \Vert _ {\mathbf{H} _ {0:k^* }^{-1}}^2+\Vert {\mathbf{w}}^* \Vert _ {\mathbf{H} _ {k^* :\infty}}^2,\newline
>         &\mathrm{VarErr}\gtrsim\frac{{\sigma _ A^{(q)}}^2+\frac{2\beta}{N\gamma}\left(\Vert{\mathbf{w}}^* \Vert _ {\mathbf{I} _ {0:k^* }}^2+N\gamma\Vert {\mathbf{w}}^* \Vert _ {\mathbf{H} _ {k^* :\infty}}^2\right)}{1-\gamma\beta\mathrm{tr}\left(\mathbf{H}+\epsilon _ d\mathbf{I}\right)}\left(\frac{k^* }{N}+N\gamma^2\sum _ {i>k^* }\lambda _ i^2\right),
>     \end{aligned}$$ with ${\sigma _ A^{(q)}}^2=\frac{\epsilon _ o+\epsilon _ a}{B}+\beta \epsilon _ p\mathrm{tr}\left(\mathbf{H}+\epsilon _ d\mathbf{I}\right)+\frac{\sigma^2}{B}.$
>
> **Q3:** Other questions.
>
> - **Q3.1:** What does the notation $\lesssim$ mean?
>
> 	**A3.1:** Thanks for your comment. For two positive-valued functions $f(x)$ and $g(x)$, we write $f(x) \lesssim g(x)$ (and $f(x) = O(g(x))$) or $f(x) \gtrsim g(x)$ (and $f(x) = \Omega(g(x))$) if $f(x) \leq cg(x)$ or $f(x) \geq cg(x)$ holds for some absolute (if not otherwise specified) constant $c > 0$ respectively. We write $f(x) \eqsim g(x)$ (and $f(x) = \Theta(g(x))$) if $f(x) \lesssim g(x) \lesssim f(x)$. We will add this in the revised manuscript.
>
> - **Q3.2:** Lines 355-361: Please indicate where the fourth term appears.
>
>     **A3.2:** Thanks for your comment. We clarify the fourth moment term here and incorporate this into the revised manuscript. The impact of activation and gradient quantization error $\pmb{\epsilon}$ on the excess risk, captured by the term $\mathbb{E}[(\mathbf{w} _ t-{\mathbf{w}^{(q)}}^* ) \otimes (\mathbf{w} _ t-{\mathbf{w}^{(q)}}^* )]$, can be characterized as $\frac{1}{B^2}\mathbb{E}[{\mathbf{X}^{(q)}}^\top \pmb{\epsilon}\pmb{\epsilon}^\top \mathbf{X}^{(q)}]$. **For multiplicative quantization**, activation and gradient quantization error **retains the structure of full-precision activation and gradient**, yielding a dependence on the fourth moment of the data (i.e., $\frac{1}{B^2}\mathbb{E}[{\mathbf{X}^{(q)}}^\top \mathbf{X}^{(q)}{\mathbf{X}^{(q)}}^\top \mathbf{X}^{(q)}]$). In contrast, **for additive quantization where activation and gradient quantization error can be directly bounded by constant**, the dependence reduced to the second moment of the data (i.e., $\frac{1}{B^2}\mathbb{E}[{\mathbf{X}^{(q)}}^\top \mathbf{X}^{(q)}]$). Consequently, this introduces an extra factor of $1/B$ in the output gradient and activation quantization error (see Lemma D.2 and Lemma D.3 for details).

---

> ### Author Response · Authors · 2025-11-23
> **Response to Reviewer MJwg's comment (6/6)**
>
> - **Q3.3:** Line 336: Please provide more intuition on why under multiplicative quantization, the additional complexity induced from the iteration update rule can be merged.
>
>     **A3.3:** Thanks for your comment. Intuitively, under multiplicative quantization, the inherent structural similarity ensures that **the quantized terms align with their corresponding full-precision versions**. This yields an additional factor of $(1+\tilde{\epsilon})$ in the linear term of the deviation error propagation (see Lemma D.3). Crucially, $\tilde{\epsilon}$ is defined as a linear function of the individual quantization factors $\epsilon_p, \epsilon_a, \epsilon_o$, ensuring that $\tilde{\epsilon} \lesssim 1$ when these factors themselves are small (e,g., $\epsilon_p, \epsilon_a, \epsilon_o \lesssim 1$). Hence, while this scaling factor does affect the final excess risk bound as a multiplicative term (see Lemma D.11), its overall impact becomes negligible when analyzing the asymptotic bounds. We will incorporate this clarification in the revised manuscript.
>
>
> - **Q3.4:** Lines 393-397: I am not sure I understand this. The errors are not completely decoupled, as $\epsilon_a,\epsilon_o$ depends on $\epsilon_d,\epsilon_p.$
>
>     **A3.4:** Thanks for your comment. Here, the quantization error parameters $\epsilon_a$ (activation error) and $\epsilon_o$ (output gradient error) are independent of $\epsilon_d$ (data error) and $\epsilon_p$ (parameter error). This independence stems from the fact that each $\epsilon_i$ parameter (indexed by its subscript) is determined solely by the chosen quantization operator $\mathcal{Q}_i$ (see the multiplicative and additive Definition 3.1 for details).
>
> - **Q3.5:** Line 463: It would be nice to provide a clearer explanation and maybe plot the Hessian spectrum as well in both cases, to illustrate the effect.
>
> 	**A3.5** Thanks for your comment. Intuitively, multiplicative data quantization maintains the structure of spectrum as $\mathbf{H}^{(q)}=(1+\epsilon_d)\mathbf{H}.$ In contrast, additive quantization significantly distorts the spectrum especially for small eigenvalues which are dominated by $\epsilon_d$, as $\mathbf{H}^{(q)}=\mathbf{H}+\epsilon_d \mathbf{I}.$
>
> 	Empirically, we apply additive and multiplicative quantization of identical magnitude and then record the resulting empirical covariance spectra to study how each quantization type perturbs the underlying eigenvalue structure. Results are summarized in Figure 2(b) in Section G in the revised manuscript. Crucially, **multiplicative quantization largely preserves the full-precision eigenvalue structure**, whereas **additive quantization introduces a marked, irreducible spectral gap** relative to the full-precision spectrum. This empirical finding aligns with previous finding and validates our theory: **additive quantization errors distort the data Hessian spectrum, increasing risk, whereas multiplicative quantization errors diminish the spectral gap, maintaining risk despite higher error levels**.
>
> - **Q3.6:** Line 293 remove either “plays” or “relies” / Line 394 $\epsilon_0$ should be $\epsilon _o$.
>
>     **A3.6:** Thanks for your comments. We will modify in the revised manuscript.
>
> [1] Risk Bounds of Multi-Pass SGD for Least Squares in the Interpolation Regime.
>
> [2] Risk Bounds of Accelerated SGD for Overparameterized Linear Regression.
>
> [3] Finite-sample analysis of learning high-dimensional single ReLU neuron.
>
> [4] Benign Over tting of Constant-Stepsize SGD for Linear Regression.

---

> > ### Comment · Reviewer_MJwg · 2025-11-26
> >
> > Thank you for the rebuttal that addresses my questions. I keep my score.

---

> > > ### Author Response · Authors · 2025-11-26
> > >
> > > Dear Reviewer MJwg,
> > >
> > > We extend our thanks for your time, and we sincerely appreciate your positive feedback on our work. Thank you again for your effort in reviewing our work.
> > >
> > > Best,
> > >
> > > Authors

---

### Official Review · Reviewer_D5UN · 2025-10-30

**Soundness:** 1
**Presentation:** 2
**Contribution:** 1
**Rating:** 2
**Confidence:** 3

**Summary:**

The paper provides a theoretical analysis of Linear Regression with diverse quantization targets: data, labels, parameters, activations, and gradients. The paper analyzes two types of unbiased compression methods: multiplicative and additive quantization, and derives convergence guarantees for SGD under these schemes. The paper also includes a small empirical evaluation on synthetic data.

**Strengths:**

* The paper brings a theoretical analysis of different quantization targets.
* The definition, assumptions, and notations are easy to follow.

**Weaknesses:**

* The statement of Theorem 4.1 is difficult to analyze and to derive guidance on what to quantize to maximize performance within a limited resource budget; the paper would benefit from organizing and grouping the results and from plugging in an optimal learning rate.
* The bounds in Theorem 4.1 do not improve with the number of samples $N$, which is atypical for SGD analyses (even in more general convex or non-convex settings). This issue remains even without compression (i.e., when $\varepsilon=0)$.
* The paper uses a simple setup of unbiased compressions (where biased compressions are more practical and common) over a simple linear regression objective for empirical risk minimization (i.e., not guaranteeing generalization).
* Empirical results are limited to a simple synthetic dataset.

**Questions:**

* It is not clear why quantizing data would outperform quantizing parameters or gradients. Could the authors provide a concrete example or theoretical justification where data quantization provably yields better results than parameter/gradient quantization?
* Regarding the quantization of the model weights in the (Quantized SGD) update step. Wouldn’t it make more sense to quantize both $w_{t-1}$? If only one of them is quantized, it seems we still need to maintain a full-precision copy of the parameters, which weakens the memory motivation.

---

> ### Author Response · Authors · 2025-11-23
> **Response to Reviewer D5UN's comment (1/5)**
>
> Thank you for your positive feedback and we appreciate your strong support for the completeness of our work. We address your concerns in the following response:
>
> **Q1:** The paper would benefit from organizing and grouping the results and from plugging in an optimal learning rate.
>
> **A1:** Thanks for your comment. We have simplified our Theorem 4.1 in the revised manuscript. Regarding plugging an optimal learning rate, we first consider excess risk bound under polynomial decay when effective dimension $k^*\leq d$ (Corollary 4.3).
>
> - multiplicative quantization
>
> 	$$\mathbb{E}[\mathcal{E}(\overline{\mathbf{w}} _ N)]\lesssim\epsilon _ d+\frac{d _ {eff}^{(M)} }{N\gamma}+\frac{d _ {eff}^{(M)}}{N} \left(\frac{\sigma^2}{B}+\epsilon _ p+\epsilon _ o+\epsilon _ a+\frac{d _ {eff}^{(M)}}{N\gamma}\right),$$ where $d_{eff}^{(M)}=\left[N\gamma (1+\epsilon_d)\right]^{\frac{1}{a}}$. If $\frac{d_{eff}^{(M)}}{N\gamma} \lesssim \frac{\sigma^2}{B}+\epsilon_p+\epsilon_o+\epsilon_a$, then the optimal learning rate is $\gamma^* = \frac{a - 1}{\frac{\sigma^2}{B} + \epsilon_p + \epsilon_o + \epsilon_a}$. Thus, the excess risk under this condition is $$\mathbb{E}[\mathcal{E}(\overline{\mathbf{w}}_N)]\lesssim\epsilon_d+ a \cdot (1+\epsilon_d)^{\frac{1}{a}} \cdot \left( \frac{\frac{\sigma^2}{B} + \epsilon_p + \epsilon_o + \epsilon_a}{N(a-1)} \right)^{1 - \frac{1}{a}}.$$
>
> - additive quantization
>
> 	$$\mathbb{E}[\mathcal{E}(\overline{\mathbf{w}} _ N)]
>                     \lesssim {\epsilon _ d}d+\frac{d _ {eff}^{(A)}}{N\gamma}
>                     +\frac{d _ {eff}^{(A)}}{N}\left(\frac{\sigma^2}{B}+(1+d\epsilon _ d) \epsilon _ p+\frac{\epsilon _ o+\epsilon _ a}{B}+\frac{d _ {eff}^{(A)}}{N\gamma}\right),$$ where $d_{eff}^{(A)}=\big(d-\big(\frac{1}{N\gamma} -\epsilon_d\big)^{-\frac{1}{a}}\big)\epsilon_dN\gamma+\left(\frac{1}{N\gamma} -\epsilon_d\right)^{-\frac{1}{a}}.$ If $\frac{d_{eff}^{(A)}}{N\gamma} \lesssim \frac{\sigma^2}{B}+(1+d\epsilon_d) \epsilon_p+\frac{\epsilon_o+\epsilon_a}{B}$ and $\frac{1}{N\gamma} -\epsilon_d \eqsim \frac{1}{N\gamma}$ then it holds $$\gamma^* = \frac{a-1}{\frac{\sigma^2}{B}+(1+d\epsilon_d) \epsilon_p+\frac{\epsilon_o+\epsilon_a}{B}} \cdot \underbrace{\frac{1}{1 + a \cdot d\epsilon_d (N\gamma^*)^{1-\frac{1}{a}}}}_{\text{Shrinkage Factor < 1}}.$$ Take $\gamma \eqsim \frac{a-1}{\frac{\sigma^2}{B}+(1+d\epsilon_d) \epsilon_p+\frac{\epsilon_o+\epsilon_a}{B}}$ we have $$\mathbb{E}[\mathcal{E}(\overline{\mathbf{w}} _ N)] \lesssim ad\epsilon _ d+a\left(\frac{\frac{\sigma^2}{B}+(1+d\epsilon _ d) \epsilon _ p+\frac{\epsilon _ o+\epsilon _ a}{B}}{N(a-1)}\right)^{1-\frac{1}{a}}.$$
>
> **The excess risk bounds under polynomial decay with (nearly) optimal learning rate simplify original bounds under general learning rate**. These simplified bounds provide the same implication as indicated from general learning rate bounds: under additive quantization, the impacts of both data quantization and parameter quantization scale with the data dimension $d$, while under multiplicative quantization, data quantization remains independent of $d$.
>
> Under general quantization (Theorem 4.1), we would like to remark that, **excess risk bounds under optimal learning rate $\gamma^*$ will be even more complicated than bounds under general learning rate**, as $\gamma^* $ is a very complex function of quantized data spectrum $\{\lambda _ i^{(q)}\} _ {i\geq 1}$ and quantization errors. Specifically, $\gamma^* $ satisfies$$\left[ \frac{C_1 C_3 + \frac{2 C _ 2 C _ 3}{\gamma^* } - \frac{C _ 2}{{\gamma^* }^2}}{(1 - C _ 3 \gamma^* )^2} \right] \left( C _ 4 + C _ 5 {\gamma^* }^2 \right) + \left[ \frac{C _ 1 + \frac{C _ 2}{\gamma^* }}{1 - C _ 3 \gamma^* } \right] (2 C _ 5 \gamma^* ) = \frac{2 C _ 6}{{\gamma^* }^3},$$ where $$C _ 1 = {\sigma _ G^{(q)}}^2 + 2\alpha _ B \Vert{\mathbf{w}^{(q)}}^* \Vert _ {\mathbf{H} _ {k^* :\infty}^{(q)}}^2, C _ 2 = \frac{2\alpha _ B}{N} \Vert{\mathbf{w}^{(q)}}^* \Vert _ {\mathbf{I} _ {0:k^* }^{(q)}}^2, C _ 3 = \alpha _ B\mathrm{tr}\left(\mathbf{H}+\mathbf{D}\right), $$$$C _ 4=\frac{k^* }{N}, C _ 5 = N \sum _ {i>k^* } {\lambda _ i^{(q)}}^2, C _ 6 = \frac{1}{N^2} \Vert{\mathbf{w}^{(q)}}^* \Vert _ {(\mathbf{H}_{0:k^* }^{(q)})^{-1}}^2.$$ Regarding to the complexity in computation, we do not plug in the optimal learning rate under general quantization.

---

> ### Author Response · Authors · 2025-11-23
> **Response to Reviewer D5UN's comment (2/5)**
>
> **Q2:** The bounds in Theorem 4.1 do not improve with the number of samples $N$.
>
> **A2:** Thanks for your comment. **Current bounds in Theorem 4.1 does indeed improve (decrease) as the number of samples $N$ increases**, except for the $N$-independent $\mathrm{ApproxErr}$ (exactly zero in absence of quantization) introduced by quantization error.
>
> - Approximation  error
>
> 	The approximation error, resulting from quantization of data, corresponds precisely to the discrepancy between the optimal solution in non-quantized data space and quantized data space, i.e., $$\frac{1}{2}\mathbb{E}\left[(\mathcal{Q} _ l(y)-\langle {\mathbf{w}^{(q)}}^* ,\mathcal{Q} _ d(\mathbf{x})\rangle)^2\right]-\frac{1}{2}\mathbb{E}\left[\left(y-\langle \mathbf{w}^* ,\mathbf{x} \rangle \right)^2\right].$$ The approximate error is **independent of the optimization algorithm** and thus is not related to the data size $N$.
>
> - Bias and Variance error
>
> 	$$\mathrm{VarErr}=\frac{{\sigma _ G^{(q)}}^2+\frac{2\alpha _ B}{N\gamma}\left(\Vert{\mathbf{w}^{(q)}}^* \Vert _ {\mathbf{I} _ {0:k^* }^{(q)}}^2+N\gamma\Vert{\mathbf{w}^{(q)}}^* \Vert _ {\mathbf{H} _ {k^* :\infty}^{(q)}}^2\right)}{1-\gamma\alpha _ B\mathrm{tr}\left(\mathbf{H}+\mathbf{D}\right)}\left(\frac{k^* }{N}+N\gamma^2\sum _ {i>k^* }{\lambda _ i^{(q)}}^2\right),$$$$\mathrm{BiasErr}=\frac{1}{\gamma^2N^2}\cdot\Vert{\mathbf{w}^{(q)}}^* \Vert _ {(\mathbf{H} _ {0:k^* }^{(q)})^{-1}}^2+\Vert{\mathbf{w}^{(q)}}^* \Vert _ {\mathbf{H} _ {k^* :\infty}^{(q)}}^2.$$ Building on a key observation of the effective dimension $k^* =\max\left\\{k: \lambda _ k^{(q)} \geq \frac{1}{N\gamma}\right\\}$, **both bias error and variance error can be upper bounded by some $N$-dependent terms**. These $N$-dependent terms decrease monotonically with $N$ and **converge to zero when $N$ approaches to infinity**. Specifically, it holds $$\frac{1}{\gamma^2N^2}\cdot\Vert{\mathbf{w}^{(q)}}^* \Vert _ {(\mathbf{H} _ {0:k^* }^{(q)})^{-1}}^2+\Vert{\mathbf{w}^{(q)}}^* \Vert _ {\mathbf{H} _ {k^* :\infty}^{(q)}}^2 \leq \frac{\Vert{\mathbf{w}^{(q)}}^* \Vert^2}{N\gamma},$$ $$\frac{k^* }{N}+N\gamma^2\sum _ {i>k^* }{\lambda _ i^{(q)}}^2\leq \frac{d}{N}.$$
>
> 	These two inequalities imply that both bias error and variance error decrease with $N$ and converge to zero when $N$ approaches to infinity.
>
> **Q3:** The paper uses a simple setup of **unbiased compressions** (where biased compressions are more practical and common).
>
> **A3:** Thanks for your comment. This is a good point. Theoretically, **the unbiased stochastic quantization assumption serves as a starting point** to isolate and analyze the effects of quantization on excess risk bounds, as the unbiased nature of the estimator helps decouple the impact of quantization error variance from other sources of uncertainty. This setup facilitates clearer theoretical analysis and makes the conclusions more interpretable, which is widely used in theoretical analysis under quantization [2,4,5].
>
> From empirical perspective, **stochastic rounding is commonly adopted in practical low-precision large-scale model training** due to its empirical and theoretical advantages over deterministic rounding. Specifically, it mitigates accumulated quantization errors and often leads to better generalization performance, as supported by recent literature [2,3].
>
> We would like to remark that **this analytic framework can be readily extended to general biased case**. The key technical barrier of the extension is the presence of extra terms introduced by quantization bias:
>
> - additional gap $I_1$ when running average iterates $\overline{\mathbf{w}}_N$ and quantized minima ${\mathbf{w}^{(q)}}^*$ between full-precision data space $(\mathbf{x},y)$ and quantized data space $(\mathcal{Q}_d(\mathbf{x}),\mathcal{Q}_l(y))$
>
> 	$$I_1=2\mathbb{E}\left[\overline{\pmb{\eta}}_N^\top\left(\mathcal{Q}_l(y)\mathcal{Q}_d(\mathbf{x})-y\mathbf{x}\right)\right].$$
>
> - additional first-order term $I_2$ arising in the analysis of the average iterate covariance $\mathbb{E}[\overline{\pmb{\eta}} _ N\otimes\overline{\pmb{\eta}} _ N]$
>         $$I_2=\frac{1}{2N^2}\cdot\sum _ {t=0}^{N-1}\sum _ {k=t}^{N-1}\sum _ {i=0}^{t-k-1}\left\langle \mathbf{H}^{(q)}, (\mathbf{I}-\gamma\mathbf{H}^{(q)})^i\mathbb{E}\left[Q _ {t-i}\otimes {\pmb{\eta}} _ t\right]+\mathbb{E}\left[\pmb{\eta} _ t\otimes Q _ {t-i}\right]  (\mathbf{I}-\gamma\mathbf{H}^{(q)})^i \right\rangle.$$ Here $Q _ t=\gamma\frac{1}{B}{\mathcal{Q} _ d(\mathbf{X}_t)}^\top \left(\pmb{\epsilon} _ t^{(o)}-\pmb{\epsilon} _ t^{(a)}-\mathcal{Q} _ d(\mathbf{X} _ t)\pmb{\epsilon} _ {t-1}^{(p)}\right)$.
>
> - additional quantization bias terms in the update rule of $\mathbb{E}[{\pmb{\eta}}_t\otimes{\pmb{\eta}}_t]$

---

> ### Author Response · Authors · 2025-11-23
> **Response to Reviewer D5UN's comment (3/5)**
>
> For the first two terms $I_1$ and $I_2$ that directly affect the excess risk, the following inequality: for any vector $\mathbf{u},\mathbf{v}$, $$\mathbf{u}\mathbf{v}^\top+\mathbf{u}\mathbf{v}^\top \preceq \mathbf{u}\mathbf{u}^\top+\mathbf{v}\mathbf{v}^\top$$ allows us to merge them into expressions involving $\mathbb{E}[{\pmb{\eta}}_t\otimes{\pmb{\eta}}_t]$ and quantization bias. Therefore, the only remaining technical challenge is to deal with the update rule of $\mathbb{E}[{\pmb{\eta}}_t\otimes{\pmb{\eta}}_t]$.
>
> Consider the biased quantization $\mathcal{Q}(\cdot)$: $$\mathcal{Q}(u) = u + \underbrace{B(u)} _ {\text{Deterministic Bias}} + \underbrace{\xi(u)} _ {\text{Zero-mean Noise}}.$$
>
> For **multiplicative quantization**, the quantization deterministic bias can be estimated by $\underline{b}\cdot u \leq B(u) \leq \overline{b}\cdot u, \quad \underline{b},\overline{b}\in \mathbb{R}.$ Therefore, the deviation error propagation in Lemma A.1 transforms to $$\pmb{\eta} _ t = \left(\mathbf{I}-\frac{b _ {p,a,o}}{B}\gamma {\mathcal{Q} _ d(\mathbf{X} _ t)}^\top \mathcal{Q} _ d(\mathbf{X} _ t) \right)\pmb{\eta} _ {t-1}+\gamma\frac{1}{B}{\mathcal{Q} _ d(\mathbf{X} _ t)}^\top \left(\pmb{\xi} _ t+\pmb{\xi} _ t^{(o,a,p)}\right),$$ where $b_{p,a,o}$ is a function of $\underline{b} _ p,\underline{b} _ a,\underline{b} _ o,\overline{b} _ p,\overline{b} _ a,\overline{b} _ o$ which capture the deterministic bias of parameter, activation and output gradient quantization respectively, and $\pmb{\xi} _ t^{(o,a,p)}$ is a linear function of $\pmb{\xi} _ t^{(a)},\pmb{\xi} _ t^{(o)},\mathcal{Q} _ d(\mathbf{X} _ t)\pmb{\xi} _ {t-1}^{(p)}$ with zero mean. With this formulation, we can directly apply the technical framework in unbiased quantization setting. It is worth noting that, a key difference from unbiased assumption is the difference in effective quantized data Hessian: $\widetilde{\mathbf{H}}^{(q)}=b_{p,a,o}\mathbf{H}^{(q)}$, which indicates that multiplicative quantization bias affects both bias error and variance error via influencing effective data Hessian. In particular, final excess risk bound can be formulated as: $$\mathbb{E}[\mathcal{E} _ N]\lesssim \mathrm{VarErr}+\mathrm{BiasErr}+\mathrm{ApproxErr},$$ where $$\mathrm{BiasErr}=\frac{1}{\gamma^2N^2}\cdot\Vert {\mathbf{w}^{(q)}}^* \Vert _ {(\widetilde{\mathbf{H}} _ {0:k^* }^{(q)})^{-1}}^2+\Vert {\mathbf{w}^{(q)}}^* \Vert _ {\widetilde{\mathbf{H}} _ { k^* :\infty}^{(q)} }^2,$$$$\mathrm{VarErr}=\frac{{\sigma^{(q)}}^2+\frac{2\alpha_B}{N\gamma}\left(\Vert {\mathbf{w}^{(q)}}^* \Vert _ {\widetilde{\mathbf{I}} _ { 0:k^* }^{(q)} }^2+N\gamma\Vert { \mathbf{w}^{(q)} }^* \Vert _ { \widetilde{\mathbf{H}} _ {k^* :\infty}^{(q)} }^2\right) }{1-\gamma\alpha _ B\mathrm{tr}\left(\widetilde{\mathbf{H}}^{(q)}\right) }\left(\frac{k^* }{N}+N\gamma^2\sum _ {i>k^* }{ \widetilde{\lambda} _ i^{(q)} }^2\right),$$$$\mathrm{ApproxErr}=\frac{1}{2}\langle\mathbf{H},\mathbb{E}[({\mathbf{w}}^* -{\mathbf{w}^{(q)}}^* )\otimes({\mathbf{w}}^* -{\mathbf{w}^{(q)}}^* )]\rangle+\frac{1}{2}\mathbb{E}\left[\langle {\mathbf{w}^{(q)}}^* ,\mathcal{Q} _ d(\mathbf{x})-\mathbf{x}\rangle^2\right].$$
>
> **Remark:** The additional approximation error $I_1$ is incorporated into bias error and variance error. Label quantization and data quantization affect ${\mathbf{w}^{(q)}}^*$, and thereby influence excess risk.
>
> For **additive quantization**, the quantization deterministic bias can be estimated by $\underline{\pmb{b}}\leq B(u) \leq \overline{\pmb{b}}.$ The deviation error propagation in Lemma A.1 then includes extra bias terms: $$\pmb{\eta} _ t = \left(\mathbf{I}-\frac{1}{B}\gamma {\mathcal{Q} _ d(\mathbf{X} _ t)}^\top \mathcal{Q} _ d(\mathbf{X} _ t) \right)\pmb{\eta} _ {t-1}+\gamma\frac{1}{B}{\mathcal{Q} _ d(\mathbf{X} _ t)}^\top \left(\pmb{\xi} _ t+\pmb{\xi} _ t^{(o,a,p)}+\pmb{b}^{(o,a,p)}\right).$$ The key implication is that although both multiplicative and additive quantization biases contribute to bias and variance errors, they do so qualitatively differently: **multiplicative quantization bias acts by modifying the effective data Hessian, while additive quantization bias affects the bias through an additional first-order term in the update rule of $\mathbb{E}[{\pmb{\eta}}_t\otimes{\pmb{\eta}}_t]$ (arises from non-zero mean $\pmb{b}^{(o,a,p)}$) and affects the variance by directly increasing the noise**. We defer the comprehensive study of additive biased quantization to future work.
>
> In conclusion, our analytic framework can extend to biased quantization where the quantization of labels, data, parameters, activations, and gradients will all affect bias and variance error.

---

> ### Author Response · Authors · 2025-11-23
> **Response to Reviewer D5UN's comment (4/5)**
>
> **Q4:** The paper uses a simple linear regression objective for empirical risk minimization (i.e., not guaranteeing generalization).
>
> **A4:** Thanks for your comment. We would like to emphasize that **the goal of manuscript is to analyze the generalization error, i.e., excess risk** under a comprehensive range of quantization targets. While we employ Empirical Risk Minimization (ERM) as the learning algorithm, our theoretical analysis explicitly focuses on the excess (generalization error) defined over the population distribution: $$\mathcal{E}(\overline{\mathbf{w}} _ N) = L(\overline{\mathbf{w}} _ N) - L(\mathbf w^* ),\tag{Excess Risk}$$ where $L(\mathbf{w})=\frac{1}{2}\mathbb{E}_{\mathbf{x},y}\left[\left(y-\langle \mathbf{w},\mathbf{x} \rangle \right)^2\right]$ denotes the population risk (expected loss over the data distribution), and $\mathbf{w}^*$ represents the optimal parameters.
>
> **Q5:** Regarding the quantization of the model weights in the (Quantized SGD) update step. Wouldn’t it make more sense to quantize both $\mathbf{w}_{t-1}$?
>
> **A5:** Thanks for your comments. We **follow the practical large-scale deep learning to maintain the master weights $\mathbf{w}_{t-1}$ in full precision**, which is crucial for the stability of the training process. As described in [1], mixed-precision training framework typically performs forward and backward passes in low-precision formats, while maintaining master weights in high precision (full precision). **If the master weights themselves are quantized, the small gradient updates would be lost (i.e., rounded to zero), preventing the model from converging.**
>
> On technical level, even if we quantize both $\mathbf{w} _ {t-1}$, **our analytic framework can also generalize to this case**. Specifically, the quantization of $\mathbf{w} _ {t-1}$ introduces an additional weight quantization error $\mathbf{\epsilon} _ {t-1}^{(p)}$ in the deviation error propagation. Thus, Lemma A.1 will be coverted into
> $$\pmb{\eta} _ t = \left(\mathbf{I}-\frac{1}{B}\gamma {\mathcal{Q} _ d(\mathbf{X} _ t)}^\top \mathcal{Q} _ d(\mathbf{X} _ t) \right)(\pmb{\eta} _ {t-1}+\pmb{\epsilon} _ {t-1}^{(p)})+\gamma\frac{1}{B}{\mathcal{Q} _ d(\mathbf{X} _ t)}^\top \left[\pmb{\xi} _ t+\pmb{\epsilon} _ t^{(o)}-\pmb{\epsilon} _ t^{(a)}\right],$$and the second order error propagation results (Lemma D.2 and Lemma D.3) will retain their current structure with only minor coefficient adjustments. Therefore, our techniques remain applicable when the master weights $\mathbf{w} _ {t-1}$ are quantized.
>
> **Q6:** It is not clear why quantizing data would outperform quantizing parameters or gradients. Could the authors provide a concrete example or theoretical justification where data quantization provably yields better results than parameter/gradient quantization?
>
> **A6:** Thanks for your comments. We did not claim that quantizing the data always outperforms quantizing parameters or gradients. Rather, our manuscript’s core contribution is the comprehensive analysis of **the effects of different quantization targets** (data, labels, parameters, activations and gradients) on learning performance. In particular, we demonstrate how **the quantization errors** introduced by each quantization target **are sensitive to critical parameters** such as data dimension and batch size. However, we do not make any claims that any quantization target outperforms others.

---

> ### Author Response · Authors · 2025-11-23
> **Response to Reviewer D5UN's comment (5/5)**
>
> **Q7:** Empirical results are limited to a simple synthetic dataset.
>
> **A7:** Thanks for your comment. We additionally evaluate on the publicly available $\texttt{Communities and Crime}$ dataset, which contains community-level statistics from across the United States.
>
> - Performance on $\texttt{Communities and Crime}$ dataset under multiplicative and additive quantization
>
> 	We apply both additive and multiplicative quantization schemes to the regression task on $\texttt{Communities and Crime}$ dataset. Results are shown in Figure 2(a) in Section G in the revised manuscript. Specifically, **under multiplicative quantization, excess risk is comparable to that of full-precision training**. In contrast, **additive quantization yields substantially higher excess risk than both full precision and multiplicative quantization**.
>
> - Effects on data spectrum under multiplicative and additive quantization
>
> 	To explain the different generalization behaviors, we examine the empirical covariance spectra produced by each quantization scheme. Results are shown in Figure 2(b) in Section G in the revised manuscript. Crucially, **multiplicative quantization largely preserves the full-precision eigenvalue structure**, whereas **additive quantization introduces a marked, irreducible spectral gap** relative to the full-precision spectrum.
>
> Overall, these empirical findings align with previous finding and validate our theory: **additive quantization errors distort the data Hessian spectrum, increasing risk, whereas multiplicative quantization errors diminish the spectral gap, maintaining risk despite higher error levels**.
>
> [1] Deepseek-v3 technical report.
>
> [2] Stochastic Rounding for LLM Training: Theory and Practice.
>
> [3] Training with Fewer Bits: Unlocking Edge LLMs Training with Stochastic Rounding.
>
> [4] Microadam: Accurate adaptive optimization with low space overhead and provable convergence.
>
> [5] Qsgd: Communication-efficient sgd via gradient quantization and encoding.

---

### Official Review · Reviewer_WYgy · 2025-11-03

**Soundness:** 3
**Presentation:** 3
**Contribution:** 3
**Rating:** 6
**Confidence:** 3

**Summary:**

The paper provides theoretical foundation for understanding how quantization impacts generalization in SGD. The authors bridge the gap between optimization convergence analyses and generalization theory under hardware-limited precision. The paper specifies precise conditions on quantization errors needed for quantized SGD to match the learning performance of full-precision SGD. The authors also show an analytical framework that quantifies how quantization on various components.

**Strengths:**

1. The authors provide a unified framework that decomposes the excess risk of quantized SGD into interpretable components.
2. The paper is mathematically solid and demonstrates excess risk bounds explicitly with clear decomposition and scaling behavior under both additive and multiplicative quantization.
3. The authors provide interesting insights into precision and generalization trade off, and also provide theoretical link between quantization type, like FP vs. INT, can be beneficial for scaling law and safe precision reduction.

**Weaknesses:**

1. The experimental section is only on synthetic Gaussian data, it could be more convincing if validated on real world dataset.
2. The analysis relies on idealized assumptions, such as unbiased stochastic quantization, but these may not hold in practical low-precision systems. Discussion or relaxation of these assumptions would strengthen the generality.
3. While motivated by scaling-law literature, the link between derived quantization effects and empirical scaling behaviors remains largely qualitative.

**Questions:**

1. How do non-uniform or mixed precision quantization methods fit into the framework here?
2. Can it be generalized to multiple epochs and adaptive optimizers, would the excess risk decomposition change qualitatively?
3. Since the case study is on polynomials decaying, I am curious how would the results change for exponentially decaying or heavy tail spectra which also happen in real world embeddings.
4. In Fig 1, how sensitive are the results to batch size or step size, can authors also show some results other than B=1?

---

> ### Author Response · Authors · 2025-11-23
> **Response to Reviewer WYgy's comment (1/7)**
>
> Thank you for your positive feedback and we appreciate your strong support for the completeness of our work. We address your concerns in the following response:
>
> **Q1:** Unbiased stochastic quantization assumption. Discussion or relaxation of these assumptions would strengthen the generality.
>
> **A1:** Thanks for your comment. This is a good point. Theoretically, **the unbiased stochastic quantization assumption serves as a starting point** to isolate and analyze the effects of quantization on excess risk bounds, as the unbiased nature of the estimator helps decouple the impact of quantization error variance from other sources of uncertainty. This setup facilitates clearer theoretical analysis and makes the conclusions more interpretable, which is widely used in theoretical analysis under quantization [1,8,9].
>
> From empirical perspective, **stochastic rounding is commonly adopted in practical low-precision large-scale model training** due to its empirical and theoretical advantages over deterministic rounding. Specifically, it mitigates accumulated quantization errors and often leads to better generalization performance, as supported by recent literature [1,2].
>
> We would like to remark that **this analytic framework can be readily extended to general biased case**. The key technical barrier of the extension is the presence of extra terms introduced by quantization bias:
>
> - additional gap $I_1$ when running average iterates $\overline{\mathbf{w}}_N$ and quantized minima ${\mathbf{w}^{(q)}}^*$ between full-precision data space $(\mathbf{x},y)$ and quantized data space $(\mathcal{Q}_d(\mathbf{x}),\mathcal{Q}_l(y))$
> $$I_1=2\mathbb{E}\left[\overline{\pmb{\eta}}_N^\top\left(\mathcal{Q}_l(y)\mathcal{Q}_d(\mathbf{x})-y\mathbf{x}\right)\right].$$
>
> - additional first-order term $I_2$ arising in the analysis of the average iterate covariance $\mathbb{E}[\overline{\pmb{\eta}} _ N\otimes\overline{\pmb{\eta}} _ N]$$$I_2=\frac{1}{2N^2}\cdot\sum _ {t=0}^{N-1}\sum _ {k=t}^{N-1}\sum _ {i=0}^{t-k-1}\left\langle \mathbf{H}^{(q)}, (\mathbf{I}-\gamma\mathbf{H}^{(q)})^i\mathbb{E}\left[Q _ {t-i}\otimes {\pmb{\eta}} _ t\right]+\mathbb{E}\left[\pmb{\eta} _ t\otimes Q _ {t-i}\right]  (\mathbf{I}-\gamma\mathbf{H}^{(q)})^i \right\rangle.$$ Here $Q _ t=\gamma\frac{1}{B}{\mathcal{Q} _ d(\mathbf{X}_t)}^\top \left(\pmb{\epsilon} _ t^{(o)}-\pmb{\epsilon} _ t^{(a)}-\mathcal{Q} _ d(\mathbf{X} _ t)\pmb{\epsilon} _ {t-1}^{(p)}\right)$.
>
> - additional quantization bias terms in the update rule of $\mathbb{E}[{\pmb{\eta}}_t\otimes{\pmb{\eta}}_t]$
>
> For the first two terms $I_1$ and $I_2$ that directly affect the excess risk, the following inequality: for any vector $\mathbf{u},\mathbf{v}$, $$\mathbf{u}\mathbf{v}^\top+\mathbf{u}\mathbf{v}^\top \preceq \mathbf{u}\mathbf{u}^\top+\mathbf{v}\mathbf{v}^\top$$ allows us to merge them into expressions involving $\mathbb{E}[{\pmb{\eta}}_t\otimes{\pmb{\eta}}_t]$ and quantization bias. Therefore, the only remaining technical challenge is to deal with the update rule of $\mathbb{E}[{\pmb{\eta}}_t\otimes{\pmb{\eta}}_t]$.
>
> Consider the biased quantization $\mathcal{Q}(\cdot)$: $$\mathcal{Q}(u) = u + \underbrace{B(u)} _ {\text{Deterministic Bias}} + \underbrace{\xi(u)} _ {\text{Zero-mean Noise}}.$$

---

> ### Author Response · Authors · 2025-11-23
> **Response to Reviewer WYgy's comment (2/7)**
>
> For **multiplicative quantization**, the quantization deterministic bias can be estimated by $\underline{b}\cdot u \leq B(u) \leq \overline{b}\cdot u, \quad \underline{b},\overline{b}\in \mathbb{R}.$ Therefore, the deviation error propagation in Lemma A.1 transforms to $$\pmb{\eta} _ t = \left(\mathbf{I}-\frac{b _ {p,a,o}}{B}\gamma {\mathcal{Q} _ d(\mathbf{X} _ t)}^\top \mathcal{Q} _ d(\mathbf{X} _ t) \right)\pmb{\eta} _ {t-1}+\gamma\frac{1}{B}{\mathcal{Q} _ d(\mathbf{X} _ t)}^\top \left(\pmb{\xi} _ t+\pmb{\xi} _ t^{(o,a,p)}\right),$$ where $b_{p,a,o}$ is a function of $\underline{b} _ p,\underline{b} _ a,\underline{b} _ o,\overline{b} _ p,\overline{b} _ a,\overline{b} _ o$ which capture the deterministic bias of parameter, activation and output gradient quantization respectively, and $\pmb{\xi} _ t^{(o,a,p)}$ is a linear function of $\pmb{\xi} _ t^{(a)},\pmb{\xi} _ t^{(o)},\mathcal{Q} _ d(\mathbf{X} _ t)\pmb{\xi} _ {t-1}^{(p)}$ with zero mean. With this formulation, we can directly apply the technical framework in unbiased quantization setting. It is worth noting that, a key difference from unbiased assumption is the difference in effective quantized data Hessian: $\widetilde{\mathbf{H}}^{(q)}=b_{p,a,o}\mathbf{H}^{(q)}$, which indicates that multiplicative quantization bias affects both bias error and variance error via influencing effective data Hessian. In particular, final excess risk bound can be formulated as: $$\mathbb{E}[\mathcal{E} _ N]\lesssim \mathrm{VarErr}+\mathrm{BiasErr}+\mathrm{ApproxErr},$$ where $$\mathrm{BiasErr}=\frac{1}{\gamma^2N^2}\cdot\Vert {\mathbf{w}^{(q)}}^* \Vert _ {(\widetilde{\mathbf{H}} _ {0:k^* }^{(q)})^{-1}}^2+\Vert {\mathbf{w}^{(q)}}^* \Vert _ {\widetilde{\mathbf{H}} _ { k^* :\infty}^{(q)} }^2,$$$$\mathrm{VarErr}=\frac{{\sigma^{(q)}}^2+\frac{2\alpha_B}{N\gamma}\left(\Vert {\mathbf{w}^{(q)}}^* \Vert _ {\widetilde{\mathbf{I}} _ { 0:k^* }^{(q)} }^2+N\gamma\Vert { \mathbf{w}^{(q)} }^* \Vert _ { \widetilde{\mathbf{H}} _ {k^* :\infty}^{(q)} }^2\right) }{1-\gamma\alpha _ B\mathrm{tr}\left(\widetilde{\mathbf{H}}^{(q)}\right) }\left(\frac{k^* }{N}+N\gamma^2\sum _ {i>k^* }{ \widetilde{\lambda} _ i^{(q)} }^2\right),$$$$\mathrm{ApproxErr}=\frac{1}{2}\langle\mathbf{H},\mathbb{E}[({\mathbf{w}}^* -{\mathbf{w}^{(q)}}^* )\otimes({\mathbf{w}}^* -{\mathbf{w}^{(q)}}^* )]\rangle+\frac{1}{2}\mathbb{E}\left[\langle {\mathbf{w}^{(q)}}^* ,\mathcal{Q} _ d(\mathbf{x})-\mathbf{x}\rangle^2\right].$$
> **Remark:** The additional approximation error $I_1$ is incorporated into bias error and variance error. Label quantization and data quantization affect ${\mathbf{w}^{(q)}}^*$, and thereby influence excess risk.
>
> For **additive quantization**, the quantization deterministic bias can be estimated by $\underline{\pmb{b}}\leq B(u) \leq \overline{\pmb{b}}.$ The deviation error propagation in Lemma A.1 then includes extra bias terms: $$\pmb{\eta} _ t = \left(\mathbf{I}-\frac{1}{B}\gamma {\mathcal{Q} _ d(\mathbf{X} _ t)}^\top \mathcal{Q} _ d(\mathbf{X} _ t) \right)\pmb{\eta} _ {t-1}+\gamma\frac{1}{B}{\mathcal{Q} _ d(\mathbf{X} _ t)}^\top \left(\pmb{\xi} _ t+\pmb{\xi} _ t^{(o,a,p)}+\pmb{b}^{(o,a,p)}\right).$$ The key implication is that although both multiplicative and additive quantization biases contribute to bias and variance errors, they do so qualitatively differently: **multiplicative quantization bias acts by modifying the effective data Hessian, while additive quantization bias affects the bias through an additional first-order term in the update rule of $\mathbb{E}[{\pmb{\eta}}_t\otimes{\pmb{\eta}}_t]$ (arises from non-zero mean $\pmb{b}^{(o,a,p)}$) and affects the variance by directly increasing the noise**. We defer the comprehensive study of additive biased quantization to future work.
>
> In conclusion, our analytic framework can extend to biased quantization where the quantization of labels, data, parameters, activations, and gradients will all affect bias and variance error.

---

> ### Author Response · Authors · 2025-11-23
> **Response to Reviewer WYgy's comment (3/7)**
>
> **Q2:** While motivated by scaling-law literature, the link between derived quantization effects and empirical scaling behaviors remains largely qualitative.
>
> **A2:** Thanks for your comment. Our theoretical framework for **high-dimensional linear regression serves as a principled starting point for deriving theoretical low-precision scaling laws** and we believe **the derived quantization effects still hold in scaling laws**. The primary reason is that our analytic framework is technically applicable to sketched linear regression, which is widely used to characterize the theoretical scaling laws [4,5].
>
> Specifically, building on linear regression which allows precise, closed-form relations between generalization error and key parameters such as data dimension, sample size, and quantization error (or bit-width), sketched linear regression model $f : \mathcal{H} \rightarrow \mathbb{R}, \mathbf{x} \to \langle \mathbf{v}, \mathbf{Sx}\rangle$ introduces a sketching matrix $\mathbf{S}\in \mathbb{R}^M\times \mathbb{H}$ to further capture the model size. Here $\mathbf{v}\in \mathbb{R}^M$ is a linear learner in a Hilbert space $\mathcal{H}$ (potentially infinite-dimensional). On technical level, by decomposing the population risk of sketched linear regression into $$\mathcal{R} _ M(\mathbf{v} _ N)=\underbrace{\min \mathcal{R}(\cdot)} _ {\rm Irreducible}+\underbrace{\min \mathcal{R} _ M(\cdot)-\min \mathcal{R}(\cdot)} _ {\rm Approx}+\underbrace{\mathcal{R} _ M(\mathbf{v} _ N)-\min \mathcal{R} _ M(\cdot)} _ {\rm Excess},$$ we can apply our theoretical framework directly to analyze the algorithm-dependent excess risk ($\rm Excess$). This **extension only requires a slight modification of the effective data Hessian** from $\mathbf{H}^{(q)}$ to $\mathbf{S}\mathbf{H}^{(q)}\mathbf{S}^\top.$ Consequently, **the quantization effects** we identify (data-spectrum distortion, noise amplification from parameter/activation/gradient quantization, etc.) **carry through quantitatively to the sketched setting**.
>
>
> **Q3:** How do non-uniform or mixed precision quantization methods fit into the framework here?
>
> **A3:** Thanks for your comment. Our main Theorem 4.1 is stated for general unbiased stochastic quantization, so it already covers uniform, non-uniform and mixed-precision schemes. The key is that we **approximate the conditional second moment of the quantization error $\epsilon$ by the quantization stepsize $\delta(x)$** which fully capture the properties of quantization methods (e.g., non-uniform quantization, mixed precision quantization). This approximation is formulated as $\mathbb{E}[{\epsilon}^2|x]\approx \delta^2(x).$
>
> - non-uniform quantization
>
>     Non-uniform quantization indicates that the **quantization step varies with input**. In our theoretical framework, non-uniform quantization corresponds to the case if $\mathbb{E}[{\epsilon}^2|x]$ depends on $x$. Specifically, our multiplicative quantization is exactly the non-uniform quantization method, where $\mathbb{E}[\pmb{\epsilon}\pmb{\epsilon}^\top|\mathbf{x}]=\epsilon \mathbf{x}\mathbf{x}^\top.$
>
> - mixed-precision quantization
>
>     Mixed-precison quantization refers that the **quantization stepsize is different among different quantization components (targets)** such as data, labels, parameters, activations and output gradients. Our theoretical framework characterizes general quantization second moment level ($\epsilon _ i, i=\{d,l,p,a,o\}$) both in multiplicative case $(\mathbb{E}[\mathcal{Q}_i(\mathbf{x})\mathcal{Q}_i(\mathbf{x})^\top|\mathbf{x}]=\epsilon_i \mathbf{x}\mathbf{x}^\top)$ and additive case ($\mathbb{E}[\mathcal{Q}_i(\mathbf{x})\mathcal{Q}_i(\mathbf{x})^\top|\mathbf{x}]=\epsilon_i \mathbf{I}$), which exactly fits mixed-precision quantization methods.

---

> ### Author Response · Authors · 2025-11-23
> **Response to Reviewer WYgy's comment (4/7)**
>
> **Q4:** Can it be generalized to multiple epochs and adaptive optimizers, would the excess risk decomposition change qualitatively?
>
> **A4:** Thanks for your comment. We would like to claim that the **algorithm-independent excess risk decomposition** from Lemma A.2 **continues to hold**, where the excess risk is decomposed into approximation error (algorithm-independent) and the algorithm-dependent components (bias error and variance error). We then discuss the algorithm-dependent component in multi-pass SGD and adaptive optimizer.
> - multi-pass SGD
>
> 	We believe under multi-pass (multi-epoch) SGD, our analytic framework can be used to analyze the algorithm-dependent component $R=\frac{1}{2}\langle \mathbf{H}^{(q)},\mathbb{E}[\overline{\pmb{\eta}} _ N\otimes\overline{\pmb{\eta}} _ N]\rangle$ based on existing techniques for full-precision multi-pass SGD [3]. Concretely, $R$ can be split into three components (similar to Eq. (5.3) in [3], for simplicity we consider batchsize $B=1$ and last iterate risk):$$ R=\underbrace{\frac{1}{2}\Vert \widehat{\mathbf{w}} _ {t}-{\mathbf{w}^{(q)}}^{* }\Vert _ {\mathbf{H}^{(q)}}^{2}} _ {\mathrm{quantized\ GD\ error}}+\underbrace{\frac{\eta^{2}}{2}\sum _ {k=0}^{t-1}\left\langle\mathcal{G}^{t-1-k}\circ(C _ i\mathcal{M}-\widetilde{\mathcal{M}})\circ\mathbf{E} _ {k},\mathbf{H}^{(q)}\right\rangle} _ {\text{quantized Fluctuation error}}+\mathrm{ExtraError}, \tag{* }$$where:
>
> 	- the quantizied multi-pass SGD iterate $\mathbf{w}_{t}$ is defined by $$\mathbf{w} _ {t}=\mathbf{w} _ {t-1}+\eta\mathcal{Q}(\mathbf{x} _ {i_t})\mathcal{Q} _ o\left(\mathcal{Q} _ l(y _ {i _ t})-\mathcal{Q} _ a\left(\mathcal{Q} _ d(\mathbf{x} _ {i _ t})^\top \mathcal{Q} _ p(\mathbf{w} _ {t-1})\right)\right),$$ with $\mathbf{E} _ {t}=\mathbb{E}\left[(\mathbf{w} _ {t}-\widehat{\mathbf{w}})(\mathbf{w} _ {t}-\widehat{\mathbf{w}})^\top\right]$ denoting the second moment of the deviation from the quantized minimum norm interpolator $\widehat{\mathbf{w}}=\mathcal{Q} _ d(\mathbf{X})^\top [\mathcal{Q} _ d(\mathbf{X})\mathcal{Q} _ d(\mathbf{X})^\top]^{-1}\mathcal{Q} _ l(\mathbf{y}).$
>
> 	- the quantized GD iterate $\widehat{\mathbf{w}}_{t}$ is defined by $$\widehat{\mathbf{w}} _ {t}=\widehat{\mathbf{w}} _ {t-1}+\eta\frac{1}{n}\sum _ {i=1}^n \mathcal{Q}(\mathbf{x} _ {i})\mathcal{Q}_o\left(\mathcal{Q} _ l(y _ {i})-\mathcal{Q} _ a\left(\mathcal{Q} _ d(\mathbf{x} _ {i})^\top \mathcal{Q} _ p(\widehat{\mathbf{w}} _ {t-1})\right)\right).$$
>
> 	- $\mathrm{ExtraError}$ arises from a critical discrepancy: $$\mathbb{E} _ {\rm SGD}\left[\mathbf{x} _ {i _ t}^{(q)}{\mathbf{x} _ {i _ t}^{(q)}}^\top\right]\neq \frac{1}{n}\sum _ {i=1}^n \mathbf{x} _ {i}^{(q)}{\mathbf{x} _ {i}^{(q)}}^\top,$$ as quantization operator is non-linear.
>
>     Following the decomposition $\mathbf{(* )}$, the analysis hinges on bounding the quantized GD error, the Fluctuation error and $\mathrm{ExtraError}$. **Given the strong structural similarity between quantized GD and quantized one-pass SGD, we can directly apply our technical framework to bound the quantized GD error**. Regarding the quantized Fluctuation error, we aim to establish an upper bound for the following expression: $${\mathbf{x} _ i^{(q)}}^\top(\mathbf{I}-\eta\mathbf{\Sigma}^{(q)})^k\mathbf{H}^{(q)}(\mathbf{I}-\eta\mathbf{\Sigma}^{(q)})^k\mathbf{x} _ i^{(q)}=\underbrace{{\mathbf{x} _ i^{(q)}}^\top(\mathbf{I}-\eta\mathbf{\Sigma}^{(q)})^k\mathbf{\Sigma}^{(q)}(\mathbf{I}-\eta\mathbf{\Sigma}^{(q)})^k\mathbf{x} _ i^{(q)}} _ {\Theta _ 1}+\underbrace{{\mathbf{x} _ i^{(q)}}^\top(\mathbf{I}-\eta\mathbf{\Sigma}^{(q)})^k(\mathbf{H}^{(q)}-\mathbf{\Sigma}^{(q)})(\mathbf{I}-\eta\mathbf{\Sigma}^{(q)})^k\mathbf{x} _ i^{(q)}} _ {\Theta _ 2}.$$ A technical barrier to analyze $\Theta_1$ and $\Theta_2$ is that existing concentration techniques in random matrix analysis can not be directly applied to derive concentration bounds for quantized data i.e., $\mathbf{\Sigma}^{(q)}$. We defer the comprehensive study of these quantized random matrix analysis and the discrepancy in $\mathrm{ExtraError}$ to future work.

---

> ### Author Response · Authors · 2025-11-23
> **Response to Reviewer WYgy's comment (5/7)**
>
> - adaptive optimizer
>
> 	We believe **our theoretical framework can be effectively extended to analyze Adam by partitioning the gap between Adam and SGD into two components: momentum and preconditioning**.
>
> 	- SGD with momentum
>
> 		We can extend our theory to quantized SGD with momentum by combining our technical framework for quantized SGD with existing techniques for full-precision SGD with momentum [6]. In particular, the accelerated SGD (ASGD) scheme in [6] maintains three sequences $\mathbf{u} _ t, \mathbf{v} _ t, \mathbf{w} _ t$ which are updated as $$\mathbf{u} _ {t-1}=\alpha\mathbf{w} _ {t-1}+(1-\alpha)\mathbf{v} _ {t-1},\quad \mathbf{w} _ {t}=\mathbf{u} _ {t-1}-\delta\widehat{\nabla}L(\mathbf{u} _ {t-1}),\quad \mathbf{v} _ t=\beta\mathbf{u} _ {t-1}+(1-\beta)\mathbf{v} _ {t-1}-\gamma\widehat{\nabla}L(\mathbf{u} _ {t-1}).$$ Define the centered ASGD iterate $\pmb{\eta}_t:=\begin{bmatrix}\mathbf{w} _ t-\mathbf{w}^* \newline\mathbf{u} _ t-\mathbf{w}^* \end{bmatrix}.$ Then it holds $$\pmb{\eta} _ t=\widehat{\mathbf{A}} _ t\pmb{\eta} _ {t-1}+\pmb{\zeta} _ t,\quad\mathrm{where}\quad\widehat{\mathbf{A}} _ t:=\begin{bmatrix}\mathbf{0} & \mathbf{I}-\delta\mathbf{x} _ t\mathbf{x} _ t^\top \newline -c\mathbf{I} & (1+c)\mathbf{I}-q\mathbf{x} _ t\mathbf{x} _ t^\top\end{bmatrix},\quad\pmb{\zeta} _ t:=\begin{bmatrix}\delta\cdot\epsilon _ t\mathbf{x} _ t \newline q\cdot\epsilon _ t\mathbf{x} _ t\end{bmatrix}.$$ This propagation is similar to the update rule in one-pass SGD. Under quantization with our analytic technique, $\mathbf{x} _ t$ will be replaced with its quantized counterpart $\mathbf{x} _ t^q$ and noise term $\epsilon _ t$ will be amplified by quantization errors. Together with techniques in [6], our theoretical framework can be extended to the ASGD case, i.e., SGD with momentum.
>
> 	- Preconditioning SGD
>
> 		We can also extend our theory to preconditioning quantized SGD by combining our technical framework for quantized SGD with existing techniques for full-precision preconditioning SGD [7]. Specifically, preconditioning SGD is formulated as $$\mathbf{w} _ {t+1}=\mathbf{w} _ t-\eta\cdot\mathbf{G}\nabla l(\mathbf{w} _ t;\mathbf{x} _ t,y _ t)=\mathbf{w} _ t-\eta\cdot(\langle\mathbf{w} _ t,\mathbf{x} _ t\rangle-y _ t)\cdot\mathbf{G}\mathbf{x} _ t,$$ where $\mathbf{G}$ is the preconditioning matrix. This preconditioner can be directly applied to quantized SGD simply by incorporating $\mathbf{G}$ into the update step.
>
> 		Technically, the primary effect of this preconditioner is to **transform the effective data Hessian $\mathbf{H}$ into a rescaled Hessian $\widetilde{\mathbf{H}}=\mathbf{G}^{1/2}\mathbf{H}\mathbf{G}^{1/2}$**. In quantized SGD case, this preconditioning can be exteded with the effective quantized data Hessian becoming $\widetilde{\mathbf{H}}^{(q)}=\mathbf{G}^{1/2}\mathbf{H}^{(q)}\mathbf{G}^{1/2}$.
>
> 	By sequentially demonstrating how our framework adapts to both the momentum and the preconditioning mechanisms, we are confident that the technical tools developed in this paper are a strong foundation for extending the analysis to full adaptive optimization algorithms like Adam. In Adam, the update rule is a combination of these two components: $$\mathbf{w} _ {t+1} = \mathbf{w} _ t - \eta \cdot \underbrace{\mathbf{G} _ t} _ {\text{Preconditioning}} \cdot \underbrace{\hat{\mathbf{m}} _ t} _ {\text{Momentum}},$$ where the preconditioning matrix $$\mathbf{G} _ t = \frac{1}{\sqrt{\hat{\mathbf{v}} _ t} + \epsilon} \quad (\text{operations applied element-wise})$$ with $$\mathbf{v} _ t = \beta _ 2 \mathbf{v} _ {t-1} + (1 - \beta _ 2) (\mathbf{g} _ t)^2, \quad\hat{\mathbf{v}} _ t = \frac{\mathbf{v} _ t}{1 - \beta _ 2^t},$$ and the momentum vector $$\mathbf{m} _ t = \beta _ 1 \mathbf{m} _ {t-1} + (1 - \beta _ 1) \mathbf{g} _ t, \quad \hat{\mathbf{m}} _ t = \frac{\mathbf{m} _ t}{1 - \beta _ 1^t}.$$ To fully extend our analytic framework to Adam, it is also necessary to develop techniques for time-varying preconditioner. We defer this to future work.

---

> ### Author Response · Authors · 2025-11-23
> **Response to Reviewer WYgy's comment (6/7)**
>
> **Q5:** Since the case study is on polynomials decaying, I am curious how would the results change for exponentially decaying or heavy tail spectra which also happen in real world embeddings.
>
> **A5:** Thanks for your comment. We first remark that the key implication that additive data quantization scales with dimension $d$ while multiplicative data quantization is independent of $d$ still holds as long as $\mathrm{tr}(\mathbf{H})$ is on constant level. We then explain how different spectral decay affects excess risk under multiplicative and additive quantization. Intuitively,
>
> - **For faster spectrum decay such as exponential decay, the excess risk under multiplicative data quantization improves significantly.** A rapidly decaying $\mathbf{H}$ leads to a rapidly decaying $\mathbf{H}^{(q)}$, as quantized data spectrum scales with full-precision spectrum, i.e., $\mathbf{H}^{(q)}=(1+\epsilon_d)\mathbf{H}$.
>
> - In contrast, **additive data quantization does not see this benefit remarkably**, as constant $\epsilon_d$ adds to all eigenvalues which is irreducible regardless of how fast the decay is. This additive error dominates the small, fast-decaying eigenvalues, leading to a much poorer bound.
>
> - **For heavy tail spectra, the performance gap between multiplicative data quantization and additive data quantization diminishes**, as the intrinsic spectrum tail ($\sum_{i>k^* } \lambda_i$) is already large and the additive error from additive data quantization is no longer the dominant bottleneck. In extreme case $\mathbf{H}=\mathbf{I}$, multiplicative data quantization is equivalent to additive data quantization.
>
> Specifically, for faster decay such as exponential decay $\lambda_k=e^{-k}$,
>
> - multiplicative case
> 	$$\mathbb{E}[\mathcal{E}^{(M)}(\overline{\mathbf{w}}_N)]\lesssim\epsilon_d+\frac{\log N \wedge  d}{N}\left(1+\frac{\sigma^2}{B}+\epsilon_p+\epsilon_a+\epsilon_o\right).$$
>
> - additive case
> 	$$\mathbb{E}[\mathcal{E}^{(M)}(\overline{\mathbf{w}} _ N)]\lesssim\epsilon _ d\cdot d+\frac{d _ {eff}}{N}\left[1+\frac{d _ {eff}}{N}+\frac{\sigma^2+\epsilon _ a+\epsilon _ o}{B}+\epsilon _ p(1+\epsilon _ d\cdot d)\right],$$ where $d _ {eff}=-\log \left((\frac{1}{N\gamma}-\epsilon _ d)\vee e^{-d} \right)+\epsilon _ d N\gamma\left(d+\log \left((\frac{1}{N\gamma}-\epsilon _ d)\vee e^{-d} \right)\right).$
>
> Compared with the slower polynomial decay in Corollary 4.3, **multiplicative quantization demonstrates an improved excess risk bound (from $\frac{1}{N^{1-1/a}}$ to $\frac{\log N}{N}$)**. Additive quantization, in contrast, does not see this benefit significantly:
>
> - Case 1: $e^{-d} \geq \frac{1}{N\gamma}-\epsilon_d$
>
> 	$$\mathbb{E}[\mathcal{E}^{(M)}(\overline{\mathbf{w}}_N)]\lesssim\epsilon_d\cdot d+\frac{d}{N}\left[1+\frac{d}{N}+\frac{\sigma^2+\epsilon_a+\epsilon_o}{B}+\epsilon_p(1+\epsilon_d\cdot d)\right].$$ **The excess risk bound matches the bound in polynomial decay case** when $\frac{1}{N\gamma}-\epsilon_d$ is small.
>
> - Case 2: $e^{-d} \leq \frac{1}{N\gamma}-\epsilon_d$
>
> 	$$\mathbb{E}[\mathcal{E}^{(M)}(\overline{\mathbf{w}} _ N)]\lesssim\epsilon _ d\cdot d+\frac{d _ {eff}}{N}\left[1+\frac{d _ {eff}}{N}+\frac{\sigma^2+\epsilon _ a+\epsilon _ o}{B}+\epsilon _ p(1+\epsilon _ d\cdot d)\right],$$ where $d_{eff}=-\log \left(\frac{1}{N\gamma}-\epsilon_d \right)+\epsilon_dN\gamma\left(d+\log \left(\frac{1}{N\gamma}-\epsilon_d \right)\right)$. Though $d_{eff}$ is improved compared with polynomial decay case, the term $d\epsilon_d N\gamma$ is irreducible. **The excess bound is not improved when this term dominates**.
>
> Such a comparison, in turn, also indicates that for heavier tail spectra (slower decay), performance under multiplicative quantization worsens, and, consequently, the gap between multiplicative and additive data quantization narrows.

---

> ### Author Response · Authors · 2025-11-23
> **Response to Reviewer WYgy's comment (7/7)**
>
> **Q6:** Experiment of real world dataset / how sensitive to batch size and spectrum decay
>
> **A6:** Thanks for your comment. We additionally evaluate on the publicly available $\texttt{Communities and Crime}$ dataset, which contains community-level statistics from across the United States, and we also evaluate on the **synthetic dataset** under the case of **batch size** $B=10$ and **exponential decay**.
>
> - Performance on $\texttt{Communities and Crime}$ dataset under multiplicative and additive quantization
>
> 	We apply both additive and multiplicative quantization schemes to the regression task on $\texttt{Communities and Crime}$ dataset. Results are shown in Figure 2(a) in Section G in the revised manuscript. Specifically, **under multiplicative quantization, excess risk is comparable to that of full-precision training**. In contrast, **additive quantization yields substantially higher excess risk than both full precision and multiplicative quantization**.
>
> - Effects on data spectrum under multiplicative and additive quantization
>
> 	To explain the different generalization behaviors, we examine the empirical covariance spectra produced by each quantization scheme. Results are shown in Figure 2(b) in Section G in the revised manuscript. Crucially, **multiplicative quantization largely preserves the full-precision eigenvalue structure**, whereas **additive quantization introduces a marked, irreducible spectral gap** relative to the full-precision spectrum.
>
> - Sensitivity to batch size / spectral decay
>
> 	We complement the $B=1$ case and the polynomial-decay setting used in the main paper.
>
> 	- batch size $B=10$
>
> 		We fix $B=10$ and $d=200$, and vary the quantization error level $\varepsilon\in\{0.001,\,0.005,\,0.01\}$ for each scheme. Results are shown in Figure 3(a,b) in Section G in the revised manuscript, respectively for multiplicative and additive quantization.
>
> 	- exponential decay
>
> 		We replace the spectral structure with the exponential-decay synthetic dataset. Results are shown in Figure 3(c,d) in Section G in the revised manuscript, respectively for multiplicative and additive quantization.
>
> 	The experiments show a clear contrast: **as the quantization error level increases, the excess risk under additive quantization rises sharply, whereas under multiplicative quantization the excess risk remains close to that of full-precision training across all error levels**.
>
> Overall, these empirical findings align with previous finding and validate our theory: **additive quantization errors distort the data Hessian spectrum, increasing risk, whereas multiplicative quantization errors diminish the spectral gap, maintaining risk despite higher error levels**.
>
>
> [1] Stochastic Rounding for LLM Training: Theory and Practice.
>
> [2] Training with Fewer Bits: Unlocking Edge LLMs Training with Stochastic Rounding.
>
> [3] Risk Bounds of Multi-Pass SGD for Least Squares in the Interpolation Regime.
>
> [4] Scaling Laws in Linear Regression: Compute, Parameters, and Data.
>
> [5] Improved Scaling Laws in Linear Regression via Data Reuse.
>
> [6] Risk Bounds of Accelerated SGD for Overparameterized Linear Regression.
>
> [7] Improving Implicit Regularization of SGD with Preconditioning for Least Square Problems.
>
> [8] Microadam: Accurate adaptive optimization with low space overhead and provable convergence.
>
> [9] Qsgd: Communication-efficient sgd via gradient quantization and encoding.

---

### Official Review · Reviewer_fw7v · 2025-11-03

**Soundness:** 2
**Presentation:** 2
**Contribution:** 2
**Rating:** 4
**Confidence:** 4

**Summary:**

This paper develops a theoretical framework for understanding how low-precision quantization affects the learning performance of stochastic algorithm descent (SGD) for the problem of high-dimensional linear regression. This paper specifically focuses on the impact of quantization on the generalization (population) risk (not just the training loss/sample risk). Quantization is applied to five components of training: Data (both features and labels), model parameters, activations and gradients. Subsequently, the iterate-averaged SGD is analyzed using two quantization schemes:

1. Additive quantization, where the absolute quantization error is constant -- capturing the effect of fixed-point precision formats (e.g., INT8).

2. Multiplicative quantization, where the relative quantization error is constant (in other words, quantization error scales with the magnitude of the quantization input) -- capturing the effect of floating-point precision formats (e.g., FP8).

The key contributions of the paper include:

1. Derivation of explicit bounds on the generalization risk showing contributions from variance and bias (which are prevalent even for full-precision SGD, i.e., no quantization error), approximation error from the quantization of data/labels, and additional quantization error, arising from accumulation of quantization error from the SGD iterates.

2. Study of how the distortion is input data spectrum due to quantizing the features/labels affects the generalization risk. More specifically, the authors propose that multiplication quantization does not lead to significant spectral distortion and matches the sample complexity of full-precision SGD under mild conditions. On the other hand, for additive quantization, spectral distortion increases with dimension, and leads to worse sample complexity.

3. The work also considers a polynomial eigenvalue decay of the data spectrum, and proposes when additive vs. multiplicative quantization is preferred on the other.

**Strengths:**

The author positions this work in the context of LLMs and low-precision training, precision-scaling laws, and benign overfitting and overparameterization theory -- which is pretty relevant in today's ML landscape where bigger models and more data are preferred for better performance. The paper tackles a pretty fundamental question: How does quantization affect the generation performance of SGD for linear regression, in contrast to prior works that focus on convergence of optimization algorithms under quantization. The systematic study of quantizing data, label, parameter, activation and gradient is a major contribution as well -- prior works usually isolate one or two of these. Modeling quantization schemes as multiplicative or additive also provides an useful abstraction between this work's theoretical analysis and the data formats generally used in practice (e.g., FP8 and INT8).

**Weaknesses:**

Firstly, I have a semi-major concern: The paper implicitly assumes quantization is purely detrimental and focuses on bounded degradation relative to full precision. But quantization introduces stochasticity that also plays a role analogous to implicit regularization similar to SGD noise, weight-decay, etc. (Ref: https://arxiv.org/abs/2101.12176). There are also some empirical works in deep-learning where low-precision improves generalization slightly (e.g., https://arxiv.org/abs/2206.12372). The theory for linear regression is exactly the setting where we would want to understand whether/when this can occur. In my opinion, this is a key conceptual gap, which is not apparent from the main body of the paper (pl. correct me if I have missed anything). It would be highly appreciated if the authors could comment on the following:

1. Discuss whether their analysis explicitly take into account this implicit regularization? If yes, where exactly? If not, is it a limitation of the current analysis, or is studying the implicit regularization not relevant?
2. If the authors agree with the potential benign regularization effects of quantization, under what conditions might they emerge?

Secondly, I think the writing of the paper can be significantly improved. Currently, the theorems statements in the main paper have pretty long expressions. If it is possible to state simplified/informal versions of the theorem and/or corollaries wherever relevant to extract the essence of the theorem, I think it would make the paper highly approachable.

Real world linear regression experiments would also be appreciated, where the assumptions on the eigen-decay of the data matrix are justified. I believe this is important because the section on **Implications to integer and FP quantization** is quite relevant to practitioners in determining the choice of precision formats to be used for quantization. However, in practice, INT8 and FP8 perform for or less similar to each other (perhaps because for most recent models, dimension is high?) Sometimes, INT8 is even preferred because INT8 matmuls are faster than FP8 matmuls.

**Questions:**

I have a few questions/suggestions are would appreciate it if they are addressed:

1. Why is $Q_o$ referred to as the quantizer for *output gradient*? Shouldn't the input to $Q_o$ also include the term $Q_d(\mathbf{X}_t)^\top$, and not just the second factor?

2. Recent techniques for quantized SGD like error feedback (https://arxiv.org/abs/1909.05350) are not considered in the analysis. This somewhat limits the applicability of the results, since error feedback has been around for a few years now and is pretty widely used for optimization with compressed gradients. This should be explicitly highlighted in the limitations section.

---

> ### Author Response · Authors · 2025-11-23
> **Response to Reviewer fw7v's comment (1/3)**
>
> Thank you for your positive feedback and we appreciate your strong support for the completeness of our work. We address your concerns in the following response:
>
> **Q1:** Relation to implicit regularization of quantization.
>
> - **Q1.1:** Discuss whether this analysis capture this implicit regularization? If not, is that a limitation?
>
>     **A1.1:** Thanks for your comment. That's a good point. In fact, the noisy update (including quantized update) **can introduce implicit regularization** in the setting of optimizing empirical risk, i.e., $\tilde{\mathcal{L}}=\frac{1}{m}\sum_{i=1}^{m}\Vert\hat{y_i}^{(q)}-y_i\Vert_2^2$. As established in Theorem 3.1 of [1], minimizing the quantized empirical risk ($\tilde{\mathcal{L}}$) is approximately equivalent to minimizing its full-precision counterpart ($\mathcal{L}=\frac{1}{m}\sum_{i=1}^{m}\Vert\hat{y_i}-y_i\Vert_2^2$) plus a gradient norm regularization term:$$\tilde{\mathcal{L}}\approx\mathcal{L}+\frac{\sigma\delta^2}{m}\sum_{i=1}^m\Vert\nabla_w\hat{y}_i\Vert_2^2. \tag{*}$$ This gradient norm penalty regularizes the empirical risk and **helps alleviate overfitting that typically occurs when minimizing the empirical risk**, thereby improving generalization behavior.
>
> 	In our **quantized one-pass online SGD algorithm** where each sample is processed essentially only once however, the model is naturally less prone to overfitting the limited training data compared to the multi-pass setting. Consequently, the optimization goal shifts from minimizing the empirical risk to minimizing the quantized population risk to some extents. That said, while the empirical risk can still be theoretically decomposed as in $(*)$, the presence of the gradient norm term now primarily contributes to the gap between the quantized population risk and the full-precision population risk, instead of the implicit regularization bias. This gap is analyzed as an **extra generalization error bias** rather than an explicit regularization intended to curb overfitting.
>
> 	Therefore, our analysis focuses on bounded degradation relative to full precision is not a limitation. Instead, it is a property induced by the nature of the one-pass online SGD algorithm.
>
> - **Q1.2:** If the authors agree with the potential benign regularization effects of quantization, under what conditions might they emerge?
>
>     **A1.2:** Thanks for your comments. The potential for quantization to act as an implicit regularizer emerges primarily **under conditions that are prone to overfitting the training data**. For example, in **multi-pass SGD** where the same training set is reused many times, the noise introduced by quantization effectively behaves like an explicit regularizer as the optimization process is focused on minimizing the empirical risk.
>
>
> **Q2:** Simplified/informal versions of Theorems and Corollaries
>
> **A2:** Thanks for your comment. We have simplified bounds in Theorem 4.1 by compressing the non-dominant terms into the dominant terms, and introducing auxiliary notations in the revised manuscript.
>
> **Q3:** Real-world linear regression experiments
>
> **A3:** Thanks for your comment. We additionally evaluate on the publicly available $\texttt{Communities and Crime}$ dataset, which contains community-level statistics from across the United States.
>
> - Performance on $\texttt{Communities and Crime}$ dataset under multiplicative and additive quantization
>
> 	We apply both additive and multiplicative quantization schemes to the regression task on $\texttt{Communities and Crime}$ dataset. Results are shown in Figure 2(a) in Section G in the revised manuscript. Specifically, **under multiplicative quantization, excess risk is comparable to that of full-precision training**. In contrast, **additive quantization yields substantially higher excess risk than both full precision and multiplicative quantization**.
>
> - Effects on data spectrum under multiplicative and additive quantization
>
> 	To explain the different generalization behaviors, we examine the empirical covariance spectra produced by each quantization scheme. Results are shown in Figure 2(b) in Section G in the revised manuscript. Crucially, **multiplicative quantization largely preserves the full-precision eigenvalue structure**, whereas **additive quantization introduces a marked, irreducible spectral gap** relative to the full-precision spectrum.
>
> Overall, these empirical findings align with previous finding and validate our theory: **additive quantization errors distort the data Hessian spectrum, increasing risk, whereas multiplicative quantization errors diminish the spectral gap, maintaining risk despite higher error levels**.

---

> ### Author Response · Authors · 2025-11-23
> **Response to Reviewer fw7v's comment (2/3)**
>
> **Q4:** Why is $\mathcal{Q}_o$ referred to as the quantizer for output gradient? Shouldn't the input to $\mathcal{Q}_o$ also include the term $\mathcal{Q}_d(\mathbf{X}_t)^\top$, and not just the second factor?
>
> **A4:** Thanks for your comment. Our definition of $\mathcal{Q}_o$ is based on the widely adopted **mixed-precision training framework** introduced in [2]. In standard backpropagation, the full-precision gradient of the loss $\mathcal{L}$ with respect to the parameter $\mathbf{W}$ (used to update the master weight) is computed as the product of two factors: $$\nabla \mathcal{L} = \frac{\partial \mathcal{L}}{\partial \mathbf{W}} = \mathbf{X}_t^\top \cdot \frac{\partial \mathcal{L}}{\partial \mathbf{A}_t},$$ where $\mathbf{X}_t$ is the input and $\partial \mathcal{L}/\partial \mathbf{A}_t$ is the gradient of the loss with respect to the activation output. **The mixed-precision paradigm quantizes these two factors to low-precision format separately** (as detailed in Figure 6 of [2]):
> - The input data $\mathbf{X}_t$ is quantized by $\mathcal{Q}_d(\cdot)$.
> - The gradient w.r.t. activation output $\partial \mathcal{L}/\partial \mathbf{A}_t$ is quantized by $\mathcal{Q}_o(\cdot)$.
>
> The primary reason is that **the gradient is used for master weight update** which is in full precision FP32. Therefore, $\mathcal{Q}_o(\cdot)$ is specifically defined to quantize the second factor $(\partial \mathcal{L}/\partial \mathbf{A}_t)$, which we refer to as the output gradient component, aligning with the low-precision training implementation.
>
> We confirm that **our analytic technique remains fully applicable** even under the alternative setting where $\mathcal{Q}_o$ also includes the term $\mathcal{Q}_d(\mathbf{X}_t)^\top$. Specifically, directly quantizing the complete parameter gradient modifies the error propagation term ($\pmb{\eta}_t$) in Lemma A.1 to: $$\pmb{\eta}_t = \left( \mathbf{I}-\frac{1}{B}\gamma {\mathcal{Q}_d(\mathbf{X}_t)}^\top\mathcal{Q} _ d (\mathbf{X} _ t) \right)\pmb{\eta} _ {t-1}+\gamma\frac{1}{B}\left[\pmb{\epsilon} _ t^{(o)}+\mathcal{Q}_d(\mathbf{X} _ t)^\top\left( \pmb{\xi}_t-\pmb{\epsilon}_t^{(a)}-\mathcal{Q}_d(\mathbf{X} _ t) \pmb{\epsilon} _ {t-1}^{(p)}\right)\right].$$ Crucially, this modified error propagation has the same structure as previous update rule (Lemma A.1) with a **main linear term and zero-mean noise terms**. Therefore, our established excess risk analysis techniques remains entirely valid, ensuring the generality of our theoretical findings.

---

> ### Author Response · Authors · 2025-11-23
> **Response to Reviewer fw7v's comment (3/3)**
>
> **Q5:** Recent techniques for quantized SGD like error feedback are not considered in the analysis.
>
> **A5:** Thanks for your comment. That's a good point. We follow current mixed-precision training pipelines in industry practice that **do not require error feedback mechanisms** [2]. The primary reason for this exclusion is that error feedback mechanisms **introduce significant memory** and bandwidth overheads, which directly conflict with the core objective of quantization (i.e., reducing resource consumption). Concretely, standard error-feedback implementations maintain a full-precision residual (error) tensor of the same dimensionality as the model parameters or updates in order to accumulate quantization error across iterations. This residual is an extra parameter-sized buffer that must be stored and frequently read/written during training, inducing a substantial extra memory and bandwidth burden that is undesirable in the large-scale settings.
>
> On the other hand, we believe **our analytic framework can extend to the quantized SGD with error feedback mechanism**. Specifically, quantized SGD with error feedback mechanism [3,4] is typically expressed as the following coupled update rules:$$\mathbf{w} _ {t+1}=\mathbf{w} _ t - \mathcal{Q}(\gamma \mathbf{g} _ t(\mathbf{w} _ {t})+\mathbf{e} _ t),$$$$\mathbf{e} _ {t+1}=\mathbf{e} _ t+\gamma \mathbf{g} _ t (\mathbf{w} _ {t})-\mathcal{Q}(\mathbf{e} _ t+\gamma \mathbf{g} _ t(\mathbf{w} _ {t})),$$where $\mathbf{g} _ t(\mathbf{w} _ {t})$ is the quantized gradient which is the same as that in our quantized SGD setting, and $\mathbf{e}_t$ captures the accumulated error. Existing technique [3,4] started from virtual iterates $$\widetilde{\mathbf{w}} _ {t+1}=\mathbf{w} _ {t+1}-\mathbf{e} _ {t+1}=\widetilde{\mathbf{w}} _ {t}-\gamma \mathbf{g} _ t(\mathbf{w} _ {t})$$and demonstrated that under certain assumptions, $\mathbb{E}\Vert\mathbf{e}_t\Vert^2$ is bounded and $\mathbf{g} _ t(\mathbf{w} _ {t})\approx \mathbf{g} _ t(\widetilde{\mathbf{w}} _ {t}).$ Therefore, the key technical challenges for a formal extension would be: (i) analyze the quantized virtual iterates and (ii) approximate the error $\mathbf{e}_t$ and the gradient gap between original iterates and virtual iterates under quantization.
>
> We would like to remark that the similarity between our quantized SGD iterates and the quantized virtual iterates $\widetilde{\mathbf{w}}_{t+1}$ strongly suggests that the core techniques developed in our paper are applicable to analyze challenge (i). Regarding challenge (ii), we believe current full-precision approximation analysis can apply to low-precision setting, by including additional quantization errors into the approximation error. Overall, our theoretical framework remains applicable under the error feedback mecahnisms.
>
>
> [1] QReg: On Regularization Effects of Quantization.
>
> [2] DeepSeek-V3 Technical Report.
>
> [3] The Error-Feedback Framework: Better Rates for SGD with Delayed Gradients and Compressed Updates.
>
> [4] Error Feedback Fixes SignSGD and other Gradient Compression Schemes.

---

### Meta-Review · Area_Chair_piaT · 2026-01-07

**Summary:**

This paper provides upper bounds on the excess risk of the parameter learned by SGD. Here, the parameter is real, and the intermediate values can be quantized via additive/multiplicative quantization schemes. The paper is well-written, and address an important problem that has been overlooked in recent theory. I think the authors did not properly address some of reviewers' comments about stochastic rounding and unquantized model parameters; however, other than this, they generally responded well. Hence, I recommend the acceptance of this paper.

If this paper is accepted, I recommend the authors to revise the manuscript to include the bounds when $w_t$ is quantized. In addition, I found that the authors did not properly write values of $R1$ and $R3$ in the proof of Lemma A.2. This should be fixed.

**Reviewer Concerns:**

I think the authors did not properly address some of the Reviewer D5UN's comments:
- On the stochastic rounding. The authors answered that stochastic rounding is commonly adopted in practical low-precision large-scale model training. However, I think stochastic rounding is not commonly adopted at this stage since only a few modern hardwares (e.g., AWS trainium) start to support the stochastic rounding, while GPUs only support the round-to-nearest mode.
- On quantizing the weight parameter $w$. The authors claimed that If the master weights themselves are quantized, the small gradient updates would be lost (i.e., rounded to zero), preventing the model from converging. I disagree with this statement. The authors are considering stochastic rounding in this paper, and with the stochastic rounding, small gradient updates will not be lost but they will incur update with small probability.

**Reviewer Scores:**

I think Reviewers MJwg and WYgy would maintain their original positive scores, and Reviewers fw7v and D5UN may increase their scores if the authors' revision and response are correct.

---

### Decision · Program_Chairs · 2026-01-26

Accept (Poster)